# Beyond 2D landslide inventories and their rollover: synoptic 3D inventories and volume from repeat LiDAR data

Thomas G. Bernard, Dimitri Lague, Philippe Steer

Univ Rennes, CNRS, Géosciences Rennes - UMR 6118, 35000, Rennes, France

Correspondence: Thomas Bernard (thomas.bernard@univ-rennes1.fr)

**Abstract.**

Efficient and robust landslide mapping and volume estimation is essential to rapidly infer landslide spatial distribution, to

quantify the role of triggering events on landscape changes and to assess direct and secondary landslide-related geomorphic hazards. Many efforts have been made to develop landslide mapping methods, based on 2D satellite or aerial images, and to constrain the empirical volume-area (V-A) relationship allowing in turn to offer indirect estimates of landslide volume. Despite these efforts, major issues remain including the uncertainty of the V-A scaling, landslide amalgamation and the under-detection of landslides. To address these issues, we propose a new semi-automatic 3D point cloud differencing method to detect

geomorphic changes, filter out false landslide detections dues to LiDAR elevation errors, obtain robust landslide inventories with an uncertainty metric and directly measure the volume and geometric properties of landslides. This method is based on the M3C2 algorithm and was applied to a multi-temporal airborne LiDAR dataset of the Kaikoura region, New Zealand, following the $M_w$ 7.8 earthquake of 14 November 2016.

In a 5 km² area, the 3D point cloud differencing method detects 1118 potential sources. Manual labelling of 739 potential

sources, shows a prevalence of false detections of 24.4 % in forest-free area due to spatially correlated elevation errors, and 80 % in forested areas related to ground classification errors of the pre-EQ dataset. Combining the distance to the closest deposit and a signal-to-noise ratio metrics, the filtering step of our workflow reduces the prevalence of false source detections below 1 % in terms of total area and volume of the labelled inventory. The final predicted inventory contains 433 landslide sources and 399 deposits with a lower limit of detection size of 20 m² and a total volume of $724,297 \pm 141,087$ for sources and

$954,029 \pm 159,188$ m³ for deposits. Geometric properties of the 3D source inventory, including the V-A relationship, are consistent with previous results, except for the lack of the classically observed rollover of the distribution of source area. A manually mapped 2D inventory from aerial image comparison has a better lower limit of detection (6 m²) but only identified 258 landslide scars, exhibits a rollover in the distribution of source area around 20 m² and underestimates the total area and volume of 3D detected sources by 72 % and 58 %, respectively. Detection and delimitation errors in 2D occur in areas with

limited texture change (bare rock surfaces, forests) and at the transition between sources and deposits that the 3D method accurately captures. Large rotational/translational landslides and retrogressive scars can be detected in 3D whatever the vegetation cover but are missed in the 2D inventory owing to the dominant vertical topographic change. The 3D inventory

misses shallow (< 0.4 m depth) landslides detected in 2D, corresponding to 10 % of the total area and 2% of the total volume of the 3D inventory. Our data show a systematic size-dependent under-detection in the 2D inventory below 200 m² that may explain all or part of the rollover observed in 2D landslide source area distribution. While the 3D segmentation of complex clustered landslide sources remains challenging, we demonstrate that 3D point cloud differencing offers a greater sensitivity to detect small changes than a classical difference of digital elevation models (DEMs). Our results underline the vast potential of 3D-derived inventories in quantifying exhaustively and objectively the impact of extreme events on topographic change in regions prone to landsliding, detect a variety of hillslope mass movements that cannot be captured by 2D landslide mapping and to explore in new ways the scaling properties of landslides.

## 1. Introduction

In mountain areas, extreme events such as large earthquakes and typhoons can trigger important topographic changes through landsliding. Landslides are a key agent of hillslope and landscape erosion (Keefer, 1994; Malamud et al., 2004) and represent a significant hazard for local populations (e.g. Pollock and Wartman, 2020). Efficient, rapid and exhaustive mapping of landslides is required to robustly infer their spatial distribution, their total volume and the induced landscape changes (Guzzetti et al., 2012). Such information is crucial to understand the role of triggering events on landscape evolution and to manage direct and secondary landslide-related hazards. For instance, total volume produced by landsliding is essential to evaluate if earthquakes tend to build or destroy topography (e.g. Marc et al., 2016; Parker et al., 2011), to quantify the contribution of extreme events to long-term denudation (Marc et al., 2019) or to predict hydro-sedimentary hazards such as river avulsion related to the downstream transport of landslide debris (Croissant et al., 2017). Following a triggering event, total landslide volume over a regional scale is classically determined in two steps: (i) individual landslide mapping using 2D satellite or aerial images (e.g. Behling et al., 2014; Fan et al., 2019; Guzzetti et al., 2012; Li et al., 2014; Malamud et al., 2004; Martha et al., 2010; Massey et al., 2018; Parker et al., 2011) and (ii) indirect volume estimation using a volume-area relationship (e.g. Larsen et al., 2010; Simonett, 1967):

$$V = \alpha A^{\gamma} \tag{1}$$

with $V$ and $A$ the volume and area of individual landslides, $\alpha$ a prefactor and $\gamma$ a scaling exponent usually ranging between 1.1 and 1.6 (e.g. Larsen et al., 2010; Massey et al., 2020; Pollock and Wartman, 2020)

A first source of error comes from the uncertainty on the values of $\alpha$ and $\gamma$ which tend to be site specific and potentially process specific (e.g. shallow versus bedrock landsliding). This uncertainty could lead to an order magnitude of difference in total estimated volume given the non-linearity of eq. (1) (Larsen et al., 2010). Two other sources of error arise from the detectability of individual landslides themselves and the ability to accurately measure the distribution of landslide areas due to landslide amalgamation and under-detection of landslides. Landslide amalgamation can produce up to 200 % error in the total volume estimation (Li et al., 2014; Marc and Hovius, 2015) and occurs because of landslide spatial clustering or incorrect mapping

due, for instance, to automatic processing. Indeed, automatic landslide mapping (Behling et al., 2014; Marc et al., 2019; Martha et al., 2010; Pradhan et al., 2016) relies on the difference in texture, color and spectral properties such as NDVI (normalized difference vegetation index) between pre- and post-landslide images, assuming that landslides lead to vegetation removal or significant texture change. During this process, difficulties in automatic segmentation of landslide sources can result in amalgamation of individual landslide area, which propagate into a much larger error of volume owing to the non-linearity of eq. (1). Manual mapping and automatic algorithms based on geometrical and topographical inconsistencies can reduce the amalgamation effect on landslide volume estimation (Marc and Hovius, 2015), but it remains a source of error due to the inherent spatial clustering of landslides and the overlapping of landslide deposits and sources (Tanyaş et al., 2019).

Under-detection of landslides can occur because the spectral signature of images is not sufficiently altered by a new failure to be detected by the algorithm or person identifying the landslides. Notably, under-detection of small landslides is one hypothesis put forward to explain the divergence of small landslides from the power-law frequency-area distribution observed for medium to large landslides (e.g., Bellugi et al., 2021; Stark and Hovius, 2001; Tanyaş et al., 2019). A rollover point below which frequencies decreases for smaller landslides is observed, and varies between 40 to 4000 m² for different inventories (Tanyaş et al., 2019). Beyond the under-detection of small landslides, other explanations for the occurrence of a rollover have been put forward, notably the transition from a friction-dominated mode of rupture for large landslides to a cohesion dominated mode for small landslides (e.g. Jeandet et al., 2019; Tanyaş et al., 2019 and references therein). Under-detection can be particularly common in areas with thin soils and sparse or missing vegetation (Barlow et al., 2015; Behling et al., 2014; Brardinoni and Church, 2004; Miller and Burnett, 2007). It can be further complicated when using different image sources with different resolution, spectral resolution, projected shadows and consequent ability to detect surface change. Yet, the level of under-detection of landslide in a given inventory remains generally largely unknown. New method to detect landslides in areas with poor or total lack of vegetation area therefore critically needed. To deal with poor vegetated areas, Behling et al. (2014, 2016) developed a method using temporal NDVI-trajectories which describes the temporal footprints of vegetation changes but cannot fully address complex cases when texture is not significantly changing such as bedrock landsliding on bare rock hillslopes.

Addressing these three sources of uncertainty - volume-area scaling uncertainty, landslide amalgamation and the under-detection of landslides – is required. In the last decade, the increasing availability of multi-temporal high resolution 3D point cloud data and digital elevation models (DEMs), based on aerial or satellite photogrammetry and Light Detection and Ranging (LiDAR), has opened the possibility to better quantify surface change and displacements (e.g. Bull et al., 2010; Mouyen et al., 2019; Okyay et al., 2019; Passalacqua et al., 2015; Ventura et al., 2011).

The most commonly used technique is the difference of DEM (DoD) which computes the vertical elevation differences between two DEMs of different time (Corsini et al., 2009; Giordan et al., 2013; Mora et al., 2018; Wheaton et al., 2010). Even though this method is fast and works properly on horizontal surfaces, a vertical difference can be prone to strong errors when used to quantify changes on vertical or very steep surfaces where landsliding typically occurs (e.g., Lague et al., 2013). The "Multiscale model-to-model cloud comparison" (M3C2) algorithm implemented by Lague et al. (2013) rather considers a

direct 3D point cloud comparison. This algorithm has three main advantages over a DoD: (i) it operates directly on 3D point clouds, avoiding a phase of DEM creation that is conducive to a loss of resolution imposed by the cell size and potential data interpolation, (ii) it computes 3D distances along the normal direction of the topographic surface, allowing better capture of subtle changes on steep surfaces, and (iii) it computes a spatially variable confidence interval that accounts for surface roughness, point density and uncertainties in data registration. Applicable to any type of 3D data to measure the orthogonal distance between two point clouds, this approach has generally been used for terrestrial lidar and UAV photogrammetry over sub-kilometer scales. In the context of landsliding, it has been used to infer the displacement and volume of individual landslides, using point clouds obtained by UAV photogrammetry (e.g., Esposito et al., 2017; Stumpf et al., 2015), as well as for rockfall studies (Benjamin et al., 2020; Williams et al., 2018) and sediment tracking in post-wildfire conditions (DiBiase and Lamb, 2020). To our knowledge, systematic detection and segmentation of hundreds of landslides from 3D point clouds have not yet been attempted.

Here, we produce an inventory map of landslide topographic changes using a semi-automatic 3D point cloud differencing (3D-PcD) method based on M3C2 and applied to multi-temporal airborne LiDAR data. We use the generic term of "landslide" to define the spatially coherent changes detected by our method on hillslopes that result in at least several decimeters of negative topographic change associated with a downstream positive topographic change. Patches of negative (resp. positive) topographic change are called sources (resp. deposits) and correspond to erosion (resp. sedimentation) for landslides producing debris, or subsidence (resp. accumulation) for landslides involving movement of largely intact hillslope material. This definition therefore includes all the types of mass wasting processes involving the downward or outward movement of soil, rocks and debris under the influence of gravity, occurring on discrete boundaries and taking place initially without the aid of water as a transportational agent (Crozier, 1999). An objective of this work is to provide a first evaluation of the type of landslides produced during an earthquake that 3D point cloud differencing can detect.

Our workflow was designed to be as automated as possible in order to be applied in the future to very large multi-temporal 3D datasets. As any topographic data will contain elevation errors (Anderson, 2019; Joerg et al., 2012; Passalacqua et al., 2015) that may result in false detections of sources and deposits, our workflow combines two steps to filter them out: we first isolate patches of topographic significant change using the statistical model accounting for point cloud roughness, density and registration error defined in the M3C2 algorithm (Lague et al., 2013), and then use patch-based metrics to detect the remaining false detections. The workflow efficiency is tested against a set of sources manually labelled as actual landslides or false detections. We apply our method to a complex topography located near Kaikoura, New Zealand, where a 2016 $M_w$ 7.8 earthquake triggered nearly 30,000 landslides over a 10,000 km² area (Massey et al., 2020). We choose a 5 km² area characterized by a high landslide spatial density along the Conway segment of the Hope fault, inactive during the earthquake, where pre- and post-earthquake LiDAR and aerial images were available (Fig. 1). This area has a variety of vegetation cover (e.g. dense evergreen forest, sparse or low vegetation shrubs and grass, bare bedrock) and typically represents a challenge for conventional 2D landslide mapping. We apply our workflow to obtain a 3D landslide inventory that is compared to a traditional manually mapped inventory of landslide scars based on aerial image comparison, hereafter called the 2D inventory. We

illustrate the benefits of working directly on 3D data to generate landslide source and deposit inventories, and discuss the methodological advantages to operate directly on point clouds with M3C2 compared to DoD in terms of detection accuracy and error for total landslide volume.

The paper is organized as followed: first, the LiDAR dataset is presented followed by a detailed description of the 3D-PcD method. Second, results of the geomorphic change detection and identification of individual landslides in the studied area are presented. The remaining part of the paper focuses only on landslide sources. First, we evaluate the prevalence of false detections and define optimal filtering parameters to be used to limit their occurrence. Second, the comparison with conventional 2D landslide mapping is presented. Then, the statistical properties of the 3D and 2D landslide source inventories

are investigated in terms of area and volume. Finally, current limitations of the method are discussed as well as knowledge gained on the importance of landslide under-detection on the co-seismic landslide inventory budget, the variety of landsliding processes that can be detected by our workflow and landslide source geometry statistics.

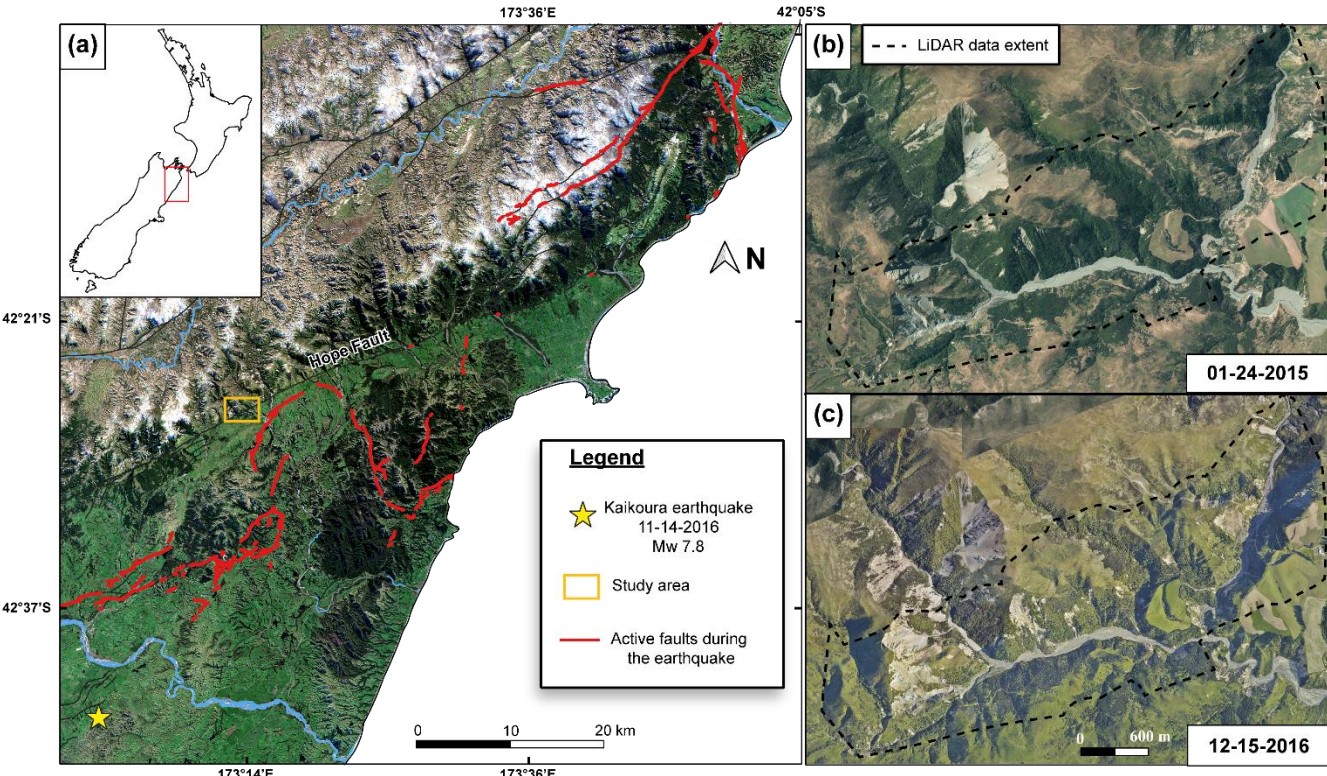

**Figure 1 : Maps of the regional context and location of the study area. (a) Regional map of Kaikoura with the location of the 2016**
**M$_w$ 7.8 earthquake, associated active faults and the study area. (b-c) Orthoimages focused on the study area dated before and after the earthquake with the 5 km² LiDAR dataset extent used in this paper (all images are available at https://data.linz.govt.nz/set/4702-nz-aerial-imagery/, Aerial survey 2017).**

## 2. Data description

In this study, we compare two 3D point clouds obtained from airborne LiDAR data collected before and after the November 14 2016 Kaikoura earthquake (Tab. 1). Both airborne LiDAR surveys were acquired during summer. Pre-earthquake (pre-EQ) LiDAR data were collated over six flights performed from March 13, 2014 to March 20, 2014 for a resulting ground point density of $3.8 \pm 2.1$ pts/m². The vertical accuracy of this dataset has been estimated at 0.068 m to 0.165 m as the standard deviation of the difference between the elevation of GPS points located on highways and the nearest neighbour LiDAR shot elevation (Dolan, 2014). However, these control points were not on the survey area. We thus have weak independent constraints on the vertical accuracy of the pre-EQ LiDAR on the survey area. The post-earthquake (post-EQ) LiDAR survey took place soon after the earthquake from December 3, 2016 to January 6, 2017 for an average ground point density of $11.5 \pm 6.8$ pts/m². The vertical accuracy of this dataset has been estimated following the same protocol as the pre-earthquake LiDAR data with a mean of 0.00 m and a standard deviation of 0.04 m (Aerial survey, 2017). The difference in acquisition dates represents a period of 2 years and 8 months. For both LiDAR point clouds, only ground points defined by the data providers are selected. For the pre-EQ LiDAR, they were classified automatically using the Terrascan software (Dolan, 2014), while the post-EQ data were classified using an unspecified algorithm followed by manual validation by the data provider (Aerial survey, 2017). Manual quality control shows that ground classification is excellent for the post-EQ data but that some points corresponding to vegetation remains in the pre-EQ data. As the incorrectly classified points are located a few meters above the ground, they can lead to false landslide source detection as they translate into a negative topographic change (i.e., apparent spatially correlated erosion). We thus reprocess this dataset to remove as many incorrectly classified points as possible using a method similar to surface-based filtering that remove points or patches of points significantly higher than the locally interpolated ground (e.g. Kraus and Pfeifer, 1998) (details in section S1 in the Supplement). This operation removes 0.3% of the pre-EQ original point cloud, but our results shows that classification errors still remain. We did not attempt to further improve the classification as these errors are expected to occur in low point-density LiDAR survey of evergreen forested areas and will generate false landslide sources that our workflow should detect and filter out. We note that the classification refinement is not a critical component of our workflow and that other classifications algorithms (Sithole and Vosselman, 2004) could be used to improve or check the quality of the LiDAR ground points before the application of the workflow.

In addition, orthoimages are used to perform a manual mapping of landslides to compare the detection of landslides from the 3D approach and a more classical approach. The pre-EQ orthoimage was obtained on January 24 2015 (available at https://data.linz.govt.nz/layer/52602-canterbury-03m-rural-aerial-photos-2014-2015) and the post-EQ image on December 15 2016. The resolutions are 0.3 and 0.2 m, respectively.

**Table 1: Information about LiDAR data used in this study**

|  | Pre-earthquake LiDAR | Post-earthquake LiDAR |
| --- | --- | --- |
| **Date of acquisition** | 13/03/2014 – 20/03/2014 | 03/12/2016 – 06/01/2017 |

| Commissioned by/provided by | USC-UCLA-GNS science/NCALM | Land Information New Zealand/AAM NZ |
|---|---|---|
| Availability | https://doi.org/10.5069/G9G44N75 | On request at gisbasemap.ecan.govt.nz |
| Original point density (pts/m²) | 9.02 | $19.2 \pm 11.7$ |
| Number of ground points | 10,660,089 | 63,729,096 |
| Ground point density (pts/m²) | $3.8 \pm 2.1$ | $11.5 \pm 6.8$ |
| Vertical accuracy (m, 1 std) | 0.068 – 0.165 | 0.04 m |
| Study area (m²) | 5,253,133 | 5,253,133 |

## 3. Methods and parameter choice

### 3.1. 3D point cloud differencing with M3C2 and distance uncertainty model

The method developed here to detect landslides consists of 3D point cloud differencing between two epochs using the M3C2 algorithm (Lague et al., 2013) available in the Cloudcompare software (EDF R&D, 2011). This algorithm estimates orthogonal distances along the surface normal directly on 3D point clouds without the need for surface interpolation or gridding. While M3C2 can be applied on all points, the algorithm can use an accessory point cloud, called core points. In our case, core points constitute a regular grid with constant horizontal spacing generated by the rasterization of one of the two clouds. In the following, all the M3C2 calculations are done in 3D using the raw point clouds, but the results are "stored" on the core points. The use of a regular grid of core points has four advantages: (i) a regular sampling of the results allows computation of robust statistics of changes, unbiased by spatial variations in point density; (ii) it facilitates the volume calculation and the uncertainty assessment; (iii) it can be directly reused with 2D GIS as a raster (rather than a non-regular point cloud); and (iv) it speeds up calculations, although in the proposed workflow, computation time is not an issue and can be done on a regular laptop.

The first step of M3C2 consists in computing a 3D surface normal for each core point at a scale $D$ (called the normal scale) by fitting a plane to the core points located within a radius of size D/2. Once the normal vectors are defined, the local distance between the two clouds is computed for each core point as the distance along the normal vector of the arithmetic mean positions of the two point clouds at a scale $d$ (projection scale). This is done by defining a cylinder of radius d/2, oriented along the normal with a maximum length $p_{max}$. Distances are not computed if no intercept is found in the second point cloud, that is in areas where the two point clouds do not overlap, or if one cloud has missing ground data (e.g., below dense forest cover). $p_{max}$ must be chosen larger than the largest topographic change to be measured. In landslide inventories, $p_{max}$ can thus be as large as several tens of meters. This poses a potential issue in highly curved features of the landscape such as narrow ridges or gorges with steep flanks where the cylinder can intercept the same point cloud twice, resulting in an incorrect distance calculation. A preliminary analysis showed that this resulted in about 1 % of false landslide detections. We have thus modified the M3C2

algorithm to avoid the double intercept issue. A new iterative procedure progressively increases the depth of the cylinder up to $p_{max}$, by intervals of 1 m for each core point and checks for the stability of the measured distance: if the distance is stable for two successive iterations, it is considered as the final M3C2 distance for this core point. This modification solved the double intercept issue.

M3C2 has the option to compute the distance vertically which bypasses the normal calculation, and we use this option several times in the workflow. We use the abbreviation vertical-M3C2 in that case and 3D-M3C2 otherwise. M3C2 also provides uncertainty on the computed distance at 95% of confidence based on local roughness, point density and registration error as follows:

$$LoD_{95\%}(d) = \pm\, t(DF).\left(\sqrt{\frac{\sigma_1(d)^2}{n_1} + \frac{\sigma_2(d)^2}{n_2}} + reg\right) with\, DF = \frac{\left(\frac{\sigma_1(d)^2}{n_1} + \frac{\sigma_2(d)^2}{n_2}\right)^2}{\left(\frac{\frac{\sigma_1(d)^4}{n_1^2}}{(n_1-1)} + \frac{\frac{\sigma_2(d)^4}{n_2^2}}{(n_2-1)}\right)} \qquad (2)$$

where $LoD_{95\%}$ is the Level of Detection, $t$ is the two-tailed $t$-statistics with a confidence level of 95% and a degree of freedom DF (Borradaile, 2003) $\sigma_1(d)$ and $\sigma_2(d)$ are the standard deviation of distances of each cloud, at scale $d$, measured along the normal direction, $n_1$ and $n_2$ are the number of points in each cloud at that scale and $reg$ is the co-registration error between the two epochs When the M3C2 distance is larger than the $LoD_{95\%}$, the topographic change is considered statistically significant. In equation (2), the first two terms assume that $\sigma_1(d)$ and $\sigma_2(d)$ empirically characterize two uncorrelated random errors

depending on the topographic surface roughness and the survey precision. On perfectly flat surfaces, $\sigma(d)$ is minimal and characterizes the instrument precision, but on rough surfaces $\sigma\,(d)$ will increase above this value and becomes the dominant source of uncertainty (Lague et al., 2013). As we consider these sources of uncertainty as uncorrelated random errors, then increasing the number of samples $n_1$ and $n_2$, reduces the $LoD_{95\%}$. The original M3C2 algorithm uses a value of $t$ equal to 1.96, the asymptotic value of the two-tailed $t$-statistics when $F$ is infinite, and the $LoD_{95\%}$ is only computed if $n_1 > 4$ and $n_2 > 4$. As

will be shown later, the low point density of the pre-EQ data in forested areas resulted in values of $n_1$ varying between 5 to 15, in which case $t(F(n_1, n_2))$ is significantly larger than 1.96. For instance, when $n_1 = n_2 = 5$, $F=4$ and $t=2.776$. To avoid under predicting $LoD_{95\%}$ in low point density areas, we choose to apply the strict two-tailed t-statistics rather than the simplification used in the Cloudcompare implementation of M3C2. As point density and surface roughness are spatially variable, $LoD_{95\%}$ is also spatially variable. For instance, in forested steep hillslopes, points located under the canopy, with a lower point density,

or vegetation points that are incorrectly classified as ground and create locally high roughness, result into a higher $LoD_{95\%}$ and therefore require a larger topographic change to be detected as significant change.

The co-registration error $reg$ in eq. (2) is treated as a systematic spatially uniform error encompassing all the errors that are not uncorrelated random errors. This is a simplification as elevation errors related to intra-flight line time-dependent attitude and position uncertainties combined with intra-survey registration errors of flight lines and to the inter-survey rigid registration

error to make *reg* theoretically spatially variable (Joerg et al., 2012; Passalacqua et al., 2015). These error sources may create apparent topographic change of low amplitude and variable wavelength that could be mistakenly considered as a significant change resembling a landslide source or deposit if *reg* is not high enough. Predicting the spatial pattern of registration error can be done in two ways: first, by using a spatially explicit direct error propagation model that accounts for all elevation errors in the LiDAR survey (e.g., Joerg et al., 2012). This approach is complex and requires detailed information on the survey,

including the trajectory file with position and attitude uncertainties. This file is rarely available in data repositories, and was not available for the pre or post-EQ dataset. Second, by studying patterns of topographic change on flat, stable and near-horizontal surfaces (e.g., Anderson, 2019) to derive amplitude and spatial correlation characteristics of the registration error. The stable area must have not changed between the survey and be much larger than the expected spatial correlation scale. It also has to be as flat as possible to limit the effect of surface roughness, which being sampled differently between each survey

may obscure the correct estimate of registration errors. This approach only captures the spatial patterns of registration error in a statistical sense (using for instance a semi-variogram, Anderson, 2019) and thus corresponds to a spatially uniform *reg*. This approach cannot be applied in our case as we lack extensive, flat, smooth and good stable area such as human infrastructures (e.g., roads, parkings…).

Hence, we assume that *reg* is uniform and isotropic. A critical aspect of the workflow is to choose the lowest *reg* possible that

does not result in too many false detections. A first estimate of *reg* can be evaluated from the vertical accuracies provided with the LiDAR datasets assuming that no systematic bias remains after the registration process (Tab. 1). If we assume that the two vertical elevation errors are uncorrelated, then *reg* is the square root of the sum of the square accuracies and varies between 0.08 and 0.17 m. If we assume that the two accuracies are perfectly correlated, a worst case scenario, then *reg* is simply the sum of the two accuracies and would vary between 0.12 and 0.21 m. In both cases, it is largely set by the accuracy of the pre-

EQ survey (Table 1). Since no GCP used to evaluate the LiDAR survey accuracy are applied in our study area, and that it will be generally the case in steep mountain areas where landslide inventory creation will be meaningful, we propose not to rely on the stated LiDAR accuracies. Instead, we define *reg* empirically as the standard deviation of 3D-M3C2 distances calculated on stable areas that are manually delimited (see section 3.4.1). However, to better account for the potentially poor quality of intra-survey registration error, we define *reg* as the maximum of the intra-survey and inter-survey registration errors. The intra-

survey *reg*, is computed on the overlapping parts of flight line (see section 3.4.1). Finally, the M3C2 definition of the $LoD_{95\%}$ makes the conservative choice of adding *reg* to the combined standard error related to point cloud roughness, rather than taking the square root of the sum of squared standard error and squared registration error (e.g., Anderson, 2019; Joerg et al., 2012). This arbitrary choice similar to Lague et al. (2013) ensures that the frequency of false detection of statistically significant change is below 5%, at the expense of a reduced capacity to detect real small topographic changes close to the $LoD_{95\%}$.

## 3.2. The Same Surface Different Sampling test

Following the approach proposed in Lague et al. (2013), we use a test based on using different sampling of the same natural surface to tune parameters of the workflow. To this end, we create two randomly sub-sampled versions of the post-earthquake LiDAR data (which has the largest point density) with an average point density equal to the pre-EQ data. The resulting point clouds correspond exactly to the same surface (i.e., $reg=0$), with roughness characteristics typical of the studied area, but with different point sampling. We subsequently refer to this type of approach as a Same Surface Different Sampling (SSDS) test.

## 3.3. Parameter selection and 3D point cloud differencing performance

In this section, we explain how to select the appropriate normal scale $D$ and projection scale $d$ to detect landslides using M3C2. The normal scale $D$ should be large enough to encompass enough points for a robust calculation, and smooth out small-scale point cloud roughness that results in normal orientation flickering and overestimation of the distance between surfaces (Lague et al., 2013). However, $D$ should also be small enough to track the large-scale variations in hillslope geometry. By studying roughness properties of various natural surfaces, Lague et al. (2013) proposed that the ratio of the normal scale and the surface roughness, measured at the same scale, should be larger than about 25. We thus set $D$ as the minimum scale for which a majority of core points verify this condition. As roughness is a scale and point density dependent measure, we explore a range for $D$ from 2 m to 15 m for the pre-EQ dataset, which has the lowest point density (Fig. 2a). We found that $D \sim 10$ m represents a threshold scale below which the number of core points verifying this condition significantly drops.

The projection scale $d$ should be chosen such that it is large enough to compute robust statistics using enough points, but small enough to avoid spatial smoothing of the distance measurement. Following Lague et al. (2013), M3C2 computes eq. (2) only if 5 points are included in the cylinder of radius $d/2$ for each cloud. In our case, the pre-EQ data with the lowest point density will thus set the value of $d$. We use a SSDS test applying M3C2 with $D=10$ m and $d$ varying from 1 to 40 m. Results show that (Fig. 2b): (i) when it can be computed the $LoD_{95\%}$ actually predicts no significant change for at least 95 % of the time, indicating that the statistical model behind the uncorrelated random error component of eq. (2) (Lague et al., 2013) is correct for this dataset; (ii) the fraction of core points for which the $LoD_{95\%}$ can be calculated rapidly increases between d = 1 and 8 m at which point it reaches 100 %. We choose $d=5$ m as it represents a good balance between the ability to compute a $LoD_{95\%}$ on most core points (here, ~ 97 %) and the smallest projection scale possible. To be able to generate M3C2 confidence intervals for as many points as possible, in particular on steep slopes below vegetation, we use a second pass of M3C2 with $d=10$ m using the core points for which no confidence interval was calculated at $d=5$ m. We note that $d$ could theoretically be set as a function of the lowest mean point density of the two LiDAR datasets, $res,$ by $d \sim 25/\pi res$. In our case the pre-EQ dataset has $res = 3.8$ pts/m² and would predict $d = 1.3$ m. However, the presence of vegetation significantly reduces the ground point density in some parts and the overlapping of flight lines creates localized high point density. Examining the mean ground point density of the entire dataset thus gives an incomplete picture of the strong spatial variations in point density. These changes in point

density, critical to the correct evaluation of the $LoD_{95\%}$ (eq. (2)), are generally lost when working on a raster of elevation (e.g., DEM).

The spacing of the core point grid should be smaller than half the projection scale $d$ to ensure that all potential points are covered by at least one M3C2 measurement, while being larger than the typical point cloud spacing of the lowest resolution dataset. Because the ground point density on steep forested hillslope of the 2014 survey is of the order of 1 pt/m², we set a core point spacing of 1 m.

Finally, the maximum cylinder length $p_{max}$ was set to 30 m as it encompassed the maximum change observed in the study area. This is generally obtained by trial and error. Setting $p_{max}$ too large increases computation time significantly.

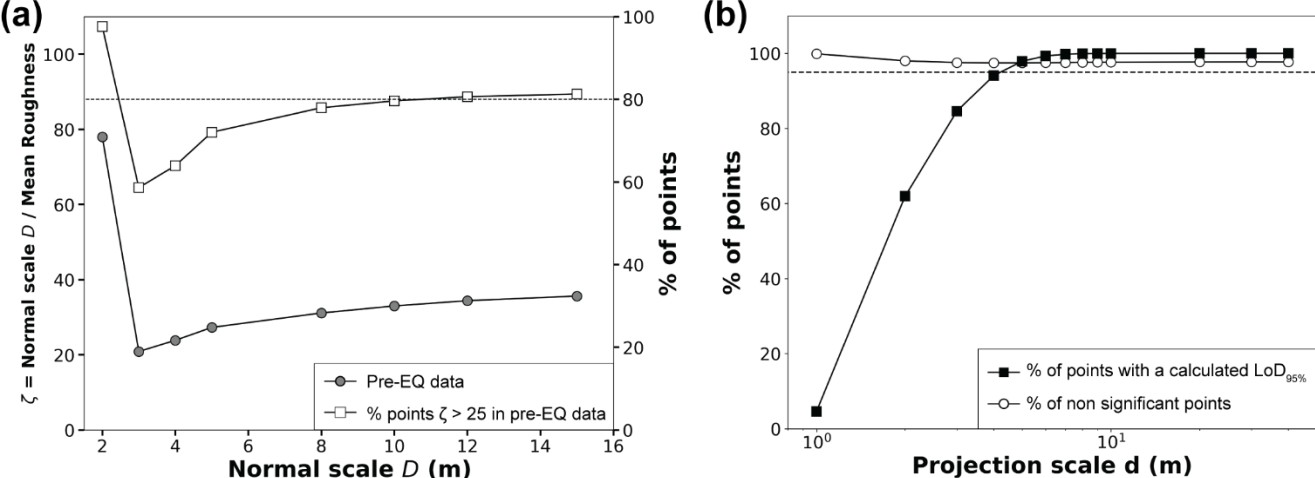

Figure 2 : Analysis of two main parameters of the M3C2 algorithm: the normal scale $D$ and the projection scale $d$. (a) Ratio between normal scale and mean roughness for different normal scale values (circles, left y-axis), and fraction of the pre-earthquake core points for which the normal scale is 25 times larger than the local roughness (squares, right y-axis). The dashed line highlights the percentage of points with $\zeta > 25$ in the pre-EQ data for D = 10 m. (b) Percentage of computed points with a confidence interval of 95% versus projection scale d. The percentage of non-significant points is represented as well as the percentage of points where the Level of Detection (LoD$_{95\%}$) was computed (i.e., with at least 5 points on each point cloud). The dashed line is set to 95% and highlights the threshold above which the projection scale is large enough to compute the $LoD_{95\%}$ on most of the core points.

### 3.4. 3D landslide mapping workflow and parameter selection

Our 3D landslide mapping workflow is divided in five main steps (Fig. 3).

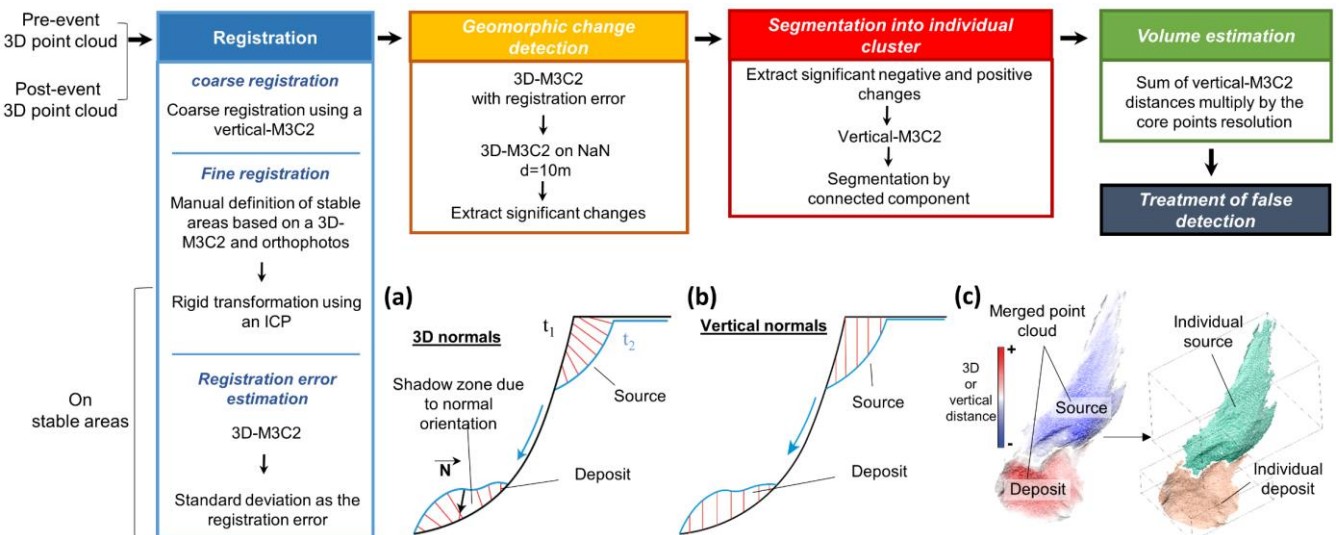

**Figure 3: Workflow of the 3D point cloud differencing method for landslide detection and volume estimation with schematic representations of the different steps (a,b,c). (a) 3D measurement step with the shadow zone effect, where the red lines show the normal orientation. (b) Vertical-M3C2 step. (c) Segmentation by connected component. The resulting sources and deposits are individual point clouds illustrated in the figure by different colours.**

### 3.4.1. Registration of the datasets and registration error estimate

To detect geomorphic changes and landslides, the two datasets need to be co-registered as closely as possible and any large-scale tectonic deformation needs to be corrected. The registration error to be used in eq. (2) must also be estimated.

First, a preliminary quality control is performed to evaluate the intra-survey registration quality of each dataset. This is feasible if the individual flight lines can be isolated, by using for instance, the pointID information specific to each line and provided in the las file format. The intra-survey registration quality can be investigated with 3D-M3C2 measurements of overlapping

flight lines using a 1 m regular grid of core points, from which we define the registration bias and error as the mean and standard deviation of the 3D-M3C2 distances, respectively. The point cloud of the pre-EQ dataset results from 12 flight lines that have overlaps ranging from 60 to 90 % (Fig S2 in the Supplement), while the post-EQ point cloud corresponds to 5 flight lines with an overlap of 50 %. For each dataset, no significant bias with respect to registration is measured between lines (maximum of 3 cm for the pre-EQ survey and 1 cm for the post-EQ survey; Tab. S3 in the Supplement), but the registration

error ranges from 13 to 20 cm for the pre-EQ survey, and is typically around 6 cm for the post-EQ survey, with one pair of overlapping flight lines having a registration error of 12 cm. Hence, the internal registration quality of the pre-EQ dataset is significantly worse than the post-EQ dataset, a likely consequence of differences in instrument precision and post-processing methods.

Second, the registration between the two surveys must be evaluated, and in general improved. As delivered, the LiDAR datasets

have a vertical shift between 1 and 2 m relative to each other. To correct for this shift, a grid of core points is first created by rasterizing the dataset with the largest point density - here the post-EQ dataset - with a 1 m grid spacing. Then, a vertical-

M3C2 calculation is performed and the mode of the resulting distribution is used to adjust the two datasets by a vertical shift of 1.36 m. This approach is valid only when the fraction of the surface affected by landsliding is small. A subsequent 3D-M3C2 calculation is performed to obtain a preliminary map of geomorphic change. At this stage, a visual inspection of the

335 pre-EQ and post-EQ orthoimages and of the preliminary 3D-M3C2 distances allows us to determine that there is no significant internal tectonic displacement. Then, we manually define areas deemed stable, 25 % of the studied area (Fig. 4a), to perform a cloud matching registration. The stable areas area defined as coherent surfaces (1) with a 3D-M3C2 distance smaller than 1 m, (2) where visual assessment of orthoimages suggested no change had occurred, and (3) away from visible mass-wasting processes and forested areas deduced from the analysis of the 3D-M3C2 distance map. Attention has been paid to select areas

340 uniformly distributed in terms of location and slopes in the studied region to maximize the registration quality.

An Iterative Closest Point (ICP) algorithm (Besl and McKay, 1992) is then performed on the stable areas, and the obtained rigid transformation is applied to the entire post-earthquake point cloud to align it with the pre-earthquake one (Tab. S4). The mean 3D-M3C2 distance on stable areas is -0.01 m, showing that there is almost no bias left in the registration, and the standard deviation of 3D-M3C2 distances is 0.17 m (Fig. 4b). At this stage, the two datasets are considered optimally registered for the

345 stable areas but with an unknown registration error *reg*. We propose to define *reg* in eq. (2) as the maximum of the standard deviation of the intra-survey and inter-survey 3D-M3C2 distances. In the ideal case of two very high quality LiDAR datasets, *reg* would be equal to the inter-survey registration error. In the studied case, the pre-EQ intra-survey registration error is locally worse (0.2 m) than the inter-survey registration error (0.17 m). We thus set *reg*=0.2 m, which is consistent with an estimate of *reg* that would be based on the combined LiDAR accuracy derived from GCP assuming complete correlation of errors (see

350 section 3.1). Consequently, and according to eq. (2), with *reg*=0.2 m, our workflow cannot detect a 3D change that is smaller than 0.40 m in the ideal case of a negligible roughness surface. At this stage, a 3D map of topographic change is available, but the significant geomorphic changes and individual landslides have not been isolated.

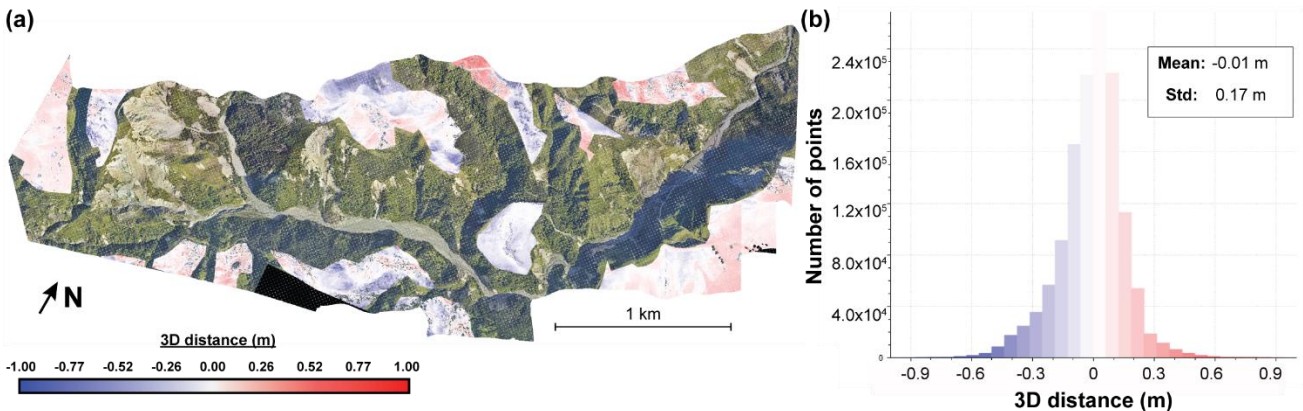

**Figure 4: (a) Map of 3D-M3C2 distances on stable areas and (b) the associated histogram. The map only displays 3D-M3C2 distance**

355 **on the areas chosen as stable for ICP registration and shows the post-earthquake orthoimage otherwise (Aerial survey, 2017).**

### 3.4.2. Geomorphic change detection

The registration error *reg* is then used in a first application of 3D-M3C2, using the pre-determined projection scale $d$=5 m, to estimate the spatially variable $LoD_{95\%}$ according to eq. (2). For core points in low point-density areas, where a confidence interval could not be estimated due to insufficient points, a second application of 3D-M3C2 is performed at a larger projection scale $d$=10 m. These core points generally correspond to ground points under canopy on steep slopes and represent 9.5% of the entire area and 12% of steep slopes prone to landsliding. Significant geomorphic changes at the 95% confidence interval are then obtained by considering core points with a 3D-M3C2 distance larger than the $LoD_{95\%}$. Significant geomorphic changes can be associated to any geomorphic processes, including landsliding, but also fluvial erosion and deposition. Changes located in the river bed, and likely specifically related to river dynamics and not to landslide deposits, are manually removed using the post-EQ orthoimage. Along with the selection of the stable areas, this is the only manual phase of the workflow.

### 3.4.3. Landslide source and deposit segmentation

Core points with negative and positive significant changes are first separated into two distinct point clouds of sources and deposits, respectively. A vertical-M3C2 is performed on each of these point clouds to estimate the volume of landslide sources and deposits (see section 3.3.4). As for any 2D landslide inventory, a critical component of the workflow is to segment each point cloud into individual landslide sources and areas. Segmenting complex patterns of erosion and deposition in 3D, with a very wide range of sizes, is still a challenge. Here, for the sake of simplicity we use a classical clustering approach by a 3D label connected component algorithm (Lumia et al., 1983), available in CloudCompare (Fig. 3c). The point cloud is segmented into individual clusters based on two criteria: a minimum number of points or surface area (in our case) $A_{min}$ defining a cluster and a minimum distance $D_m$ below which neighbouring points, measured in a 3D Euclidean sense, belong to the same cluster (Lumia et al., 1983). $A_{min}$ was set to 20 m² to be consistent with the area of the projection cylinder used to average the point cloud position in the M3C2 distance calculation, $\pi(d/2)^2 = 19.6$ m² with $d$=5 m. $D_m$ is an important parameter which, if chosen too large, will favour landslide amalgamation in identical clusters, and if too small, in relation to the core point spacing, may over-segment landslides. In any case, $D_m$ must be larger than the core point spacing. As there is no objective way to a priori choose $D_m$, we explore various values and choose $D_m$=2 m as an optimal value between landslide amalgamation and over-segmentation. The impact of $D_m$ on the statistical distribution of landslide sources is addressed in the discussion.

We note that density based clustering algorithms based on DBSCAN (Ester et al., 1996) have been used for 3D rockfall inventory segmentation (e.g., Benjamin et al., 2020; Tonini and Abellan, 2014). These algorithms separate dense clusters of points, considered as areas of coherent topographic change, from areas of low point density, considered as noise. As shown in the Supplement (Section S5), density based clustering approaches do not yield a significantly better segmentation than a connected component algorithm. However, they have several drawbacks ranging from slow computation time, to less intuitive selection of parameters. We have therefore not used density based clustering in our analysis.

### 3.4.4. Landslide area and volume estimation

While 3D normal computation is optimal to detect geomorphic changes, it is not suitable for volume estimation which requires consideration of normals with parallel directions for a given landslide. Considering 3D normals can lead to "shadow zones",
due to surface roughness, which would result in a biased volume estimate (Fig. 3a). Therefore, distances and in turn volumes are computed by using a vertical-M3C2 on a grid of core points corresponding to the significant changes (Fig. 3b). As the core points are regularly spaced by 1 m, the landslide volume is simply the sum of the vertical-M3C2 distances estimated from the individualized landslides. While the distance uncertainty predicted by the vertical-M3C2 could be used as the volume uncertainty, it significantly overpredicts the true distance uncertainty due to non-optimal normal orientation for the estimation
of point cloud roughness on steep slopes (i.e., the roughness is not the detrended roughness). For each landslide source and deposit, we thus compute the volume uncertainty from the sum of the 3D-M3C2 uncertainty measured at each core point, not the vertical-M3C2 uncertainty. The volume uncertainty is specific to each landslide source and deposit and depends on the local surface properties such as roughness, the number of points considered and the global registration error, but not on the volume itself. For each individual landslide source, the area $A$ is obtained by computing the number of core points inside the
source region. This represents the vertically projected area which is also consistent with the existing literature based on 2D studies of landslide statistics. The difference between planimetric area and true surface area (i.e., measured parallel to the surface) is addressed in the discussion.

### 3.5. Treatment of false detections

Owing to the simplified formulation of the $LoD_{95\%}$ (eq. (2)), it is possible for spatially correlated errors to create patches of
statistically significant change that would appear after segmentation as false landslide detection (Fig. 5). The inventory after segmentation is thus provisional. The workflow has a classification step aiming at separating real landslides from false detections using patch based metrics. As the pre-EQ LiDAR shows ground classification errors that would create false landslide sources (i.e., apparent negative change, Fig. 5a), and that we are specifically interested in the scaling relationships of sources, we focus on obtaining the best classification for sources, and then simply use the proximity to the predicted true landslides
sources to select real deposits. To construct the final landslide inventory, we apply the following steps:

      1. Labelling of at least 60 % of the provisional source inventory as actual landslide sources and false detections.

      2. Evaluation of the classification potential of various filtering metrics.

      3. Determination of the optimal filtering metrics based on a classification performance index.

      4. Application of the optimal filtering metrics to classify the provisional landslide source inventory in predicted
landslides and predicted false detections.

As the pre-EQ LiDAR data quality (point density and classification) is significantly worse in forests than in forest-free areas, we carry out step 2 and 3 for provisional landslide sources located in forest and forest-free areas separately. Forest areas are defined based on the number of laser returns of the post-EQ dataset (Fig. S9). This corresponds to the number of targets a laser

pulse has intercepted. For forest-free areas, this number is 1 has the laser only hit the ground. However, in forest areas this number is expected to be greater than one, as tree elements creates additional echoes before the laser hit the ground. For each core points, we thus calculate the average number of laser returns in a neighbourhood of 2.5 m, to be consistent with the projection scale *d,* using the post-EQ LiDAR point cloud which has the best canopy penetration. We then consider that any core point with an average number of laser returns equal or higher than 2 is in forested areas.

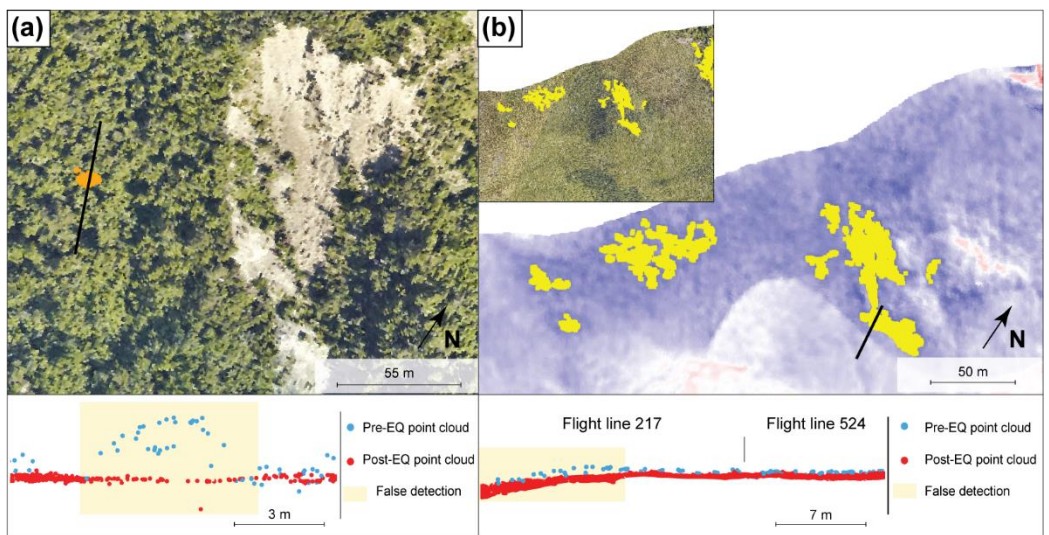

**Figure 5: Illustration of 2 types of labelled false detections. (a) False detection located in forest due to vegetation incorrectly classified as ground in the pre-EQ point cloud. The apparent negative topographic change creates a source. Note the limited penetration of the pre-EQ LiDAR in the dense evergreen forest that makes ground classification extremely difficult. (b) False detection due to pre-EQ intra-survey registration errors. Yellow points on orthoimagery and the M3C2 distance field indicates patches of significant change that are a false detection. They occur due to a complex combination of intra-line errors related to time-dependent attitude and position errors and intra-survey flight line registration error. Flight line 524 appear correctly registered to the post-EQ data, but flight line 217 is slightly misaligned which increases the likelihood of significant change detection.**

### 3.5.1. Construction of a labelled source inventory

The reference labelled source inventory is created with two classes, actual landslide source and false detection, according to the following procedure. We first manually label all the provisional landslide sources with an area higher than 200 m², as they are expected to correspond to the largest part of the total volume, and are thus critical. Provisional landslide sources with A < 200 m² are then divided into ranges of area of 20 m². Then, we choose to sample and label 60% of the provisional landslide sources located in each range of areas to be representative of the provisional inventory and avoid a size bias. Attention has been paid to have a spatially uniform and equally distributed sampling between provisional landslide sources located in forest and forest-free areas.

The labelling of actual landslide sources and false detections is based on a visual inspection of the pre-EQ and post-EQ orthophotos, of the pre-EQ and post-EQ LiDAR point clouds, of the 3D-M3C2 field and the provisional deposit inventory. We consider an actual landslide source according to the following criteria:

1. One of the following sign of mass movement is visible on orthoimagery: (1) a drastic change of color between pre-EQ and post-EQ orthophotos due to avalanches, debris flows, landslides or rockfalls or (2) the presence of scars.

2. The structure of the two point clouds does not show high local points due to misclassification of vegetation (Fig. 5a).

3. The surrounding 3D-M3C2 field does not show a large constant value indicative of a locally incorrect registration (Fig. 5b)

4. The provisional source can be associated to at least one downstream provisional deposit within a radius of 30 m.

Not all criteria have to be met simultaneously. Uncertain provisional landslide sources have been labelled as false detection. The resulting labelled inventory is then used as a reference to evaluate the filtering performances.

### 3.5.2. Definition of filtering metrics

As false detections mainly emerge from the errors in the data in relation to the amplitude of a real topographic change that we aim to capture, we first choose to analyse 3 metrics based on the 3D-M3C2 calculation: (1) the maximum 3D distance, (2) the mean $LoD_{95\%}$ and (3) the mean signal-to-noise ratio ($SNR$). We expect the maximum 3D distance to discriminates between deep actual landslide sources and low amplitude false detections arising from flight line misalignments and residual registration errors characteristic of false detections in forest-free areas (Fig. 5b). As the $LoD_{95\%}$ is a direct measure of the quality of the data (point density and roughness), classification errors of vegetation should be characterized by a significantly higher mean $LoD_{95\%}$ than actual landslide sources. The $SNR$ is defined as the ratio between the 3D-M3C2 distance and the associated $LoD_{95\%}$ for each core point. This measure can be used as a confidence metric for each source.

We also choose to take advantage of the ability of the 3D differencing approach to detect deposit areas to analyse the closest deposit distance ($CDD$). The $CDD$ is defined for each provisional source as the closest downslope distance to a provisional deposit along the flow path using a D8 algorithm (Fairfield and Leymarie, 1991) .This distance is calculated from the post-EQ DEM with the MATLAB-based software TopoToolbox (Schwanghart and Scherler, 2014).

Metrics with the best potential are then tested to determine an optimal configuration of filtering metrics that best remove false detections while keeping a maximum of actual landslide sources. The resulting predicted source inventory is then used to filter the provisional landslide deposit inventory by selecting the deposits that are connected to an upstream predicted landslide source along the flow path using TopoToolbox (Schwanghart and Scherler, 2014). The resulting inventory is called the predicted deposit inventory.

### 3.5.3. Definition of a classification performance index

To estimate the performance of the filtering metrics, we use the balanced accuracy ($BA$) defined (Brodersen et al., 2010; Brodu and Lague, 2012) as the average accuracy obtained on the two predicted classes:

$$BA = \frac{1}{2}\left(TP_{rate} + TN_{rate}\right) \tag{3}$$

where in our case $TP_{rate}$ represents the percentage of correctly classified actual sources compare to the total labelled actual sources. Similarly, $TN_{rate}$ represents the percentage of correctly classified false detections compare to the total labelled false detections. This index not only reflects the overall performance of the filtering but also how $TP_{rate}$ and $TN_{rate}$ are balanced avoiding a biased representation of the filtering accuracy by the most frequent class. High values of $BA$ are obtained when $TP_{rate}$ and $TN_{rate}$ are high and balanced. The $BA$ can be estimated based on the number, the area or the volume of the predicted landslides ($BA_n$, $BA_a$ and $BA_v$ respectively), and we define $BA_{n,a,v}$ as the mean of the $BA_n$, $BA_a$ and $BA_v$. By exploring a range of values for each filtering metrics, we find the value that maximises $BA$. Given the limited number of metrics we use at once (a maximum combination of 2), we did not use machine learning approaches to train the classifier.

### 3.6. Comparison with a manually mapped inventory based on orthoimagery

To estimate the potential in terms of landslide topographic change detection between the 3D-PcD method (3D predicted inventory) and a traditional approach, we created a second inventory (2D inventory) by manually delineating landslide sources based on a visual interpretation of the pre- and post-EQ orthoimages, looking for texture change consistent with landslide scars. The LiDAR data were not used in the process, and the mapmaker did not have a detailed knowledge of the 3D predicted inventory. Deposits were not mapped. The 2D and 3D landslide source inventories are then compared in terms of number of landslides and the intersection of mapped surfaces in planimetric view using GIS software. For source areas only detected by manual mapping, we define 4 classes: (1) areas located on deposit zones detected by the 3D-PcD method, (2) areas under the $LoD_{95\%}$, (3) areas filtered by the minimum area of 20 m² and (4) areas filtered by the application of the optimal filtering metrics. For areas only detected by the 3D-PcD method, we distinguish landslide areas located in three land cover classes: (1) forest, (2) bare-rock and (3) other land covers. Forest areas are defined according to the number of returns of the post-EQ LiDAR (see section 3.5). Bare rocks are delineated manually on the orthoimages. We finally analyse the proportion of areas only detected with the 3D-PcD approach that are connected to a landslide source in the 2D inventory.

## 4. Results

### 4.1. Geomorphic change and results of the segmentation

The map of 3D-M3C2 distances (Fig. 6a) prior to statistically significant change analysis and segmentation provides a rare insight into topographic changes following a large earthquake. At first order, it highlights areas of coherent patterns of large (3D-M3C2 > 4 m) erosion (i.e. negative 3D distances) and deposition (i.e. positive 3D distances) located on hillslopes and corresponding to major landslides. Simple configurations with one major source area and a single deposit area can be easily recognized. A more complex pattern of intertwined landslides and rockfalls occur on a bare rock surface in the western part of the study area with a large variety of source sizes and apparent aggregation of deposits. Most of the deposits are located on hillslopes while the deposits of three large landslides have reached the river and altered its geometry. At second order, a variety of patches of smaller amplitude (< 2 m) are visible on hillslopes. Erosion/deposition patterns in relation to fluvial activity can

be documented on the river bed. The flight line mismatch, identified during the preliminary quality control, leads to low amplitude and long wavelength patterns of negative and positive 3D distances, notably visible on the central northern part of the study area.

The area extent of significant changes, where the absolute amplitude of change is greater than $LoD_{95\%}$, represents 13 % of the study area (Fig 6b). Using the strict definition of the two-tailed statistics in eq. (2) reduces by ~50000 pts the number of statistically significant points, that is a reduction of 6.6 % of the statistically significant area of change. After the manual removal of changes in the fluvial domain related to fluvial processes, the maximum 3D-M3C2 distance on significant change areas is -29.46 ± 1.00 m. Due to surface roughness, the minimum $LoD_{95\%}$ observed is 0.40 m.

The point cloud of significant changes is segmented to identify the provisional landslide sources and deposits. During this step, clusters smaller than the detection limit of 20 m² are removed. They account for an area of 25,007 m², that is 3.5 % of the total area of significant change. The provisional inventory contains 1118 sources and 698 deposits for a total of 320,170 and 312,471 m² respectively. The resulting provisional landslide volume ranges from 2.27 ± 17.4 m³ to 169,843 ± 20,598 m³ for source areas, with a total of 784,689 ± 179,608 m3, and from 1.3 ± 24.08 m³ to 151,535 ± 15,301 m³ for deposits, with a total of 975,309 ± 171,578 m³. Considering that the minimum $LoD_{95\%}$ observed in 3D is 0.40 m, and that the minimum landslide area is 20 m², the minimum volume that we can confidently measure should be 8 m³, a value higher than the observed minimum volumes. 5 provisional landslide source and 16 provisional deposits are smaller than 8 m³. They correspond to peculiar cases of very small landslides where either positive or negative 3D distances close to the $LoD_{95\%}$ are positive or negative when measured vertically and thus reduce the apparent volume of the material. The uncertainty on total volume estimation represents 23% for sources and 18% for deposits.

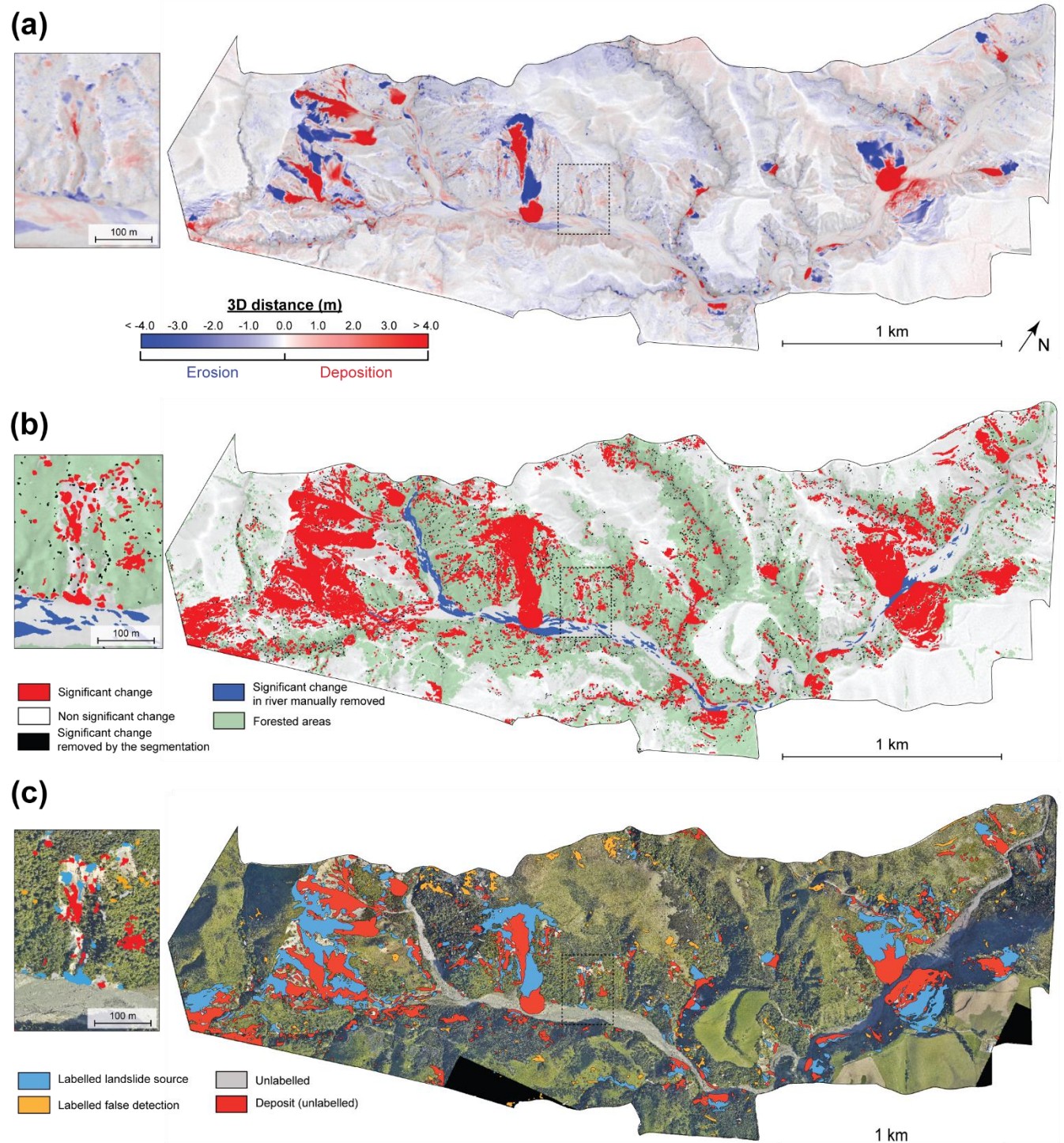

**Figure 6: Maps of the different steps of the workflow to generate the provisional landslide inventory. (a) 3D-M3C2 distances from the geomorphic change detection step. (b) Significant changes (>*LoD95%*) with indication of areas filtered in the river and by the segmentation procedure. The forested areas are calculated as a function of the number of LiDAR returns on the post-EQ data are**

also indicated in green (see section 3.5). (c) Provisional inventory after application of a minimum area of 20 m². The labelled source inventory is also shown. Results are overlaid on the post-earthquake orthoimagery (12-15-2016, Aerial survey, 2017).

## 4.2. Removal of false detections and the 3D predicted inventory

### 4.2.1. Labelled inventory characteristics and potential of filtering metrics

The labelled inventory contains 66% of the provisional landslide sources with 384 actual landslide sources and 355 false detections. In forest-free areas, 321 actual landslide sources have been labelled and 104 false detections indicating a prevalence of false detection of 24.5 %. The mean area of false detections is 174 m² with a minimum of 20 m² and a maximum of 2417 m². In forested areas, only 63 actual sources have been labelled for 251 false detections, resulting in nearly 80 % of false detection prevalence. The mean area of false detections is 90 m² with a minimum of 20 m² and a maximum of 1039 m². The prevalence of 24.5% in non-forested area is indicative of the combined effect of elevation errors due to the combination of intra flight-line elevation errors, intra-survey flight line registration errors, and the inter-survey registration error. The increased prevalence in forest is due to ground classification errors. Considering the entire labelled inventory, the prevalence of false detections decreases with size from 60 % for the class 20-40 m² to 10 % for A > 2000 m² (Fig. 7). If false detections were not removed, their prevalence and size dependency would strongly bias the landslide source area distribution towards small

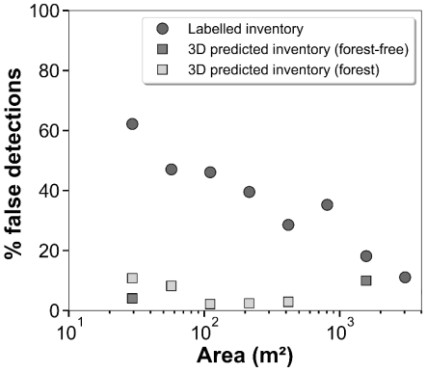

Figure 7: Proportion of false detections in the labelled inventory and the 3D predicted inventory (forest-free and forested areas) with source area.

sizes.

To determine the potential of the patch metrics to remove false detections, we analyse the cumulative density distributions of each metric for actual landslide sources and false detections (Fig. 8). In forest-free areas, the maximum 3D distance and the $LoD_{95\%}$ of the labelled inventory show the weakest potential to remove false detections as the cumulative density functions (CDFs) of false detections and landslide sources are similar (Fig. 8a and 8b). However, the CDFs of $SNR$ and $CDD$ show different behaviour for landslide sources and false detections (Fig. 8c and 8d). About 60% of landslide sources are characterized by a mean $SNR$ higher than 1.5 and about 75% have a $CDD$ lower than 30 m. On the contrary, all false detections are characterized by a mean $SNR$ below 1.5 and about 90% have a $CDD$ higher than 30 m. The low $SNR$ observed for false detections is due to the low amplitude (70% of false detection area have 3D-M3C2 < 1 m) of the type of elevation errors found in forest-free areas (Fig. 5b). Similarly, these type of errors are not expected to produce coherent pattern of upslope erosion and downslope deposition over short distances similar to what landslide produce. In forest-free areas, the $SNR$ and the $CDD$ thus constitute the best metrics to differentiate between actual landslide sources and false detections.

In forested areas, where ground classification errors are present (Fig. 5a), false detections are characterized by a higher maximum 3D distance and mean $LoD_{95\%}$ than landslide sources (Fig. 8a). This is related to ground classification errors which correspond to 3D-M3C2 distances of the order of the forest canopy height (i.e., several meters), with high $LoD_{95\%}$ due low

point density and high point cloud roughness. Yet the CDFs of maximum 3D distance is not distinct enough for the two classes for this metric to be used in the classification. The CDFs of the mean $LoD_{95\%}$ show that actual sources are characterized by a maximum mean $LoD_{95\%}$ of 0.9 while about 15% of the false detections have higher mean $LoD_{95\%}$. The *SNR* is in this context not an interesting classifying metric as false detections in forest have a larger M3C2-distance compared to non-forested areas, resulting in a similar CDF of sources and false detections. The CDFs of *CDD* are similar to forest-free areas highlighting the good classification potential of this metric.

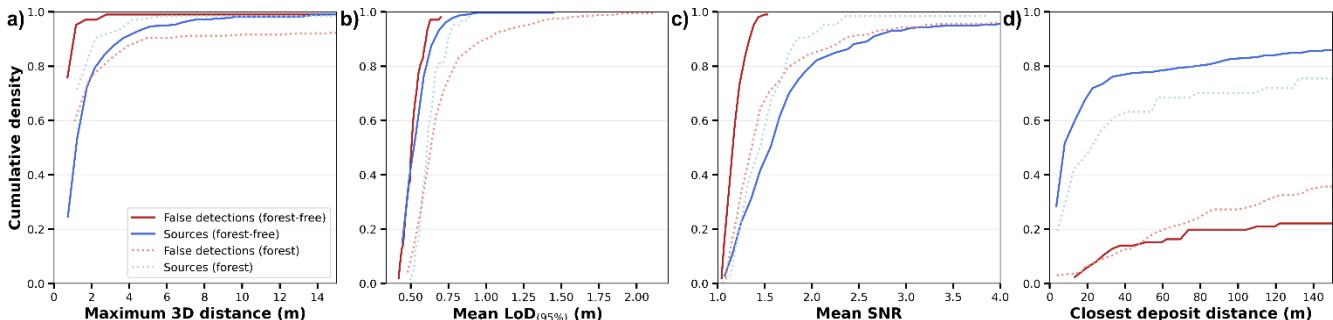

**Figure 8: Cumulative density functions of the different metrics introduced in section 3.5.2 using the labelled inventory. The distributions are presented according to landslide sources and false detections in forest-free and forested areas.**

We thus choose to test the mean $LoD_{95\%}$, the mean *SNR* and the *CDD* metrics to classify the provisional landslide source inventory. For each test, we select provisional landslide sources below a threshold $T_{LoD}$ and $T_{CDD}$ when using the mean $LoD_{95\%}$ and the *CDD* respectively and above a threshold $T_{SNR}$ when the mean *SNR* is considered. Different combinations of these metrics have been tested to determine an optimal classification (Tab. 2).

### 4.2.2. Optimal filtering metrics

Table 2 reports the *BA* for the various metrics and forest environments, as well as the true positive rate (i.e. the fraction of landslide source preserved in the predicted inventory) and the false positive rate (i.e., the fraction of false detections that are not removed out of all false detections). We choose the optimal configuration of filtering metrics corresponding to the highest $BA_{n,a,v}$ (Tab. 2) as it reflects the best balance between false detections removal and preservation of true landslide sources in terms of number, area and volume.

In forest-free areas, the lowest performances are obtained when applied the mean $LoD_{95\%}$ alone or in combination with the mean *SNR* and the *CDD* (mean $BA_{n,a,v} < 88$ %) as expected by the analysis of CDFs (Fig. 8b). On the opposite, the mean *SNR* and the CDD provide higher performances (mean $BA_{n,a,v} > 88$ %) applied alone or in combination. When applied alone, they preserve 96% of the total actual landslide sources. However, while the mean *SNR* ($T_{SNR}=1.6$) removes all false detections ($FP_{rate}=0$), the *CDD* ($T_{CDD}=18$ m) better preserves actual landslide sources number ($TP_{rate}= 60$ %) and area ($TP_{rate}= 90$ %). The optimal combination is obtained when first applying $T_{CDD} = 18$ m followed by $T_{SNR} = 1.45$ resulting in a mean $BA_{n,a,v}$ of 93 %. This combination best preserves the area and volume of labelled landslide sources with a $TP_{rate}$ of 79.1%, 97.7% and 99.4%

respectively. The number, area and volume of false detections remains low with a $FP_{rate}$ of 3.9%, 6.5% and 7.7% respectively. The prevalence of false detection after classification of the inventory is 1.5% in number, 0.48 % in area and 0.15 % in volume.

In forested areas, the lowest performances are obtained for the *SNR* with a $BA_{n,a,v}$ of 58 %. However, in combination with the *CDD*, it best preserves the number, area and volume of the labelled landslide sources with a $TP_{rate}$ of 76%, 94% and 97% respectively but fails to remove false detections ($FP_{rate}$ in volume = 44%). The combination of filtering metrics that best remove false detections is obtained when first applied $T_{LoD95\%} = 0.65$ and $T_{CDD} = 28$ m but at a higher filtering cost of labelled landslide sources ($TP_{rate} = 1.2\%$). The best performance is obtained by applying only $T_{CDD} = 28$ m with a mean $BA_{n,a,v}$ of 0.80 and corresponds to the best balance in terms of number, area and volume of landslide sources preserved and false detections removed. The prevalence of false detection after classification of the inventory is 33 % in number, 13.3 % in area and 12.5 % in volume.

Finally, after application of the optimal configuration of filtering metrics in forested and forest-free areas, we obtain a total labelled predicted source inventory preserving 75% of landslide sources and 96% and 99 % of the corresponding total area and volume, respectively (Tab. 2). False detections remaining in the labelled predicted inventory represents 6.5% of the total number of labelled predicted sources and only 1% of the total area and 0.45% of the total volume.

**Table 2: Statistics of the labelled inventories according to each combination of filtering metrics. $FP_{rate}$ is given as $FP_{rate} = FP/(FP+TN)$ where FP represent the false detection present in the labelled predicted inventory.**

| | Metrics | Threshold | $BA_n$ | $BA_a$ | $BA_v$ | Mean $BA_{n,a,v}$ | Nb $TP_{rate}$ | Area $TP_{rate}$ | Volume $TP_{rate}$ | Nb $FP_{rate}$ | Area $FP_{rate}$ | Volume $FP_{rate}$ |
|---|---|---|---|---|---|---|---|---|---|---|---|---|
| **Forest-free areas** | LoD | 0.7 | 47.0 | 54.0 | 57.0 | 53.0 | **91.3** | 94.2 | 92.8 | 98.1 | 86.0 | 78.7 |
| | SNR | 1.6 | 75.0 | 94.0 | **98.0** | 89.0 | 49.5 | 87.1 | 96.4 | **0.0** | **0.0** | **0.0** |
| | CDD | 18 | 79.0 | 95.0 | **98.0** | 91.0 | 60.1 | 89.7 | 96.7 | 1.9 | 0.3 | 0.2 |
| | LoD + CDD | 0.7 / 18 | 76.0 | 92.0 | 95.0 | 88.0 | 53.6 | 84.4 | 89.8 | 1.9 | 0.3 | 0.2 |
| | CDD + LoD | 18 / 0.55 | 60.0 | 78.0 | 77.0 | 72.0 | 32.4 | 58.4 | 55.7 | 12.5 | 2.5 | 1.9 |
| | SNR + CDD | 1.6 / 23 | 86.0 | 94.0 | 95.0 | 92.0 | 77.6 | 97.4 | 99.2 | 4.8 | 8.6 | 9.5 |
| | **CDD + SNR** | **18 / 1.45** | **88.0** | **96.0** | 96.0 | **93.0** | 79.1 | **97.7** | **99.4** | 3.9 | 6.5 | 7.7 |
| **Forested areas** | LoD | 0.65 | 60.0 | 68.0 | 73.0 | 67.0 | 69.8 | 73.5 | 72.5 | 49.4 | 37.5 | 26.1 |
| | SNR | 1.8 | 48.0 | 62.0 | 63.0 | 58.0 | 15.9 | 54.9 | 68.1 | 20.3 | 31.0 | 42.4 |
| | **CDD** | **28** | **72.0** | **83.0** | **86.0** | **80.0** | 50.8 | 71.0 | 78.4 | 6.4 | 5.9 | 7.4 |
| | LoD + CDD | 0.65 / 28 | 68.0 | 78.0 | 81.0 | 75.0 | 36.5 | 56.6 | 61.6 | **1.2** | **0.6** | **0.5** |
| | CDD + LoD | 28 / 0.65 | 68.0 | 78.0 | 81.0 | 76.0 | 44.4 | 60.0 | 64.4 | 8.7 | 4.0 | 2.8 |
| | SNR + CDD | 1.8 / 36 | 69.0 | 74.0 | 72.0 | 72.0 | 61.9 | 82.0 | 88.3 | 24.7 | 33.1 | 44.5 |
| | CDD + SNR | 28 / 1.45 | 66.0 | 64.0 | 60.1 | 63.0 | **76.2** | **94.2** | **97.2** | 44.2 | 65.9 | 76.9 |

| | | | | | | | | | | |
|---|---|---|---|---|---|---|---|---|---|---|
| **Total area** | - | - | - | - | - | - | 74.5 | 96.5 | 98.9 | 5.6 | 6.1 | 7.5 |

### 4.2.3. The 3D predicted inventory

The optimal filtering metrics are then applied on the entire provisional source inventory. We hereafter call the resulting filtered inventory the 3D predicted inventory on which we perform subsequent analysis. This inventory contains a total of 433 sources and 399 deposits, with many sources sharing the same deposits at the toe of hillslopes (Fig. 9, Tab. 3). The filtering step successfully removed false detections located on stable areas, so that no landslide source is predicted on stable areas. For sources, the mean absolute vertical-M3C2 distance is 2.82 m, the standard deviation 2.99 m and the maximum absolute value $23.06 \pm 0.70$ m. For deposits, the mean absolute vertical-M3C2 distance is 3.33 m, the standard deviation 3.66 m and the maximum absolute value is $27.9 \pm 0.50$ m. The area of detected landslides ranges from 20 to 40,475 m² for sources and from 20 to 27,782 m² for deposits, and the total source and deposit areas are 259,415 and 289,278 m², respectively. The resulting individual landslide volume ranges from $0.58 \pm 11.53$ m$^3$ to $169,725 \pm 20,598$ m$^3$ for source areas, with a total of $724,297 \pm 141,087$ m$^3$, and from $7.95 \pm 9.83$ m$^3$ to $151,717 \pm 15,301$ m$^3$ for deposits, with a total of $954,029 \pm 159,188$ m$^3$ (Tab. 3). The uncertainty on total volume estimation represents about 19% for sources and 17% for deposits. The filtering approach removes 685 provisional sources and 299 provisional deposits. This represents 19% and 7.4 % of the total provisional source and deposit area and 7.7% and 2.2% of the total volume respectively.

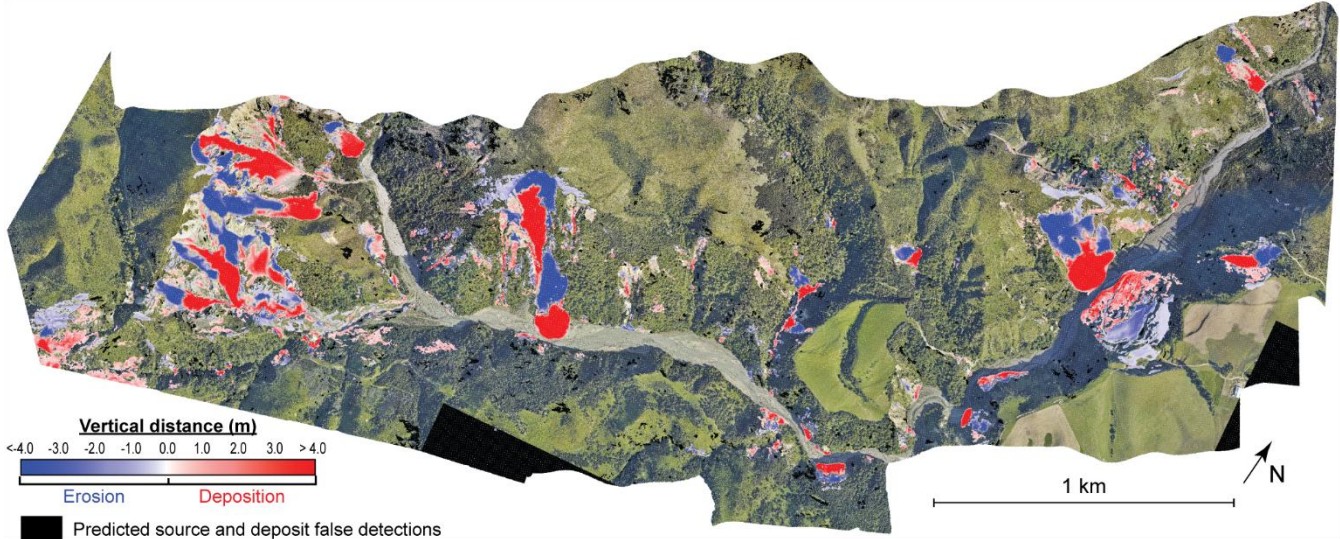

**Figure 9: Map of the 3D predicted inventory with vertical-M3C2 distances after application of $T_{CDD}$=18 m and $T_{SNR}$=1.45 in forest-free areas and $T_{CDD}$= 28 m in forested areas. The landslide inventory is overlaid on the post-EQ orthoimagery (12-15-2016, Aerial survey, 2017). Landslide sources are in blue and landslide deposits are in red.**

**Table 3: Statistics of the predicted inventory with the percentage located in forest-free areas**

| | Landslide sources | | Landslide deposits | |
|---|---|---|---|---|
| | **Predicted sources** | **Predicted false detection** | **Predicted deposits** | **Predicted false detection** |
| **Number** | 433 (84%) | 685 (38%) | 399 (79%) | 299 (67%) |
| **Area (m²)** | 259,415 (96%) | 60,755 (45%) | 289,278 (95%) | 23,193 (77%) |
| **Volume (m³)** | 724,297 ± 141,087 (97%) | 60,392 ± 38,521 (32%) | 954,029 ± 159,188 (97%) | 21,280 ± 12,391 (75%) |

### 4.3. 3D predicted vs 2D landslide source inventory

In the following analysis we separate two types of errors: detection errors, corresponding to landslides present in only one of the 2D and 3D inventories, and delimitation errors, corresponding to differences in the planimetric outline of landslides. 258 landslide sources, called hereafter *2D-sources* as opposed to *3D-sources* derived from 3D-PcD, were mapped from visual inspection of pre-EQ and post-EQ orthoimages (Fig. 10b). The *2D-sources* represent a total area of 146,641 m² (Tab. 4) with a minimum area of 6 m² and a maximum of 19,784 m². The minimum area detected shows that the resolution capability of the 2D inventory is finer than the 3D-PcD workflow. From the 258 *2D-sources*, 193 intersect *3D-sources* and 65 are not detected by the 3D-PcD method. These 65 *2D-sources* range from 12 m² to 645 m² with 69% smaller than 100 m². However, 22 are actual deposits in the 3D inventory, highlighting detection errors in the 2D inventory. These detection errors represent 14.8 % of the surface of the 2D inventory and are removed in the following, leading to 43 *2D-sources* not detected by the 3D-PcD method. The 3D-PcD method thus detects 74.8 % of the *2D-sources*, and 57.2 % of the total surface of the 2D inventory (Tab. 4). The 43 *2D-sources* not detected by the 3D-PcD method correspond to 39 *2D-sources* located in areas with no statistically significant change (i.e., 3D-M3C2 distance < $LoD_{95\%}$) and 4 *2D-sources* removed by the filtering step of the workflow. In terms of planimetric surface area, the area not captured by 3D-PcD is overwhelming dominated by non-statistically significant change (42 % of the total *2D-sources* surface are < $LoD_{95\%}$), as opposed to the filtering based on the *CDD* and *SNR* (0.4%) or the minimum detectable size (0.3%). The surface of non-statistically significant change corresponds to delimitation errors located on the edges of sources and deposits, owing to the averaging effect of the M3C2 approach, and the transition between landslide sources and deposits (Fig. 10c). The volume missed in *3D*-sources was computed by using the intersection between the outline of the *2D-sources* not shared in 3D and the vertical M3C2 field of the core points. We find that the volume that would be missed in the 3D inventory is 2.2% of the total volume of *3D-sources.*

While 193 *2D-sources* are common to *3D-sources*, this corresponds to 182 *3D-sources* owing to the difference in landslide segmentation in the two inventories (Fig. 10a). The 2D inventory misses 42% (251) of the landslide sources detected in 3D, including landslides as large as 11,902 m² (blue polygon in the frame of Fig. 10a). The detection errors are predominantly in bare rock areas (116, 46% of detection errors) and other land covers (87, 35% of detection errors) where the prevalence of false detections in the 3D-PCD dataset is expected to be extremely low (1.5 %) so that the missing landslides in the 2D are not

potential false detections. The missed landslides can be very large landslides occurring within pronounced shadows in the post-EQ orthoimage, and where the topographic change is mostly vertical (Fig. 10c). Missed landslides also occur in forested areas in lower proportion (48, 19 % of detection errors), although we could expect from the classification of the labelled data that a third of these missing landslides may be false detection of the 3D inventory. 72% of the total surface of *3D-sources*, is not detected in the 2D inventory, with a small fraction (17%) under forest, 39% on bare rock and 44% on other land covers (Tab.

4 and Fig. S10). The landslide sources in this latter domain should be generally visible owing to strong spectral contrast between pre-EQ vegetation and post-EQ rock surfaces. However, the large source of disagreement is explained by the incorrect delimitation of upslope topographic subsidence related to large scars as well as the under-detection of vertical subsidence related to translational and rotational landslides (Fig. 13b). These vertical movements of meter scale amplitude or less do not correspond to a clear change of orthoimagery texture, or create scarps that are too small or not easily detectable if they do not

generate a shadow. Similarly, large subsiding areas corresponding to retrogressive failure plane development or reactivation can be detected under forest, but totally missed in 2D (Fig. 10c). Detection and delimitation errors contribute roughly equally to the 2D area under-detection (42 % detection error, 58 % delimitation error). The *2D-sources* misses 54 % of the total volume of the *3D-sources*. In contrast to the missed planimetric surface area, the missed volume is predominantly on bare rock (34.2 %), about three times larger than in forest (10.6 %) or other land covers (13.4 %).

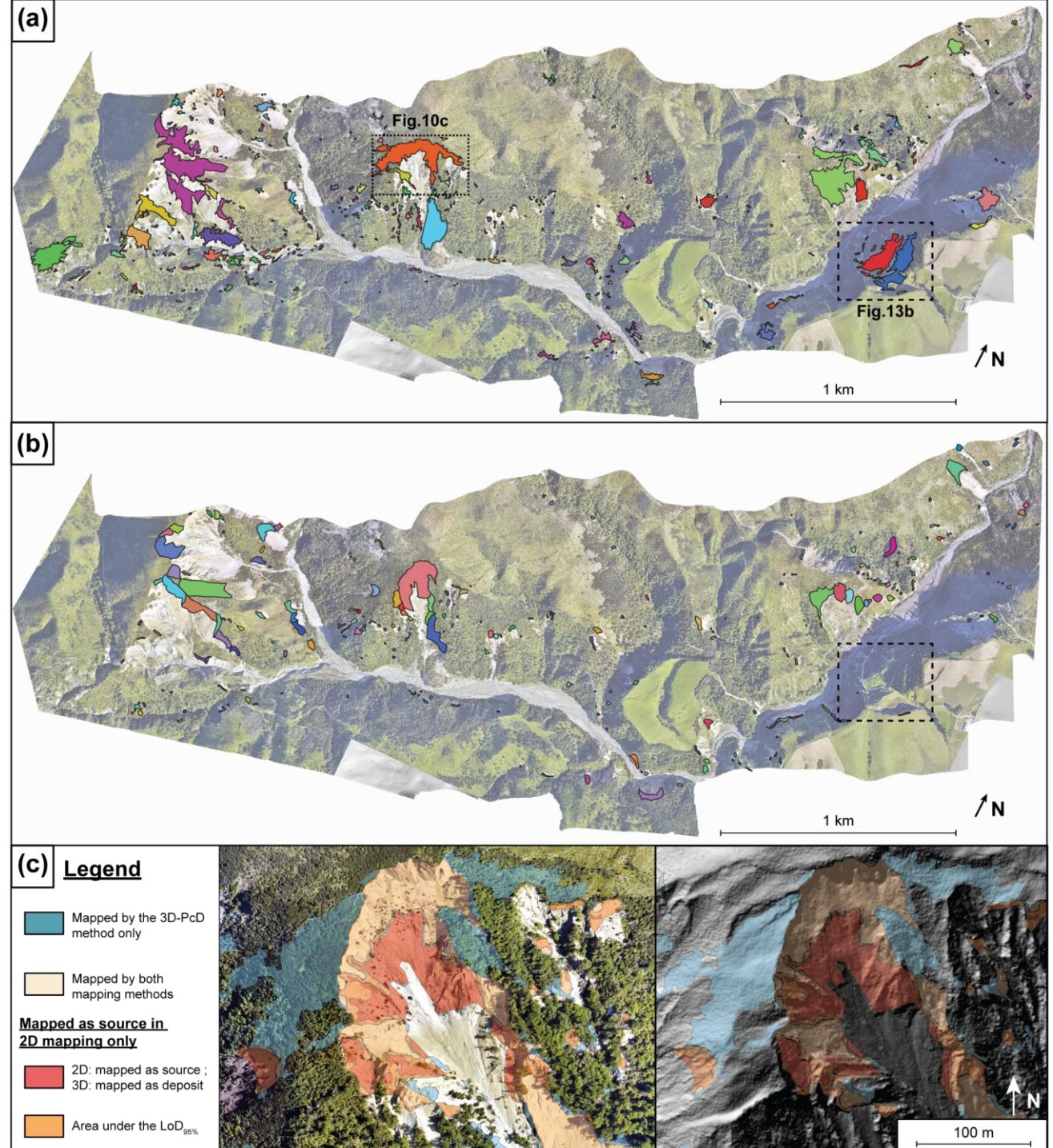

Figure 10: Comparison between (a) the 3D predicted inventory and (b) a manual mapping based on 2D orthoimage comparison. Each landslide source is shown as a single colored polygon. (c) Detailed comparison of typical mapping differences between the 2D and 3D approach. Data are overlaid on the post-earthquake orthoimagery (12-15-2016, Aerial survey, 2017) and the post-EQ DEM is shown on the right. Fig 10c shows that the detected negative topographic change by the 3D-PcD method under dense forest area


**is consistent with the development or reactivation of retrogressive failure planes with visible scarps in the hillshade view around the main landslide. 2D orthoimagery cannot detect these mainly vertical change under forest, that do not affect the texture of the image.**

**Table 4: Summary of the comparison between the 2D and the 3D predicted landslide source inventories. For the 2D inventory, the percentages are calculated with respect to the corrected total in which the 2D sources corresponding to 3D deposits are removed.**

| | Category | landslide sources | Area m² | Area % | Volume* m³ | Volume* % of 3D total |
|---|---|---|---|---|---|---|
| **2D inventory** | Total | 258 | 146,641 | | NA | |
| | *On 3D deposit* | *22* | *21,735* | *14.8* | NA | |
| | Corrected total | 236 | 124,906 | **100** | 331,621 | 45,8[+] |
| | Shared | 193 | 71,495 | 57.2 | 301,577 | 41,6[+] |
| Not in 3D | < LoD$_{95\%}$ (positive/negative changes) | 39 | 29,250 / 23,308 | 23.4 / 18.6 | 13,951 / 14,980 | 2.2[+] (negative changes) |
| | < min area (20 m²) | 0 | 376 | 0.3 | 419 | |
| | > T$_{CDD}$ and <T$_{SNR}$ | 4 | 477 | 0.4 | 541 | |
| **3D inventory** | Total | 433 | 259,415 | **100** | 724,297 | **100** |
| Not in 2D | Shared | 182 | 71,495 | 27.6 | 301,501 | 41.7 |
| | Forest | 48 | 32,236 | 12,4 | 77,066 | 10.6 |
| | Bare rock | 116 | 72,533 | 28.0 | 247,998 | 34.2 |
| | Other land covers | 87 | 83,151 | 32.0 | 97,732 | 13.5 |

*: volumes for the 2D inventory are computed from the vertical-M3C2 of core points located within the sources delimitations

+: the percentage represents the 2D volume of the class in comparison of the total volume of the 3D predicted inventory.

## 4.4. Landslide sources area, depth and volume analysis

The area distribution of landslide sources is computed as follow (Hovius et al., 1997; Malamud et al., 2004):

$$p(A) = \frac{1}{N_{LT}} \times \frac{\delta N_{LT}}{\delta A} \tag{4}$$

where $p(A)$ is the probability density of a given area range within a landslide inventory, $N_{LT}$ is the total number of landslides and $A$ is the landslide source area. $\delta N_{LT}$ corresponds to the number of landslides with areas between $A$ and $A + \delta A$. The landslide area bin widths $\delta A$ are equal in logarithmic space.

First, the area distribution of landslide sources obeys a power-law scaling relationship consistent with previous studies (e.g., Hovius et al., 1997; Malamud et al., 2004). The exponents are respectively $c = - 1.76 \pm 0.06$ and $c = - 1.64 \pm 0.03$ for the 2D

and 3D inventories, respectively (Fig. 11a). The distribution of the 2D inventory shows a cut-off from power-law behaviour at around 100 m², a peak probability at 20 m² and a rollover for smaller landslide sizes. The landslide area distribution of the 3D predicted inventory does not exhibit a rollover but slightly-deviate from the power-law behaviour around 40 m². The distribution differs from that derived by Massey et al. (2020) in the broader Kaikoura region for which a cut-off appears around 1000 m² with a rollover at 100 m².

The volume distribution of the landslide sources in the 3D inventory was defined using equation (44), replacing $A$ by the volume $V$, and also exhibits a negative power-law scaling (Fig. 11b) of the form: $p(V) = dV^e$. The exponent of the power-law relationship is $e$ = -1.54 ± 0.07. A rollover is visible on the landslide volume distribution around 20 m$^3$.

With a direct measurement of landslide volume, it is possible to compute the volume-area relationship (eq. (1); Simonett, 1967; Larsen et al., 2010) and to compare it with previous results in New Zealand (Larsen et al., 2010, Massey et al., 2020).

Here we determine V-A scaling coefficients using two methods: by fitting a linear model (1) on log-transformed data and (2) on averaged log-binned data. While the first method leads to a V-A relationship best describing the volume of each landslide, the second one is not affected by the varying number of landslides in each landslide area bin and leads to a V-A relationship that best matches the total landslide volume. Using the first approach, we find a volume-area scaling exponent of $\gamma = 1.14 \pm 0.01$ and an intercept $\log \alpha = -0.20 \pm 0.03$ m$^{0.72}$ with a determination coefficient $R^2 = 0.93$ (Fig. 11c). Using the second

method, we find $\gamma = 1.17 \pm 0.03$, an intercept $\log \alpha = -0.22 \pm 0.10$ m$^{0.66}$ and a determination coefficient $R^2 = 0.99$. We also obtain a good correlation R² of 0.86 and 0.80 with the Larsen et al. (2010) relationships derived from soil landslides and from mixed soil landslides and bedrock landslides, respectively (Tab. 5). R² of 0.92 is obtained when considering the parameters of the V-A relationships of the Kaikoura region, derived by Massey et al. (2020), including all their mapped landslides. At first order, the V-A relationships we obtained are thus consistent with previous studies. Yet, if the relationships

from Larsen et al. (2010) and Massey et al. (2020) were applied to our landslide area inventory, the total volume would vary from 0.346x10$^6$ m$^3$ to 0.940x10$^6$ m$^3$ (Tab. 5), compared to 0.724x10$^6$ ± 0.141x10$^6$ m$^3$ that we estimate directly. The closest evaluation of the total volume is based on the Massey et al. (2020) V-A relationship that predicts a total volume of 0.600x10$^6$ m$^3$. The farthest evaluation of the total volume is the V-A relationship from Larsen et al. (2010) for all landslides (0.940x10$^6$ m$^3$), while their soil-dominated landslide relationship predicts less than a half of the total volume of the 3D inventory.

We presented the V-A relationship as it is classically used in co-seismic volumes estimate from 2D inventories, however because the volume being is the product of mean depth and area, the V-A relationship hides an indirect correlation with area which may hinder obscure subtle variations of depth with landslide size. The raw mean depth-area data shows a large scatter for nearly all landslide areas. The log-binned data shows a slight increase of depth with area. Using a power-law model which is actually weakly constrained here, an exponent 0.18 ± 0.06 (R²=0.87) is derived consistent with the V-A relationship. The

soil landslide relationship of Larsen et al. (2010) and Massey et al., (2020) are broadly consistent with the log-binned data (R² =0.81 and R²=0.75, respectively), but the mixed soil and bedrock landslide relationship of Larsen et al., (2010) exhibits a much larger scaling exponent resulting in a poor correlation (R² =-0.09).

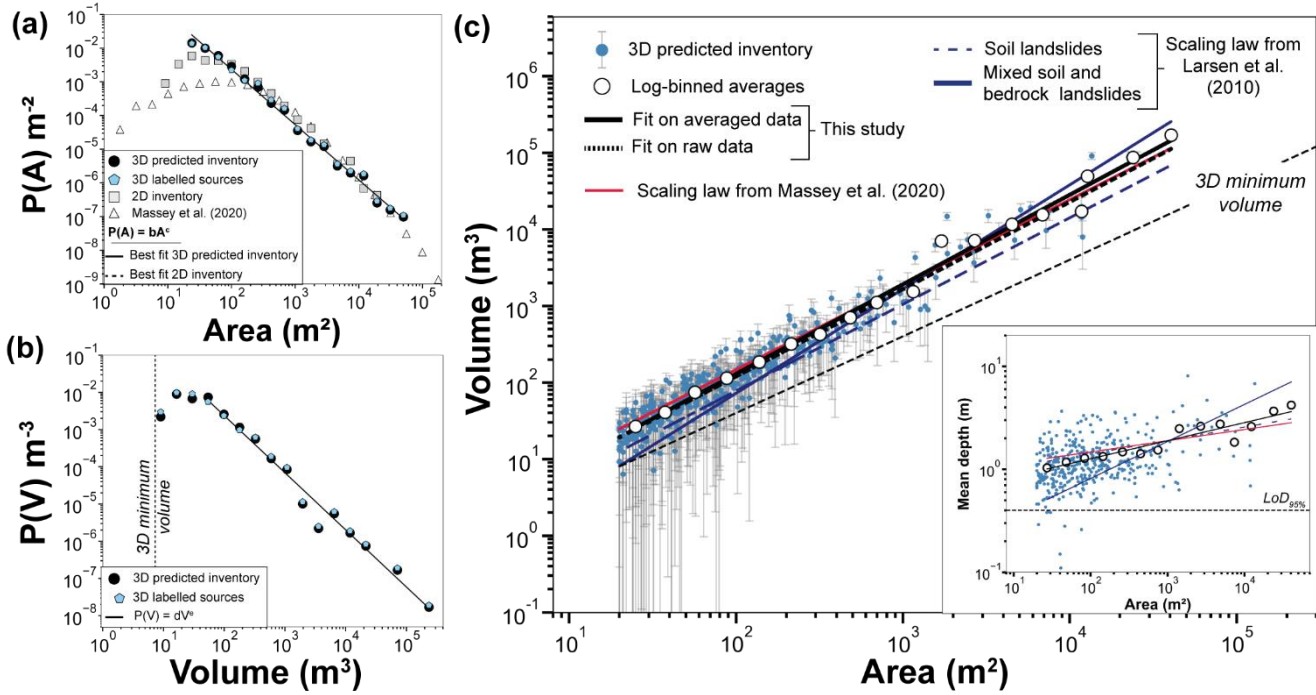

Figure 11: Landslide source inventory analysis of the study area. (a) Landslide area distribution of the 3D predicted, the labelled source, the 2D inventory and Massey et al. (2020). (b) Landslide volume distribution of both the 3D predicted and the labelled source inventory. (c) Volume-area scaling relationships with uncertainty on volume and comparison with Larsen et al. (2010) and Massey et al. (2020) relationships obtained in New Zealand. The landslide mean depth vs area is also presented in inset. All scaling parameter values are summarized in Table 5. Fits correspond to least-square fitting of a power-law to log-binned data above a manually selected cutoff.

Table 5: Power-law scaling parameter values of the relations show in Figure 8. Log α and γ are scaling parameters from the landslide area-volume relationship (eqn. 1). Units of α are [L(3-2γ)] with L in meters. Landslide source area and volume distribution coefficients are b and d while exponents are c and e respectively. The coefficient of determination R² is also given for each power-law fit function. The total volume refers to the application of the V-A relationship to the landslide areas of the 3D predicted inventory.

| | | log b, log d or log α | c, e or γ | R² | Total Volume (m³) |
|---|---|---|---|---|---|
| Area distribution | 3D predicted inventory | $0.65 \pm 0.11$ | $-1.64 \pm 0.03$ | 0.99 | - |
| | 2D inventory | $1.20 \pm 0.18$ | $-1.76 \pm 0.06$ | 0.97 | - |
| Volume distribution | 3D predicted inventory | $0.45 \pm 0.23$ | $-1.54 \pm 0.07$ | 0.98 | $0.724 \times 10^{6*}$ |
| V-A relationship | Averaged log-binned data (this study) | $-0.22 \pm 0.10$ | $1.17 \pm 0.03$ | 0.99 | $0.702 \times 10^6$ |
| | Log-transformed data (this study) | $-0.20 \pm 0.03$ | $1.14 \pm 0.01$ | 0.93 | $0.561 \times 10^6$ |
| | Soil landslides (Larsen et al., 2010) | $-0.37 \pm 0.06$ | $1.13 \pm 0.03$ | 0.86 | $0.346 \times 10^6$ |
| | Mixed soil and bedrock landslide (Larsen et al., 2010) | $-0.86 \pm 0.05$ | $1.36 \pm 0.01$ | 0.80 | $0.940 \times 10^6$ |
| | Landslide inventory Massey et al., 2020 | $-0.05 \pm 0.02$ | $1.109 \pm 0.01$ | 0.92 | $0.600 \times 10^6$ |

| D-A relationship[+] | | | | | |
|---|---|---|---|---|---|
| | Averaged log-binned data (this study) | $-0.25 \pm 0.02$ | $0.18 \pm 0.06$ | 0.87 | - |
| | Log-transformed data (this study) | $-0.26 \pm 0.04$ | $0.16 \pm 0.02$ | 0.17 | - |
| | Soil landslides (Larsen et al., 2010) | $-0.11 \pm 0.02$ | $0.13 \pm 0.03$ | 0.81 | - |
| | Mixed soil and bedrock landslide (Larsen et al., 2010) | $-0.81 \pm 0.05$ | $0.36 \pm 0.01$ | -0.09 | - |
| | Landslide inventory Massey et al., 2020 | $-0.05 \pm 0.03$ | $0.109 \pm 0.01$ | 0.75 | - |

*: Direct measurement; +: The exponent is estimated as γ-1 and the coefficient has units of [$L^{(1-2\gamma)}$]. R² estimated on log-binned data.

# 5. Discussion

The aim of this paper is to present a semi-automatic workflow, called 3D-PcD, for the detection and geometric characterization of landslide sources and deposits from repeated airborne LiDAR data. We specifically aim to overcome the impact of issues such as under-detection of landslides in inventories based on imagery analysis, landslide amalgamation and V-A relationship biases on total volume calculation. In the following, we discuss 1) the benefits and limits of the 3D-PcD method, 2) the benefits of 3D change detection to create landslide inventories, and 3) how 3D landslide inventories shed new light on the scaling properties of landslide sources.

## 5.1. 3D point cloud differencing and landslide detection

### 5.1.1 Vertical versus 3D change detection capability, and the M3C2 algorithm

The importance of detecting changes in 3D (3D-M3C2), as opposed to vertically (vertical-M3C2), in steep slopes can be illustrated by a SSDS test applied to the post-EQ point cloud (Fig. 12). Typical of change measurement methods on rough surfaces with random point sampling (e.g., Lague et al., 2013), a non-null mean distance is often measured, even though the two point clouds are samples of exactly the same surface. The distribution of measured distances is centred near zero, with means of $-2 .10^{-4}$ and $1.10^{-4}$ m, for the vertical and 3D approaches respectively. However, the 3D approach results in a standard deviation, $\sigma=0.05$ m, four times smaller than using a vertical differencing, $\sigma= 0.20$ m. The map of difference shows that vertical differencing systematically results in much larger distances on steep slopes than the 3D approach, while they both yield similar low distances on horizontal surfaces.

We thus find that the 3D-PcD method offers a greater sensitivity to detect changes compared to classical vertical DoD. This difference is particularly important as it propagates into a lower level of detection and uncertainty on volume calculations. Using the M3C2 algorithm in 3D (Lague et al., 2013) also offers the benefit of accounting for spatially variable point density and roughness in estimating a distance uncertainty for each core point, that can be subsequently used in volume uncertainty calculation. For instance, 3D-M3C2 reduces the sensitivity of change detection in vegetated areas to a lower ground point density and potentially to a higher roughness due to vegetation misclassification. In turn, this advantage prevents in part the detection of false sources or deposits by using 3D-M3C2. By using a regular grid of core points as in Wagner et al. (2017), our

workflow combines the benefits of working directly with the raw unorganized 3D data, as opposed to DoD where the relationship with the underlying higher point density data is lost. This approach also produces results with a regular sampling that can easily be used for unbiased spatial statistics, volume calculation and easy integration into 2D GIS software. Compared to DoD, if an interpolation is needed, it is performed on the results rather than on the original DEM which can lead to uncontrolled error budget management.

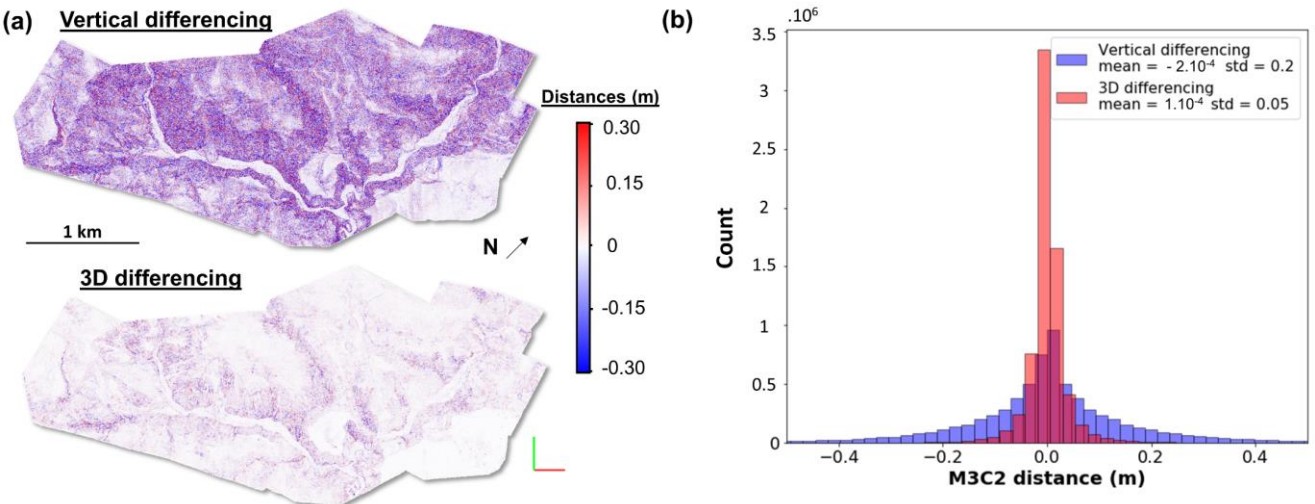

Figure 12: Comparison between vertical differencing (vertical-M3C2) and 3D differencing (3D-M3C2) on the post-EQ point cloud, sub-sampled randomly two times to generate two point clouds of the same surface with a different sampling (Same Surface Different Sampling test). (a) Resulting change detection maps of the two different techniques. (b) Histogram of the computed distances with the two techniques.

### 5.1.2. Current limits of the method

**Registration and elevation errors:** A critical aspect of the comparison of 3D point clouds is their co-registration, in particular in the context of co-seismic landsliding. In this study, a rigid transformation is applied to the datasets using an ICP algorithm (Besl and McKay, 1992), assuming that internal deformation induced by the earthquake is negligible. The 3D-M3C2 map does not exhibit any systematic horizontal shift either north or south of the Hope fault. We thus conclude that internal deformation, if any, was below the typical registration error in our study area. For larger studied regions with internal deformation and in the absence of a 3D co-seismic deformation model that could be applied to the post-EQ point cloud (e.g., Massey et al., 2020), our workflow should be applied in a piecewise manner with boundaries corresponding to the main identified faults or deformation zones. For landslide inventories following climatic events, the application to large datasets should be straightforward as no significant internal deformation is expected. However, we note internal flight line height mismatches of 0.13-0.20 m in the pre-EQ survey that are difficult to correct after data delivery and generate some apparent large scale, low amplitude topographic changes (Fig. 4, Tab. S3 and section S2 in the Supplement). Interestingly, in the M3C2 calculation, flight line mismatches are averaged out in the distance measurement but lead to a higher local point cloud standard deviation, and thus to an increase of the $LoD_{95\%}$ and to a lower probability of incorrect topographic change detection. Despite significant

flight line mismatches in the pre-EQ dataset, using the *SNR* and *CDD* filtering approach efficiently removes the false detection source areas related to this issue in non-forested area. This highlights 1) the need for a detailed quality control (e.g., by applying M3C2 on overlapping lines) to ensure the highest accuracy of the LiDAR data, 2) the importance of the statistical significance tests performed at the core point scale, and 3) the need for confidence metrics at the landslide scale, to filter out a variety of potential false landslides. Ideally, a spatially variable model for point cloud errors and registration should be developed for each survey and combined into a more accurate and complete form of *LoD* than what the M3C2 approach currently offers (e.g., Glennie, 2008; Passalacqua et al., 2015). However, the position and attitude information of the sensor (e.g., Smoothed Best Estimate of Trajectory file) and raw LiDAR data are rarely available from LiDAR data repositories. Additionally, a dense network of ground control points is hard to get in mountainous environments. It is thus frequently impossible either to reprocess the LiDAR data to improve their quality (e.g., Glennie et al., 2014), or to create a spatially variable registration and point cloud error model. This explain our choice of assuming a uniform registration error and to estimate it empirically from intra-survey flight line overlapping errors or inter-survey 3D-M3C2 distances over stable areas. As a result, we analyse the influence of the *reg* value on the landslide source population (see section 5.2.2).

**Landslide segmentation:** The connected component segmentation is a simple, objective and rapid way to separate landslides in 3D that can be scaled up to much larger datasets. However, given the complexity of the 3D data, and in particular the very large range of landslide sizes (i.e., 4 orders of magnitude in the studied case), it inevitably exhibits some drawbacks and is subject to improvement. In particular, landslide amalgamation occurs between two sources or deposits if two of their core points are closer than $D_m$ (2 m in this study). Hence, landslides occurring on the two sides of a collapsed divide can be erroneously connected. This is the case for the largest landslide of our database located on the rock cliffs in the western part of the study area (Fig. 10a and Fig. 13a). In this example, the landslide source could reasonably be segmented into at least 5 smaller landslides. However, there does not seem to be a unique way to segment such a complex set of amalgamated events, even manually, underlining that landslide segmentation cannot currently be fully objective. As we aim to apply our workflow to very large dataset with potentially several tens of thousands of landslides, and that false detections filtering need to operate on segmented patches, automatic segmentation is a mandatory step. We have explored the use of fast implementations of density based spatial clustering algorithm derived from DBSCAN (Ester et al., 1996) an algorithm used for segmentation of 3D point clouds of rockfalls and removal of noisy points (e.g., Benjamin et al., 2020; Tonini and Abellan, 2014; details in Section S5). We applied OPTICS (Ankerst et al., 1999), recently used for rockfall segmentation of 3D LiDAR data (Carrea et al., 2021) and HDBSCAN (McInnes et al., 2017) which has a better ability to detect clusters of various sizes compared to DBSCAN, and is much faster. However, none managed to provide a significantly better segmentation of the largest landslides of our database, and the density probability of source area they produce is very similar to the one generated by a connected component approach. These approaches are however significantly longer to run than a connected component approach in Cloudcompare (Section S5), and have parameters which are less intuitive to set than $D_m$, which is a distance directly comparable to core point spacing. New segmentation approaches accounting for normal direction, divide organization and 3D depth maps of amalgamated sources are needed to improve the segmentation of complex cases. Manual segmentation can also

be envisioned as a refinement after the predicted true landslide sources and deposits have been produced, but is non reproducible and time consuming when applied to very large datasets. We note however that segmentation issues do not affect the total landslide volume calculation in our study and that a sensitivity analysis of the impact of $D_m$ shows that landslide

source statistics are not severely affected by this parameter as long as it is close to the value we have used (see section 5.2).

**Landslide volume calculation:** Landslide volume is computed using vertical-M3C2 on regular core points. This facilitates volume calculation on potentially complex 2D landslide geometry, but may lead to incorrect volume estimates on very steep slopes. Yet, the median slope of the source core points (measured on the pre-EQ surface, Fig S11) is 33.9°, and only 1.2 % of the core points have slopes higher than 60°. We thus expect this effect to affect a very small fraction of our inventory.

Measuring landslide volume in 3D would be preferable, for instance along a constant surface normal direction defined for each source or deposits, but such simple approach is not better than a vertical measurement for the complex surface geometry of large landslides observed in the dataset and which are properly segmented (e.g., Fig. 13a and b). New approaches based on 3D mesh reconstruction have been used recently for rockfall volume estimation (Benjamin et al., 2020) and represent a future improvement of our workflow.

**Landslide surface area:** Another simplification of our approach is the calculation of planimetric surface areas, rather than true surface area. This choice was made to be consistent with previous results based on 2D inventories and to facilitate the comparison with our image-based inventory. Measuring surface parallel area with 3D data would potentially help unravel new relationships between normal depth and area that are independent of topographic slope. Yet, this calculation is not trivial for complex landslide geometries occurring, for instance on highly curved hillslopes in which assuming a unique normal

orientation to get a surface parallel area measurement could result in a bias on the sides of the scar (e.g., Fig. 3a). Approaches based on 3D mesh surface calculation could help resolve this. Given that the median of the slope distribution of our landslide source inventory is 33.9°, a back of the envelope estimate of the true area gives a total landslide source area of 334,590 m² rather than 259,415 m².

**Translational/rotational landslides with limited internal deformation:** The 3D-PcD workflow we have designed is not

intended for the measurement of landslides for which the dominant movement is parallel to the topographic surface, as is can be the case for translational or rotational landslides mobilizing mostly intact hillslope material. As in Figure 13b, these landslides will appear as negative surface elevation in the subsiding source area and positive in the downslope accumulation area with little or non-significant 3D-M3C2 distance over much of the landslide body. These landslides can be detected efficiently with the 3D-PcD workflow, even below forest, but the corresponding actively mobilized volume and area may not

be accurately estimated. For mostly translational landslides, the surface parallel component of the deformation may be evaluated with feature tracking approaches as long as there are features to track (e.g., Aryal et al., 2012; Teza et al., 2007). The only elements that could be easily tracked in the 3D-PcD workflow are the barycenter of the source and associated deposit of each landslide, to explore runout dynamics, but we have not investigated this option yet.

**Significant changes and geomorphic processes:** While not a limitation per se, the 3D-PcD workflow detects changes, but

cannot classify the nature of this change into various types of geomorphic processes. Given the current $LoD_{95\%}$ (i.e., $> 0.40$ m)

only large topographic changes corresponding to mass wasting processes on hillslopes and fluvial processes, are detected. Debris-flow processes could be detected, and may actually be part of the processes that remobilize landslide debris, but they potentially create erosion in narrow steep channels that are likely below our spatial resolution capability, or will generate very small sources. They could however generate very large deposits. Topographic change due to fluvial processes are removed by the only manual operation performed in 3D-PcD, deemed necessary to preserve landslide deposits that have reached the river. Our approach does not directly resolve the typology of the landslides, including their failure mechanism (sliding, flow, fall), the failed materials (rock, soil, debris) and the velocity of the displacement (Hungr et al., 2014). Yet, combining the 3D-M3C2 distance field with orthoimages (Fig. 13), we have identified the presence of rock avalanches, slumps (rotational failures), debris slides (translational failure) and we suspect the occurrence of some large rockfalls, although pre-EQ slopes steeper than 60° are extremely rare in the detected sources. We did not try to separate these as: (1) we were primarily interested in co-seismic volumes rather than detailed landslide mechanics which would have required field data; (2) there is no way to unequivocally identify, for the vast majority of our sources, the dominant landslide mechanism with either the 3D-M3C2 distance field and/or the orthoimages; and (3) large landslides for which a dominant mechanism can be identified are too few in our inventory to draw a robust inference on scaling properties and geometry .

### 5.1.3. False detections and filtering metrics

We show that performing 3D point cloud differencing can lead to many false detections that the $LoD_{95\%}$ statistical model fails to remove. The labelled inventory shows a prevalence of false detections around 50% with a strong size dependency and land cover type: false detections are much more frequent in forests (80%) owing to ground classification error of the pre-EQ LiDAR data, than in forest-free areas (25%). As we operate at a confidence level of 95%, incorrect false detections may occur, but the prevalence we observe in the provisional inventory showed that our $LoD_{95\%}$ model is too optimistic. Yet, given the complexity of building a spatially explicit model of elevation errors (see section 5.1.2), we propose that our formulation of the $LoD_{95\%}$ is complex enough, and that the critical point of the workflow is the filtering of false detections alongside automatic segmentation. While fewer false detections would happen by increasing *reg*, it would be at the expense of further censoring the detection of shallow landslides.

Our work demonstrates that outside forested areas, spatially correlated elevation errors resulting in false detection can be filtered out by a combination of the *SNR* and the *Closest Deposit Distance*, while preserving most of the true landslides (balance accuracy= 0.98). The predicted inventory has a prevalence of false detections in the final inventory of less than 1.5 % in number, and a negligible fraction in terms of area and volume ($< 1$ %).  The optimal *CDD* that best remove false detections is low (18 m) consistent with the expectation that long wavelength, spatially correlated errors are unlikely to produce coherent patterns of negative and positive topographic change at close distance and along the downslope direction. This low value may also reflect the criteria used to label the provisional source inventory but the value of 18 m we obtain in forest-free area is much lower than our manual labelling criteria (30 m). While the thresholds we derive for *CDD* and *SNR* are optimal for this

region, manual labelling of a fraction of the provisional data may still be needed to evaluate the optimal threshold values in other regions and for other data registration and quality.

It should be emphasized that the pre-EQ LiDAR data we use probably represents a worst case scenario for topographic change detection for two reasons: first, it has relatively large internal geometrical errors compared to typical modern dataset such as the post-EQ data which translates in a poor registration error, limiting our ability to detect change at ~ 40 cm. Second, it suffers from ground classification errors related to the limited laser shot density and penetration capability of older generation sensors in dense forest: 4 echoes maximum for the pre-EQ data, while the post-EQ data can have up to 7 echoes for a single laser shot,

which improves the likelihood of hitting the ground under dense canopy. We expect many legacy LiDAR data in mountain belts to suffer from similar issues (e.g., Glennie et al., 2014). The prevalence of false detections in forest (80 %) is a direct consequence of ground classification errors and results in potentially many small false detections that should be properly filtered out. The optimal *CDD* results in a good balanced accuracy (0.8), yet this result could probably be improved by using additional metrics related to the geometry of the provisional landslide sources. We expect the proportion of false detection to

decrease with the increase of LiDAR data quality with higher point density and canopy penetration, offering greater insights into the mass wasting processes that may occur in forested area.

## 5.2. Benefits of the 3D-PcD approach to create landslide inventories

### 5.2.1. Landslide topographic change detection compared to manual passive imagery mapping

We presented for the first time a comparison between a classical manual inventory of landslide sources from 2D orthoimagery

comparison, and a 3D inventory based on LiDAR change detection where landslides are detected according to the topographic change they produce. Results show how different two landslide inventories of the same region, constructed from fundamentally different data sources (passive vs active remote sensing), can be. While the 3D inventory cannot be considered exhaustive, as it has a non-null $LoD_{95\%}$ and a lower limit of size detection of 20 m², it nonetheless detects roughly 2 times more landslides than the 2D image-based approach and a planimetric area affected by landsliding nearly two times larger. Most importantly,

the detection limit for the 3D-PcD workflow is known, as one of its outcome is a spatially variable confidence interval ($LoD_{95\%}$) and confidence metrics ($SNR$) for each segmented source and deposit. While the resolution capability of 2D image analysis can be evaluated based on pixel size and is better than the LiDAR based approach in our study case, the detection capability is much more difficult to quantify, especially if the inventory is manually performed.

Both detection and delimitation errors roughly equally contribute to under-detection of the total area in the 2D inventory. They

are, as expected (Zhong et al., 2020), frequent in areas with poor spectral contrasts between successive orthoimages such as bare rock surfaces. But under-detection also occurs in sparse or low vegetation zones (other land covers) where some very large areas corresponding to vertical subsidence at the top of rotational or translational landslides were not detected (e.g., Fig. 13b) or incorrectly mapped (e.g., Fig. 10c). Our results also suggest that under-detection can occur in forest: based on the manually labelled 3D data, 44 landslides are not detected in 2D, representing 11% of the labelled 3D source inventory. Yet,

we also show that a high proportion of 3D false detections can remain in forested area (33%). We also demonstrate that large scale subsidence areas associated to new or reactivated retrogressive slip planes (Fig. 10c) can be detect in forest and may prove important for subsequent landslide hazard management.

Delimitation and detection errors are dominant on sparse and bare rock surfaces corresponding to 60% of the total landslide area. In particular, it is extremely difficult to map the transition between sources and deposits, especially on large and

amalgamated landslides (e.g., Fig 10c). Here the ability of the 3D-PcD approach to not only detect sources but also deposits is essential. Our results thus indicate that existing landslide inventories, manually mapped from 2D images, may significantly suffer from under-detection of landslide area at least in regions dominated by sparse or absent vegetation cover. Yet, by capturing landslide mechanisms such as rotational/translational landslides or rockfalls on steep hillslope more systematically, the 3D inventory statistical properties may not be fully comparable to traditional 2D inventories.

We show that the main reason the 3D-PcD method did not detect surfaces mapped on the 2D inventory is that these surfaces are located in areas where the 3D-M3C2 distance is below the $LoD_{95\%}$. The detection limits of the 3D-PcD will improve in future years, by using the latest generation of LiDAR instruments generating dense (> 10 pts/m²) and more accurate 3D point clouds (< 5 cm Z error). With such data, the registration error could become of the order of 5 cm or less, further improving the detection capability of 3D-PCD both in terms of spatial resolution and $LoD_{95\%}$.


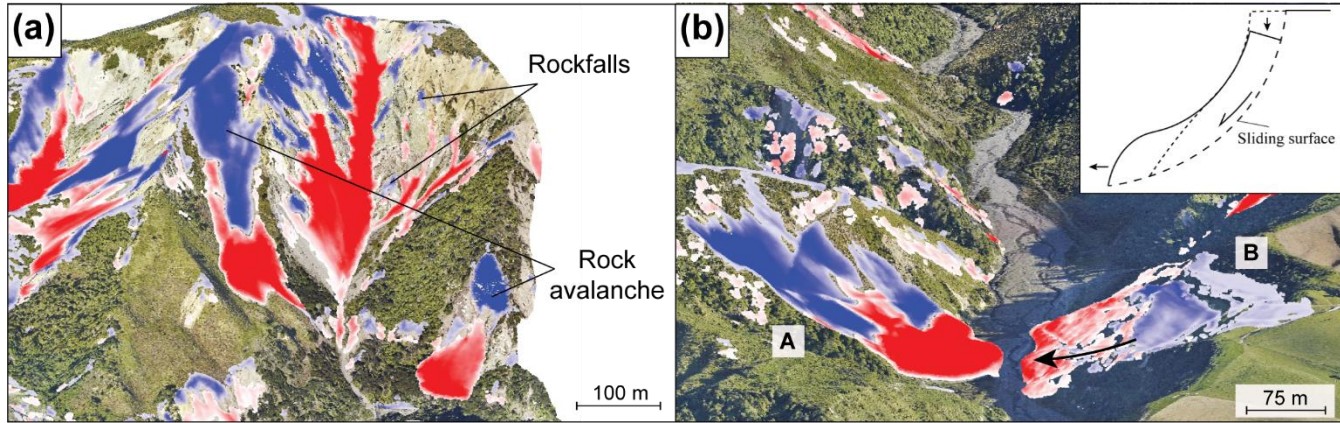

**Figure 13: Two different points of interest in the 3D predicted inventory illustrating various type of landslide mechanisms (blue is erosion, red is deposition). (a) Area mostly dominated by rock avalanche, where large rockfalls are also expected. (b) Debris slide with mostly translational movement (A) and slump with likely rotational to translational displacement (B). The post-earthquake**
**orthoimage is overlaid on the point cloud (December 15, 2016; Aerial survey, 2017)**

### 5.2.2. Toward a minimization of amalgamation and under-detection biases on total landslide volume estimation

By direct measurement from topographic data, the amalgamation effect is no longer an issue for total landslide volume estimation of an inventory even though our segmentation approach cannot resolve the amalgamation of individual landslides perfectly. Bypassing the use of a non-linear V-A relationship also avoids uncertainty inherent to the choice of the best suited

scaling parameters. As we show, the total landslide volume varies significantly (from $0.346 \times 10^6$ m³ to $0.940 \times 10^6$ m³ ; Tab. 5)

depending on the V-A scaling relationship applied to our landslide inventory. We also observe a difference of 20% in total volume estimation only due to the method used to fit data (i.e. on log-transformed or on averaged log-binned data). We also note that total landslide volume estimation from such relationships can get close to the volume estimated from the 3D-PcD but for the wrong reasons. Applying our V-A relationship to both versions of the 2D inventory with and without deposit areas (Tab. 4) leads to a difference of 15% in total volume. These results highlight the overarching sensitivity of the total volume of eroded material to the V-A relationship biases (Li et al., 2014; Marc and Hovius, 2015).

Our 3D-PcD approach also allows estimation of total landslide volume without the issue of under-detection of landslides. Due to the difference in the type of under-detection and delimitation errors between both 2D and 3D inventories, these issues do not propagate into total landslide volume estimate in similar ways. The volume not detected by the 3D-PCD method but in the 2D inventory represents only 2.2 % of the total. This is a negligible component owing to the fact that only very shallow landslides, or shallow parts of very large landslides, are missed. In contrast, the area not detected by the 2D inventory represents 54 % of the total volume, highlighting the pronounced underestimation of total volume estimate if one uses image based detection followed by volume calculation. Most of this missed volume is due to the landslide delimitation errors on bare rock and sparse vegetation cover surfaces (other land covers) which represent 34.3% of the total volume while the under-detection of entire landslides only represents 13.3%. We also note that a third of the total volume is missed on bare rock surfaces. Our study area was chosen based on LiDAR data availability and contains a particularly high proportion of under-detected landslides in the 2D inventory due to the presence of actively eroding bare bedrock hillslopes. We expect this proportion to significantly vary when considering other landscapes with potentially varying proportions of vegetation cover, vegetation density and type (e.g. grass, shrubs, trees), lithology and ground shaking intensity. Nonetheless, our finding represents a first approach to the issue of considering the under-detection of landslides in total landslide volume estimates. We show that extreme caution should be put on co-seismic volumes estimated on landscapes where a large fraction of bare rock surfaces and sparse vegetation cover are present before earthquake, such as the Kaikoura ranges.

## 5.3. Landslide source scaling properties

The use of 3D data opens up a very large range of new geometric analysis of landslide sources and deposits. Here, we revisit traditional size-distributions and scaling relationships of landslide sources generated from 2D inventories, even though, owing to the detection methods, both type of inventories may not be fully comparable in the type of landslide mechanism they capture. We use the relationships derived from the manually labelled 3D inventory and from the 3D predicted inventory which is slightly more complete but contain a few false detections. We then perform sensitivity analysis of these relations to the main parameters of the 3D-PcD workflow: the registration error $reg$ and the minimum distance for segmentation $D_m$ (See Fig.14, S12 and S13).

### 5.3.1. Total volume of landslide sources and deposits

Over the studied area of ~5 km$^2$, 433 landslide sources and 399 landslide deposits were detected with the 3D-PcD workflow. The scaling of pdf($V$), with an exponent of -1.54 ± 0.07, indicates a slight tendency for the overall eroded volume to be dominated by the largest landslide (169,725 m$^3$, that is 23 % of the total volume). The uncertainty on total landslide volume, 17% to 19 % for deposits and sources, respectively, might appear large, as it is based on a conservative 95% confidence interval that we use throughout our analysis. These uncertainties are dominated by the registration error ($reg$ = 0.2 m) and by the lower point cloud density of the pre-earthquake LiDAR data (Tab. 1). Considering both forested and forest-free areas, false detections only represent 0.44% of the total volume of the predicted inventory. The uncertainty on volume is thus mainly controlled by the M3C2 distance uncertainty rather than the presence of false detections. Within these uncertainties, the total volume of sources (724,297 ± 141,087 m$^3$) and deposits (954,029 ± 159,188 m$^3$) are not statistically different. The larger volume of deposits is however consistent with rock decompaction during landsliding, which could be constrained by using a joint gravity survey in future studies (Mouyen et al., 2020).

### 5.3.2. Distribution of landslide source area and lack of rollover

We obtain a range of landslide areas over 3 to 4 orders of magnitude (20 to 42,475 m²) which obey a power-law relationship for A > 40 m² with an exponent c = -1.64 ± 0.03 (Fig. 11a). The negative power law behaviour for landslide area is generally observed for 2D landslide inventories, although only for source areas typically larger than 500- 5000 m² (Guzzetti et al., 2002; Malamud et al., 2004; Malamud and Turcotte, 1999;), and although the power-law nature of the tail is debated (Jeandet et al., 2019; Medwedeff et al., 2020). Our exponent is roughly consistent with the exponents obtained over the entire Kaikoura coseismic landslide inventory of -1.88 ($N_{LT}$ = 10,195; Massey et al., 2018) but differs significantly from a more recent estimate of -2.10 (N$_{LT}$ = 29,557; Massey et al., 2020, Fig. 11a) for which the power-law scaling is expressed for A > 500 m². A sensitivity analysis of the impact of the workflow parameters (Fig. 14), in particular $D_m$ which affects the level of amalgamation in the dataset, does not yield values of $c$ smaller than -1.67 and cannot reconcile our results with those of Massey et al. (2020). Either our limited study area overemphasizes, by chance, the occurrence of large landslides generating a smaller value of c, or the manual inventory of Massey et al., 2020 may miss a large fraction of intermediate and small landslides, especially on bare rock hillslopes which are frequent in the high mountains of the Kaikoura range.

Most importantly, the landslide area distribution that we derive does not exhibit a rollover classically observed in 2D landslide inventories. Only a small deviation of the power-law behaviour appears for A < 40 m². The same behaviour is observed for the labelled source inventory without any false detections (Fig. 14a). Varying $reg$ or $D_m$ does not change this behaviour (Fig. 14b and 14c), nor does the use of a density based clustering approach (Fig. S8). In addition, setting reg to 0.5 m as opposed to 0.2 m, implies that only changes larger than 0.98 m are statistically significant. This leaves only 251 sources out of 433, yet p(A) does not exhibit a rollover or even a significant deviation from the power-law behaviour for small landslides. Hence, we are confident that our probability density of source area, generated by a purely objective and automatic approach, does not

exhibit a rollover. If there is any, it would occur for sizes smaller than 20 m². We note that p(A) for the labelled false detection obey an approximate power-law with a higher exponent (fig 14a) emphasizing the critical role of false detection removal to

correctly capture the scaling behaviour of real landslides at small sizes.

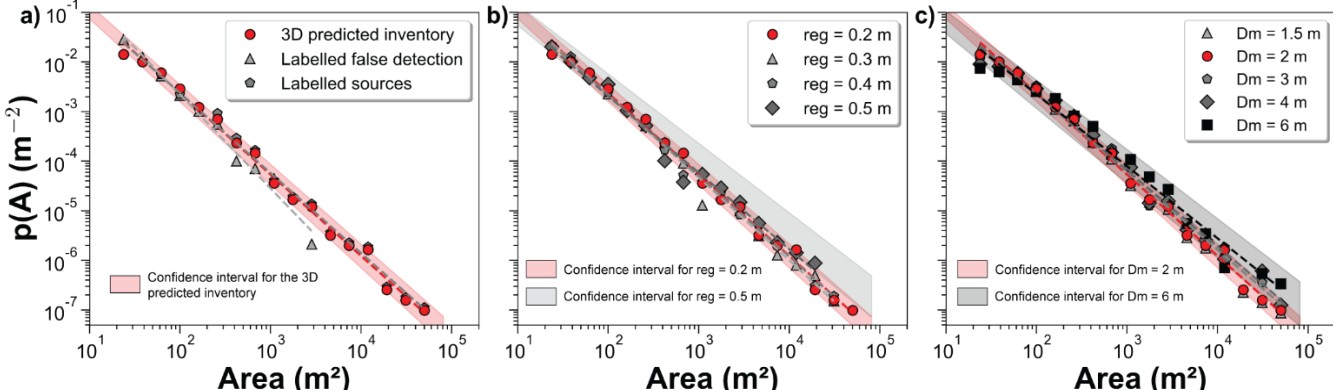

**Figure 14: Landslide source area distributions for different inventories. (a) Comparison between the 3D predicted, the labelled false detections and sources inventories. The two other comparisons are based on the resulting inventories with different value of (b) registration errors *reg* and (c) minimum segmentation distances *Dm*. All plots share the same y-axis. Values of the parameters used**
**for this study are colored in red.**

Several hypotheses, related to landslide mechanics or to landslide detection capabilities, have been put forward to explain the rollover behaviour for small landslide area. These include the transition to a cohesion-dominated regime reducing the likelihood of rupture (Frattini and Crosta, 2013; Jeandet et al., 2019; Stark and Guzzetti, 2009), a cohesion gradient with depth
(Frattini and Crosta, 2013), landslide amalgamation (Tanyas et al., 2019) or the under-detection of small landslides (Hovius et al., 1997; Stark and Hovius, 2001). Our segmentation approach tends to amalgamate landslides rather than over-segment large ones and cannot explain the lack of rollover. On the contrary, this would create or accentuate a rollover by suppressing small landslides through amalgamation. Changing $D_m$ does alter the scaling exponent of the p(A) (Fig. 14c), but no rollover is observed. The lack of rollover may also hint at a transition towards a different landsliding process, where rockfall dominates
for instance. However, core points in sources with slopes > 60° represent only 1.2 % of the source area, pointing at an extremely limited contribution of rockfall processes originating from near-vertical cliffs.

To evaluate the degree of under-detection as a function of landslide size, we can leverage the two inventories we have created. For this, we compute a completeness ratio as the number of detected sources in the 2D inventory (*corrected*, Tab.4) over the number detected in 3D, per range of source area. Fig.15 shows that the completeness ratio is around 0.25 for areas ~20-40 m²
and increases with landslide size up to 0.8-0.9 for sizes larger than 200-500 m² with one exception for which the number of 2D-sources is higher than the number of 3D-sources. We observe the same behaviour when the completeness ratio is calculated with the 3D labelled sources which is non-exhaustive but without false detections. As some very shallow landslides detected in 2D are not detected in 3D, we cannot consider the 3D inventory as complete for small sizes and the true completeness ratio may actually be slightly overestimated at very small sizes. Yet, the 3D inventory is however far more complete than the 2D

inventory. As such, our results demonstrate that in this study area, the deviation from the power-law trend around 100 m² is caused by a size-dependent under-detection of small landslides existing even when using high resolution imagery with a better resolving capability than our 3D-PcD workflow (6 m² vs 20 m²) (Hovius et al., 1997; Stark and Hovius, 2001). Because the rollover of the 2D inventory occurs at 20 m², the lower limit of size detection of the 3D inventory, we cannot formerly demonstrate that the rollover is strictly due to under-detection. However, we expect this size-dependent under-detection of

small landslides to be systematically present in other image-based landslide inventories, even if carefully hand-crafted (Tanyaş et al., 2019). Whether this effect systematically explains all the rollovers observed in past landslide inventories, or if other hypotheses such as a transition to a cohesion-dominated regime also contribute or are only expressed at even smaller scales, remains to be explored. In any case, the number of landslides potentially missed in previous studies can be important given the level of under-detection that we report for small sizes. The volume corresponding to under-detected small landslides

(A<100 m²) may actually not matter in terms of total volume produced by earthquake derived landsliding (2% of total volume). However, the presence or not of a rollover significantly matters in terms of hazard management (i.e. impact on the exposed population, infrastructure damage etc.) owing to the very large differences in the probability of small landslides.

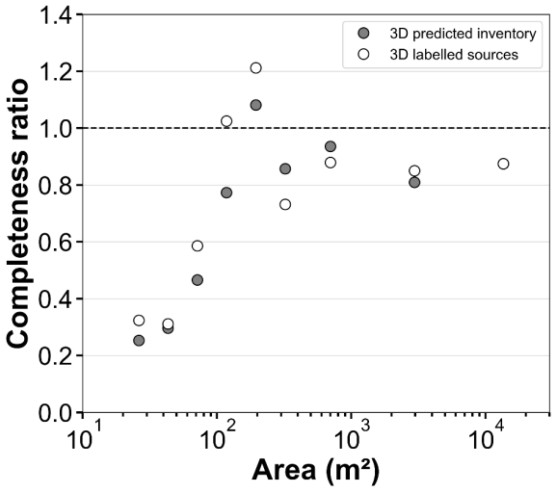

**Figure 15: Number of 2D corrected sources over number of 3D sources as a function of the source area. Assuming that the 3D**
**inventory is nearly complete, this measure represents the ratio of completeness of the 2D inventory.**

### 5.3.3. Distribution of landslide volume

    We present here one of the first co-seismic landslide volume distributions derived directly from 3D topographic data (Fig. 11b), rather than inferred from the combination of the landslide area distribution, based on 2D data, and an estimated V-A relationship. Our direct measurements show that the landslide volume distribution indeed obeys a power-law relationship for

$V > 50$ m³ with an exponent $e = -1.54 \pm 0.07$, consistent with the very broad range of exponents estimated in previous studies of $-1.0 \leq e \leq -1.9$ and $-1.5 \leq e \leq -1.9$ for rock and soil landslides, respectively (e.g., Brunetti et al., 2009; Malamud et al., 2004)). The sensitivity analysis to the workflow parameters (Fig. S12, Table S13) shows that the exponent $e$ will decrease with

*reg* as this parameter will censor progressively thinner landslides which are statistically the smallest ones. Contrary to the distribution of source area, the segmentation distance $D_m$ has little impact on *e*.

A deviation from the power-law trend occurs around 80 m³, with a rollover point around 20 m³ very close to the minimum volume that we can theoretically detect (~ 8 m³). It is thus difficult to evaluate if the deviation is a real feature, or an under-detection due to the lower depth limit that the 3D-PcD method can detect given the registration error (40 cm). Because no rollover is observed in *p(A)*, our data may hint at a different landsliding process resulting in smaller depth for small landslides, but the D-A relationship is too scattered to detect a different trend. As we suspect that our inventory may contain a small

fraction of very large rockfalls, the comparison with rockfall volume statistics is relevant. The probability distribution of rockfall volume generally obeys a power-law relationship with an exponent $e_R$ ranging from -1 to -2.2 (e.g., Malamud et al., 2004; Benjamin et al., 2020) without a rollover. If we restrict existing inventories to those having at least 500 rockfalls and the largest rockfall at least of 20 m³, the range of exponent $e_R$ narrows to -1.5 to -2 with a majority of inventories around -1.6 ± 0.1 (Benjamin et al., 2020), very close to our scaling exponent. Although, we do not expect rockfalls to be a dominant

mechanism in our database given the lack of very steep slopes and given that rupture mechanisms (e.g., fragmentation, sliding, slumping), rock heterogeneity and topographic constraints (e.g., hillslope size) are not expected to be similar (Dussauge et al., 2003), the consistency of the exponent we find is striking. This may suggest a much larger range of scales over which the volume of landslides, encompassing rockfalls in this definition, obeys a unique scaling behavior. Datasets specifically acquired to bridge the gap between large scale airborne LiDAR and terrestrial LiDAR are needed to get a better handle on the volume

distribution of landslides, critical information with respect to risk analysis and landslide erosion calculation.

### 5.3.4. Landslide depth and volume-area relationship

Our 3D-PcD approach opens the possibility to directly quantify the variations of landslide depth with size. We show that landslide mean depth slightly vary with landslide area on average by less than one order of magnitude for the given range of area of the 3D-predicted inventory. The same behaviour has been observed by Larsen et al. (2010) for soil landslide scars

suggesting that our landslide inventory may be relevant to shallow landsliding. This is consistent with the fact that 50% of the landslide thicknesses are lower than 1.6 m and that the landslide volume-area (V-A) scaling relationships obtained in this study are close to that of Massey et al. (2020) and Larsen et al. (2010) for soil landslides.

The sensitivity analyses to the workflow parameters show that the V-A exponent γ is not significantly affected by the variations of the *reg* values we explored. Respectively, γ varies from 1.17 ± 0.03 to 1.19 ± 0.03 (Fig S13, Table S14). It is also not

affected by the segmentation distance for $D_m < 6$ m, beyond which landslide amalgamation becomes significant and γ decreases to 1.1.

**6. Conclusion**

In this paper, we introduce a new workflow for semi-automated landslide source and deposit detection using 3D differencing based on high resolution topographic point cloud data. This method uses the M3C2 algorithm developed by Lague et al. (2013) for accurate change detection based on the 3D distance normal to the local surface. Potential landslide sources and deposits are segmented using a 3D connected component approach, and their volumes are computed by a vertical-M3C2. Spatially variable uncertainties on distance and volume are provided by the calculation and used in the workflow to evaluate if a change is statistically significant or not, for volume uncertainty estimation and to define a confidence metric per source or deposit (signal to noise ratio, *SNR*). Combined with the downslope distance to the closest deposit measured for each potential landslide source, the *SNR* is used to filter out false detections related to spatially correlated elevation errors such as intra-survey registration errors and ground classification errors in forests. We provide various tests and recipes to estimate the registration error and to choose the parameters of the M3C2 algorithm as functions of the point cloud density to ensure the lowest level of change detection, and the best resolution of the 3D map of change. Applied to a 5 km² area located in the Kaikoura region in New Zealand with pre- and post-earthquake LiDAR, we generate the first automatic inventory of landslide sources and deposits based on repeat 3D airborne LiDAR data. We show that:

- A minimum level of 3D change detection at 95% confidence of 0.40 m can be reached with airborne LiDAR data, which is largely set by the registration error. In our case, the limited quality of flight line alignment of the pre-EQ data was the dominant source of registration uncertainty. Because it operates on raw data, M3C2 accounts for characteristics such as point density and roughness that are not accounted for when working on DEMs, and results in more robust statistics when it comes to evaluate if a change is significant or not. 3D point cloud differencing is critical on steep slopes and allows a lower level of change detection compared to the traditional DoD.

- Complex correlated elevation errors may result in long wavelength low amplitude false detections of landslide sources with typical prevalence of 25 % in forest-free areas. They can be efficiently removed while preserving true landslides using a combination of *SNR* and the newly introduced closest deposit distance (*CDD*). Landslide detection in the dense evergreen forest of our study area is more challenging owing to low ground point density and ground classification errors.

- Considering 3D topographic change for landslide detection removes the amalgamation effect on the total landslide volume by directly measuring it in 3D rather than considering an *ad hoc* V-A relationship. Amalgamation in 3D is still an issue when exploring individual landslide area and volume statistics given the simplistic segmentation approach that we have used. However, our approach has the benefits of more systematically capturing small landslides than traditional approaches based on 2D imagery with manual landslide mapping.

- Landslides on surfaces with low or no vegetation cover are classically missed with 2D imagery processing due to the lack of texture or spectral change. In our study case, 72 % of landslide surface area is missed when considering a 2D inventory, corresponding to 54 % of the total volume determined with the 3D inventory. Landslide surface area is

missed due to both detection error (landslide fully missed) and delimitation error (uncertainty in outlines). Our method also shows the ability to detect subsidence related to slip failure propagation and the initiation or displacement of translational and rotational large landslides, which cannot be detected with 2D imagery.

- As this method provides direct 3D measurement, landslide geometric properties such as volume, area, depth and their distributions can be explored. Our results are broadly consistent with the V-A relationship scaling parameters
determined by Larsen et al. (2010) for soil landslides and Massey et al. (2020), with a scaling exponent of 1.17.

- No rollover is observed in the landslide area distribution down to 20 m², our conservative resolution limit, using the 3D landslide inventory. However, we demonstrate, a size-based under-detection in landslides mapped from repeat 2D images, which in turn results in a cut-off of the power-law behaviour of the 2D inventory, and contributes to the rollover occurring at 20 m². This result lends credit to the hypothesis that the rollover observed in landslide area
distributions generated from 2D images is entirely or partially related to an under-detection of small landslides (Stark and Hovius, 2001).

Our 3D processing workflow is a first step towards harnessing the full potential of repeated 3D high resolution topographic surveys to automatically create complete and accurate landslide inventories. However, high density LiDAR flights are not always available in landslide-prone regions for which a 2D image-based approach remains the most suited approach.
Nevertheless, we recommend to systematically perform a 3D-PcD approach where repeat LiDAR data exist. This is critically needed to improve landslide science and managing the cascade of hazards following large earthquakes or storm events, by automatically identifying landslide deposits, and subtle features such as subsidence developing around landslides missed in 2D inventories. Current bottlenecks to apply this workflow over larger scales, beyond the availability of high-quality 3D data itself, are the registration of pre- and post-EQ data when complex co-seismic deformation patterns occur, and limitations of
the segmentation method in high landslide density areas. While airborne LiDAR is best suited to vegetated environments and currently results in the best precision compared to aerial or spatial photogrammetry, the workflow operates for any kind of 3D data.

**Code availability**

The code producing the landslide inventory in this study is available as a jupyter notebook at https://github.com/Thomas-
Brd/3D_landslide_detection.

**Data availability**

The landslide source and deposit information supporting the findings of this paper can be found in Zenodo: https://doi.org/10.5281/zenodo.5113770.

**Supplement link**

The supplement related to this article is available online at: .

**Author contribution**

D.L. and T.B. designed the landslide detection workflow. T.B. and all co-authors participated to the discussion, writing and reviewing of this paper. T.B produced the figures and the code. P.S. wrote the MATLAB-based code to calculate the *CDD*.

**Competing interests**

The authors declare that they have no conflict of interest.

**Acknowledgements**

This project has received funding from the European Research Council (ERC) under the European Union's Horizon 2020 research and innovation programme (grant agreement no. 803721), from the Agence Nationale de la Recherche (grant no. ANR-14-CE33-0005), from the LiDAR topobathymetric platform of the Rennes University and from the Brittany region
(project LIDAREAU). We would like to thank Alex Densmore and David Milledge for the critical review that helped to improve this manuscript.

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
