# Peer review of "Beyond 2D landslide inventories and their rollover: synoptic 3D inventories and volume from repeat LiDAR data"

_Earth Surface Dynamics, 2020_

## Referee Comment (RC1) · Dave Milledge (Referee) · 29 Oct 2020

This is a really exciting paper that takes a careful and robust approach to surface differencing in order to identify topographic changes associated with co- and post-seismic processes. In this respect the paper provides a novel and useful contribution both in developing a methodology for identifying the changes and in documenting the changes themselves. It is also (if correct) a potentially very important paper! It argues that the current form of landslide size distributions are a result of observational error for all but the right tail. Given the importance of these findings it is essential that the paper is very clear about its landslide detection process and the implications of each processing step for landslide detection (in terms of location, size and shape). Fundamentally I remain unconvinced at the end of the paper that the findings on landslide scaling and

size distribution reflect those of real landslides in the study area.

The approach to surface differencing is rigorous, the writeup is clear (subject to a few minor comments that could be easily resolved). It is the step from change detection to landslide detection that I find problematic. Landslides are detected as connected patches of surface difference above a threshold. The validity of this detection method is not tested against any independent observations. I believe that the detection process is: 1) sensitive to topographic errors; and 2) prone to amalgamate some landslides and break up others. The resultant landslide inventory is then used to make claims about the size distribution and scaling properties of landslides, both of which are extremely sensitive to the errors detailed above. The conclusions of the paper are then built around these later size distribution and scaling findings, which I don't think the data are currently capable of supporting.

To me, the two main missing elements of the manuscript are: 1) a very clear definition of the range of processes and landforms that the authors would include within the category of 'landslide' (and therefore what set of processes their inventory represents); and 2) a detailed comparison of the landslide inventory generated here against independent observations within the study area, these are likely to include optical imagery but would ideally also include field investigation. These two elements are essential if the authors are to support their conclusions on landslide scaling and landslide size-frequency distributions.

The estimates of total landslide volume and of net mass loss from the study area are important contributions on their own. However, I think that more work is needed to post-process that volume estimate to account for counter-factual observations (such as deposition areas with no upslope erosion). It would also be very interesting to investigate the work associated with the event in terms of elevation reduction for the landslide mass.

Finally, examination of the properties of the landslides that have been identified would

be valuable both because the dataset should enable interesting insight and because disagreement between findings from this dataset and more traditional inventories may highlight uncertainties or errors not only in the traditional approaches but also in this new approach.

Please also note the supplement to this comment: https://esurf.copernicus.org/preprints/esurf-2020-73/esurf-2020-73-RC1-supplement.pdf

**Supplement:**

**Review of Bernard et al. 2020, *"Beyond 2D inventories : synoptic 3D landslide volume calculation from repeat LiDAR data"* for ESURF by David Milledge**

**Summary**

This is a really exciting paper that takes a careful and robust approach to surface differencing in order to identify topographic changes associated with co- and post-seismic processes. In this respect the paper provides a novel and useful contribution both in developing a methodology for identifying the changes and in documenting the changes themselves. It is also (if correct) a potentially very important paper! It argues that the current form of landslide size distributions are a result of observational error for all but the right tail. Given the importance of these findings it is essential that the paper is very clear about its landslide detection process and the implications of each processing step for landslide detection (in terms of location, size and shape). Fundamentally I remain unconvinced at the end of the paper that the findings on landslide scaling and size distribution reflect those of real landslides in the study area.

The approach to surface differencing is rigorous, the writeup is clear (subject to a few minor comments that could be easily resolved). It is the step from change detection to landslide detection that I find problematic. Landslides are detected as connected patches of surface difference above a threshold. The validity of this detection method is not tested against any independent observations. I believe that the detection process is: 1) sensitive to topographic errors; and 2) prone to amalgamate some landslides and break up others. The resultant landslide inventory is then used to make claims about the size distribution and scaling properties of landslides, both of which are extremely sensitive to the errors detailed above. The conclusions of the paper are then built around these later size distribution and scaling findings, which I don't think the data are currently capable of supporting.

To me, the two main missing elements of the manuscript are: 1) a very clear definition of the range of processes and landforms that the authors would include within the category of 'landslide' (and therefore what set of processes their inventory represents); and 2) a detailed comparison of the landslide inventory generated here against independent observations within the study area, these are likely to include optical imagery but would ideally also include field investigation.  These two elements are essential if the authors are to support their conclusions on landslide scaling and landslide size-frequency distributions.

The estimates of total landslide volume and of net mass loss from the study area are important contributions on their own. However, I think that more work is needed to post-process that volume estimate to account for counter-factual observations (such as deposition areas with no upslope erosion). It would also be very interesting to investigate the work associated with the event in terms of elevation reduction for the landslide mass.

Finally, examination of the properties of the landslides that have been identified would be valuable both because the dataset should enable interesting insight and because disagreement between findings from this dataset and more traditional inventories may highlight uncertainties or errors not only in the traditional approaches but also in this new approach.

**Major Comments**

**MC1) The paper needs to more clearly define: 1) landslides (i.e. what the inventory includes) and 2) what your inventory can and cannot be used for.**

Early on you introduce the idea that there are different landslide processes (L43, "process specific") but you don't follow this logic through into your results. Instead, your analysis may contain an implicit definition of landslides as all processes responsible for surface change that cannot be attributed to fluvial processes (L217-9).  You certainly need to make this definition of landslides explicit in your introduction.

The introduction needs a much clearer explanation for what you expect the inventory to be useful for. If it is for understanding landslide mechanics then it is essential that you make an effort to distinguish individual landslides on a mechanistic basis (see MC3 on amalgamation). If that is not an expected application of the inventory you should say so, otherwise there is a real risk it will be misapplied.

Appropriate uses of the inventory (e.g. volume estimation or landslide mechanics) depend not only on its purpose but also on entry criteria into, and distinctions within, the inventory. Non-fluvial surface change that might result from earthquake shaking includes: tree-throw, ravel, rockfalls and slides, slow earthflows, rapid soil slides and debris flows. The processes responsible for these surface changes differ from one another to varying degrees. If there is no distinction between them this precludes the inventory's use in analysis of landslide mechanics and therefore

prediction. It allows comment on correlation e.g. of volumes and areas of change, but makes it extremely difficult to make any inference about causation. It also opens the work to the criticism that the bulk statistics mask important differences in behaviour between processes. For example: the differing size distributions for rockfalls (where others have reported no detectable rollover) and landslides (where there usually is).

**MC2) Elevation errors need to be better quantified and more thoroughly discussed**

The manuscript needs a more thorough treatment of errors in the topographic data dealing with both: 1) the properties of the elevation errors that you have identified (e.g. spatial pattern, wavelength, covariation with landscape properties); and 2) the possible sources of error.

**RE error properties:** The amplitude and correlation length of elevation uncertainty from different sources and how they interact to generate a 2D elevation error field with a particular amplitude and wavelength is really important for this particular application, where landslides are identified by thresholding then segmenting differences. The error appears to have a fairly long wavelength in many areas (tens to hundreds of metres). It also appears to have some aspect dependence. The spatial correlation of these errors is important because you assume uniform isotropic registration errors.

**RE error sources:** It isn't clear to me what you mean by imperfect alignment, nor ICP related errors (L204-5). Identifying errors on Fig 3 and hypothesising the sources from which they derive are useful but need to be discussed in the manuscript as well. You recognise the presence of "internal flight line height mismatch" and indicate that it results in "large scale low amplitude topographic change" (L418-22). They should be introduced earlier in the article with a more complete explanation of what they are and how you found them.

**Error consequences:** It would be useful to say something about the implications of the topographic errors. I can see two implications: **First**, incorrectly assuming uniform isotropic errors will result in a confidence interval for identifying significant geomorphic change that is too strict on some slopes (e.g. some aspects) and not sufficiently so on others. This in turn will lead to false negative change detection on some slopes and false positives on others. **Second**, change detection false positives will result in false identification of landslide objects or false representation of their geometry. False negatives are equally problematic since they could result in not only changes to landslide geometry but also cluster breakup (biasing the size distribution). These problems are illustrated in Fig 3 where a number of error patches would be identified as landslides by the detection algorithm if these areas were not assumed to be stable. If such false positives exist here it is they likely also exist elsewhere. It is essential that you quantify their impact on your findings.

**Suggested additional analysis:** You could apply your landslide detection method to only the pre-defined 'stable areas' and generate landslide geometries. These geometries and their scaling relationships might indicate the impact of error on your findings (particularly: number and area density, pdfs, and scaling relationships of artefact landslides). If the results are similar to your findings for 'non-stable areas' it would be very difficult to argue that the data support your claims with any certainty (the same results could have been generated purely from topographic errors in the absence of any landslides).

**MC3) You recognise that segmentation results in amalgamation but don't quantify its extent or impact**

Severe amalgamation can result from automated segmentation of a thresholded classifier. In the landslide maps that you show here (e.g. Fig 2), amalgamation appears a severe problem for the largest landslides. The argument that it can be solved by tuning Dm (as you suggest on L426-7) is unconvincing since two separate landslides can be within millimetres of one another but have different failure mechanisms. You later say that you "cannot resolve the amalgamation" (L511) and that it "is still a potential issue" (L558). I would argue that it is not potentially but certainly an issue. Your figures show that amalgamation is present (perhaps pervasive) in your inventory but its extent or impact is not quantified. The total volume is insensitive to amalgamation but your landslide pdfs and scaling relationships are not.

**Amalgamation makes it difficult to use the inventory to understand landslide mechanics and therefore susceptibility or hazard.** If the inventory is to be useful in understanding landslide mechanics then it is essential that you make an effort to distinguish individual landslides on a mechanistic basis. In the extreme case, an individual landslide where subsets of the failed material moved in opposite directions (e.g. Fig 2) is clearly problematic for the mechanics of that failure.

**Amalgamation introduces bias towards large landslides in size distributions.** Power law exponents and even the appropriateness of any power law can be sensitive to landslide amalgamation. Observations such as that on L569-70

that "largest and deepest landslides deviate significantly from this trend" become questionable since these are the landslides most likely to be the result of amalgamation.

**Suggested additional analysis:** If you are to retain your findings on landslide scaling and size distributions, then you must examine the potential impact of amalgamation on these findings. There are methods available that quantify amalgamation in landslide mapping. You should investigate these. You should preferably also address this amalgamation, either by removing affected landslides or ideally, by decomposing amalgamated landslides into their individual failures.

**MC4) Topographic errors propagate through segmentation to introduce a bias towards small landslides. Existing experiments to quantify this bias are insufficient.**

You say on L559-60 that "our approach has the benefits of more systematically capturing small landslides than traditional approaches". However, this is one of my main problems with the paper given potential propagation of topographic errors through thresholding and segmentation. You show (in SI) that: 1) in the absence of any real topographic change your detection algorithm generates artefact landslides of 1-20 m$^2$ purely due to spatially uncorrelated topographic noise; 2) this noise generates many more small than large landslides; 3) in this experiment artefacts >20 m$^2$ were extremely rare. However, this does not demonstrate that predicted landslide size is insensitive to longer wavelength topographic errors (known to be present in the data); nor even to short wavelength noise in the presence of longer wavelength surface differences (e.g. real landslides).

**First,** even without any real topographic change (i.e. no real landslides), the size distribution of erroneous landslide-like clusters will depend on the spatial correlation length of the difference errors, which in turn depends on the correlation lengths of the errors in the surfaces being differenced. Longer error wavelengths will enable the generation of larger error clusters. Your figures show that topographic errors are clearly not uncorrelated and you recognise this yourselves (L418-22); nor do the errors appear to have a single characteristic wavelength. This is a hard problem but one that you must deal with if you are to convince the reader that the landslide inventory you have generated is not hopelessly biased by these landslide-like artefacts. **Second,** the problem is not only that clusters of erroneous negative surface difference due to roughness (or other errors) can create artefacts that appear to be landslides, but also that clusters of erroneous positive surface difference are collocated with real topographic change (e.g. due to a landslide); these can interfere negatively with real changes reducing the surface difference below the threshold for detection and breaking a single landslide into multiple patches.

**Oversampling of small landslides is important because** it undermines your most surprising and high impact claim: that rollover reported in previous inventories is due to under detection (L573-4). I am not currently convinced by this claim because you do not exclude the possibility that the lack of a rollover is solely due to detection errors. You need to quantify these detection biases before you can make these claims.

**Suggested additional analysis:** A "more advanced segmentation" (L431-3) may be out of scope for this paper. However, an indication of the impact of the simple segmentation on object based classification skill is an essential requirement of this paper if it is to retain the current approach to identifying discrete landslides. Two possible avenues could be followed to provide such an indication. **First,** your analysis of topographic changes on the stable surfaces (Fig 3) would allow you to perform the same analysis that you have performed in the SI but using the pre- and post-EQ surfaces for the stable zones identified in Fig 3. This would enable you to identify the size distribution of artefacts that can be generated from topographic errors with a spatial correlation length closer to that for the unstable parts of the study area. This though still does not account for the possibility that the landslide erosion signal itself is altered by the noise (e.g. by disconnecting clusters). **Second,** you could collect a landslide check dataset using independent observations. This might take the form of an entirely independent inventory but should certainly also involve cross-checking to confirm the existence and characteristics (e.g. area, shape, depth) of your predictions.

**MC5) Findings that differ from previous work**

There are a number of unusual findings that are worthy of comment because they are some of your most interesting and potentially important findings. It is essential though that each is carefully examined and that the critique that it might have arisen due to methodological errors is dealt with head on.

**First**, it is not unusual to identify more sources than deposits due to amalgamation of landslide deposits. However, it is unusual for deposit areas to be smaller than source areas (L281-2), and for deposit depths to be thicker than source depths (L265-70 and L460). These result should be compared with results from previous studies.

**Second**, the estimate that Massey et al. (2020) "potentially missed around 169,000" landslides (L491-2), i.e. 92% of the landslides is surprising because of its magnitude. However, to be convincing you need to demonstrate that the landslides that you have detected genuinely are landslides.

**Third**, you identify areas of deposition where there is no upslope erosion (L256 and Fig 4). I don't think these can be real deposition zones but instead must be a consequence of incorrect landslide detection. Their spatial extent and depth distribution would be a useful indication of the precision of the technique.

**Fourth**, it is extremely unusual that locations classed as vegetated in post-event optical imagery but identified as a landslide by another technique are considered by the authors to be genuine landslides (L347-8). Instead, the presence of vegetation at the location strongly suggests a false positive.

**Fifth**, you find no statistically significant difference between total landslide erosion and deposition (L459), does that mean that there was no significant mass loss due to landsliding from the study area? If so, the earthquake's role has been almost exclusively to break up the material and redistribute it within the study area. This seems to differ from the findings of previous studies on the influence of coseismic landslides on earthquake mass balance. It would be very interesting to see the elevation change of the mass (perhaps as elevation pdfs for scars and deposits). This might be a valuable contribution on the instantaneous impact of coseismic landslides on mass balance.

**MC6) Areas should be calculated surface parallel**

I agree that variation in surface normal orientation could introduce bias in the volume estimate (L238-9) and that volume should be calculated normal to the plane on which landslide area is calculated. However, it isn't clear to me that a horizontal plane is most appropriate for area calculation. This will severely underestimate landslide area on the very steep slopes typical of rockfall initiation. Surface parallel area calculations would be more appropriate and orthogonal measurements for volume calculation would still be possible. I don't find the argument based on retaining consistency with other studies (L279-80) convincing. Your inventory appears to contain a significant number of rockfalls and their area and volume are often calculated in a vertical rather than horizontal reference frame. If you do not alter the reference frame you should certainly report the distribution of surface slopes of your landslides and comment on the effect of vertically projected area given this distribution.

**MC7) Reactivation is actually under-detection on bare or sparsely vegetated slopes**

I am not convinced that the set of landslides that you are describing are really re-activated landslides (L344-5) but I'm also not convinced that reactivation is really the relevant issue. I think you demonstrate that you are able to detect landslides in vegetation sparse or vegetation free areas that have generally suffered serious underreporting. Some of these areas may be bare due to previous landsliding, others may not but the key point is that inventories have underestimated landslide density due to underdetection in these areas.

However, if you do want to focus specifically on reactivation then you need to clearly define reactivated landslides and detail how you classify them as such. For example: Does a landslide need to reoccur within the footprint of an existing landslide to be a reactivation or is retrogression included within reactivation? Within what time window following the first landslide must the second occur for it to be considered reactivation?

**MC8) Landslide object detection needs to be tested against an independent dataset**

The findings in this paper depend critically on the skill with which the proposed method can classify landslide scars and deposits. Thus it is essential that the paper reports testing results that quantify this object based classification skill. At present, "orthophotos were used to visually validate" the classifier (L115-6) but without reporting results of this analysis. I think it is essential that you explicitly explain your sampling and mapping strategy for landslide detection from orthophotos in the methods. You should then include a section in the results where you compare your orthophoto based mapping to the surface differencing approach. However, it is not enough to simply say the orthophoto mapping did not identify landslides that were identified by the surface differencing. You should then go to a carefully chosen (e.g. stratified random) subset of the landslides detected by each method that were not detected by the other (i.e. surface differencing but not ortho-photo mapping and vice versa) to establish as far as possible which of the two methods was in error and why. While finding and mapping thousands of landslides might be time-consuming (L255), confirming their existence and characteristics (e.g. area, shape, depth) would not.

**Minor Comments**

L32: You need to define landslides early in the paper, it will influence interpretation of your later findings.

L34: "spatial distribution, total volume…"? explaining why these are useful would be helpful because this sets the motivation for everything that follows. Landslide mapping has different requirements depending on its purpose. The purposes you choose here set the metrics against which your own mapping should be evaluated.

L35: "associated direct and secondary hazards" What is the connection between mapping, the first three elements of the list and these later two? It would help to make this explicit. As above, this relates to the purpose of the mapping.

L43: "process specific": here you recognise the importance of different landslide processes. This is an important point that needs to be reflected in your own analysis (see MC1).

L110&113: It isn't clear what statistic is being referred to here as a metric for vertical accuracy. RMSE? If so it would be better to quote mean error and SDE, giving an indication of the contributions of precision and bias.

L114: "using the classification provided": More detail is needed on the method used to classify ground points.

L115-6: "orthophotos were used to visually validate": The results of this validation are missing from the paper.

L135: "core point": What is the sensitivity of the results to varying core point spacing? The orientation and spacing of the grid of core points should also be described and justified either here or in section 3.2.

L136: "first dataset": Which is the first dataset? Are your results sensitive to this choice, both in terms of depth area scaling and disagreement in 1) change surfaces and 2) maps of significant positive and negative change?

L137-8: This is not clear: do you mean that you calculate the centroid of each point cloud in 3D then take the magnitude of the 3D vector that connects these two points; or that (for each point cloud) you take the arithmetic mean of differences from the reference plane (defined by D) in a direction normal to that plane? In either case you are performing a spatial averaging at length scale d/2 assuming a uniform kernel. First, is it problematic to perform averaging over length scales larger than the core point spacing? Second does it make sense to assume equal weight in the average with plane parallel radial distance from the core point, or should some form of inverse distance weighted average be used? I would have assumed a weighted average was more appropriate but it would be useful for you to explain why an unweighted mean is more appropriate.

L139: "if not intercept is found…": This is not clear to me. Do you mean 'if the cylinder does not intersect any points in the second surface? Why would this happen? Does this only occur at the boundaries of the point clouds? How do these intersection failures influence the surface differencing and how do you report them in your later analysis?

L140: "provides uncertainty": what is the basis / justification for the uncertainty estimate taking this particular form? It looks familiar as it has some similarities to a confidence interval but also some differences. This threshold is important to explain and justify in detail because it is used to threshold discrete landslides in the following analysis. Why threshold at 95% confidence? What is the impact on your findings (total volumes and scaling relationships) of thresholding at a difference CI (e.g. 99 or 90% confidence)?

L145: "detrended roughness": I think it would be useful to add that these are standard deviations if this is true.

L145: "reg" is quantified using the standard deviation of differences between the surfaces. I think it would be helpful here and elsewhere to use similar notation for the registration error to the other errors being examined here. Why are the local terms converted to standard errors but reg is left as a standard deviation? Finally, the length scale over which reg was calculated would seem to be important here.

L164: "at least 5 data points": Does removing these points from your analysis alter your results on pdfs and scaling?

L165: "not deemed interesting": I don't think that this is the right phrase, can you rephrase? What was the impact on your results of applying d=10 m throughout?

L173: "may result in…": How do you identify when this has happened? What is the objective that you are optimising?

L187: "larger than 1 m": perhaps give a range, >1 could be very large but really it is on the order of 1-2 m.

L187: "rasterizing the dataset": at what resolution?

L196: "standard deviation…" These stable areas would be an excellent test of your landslide detection method, indicating the scaling relationships, size distributions, and total volumes generated by artefacts alone.

L199-200: "The registration error…": this definition should come earlier. It is important for interpreting eqn 2.

L200-1: "manually selected": This doesn't seem consistent with L191. Was a threshold of change selected manually then all areas with changes smaller than threshold included? Or, did you take a subset of pixels from the areas with changes smaller than the threshold so that only large patches were considered?

L202-4: "The standard deviation": This is great. Clear and useful!

L215: 15-20% it might be useful to show the location of these points on one of your maps.

L217-9: "all geomorphic processes…": This assumes that all non-fluvial changes are landslides (see MC1).

L221-234: "Landslide source and deposit segmentation": This process is much more clearly articulated in the SI. When I finished this paragraph I was totally confused but when I read the SI it became clear. The SI text on this analysis is really important and needs to be included within the main manuscript.

L225: "compact": I don't understand what compact means in this context

L225: More detail on how this method works is essential here. This step is a key component of your method and it is very important that the algorithm is given the same clear treatment that others have been in previous sections. I guess that Dm sets the distance between points within which two points are considered connected. This is a tuning parameter and it is not clear to me what objective you are seeking to optimise during tuning.

L226: "amalgamation effect and the over-segmentation": These two effects and your method for quantifying them need to be explained. I don't think this can be included only in SI.

L228: "small artefacts". Do you mean that there are a small number or that they are small in magnitude?

L229: Why would it be the smallest landslides that are particularly affected by "these artefacts"? I had assumed that these were due to things like inclusion of points on either side of ridges.

L230: "minimum number of points": You haven't explained what this parameter controls. Is it the minimum number of connected components required for a patch to be retained (i.e. minimum landslide size)?

L231: I must have misunderstood Np, it does not follow that all detected changes will be artefacts if Np simply removes landslides smaller than some minimum size. On reading the SI I now understand, this is because you aren't comparing before and after point clouds. You explain this well in the SI, but need to bring some of that text in here.

L231-2: Reference to pre- and post- earthquake data here is confusing unless read in conjunction with the SI.

L233: "artefacts": What are artefacts? How do you identify them and why do they become smaller/less frequent/less important for larger areas? What does it mean that they are limited (e.g. they make up less than x% of the landslide objects)? The SI text on this analysis is really important point that needs to be included within the main manuscript.

L238-9: I agree that volume should be calculated normal to the plane on which landslide area is calculated. But disagree that a horizontal reference frame is most appropriate. This will underestimate landslide area (severely on the very steep slopes typical of rockfall initiation). Surface parallel calculations seem more appropriate (see MC6).

L245-7: "specific to each landslide": This potentially presents a nice opportunity to check your method by comparing source and deposit volumes for individual landslides over your study area.

L252-3: "Most of the detected changes on hillslopes correspond to…" How do you know this? Is it based on your definition that all non-fluvial change is due to mass movement? Or, are you drawing on additional information to interpret the change map? Ideally you would have some independent check data.

L254: "previously unstable bare rock": It is useful to know that many are located on bare rock but how do you know that it was previously unstable and what do you mean by previously (this implies a timescale)?

L255: "Their large number illustrates how difficult": Why would their large number make them difficult to manually extract? Perhaps time-consuming rather than difficult. An inventory with 1431 sources is a very modest sample size by the standards of modern landslide inventories.

L256: "deposit areas" Some of these areas do not have any upslope erosion. How do you explain this? See MC5

L259: "stable areas": This argument seems somewhat circular since stable areas were identified from surface differencing rather than independent information.

L259: What do you mean by "artificial changes" here?

L286: It is not clear what "substantial" means in this context.

L291: A rollover is considered characteristic of landslide, but not of rockfall, distributions I think. As I understand it your inventory includes both. It would be useful to comment on this amalgamation of processes within a single inventory and what it implies for interpretation of the landslide distribution. See MC1.

L292-3: "if we reduce the minimum landslide size to…" I don't think that you can make this statement if you have excluded landslides smaller than 20 m$^2$ for good reason. Your methods text makes a compelling argument that surface differencing errors will generate landslide-like artefacts that will distort the distribution and thus should be censored.

L305-7: This is connected to my earlier concern about planimetric areas and suggests a need either to address the problem as you suggest here or to include a more detailed explanation for when the problem presents itself, what fraction of the landslides in your inventory might suffer from it. See MC6.

L308: Why frame your analysis in terms of volume-area relationships rather than depth-area? Since volume is the product of depth and area the x and y variables are correlated by definition. Others e.g. Larsen have examined depth-area scaling and the translation from one to the other is straightforward. But using depth de-trends the y-axis making it easier to see the scatter in the relationship, which is a major advantage.

L315: It is not surprising that your depth area scaling relationship is gentler than that of Larsen since you censor core points with difference < 0.33, making it impossible for small landslides to also be shallow.

L325: An analysis of the uncertainty associated with total volume estimates would be very valuable. Your study is unusual in not only being able to resolve scar volumes but also deposit volumes, thus you can calculate a net volume removed from the hillslopes. This value is of considerable interest and comparing it to the estimates that have been made elsewhere using different approaches seems worthwhile. You should however deal with your own uncertainties as well and within those uncertainties should be included those areas of deposition that have no landslide source upslope.

L344-5: A clear definition of reactivated landslides is needed here. See MC7.

L347: In many areas "bare rock areas" surfaces do not necessarily indicate recent landsliding (depending on how recent is defined). They may in this landscape though, so it would be useful to say so and evidence the claim.

L349: "following classical approaches": This is not a convincing justification for the method. It argues for the method's validity because others have previously adopted it without any further indication of why it might be valid here.

Section 5.1.1: This section is excellent, this alone is a major contribution!

L384: I don't think you do explain the subsampling exercise in section 3.2 it is currently in the SI as I understand it.

L431-3: "more advanced segmentation": an indication of the classification skill of the simple segmentation of object based classification skill is an essential requirement of this paper.

L460: "deposits form more concentrated and thicker patches": This result is surprising and should be compared with expectations from other studies on landslide runout.

L461-2: I would have expected "very shallow rockfalls" to result in even shallower rockfall deposits because they are initially small in volume and spread out over a wide area during deposition. Reference to rockfalls here increases my concern around use of horizontal (i.e. planform) area calculations.

L465: It is not clear what you mean by "landslide analysis". I suggest cutting this phrase, I don't think it is necessary.

L466: Whether the right tail is a "power law" or not is debated. See Medwedeff et al. (2020) among others.

L473-4: Power law scaling can be sensitive to amalgamation errors such as those in your inventory. As a result I struggle to know how much confidence I can place in these values. See MC3.

L488-90: "rollover … is likely caused by an under detection of small landslides": This is a very important claim with significant implications for our understanding of landslide mechanics. But it depends on both: 1) your findings being robust to error in segmentation; and 2) the inventories that you are discussing having comparable definitions of landslides. Given that the landslide detection method has not been tested and that there are reasons to expect that the method introduces considerable bias in the size distribution I am not convinced that the claim is true. The lack of a rollover in your dataset might instead be due to detection errors.

L499-500: "most of our inventory is relevant to shallow landsliding": shallow landsliding is usually defined as landslides initiating above the soil-bedrock interface, and thus distinct from rockfall. Can you make this distinction in your inventory?

L511: "cannot resolve the amalgamation…": The problem of amalgamation needs a more detailed treatment. Total volume is insensitive to amalgamation but landslide size distributions and scaling relationships are not. See MC3.

L535: "a much lower detection level than optical methods": This phrase is unclear, can you rephrase?

L551: "95% confidence of 0.34 m": This is quoted here as spatially invariant but the method captures spatial variability. The value here doesn't account for the impact of local roughness on the confidence intervals.

L558: "Amalgamation in 3D is still a potential issue": Amalgamation needs a more detailed treatment, given its potential impact on your findings it cannot simply be flagged as a potential issue.

L561: "reactivation" needs to be much more clearly defined. However, your argument doesn't really depend on reactivation but on landslide detection in vegetation sparse or vegetation free areas. I suggest restating in these terms.

L566: "V-A relationship": depth-area should be examined as well as or instead of V-A because it removes spurious correlation with area. All three geometric properties are likely to be very severely affected by detection errors.

L569-70: "largest and deepest landslides deviate significantly": these slides are also most likely to suffer amalgamation.

**Figures**

Figs: 3,4&6: Though they are very well presented I find the 3D plots (Fig. 3, 4 and 6) very difficult to interpret. I feel strongly that all these Figures should be presented in 2D map view instead of as oblique images.

Fig 4: The colourmap in Fig4a does not match description in the caption or in the text. It appears to show a single colour for erosion and another for deposition. It also appears to apply a filter to areas with non-significant change. Areas of significant change are shown in Fig4b so the mask is not necessary here. By masking non-significant changes you lose the opportunity to compare the magnitude of change before significance tested. The colourbars in both Fig4a and c are a good length but contain little usable information. As I see it there is no change in colour for erosion/deposition >1 m in Fig 4a and 5 m in Fig 4c. In addition, the scaling is asymmetric in 4c making it difficult to compare erosion and deposition.

Fig 5: The 3D minimum volume line needs explaining in the caption. I would have expected this line to be oriented parallel to the depth contours since minimum volume for a given area is set by minimum detectable depth. If so the minimum detectable volume might explain the sharp lower boundary to the volume area point cloud.

Fig 6: The inventory of Massey is useful additional information but scar outlines would be more useful than controids.

Fig 6b: It would be useful to identify the largest four or five landslides in the right hand panel of Fig 6b either by giving them individual outlines or different colours this would give an indication of the degree of amalgamation.

---

## Referee Comment (RC2) · Alexander Densmore (Referee) · 10 Nov 2020

This is an exciting manuscript that reports on a promising way forward in detection and analysis of landslide inventories. The authors use the M3C2 point cloud analysis approach, previously developed by Lague and colleagues, to detect change in pre- and post-earthquake Lidar point clouds from the Kaikora earthquake area in New Zealand. This is absolutely the right thing to be doing, and the authors are clear about how their approach gets around some vexing issues with current best practice (which is to map using 2d imagery and to use volume-area scaling to get at landslide volumes). The topic is fully appropriate to the journal and I anticipate that this work will attract a good deal of attention from the journal readership.

I do have some questions and suggestions for the authors to consider before the manuscript is published. Many of these are fairly minor and are detailed in the attached PDF; I will not repeat those here. But there are a couple of wider issues that are related to the clarity of what the authors have done, and in part to the division of material between the manuscript and the supplemental information. In brief, I don't think it's possible to follow the authors' approach from the manuscript alone, and there are key parts of their analysis that can only be understood by going to the SI. I don't think that's right for a manuscript that purports to document a new methodological approach. These issues come under three headings (but please see the PDF for more detailed comments):

First, because the authors are documenting a new approach for mapping landslides, I would expect to see some quantitative comparison of their results with a landslide inventory or inventories prepared in the more traditional way. The authors show some amalgamated results (Fig 5, Table 2), but I think it would be useful to show a more systematic comparison with the Massey et al. results. Given that there are only 27 landslides in the Massey et al. inventory within the study area, this should be pretty straightforward - but I think it is important to demonstrate the extent to which their approach can match (or not) landslides detected by the alternative approach, as well as the additional landslides that they claim to be able to map. The only place I could see the Massey et al. landslides was as barely-visible centroids on a perspective view of the study area in Fig. 6.

Second, and somewhat related, the text is very unclear on how individual landslide sources and deposits are segmented and identified. Because of this, all of the resulting statistics of the area and volume distributions are uncertain in the reader's mind. This is more clearly explained in the SI... but again I don't think it's fair to require the reader to go to the SI to understand a methodological advance that is being proposed. I deliberately read the manuscript without going to the SI to see if I could follow it, and there are places (e.g., section 3.3.3) that are very difficult to understand and don't

really address what has been done. I'd really recommend that the authors review the balance between the text and SI and try to flesh out the explanations in the main text. The details of the sensitivity analysis (e.g., to the distance threshold D_m) can be left for the SI, for sure.

Third, I found the authors' use of the term 'reactivation' potentially confusing, and I'd suggest that they choose a different way to express this. Assessing reactivation (e.g., of coseismic landslides in post-earthquake storms) is definitely a big issue in multi-temporal landslide inventory analysis, and a few different approaches have been put forward, none of them very satisfying. Reactivation implies some renewed landslide activity that partly or wholly overlaps with a landslide from a previous epoch - e.g. further erosion within a pre-existing scar, or headward/lateral progression of the scar edges, or erosion within a scar coupled with deposition on a pre-existing deposit... But that's not what the authors are actually talking about here, because they only have two point clouds spanning one epoch, and there's no independent dataset of pre-existing landslides. A better way of framing this part of the analysis might be around the following question: are there landslides that would not have been recognised using the classical approach, either because they occurred on bare bedrock or because the vegetation contrast was too low, but that they can see with their method? That's an important question - but it's not the same as reactivation.

Finally, a fairly minor point: I agree that the approach outlined by the authors is the way forward and that, where suitable Lidar data are available, this should be further pursued and developed. But I think it's equally fair to recognise that (1) suitably accurate and high-resolution Lidar data aren't always available, and (2) there are problems and applications for which this approach simply isn't (yet) feasible. For example, for multi-temporal inventory creation over the full landslide-affected area, where the goal is understanding patterns of landslide occurrence and hazard but volume estimation is a secondary concern, then a traditional 2d image-based approach might be fine. New Zealand and a few other countries can fly repeat high-density Lidar surveys; this capacity doesn't (yet) exist in most landslide-prone regions of the world. I think it would be fair of the authors to acknowledge this - it doesn't detract from their analysis but perhaps places it into a better wider context.

To summarise, this is a really exciting piece of work. Once the authors have dealt with these issues, then I look forward to seeing this published.

Please also note the supplement to this comment:
https://esurf.copernicus.org/preprints/esurf-2020-73/esurf-2020-73-RC2-supplement.pdf

**Supplement:**

[revised manuscript text omitted]

---

## Author Comment (AC1) · 24 Feb 2021

We first thank David Milledge and Alex Densmore for the positive reviews and for the relevant and detailed remarks they raised that greatly helped to improve the manuscript. Following their comments, the previous manuscript has been subject to important revisions. In essence, our results are not changed, but we believe the MS now makes a more compelling and quantitative case for our previous conclusions. We propose here to give a general overview of the modifications brought to the manuscript but please see the rest of the document for a detailed answer by comment. 1. We clarified what we consider through the generic term of "landslide" which corresponds to the spatially coherent changes of several decimeters detected by our method on hillslopes. This definition has been added in the introduction of the manuscript. We have added a dedicated section that detail the different processes we observed in the discussion (section 5.1.2, L595). 2. Although it would be interesting to compare our landslide inventory to field observations, such data are not available for the studied area. However, we added a detailed comparison with a landslide inventory manually mapped from pre- and post-EQ aerial imageries (section 3.4 and 4.2, fig. 7). Given the impossibility of having the landslide extent mapped by Massey et al. (2020), due to the author's lack of response to our numerous requests, Dimitri Lague carried out a manual mapping. We used this new manually mapped landslide inventory to: (1) compare the two methods in terms of number and area of mapped landslide, (2) detail the origin of under-detection of landslides from both methods and (3) discuss the impact of landslide under-detection on landslide volume estimation using traditional approach. Following this analysis, the discussion now extensively discusses the origin of a rollover in the pdf(Area), and we provide a new analysis showing a systematic size dependent under-detection of landslides in the 2D inventory (fig. 12). This section also replaces the section about reactivated landslides present in the previous version of the manuscript for which we agree that the term "reactivation" was misused. 3. To further improve our treatment of potential errors and false detection, we have added two additional tests to our workflow and have added many elements on the discussion regarding current limits of the workflow. First, we explore in details the source of errors coming from misclassification of the original point clouds and imperfect flight line alignments. We therefore reprocessed the pre-EQ point cloud to remove incorrectly classified points and quantified residual errors for each flight line alignments (section 3.3.1). We now integrate the residual error from flight line misalignment in the estimation of the registration error. Second, we added a new analysis to deal with false detections after segmentation (resulting in a new step in the workflow, section 3.3.4). We define a new confidence metric for each segmented cluster (source or deposit) called mean signal-to-noise ratio (SNR). An optimal value of SNR, is defined to minimize the proportion of potential artefacts (new fig. 6 and 5b). This results in more than 746 sources and 748 deposits of low confidence removed from our inventory. The final inventory now contains 524 sources and 304 deposits of very high confidence (in the first version of the MS, there were 1431 sources and 853 deposits). 4. We have added further analysis of the impact of the main parameters of our workflow (registration, segmentation and SNR filtering) in the discussion, appendix and supplementary material. Our results, and in particular the lack of rollover in pdf(A) are not significantly affected by these parameters. 5. We have tested segmentation approaches based on density based clustering (DBSCAN and derivatives) which are considered state of the art for 3D rockfall segmentation. We demonstrate that these approaches (1) do not perform better than the segmentation approach we use and do not generate significantly different pdf(A); (2) may have the tendency to oversegment intermediate landslides into small ones or remove border points of large landslides; (3) are sometime much longer to run (45 min vs 3 sec) and have parameters which are not intuitive to set. Because the method part of the article was already dense, the result of this comparison exercise are in the supplementary (supplementary material S5), along with an explanation as to why density based clustering is actually not suitable to segment large landslides in our workflow. As in the first version of the article, we clearly state in the article that our segmentation has limitations, but we are not aware of other approaches that would perform better in segmenting complex 3D objects covering 3 to 4 order in size. 6. The balance between the supplementary materials and the manuscript has been revised, and many figures have been updated following the reviewers recommendation. 7. The title and abstract have been updated to feature the notion of rollover in the pdf(A) of 2D inventories. It reflects the additional data and analysis we provide that clearly demonstrate, at least in our study case, that the rollover in the pdf(A) observed in our 2D inventory but not in the 3D one, is due to a size-dependent under detection in 2D. We believe this is of critical importance for the geomorphic community working on landslide science.

We wish both reviewers a pleasant reading and thank them in advance for the time they may spent in evaluating this new version of the manuscript.

Please also note the supplement to this comment:

https://esurf.copernicus.org/preprints/esurf-2020-73/esurf-2020-73-AC1-supplement.pdf

**Supplement:**

**Author's response to referee comments of Bernard et al. 2020 by David Milledge.**

Thomas G. Bernard, Dimitri Lague, Philippe Steer

Univ Rennes, CNRS, Géosciences Rennes - UMR 6118, 35000, Rennes, France

**General response**

We first thank David Milledge and Alex Densmore for the positive reviews and for the relevant and detailed remarks they raised that greatly helped to improve the manuscript. Following their comments, the previous manuscript has been subject to important revisions. In essence, our results are not changed, but we believe the MS now makes a more compelling and quantitative case for our previous conclusions. We propose here to give a general overview of the modifications brought to the manuscript but please see the rest of the document for a detailed answer by comment.

1. We clarified what we consider through the generic term of "landslide" which corresponds to the spatially coherent changes of several decimeters detected by our method on hillslopes. This definition has been added in the introduction of the manuscript. We have added a dedicated section that detail the different processes we observed in the discussion (section 5.1.2, L595).

2. Although it would be interesting to compare our landslide inventory to field observations, such data are not available for the studied area. However, we added a detailed comparison with a landslide inventory manually mapped from pre- and post-EQ aerial imageries (section 3.4 and 4.2, fig. 7). Given the impossibility of having the landslide extent mapped by Massey et al. (2020), due to the author's lack of response to our numerous requests, Dimitri Lague carried out a manual mapping. We used this new manually mapped landslide inventory to: (1) compare the two methods in terms of number and area of mapped landslide, (2) detail the origin of under-detection of landslides from both methods and (3) discuss the impact of landslide under-detection on landslide volume estimation using traditional approach. Following this analysis, the discussion now extensively discusses the origin of a rollover in the pdf(Area), and we provide a new analysis showing a systematic size dependent under-detection of landslides in the 2D inventory (fig. 12). This section also replaces the section about reactivated landslides present in the previous version of the manuscript for which we agree that the term "reactivation" was misused.

3. To further improve our treatment of potential errors and false detection, we have added two additional tests to our workflow and have added many elements on the discussion regarding current limits of the workflow. First, we explore in details the source of errors coming from misclassification of the original point clouds and imperfect flight line alignments. We therefore reprocessed the pre-EQ point cloud to remove incorrectly classified points and quantified residual errors for each flight line alignments (section 3.3.1). We now integrate the residual error from flight line misalignment in the estimation of the registration error. Second, we added a new analysis to deal with false detections after segmentation (resulting in a new step in the workflow, section 3.3.4). We define a new confidence metric for each segmented cluster (source or deposit) called mean signal-to-noise ratio (SNR). An optimal value of SNR, is defined to minimize the proportion of potential artefacts (new fig. 6 and 5b). This results in more than 746 sources and 748 deposits of low confidence removed from our inventory. The final inventory now contains 524 sources and 304 deposits of very high confidence (in the first version of the MS, there were 1431 sources and 853 deposits).

4.  We have added further analysis of the impact of the main parameters of our workflow (registration, segmentation and SNR filtering) in the discussion, appendix and supplementary material. Our results, and in particular the lack of rollover in pdf(A) are not significantly affected by these parameters.

5.  We have tested segmentation approaches based on density based clustering (DBSCAN and derivatives) which are considered state of the art for 3D rockfall segmentation. We demonstrate that these approaches (1) do not perform better than the segmentation approach we use and do not generate significantly different pdf(A); (2) may have the tendency to oversegment intermediate landslides into small ones or remove border points of large landslides; (3) are sometime much longer to run (45 min vs 3 sec) and have parameters which are not intuitive to set. Because the method part of the article was already dense, the result of this comparison exercise are in the supplementary (supplementary material S5), along with an explanation as to why density based clustering is actually not suitable to segment large landslides in our workflow. As in the first version of the article, we clearly state in the article that our segmentation has limitations, but we are not aware of other approaches that would perform better in segmenting complex 3D objects covering 3 to 4 order in size.

6.  The balance between the supplementary materials and the manuscript has been revised, and many figures have been updated following the reviewers recommendation.

7.  The title and abstract have been updated to feature the notion of rollover in the pdf(A) of 2D inventories. It reflects the additional data and analysis we provide that clearly demonstrate, at least in our study case, that the rollover in the pdf(A) observed in our 2D inventory but not in the 3D one, is due to a size-dependent under detection in 2D. We believe this is of critical importance for the geomorphic community working on landslide science.

We wish both reviewers a pleasant reading and thank them in advance for the time they may spent in evaluating this new version of the manuscript.

**Summary**

**Major Comments from David Milledge**

We really thanks the reviewer for its numerous and detailed comments. Many of them represent area of improvement for the workflow that we have addressed. Others are beyond even an extended revision of the paper and represent what should be an ideal, fully automated 3D workflow for landslide inventory creation backed by extensive field work that we cannot perform. Scientific research is incremental, and we fully expect that our workflow will be improved in coming years by others, as it was the case for 2D landslide inventories. To this end we provide all the elements (code, data) for other researchers to apply the workflow and reproduce our results, or apply the workflow to their data. We also clearly highlights the limits of our workflow. We strongly believe that despite its limitations, it currently represents the state of the art in terms of 3D landslide detection and landslide inventory creation, and that its application in comparison with a 2D landslide inventory shows that landslide under-detection in image based inventories is extremely prevalent in our study area.

**MC1) The paper needs to more clearly define: 1) landslides (i.e. what the inventory includes) and 2) what your inventory can and cannot be used for.**

*Early on you introduce the idea that there are different landslide processes (L43, "process specific") but you don't follow this logic through into your results. Instead, your analysis may contain an implicit definition of landslides as all processes responsible for surface change that cannot be attributed to fluvial processes (L217-9). You certainly need to make this definition of landslides explicit in your introduction.*

*The introduction needs a much clearer explanation for what you expect the inventory to be useful for. If it is for understanding landslide mechanics then it is essential that you make an effort to distinguish individual landslides on a mechanistic basis (see MC3 on amalgamation). If that is not an expected application of the inventory you should say so, otherwise there is a real risk it will be misapplied.*

*Appropriate uses of the inventory (e.g. volume estimation or landslide mechanics) depend not only on its purpose but also on entry criteria into, and distinctions within, the inventory. Non-fluvial surface change that might result from earthquake shaking includes: tree-throw, ravel, rockfalls and slides, slow earthflows, rapid soil slides and debris flows. The processes responsible for these surface changes differ from one another to varying degrees. If there is no distinction between them this precludes the inventory's use in analysis of landslide mechanics and therefore prediction. It allows comment on correlation e.g. of volumes and areas of change, but makes it extremely difficult to make any inference about causation. It also opens the work to the criticism that the bulk statistics mask important differences in behaviour between processes. For example: the differing size distributions for rockfalls (where others have reported no detectable rollover) and landslides (where there usually is).*

We added the following definition of what we consider as "landslide" in the introduction of the manuscript: "We use the generic term of "landslide" to define the spatially coherent changes detected by our method on hillslopes that result in at least several decimeter erosion (i.e.,scars or sources) or deposition". The discussion now features an entire paragraph (L596) addressing the various type of landslides that can be detected by our approach. The aim of this paper is not to better understand landslide mechanics as we cannot confidently identify the different landslide processes we detect. We are mainly interested by the estimation of co-seismic volume and to overcome issues such as under-detection and amalgamation on volume estimation. The introduction now clearly integrates these two problematics. We also believe that the new filtering approach that we introduce, which results in 3 time less landslide sources compared to the initial MS, results in a much more robust inventory.

**MC2) Elevation errors need to be better quantified and more thoroughly discussed**

*The manuscript needs a more thorough treatment of errors in the topographic data dealing with both: 1) the properties of the elevation errors that you have identified (e.g. spatial pattern, wavelength, covariation with landscape properties); and 2) the possible sources of error.*

**RE error properties:** *The amplitude and correlation length of elevation uncertainty from different sources and how they interact to generate a 2D elevation error field with a particular amplitude and wavelength is really important for this particular application, where landslides are identified by thresholding then segmenting differences. The error appears to have a fairly long wavelength in many areas (tens to hundreds of metres). It also appears to have some aspect dependence. The spatial correlation of these errors is important because you assume uniform isotropic*

*registration errors.*

     **RE error sources:** *It isn't clear to me what you mean by imperfect alignment, nor ICP related errors (L204-5). Identifying errors on Fig 3 and hypothesising the sources from which they derive are useful but need to be discussed in the manuscript as well. You recognise the presence of "internal flight line height mismatch" and indicate that it results in "large scale low amplitude topographic change" (L418-22). They should be introduced earlier in the*

*article with a more complete explanation of what they are and how you found them.*

     **Error consequences:** *It would be useful to say something about the implications of the topographic errors. I can see two implications:* **First**, *incorrectly assuming uniform isotropic errors will result in a confidence interval for identifying significant geomorphic change that is too strict on some slopes (e.g. some aspects) and not sufficiently so on others. This in turn will lead to false negative change detection on some slopes and false positives on others.*

**Second**, *change detection false positives will result in false identification of landslide objects or false representation of their geometry. False negatives are equally problematic since they could result in not only changes to landslide geometry but also cluster breakup (biasing the size distribution). These problems are illustrated in Fig 3 where a number of error patches would be identified as landslides by the detection algorithm if these areas were not assumed to be stable. If such false positives exist here it is they likely also exist elsewhere.*

*It is essential that you quantify their impact on your findings.*

*Suggested additional analysis: You could apply your landslide detection method to only the pre-defined 'stable areas' and generate landslide geometries. These geometries and their scaling relationships might indicate the impact of error on your findings (particularly: number and area density, pdfs, and scaling relationships of artefact landslides). If the results are similar to your findings for 'non-stable areas' it would be very difficult to argue that the data support your claims with any certainty (the same results could have been generated purely from topographic errors in the absence of any landslides).*

The reviewer suggested many areas to explore that are extremely interesting, but which, for some of them, would constitute an entire paper by themselves, in particular when it comes to the analysis of error properties suggested by the reviewer. We also aim at developing a generic workflow applicable to a variety of cases for which users may not necessarily perform extensive error properties analysis. Hence, to improve the paper we have worked on two aspects: (i) improving and better identifying errors in our dataset, (ii) defining a new confidence metrics (the SNR) for each landslide source or deposit to filter out landslide with low confidence.

**Errors in our dataset:** entirely revisiting our data, we identified two sources of errors: (1) Remaining LiDAR point cloud misclassification in forest areas, inducing local topographic errors, and (2) imperfect flight line alignments from the pre-EQ data, inducing topographic errors of longer wavelength. To address the first issue, we first removed as much misclassified points as possible by interpolating a surface and remove outlier points (see detail in supplementary material and section 2 in the paper). To address the second, we estimated residual errors of each flight lines due to imperfect flight line alignments composing the pre-EQ point cloud and defined the registration error *reg* based on the maximum residual error of the flight line misalignments (section 3.3.1. and S3 in supplements). Compared to the previous version of the MS, the *reg* is now 3 cm larger (20 cm vs 17 cm). We also show that only 1% of points are detected as significant change in the stable area, validating our choice of $LoD_{95\%}$. While our *reg* is considered uniform over the study area, the $LoD_{95\%}$ (eq.2) also take into account the local point cloud density and roughness which are correlated to the presence of vegetation. The $LoD_{95\%}$ is thus spatially variable. In addition, we are aware that, ideally, a spatially variable model for point cloud error and registration would be preferable for each survey and combined into a more accurate and complete form of *LoD* than what the M3C2 approach currently offers. However, in the absence of the position and attitude information of the sensor (e.g., Smoothed Best Estimate of Trajectory file) and raw LiDAR data - rarely available on LiDAR data repositories -, or of dense ground control which is hard to get in mountainous environment, it is currently impossible. We now discuss this in the discussion (section 5.1.2).

**Filtering landslides with low confidence with the SNR**: To limit the false detection due to these errors, we also defined a signal-to-noise ratio threshold to efficiently remove suspicious landslides (section 3.3.4 and 4.1). This index is based on the mean ratio between the 3D-M3C2 distance and the $LoD_{95\%}$ for each landslide. We provide a way to evaluate the optimal value of SNR by comparing the number of landslides in the database to a case with no change (i.e., two versions of the same data but with different sampling, a test that we now call Same Data Different Sampling test (section 3.2). This SNR filtering removes a very large number of landslide source and deposits (fig. 5b), and in particular long wavelength low amplitude changes that
occurred due flight line misalignment in the pre-EQ data, as well as many small landslides in forested region where point
density is very low. We provide a systematic analysis of the impact of SNR (and *reg*) on pdf (A) (fig. 11), pdf (V) (fig. S 13)
and V-A relationship (fig. S14).

**MC3) You recognise that segmentation results in amalgamation but don't quantify its extent or impact**

*Severe amalgamation can result from automated segmentation of a thresholded classifier. In the landslide maps
that you show here (e.g. Fig 2), amalgamation appears a severe problem for the largest landslides. The argument
that it can be solved by tuning Dm (as you suggest on L426-7) is unconvincing since two separate landslides can
be within millimetres of one another but have different failure mechanisms. You later say that you "cannot resolve
the amalgamation" (L511) and that it "is still a potential issue" (L558). I would argue that it is not potentially but
certainly an issue. Your figures show that amalgamation is present (perhaps pervasive) in your inventory but its
extent or impact is not quantified. The total volume is insensitive to amalgamation but your landslide pdfs and
scaling relationships are not.*

***Amalgamation makes it difficult to use the inventory to understand landslide mechanics and therefore
susceptibility or hazard.*** *If the inventory is to be useful in understanding landslide mechanics then it is essential
that you make an effort to distinguish individual landslides on a mechanistic basis. In the extreme case, an
individual landslide where subsets of the failed material moved in opposite directions (e.g. Fig 2) is clearly
problematic for the mechanics of that failure.*

***Amalgamation introduces bias towards large landslides in size distributions.*** *Power law exponents and even the
appropriateness of any power law can be sensitive to landslide amalgamation. Observations such as that on L569-
70 that "largest and deepest landslides deviate significantly from this trend" become questionable since these are
the landslides most likely to be the result of amalgamation.*

***Suggested additional analysis:*** *If you are to retain your findings on landslide scaling and size distributions, then
you must examine the potential impact of amalgamation on these findings. There are methods available that
quantify amalgamation in landslide mapping. You should investigate these. You should preferably also address
this amalgamation, either by removing affected landslides or ideally, by decomposing amalgamated landslides into
their individual failures.*

We agree that the segmentation approach we use certainly do not allow to solve the amalgamation problem, and this is
highlighted in many parts of the MS. However, the problem of amalgamation is inherently subjective and plagues all
inventories. Our 3D data reveals a level of complexity, and a density of amalgamated landslides which makes the definition of a single landslide in relation to an ideal rupture mechanism extremely difficult. Even the segmentation of the 2D inventory proved to be extremely complex and is entirely not reproducible. Hence, we favour a reproducible approach, even if currently limited, that can be applied exhaustively to much larger datasets, than a non-reproducible one (2D manual mapping) that we now demonstrate misses a very large number of landslides and incorrectly map their contour.

In this new version of the paper, the landslide amalgamation can be visualized with a map of the landslide source colored by individual landslide as defined by the method (section 4.2 Fig.7). Moreover, the comparison between both inventories shows that while 171 of 2D-sources are shared with 3D-sources, it represents 144 sources 3D-sources. This highlight that 25 landslide sources are amalgamated in the 3D inventory (L-. As in the previous version of the paper, we perform a sensitivity analysis of the impact of $D_m$ showing that landslide statistics are not severely affected by this parameter for $1.5 < D_m < 3$. We also explored density based spatial clustering algorithm used in 3D rockfall segmentation, derived from DBSCAN (Ankerst et al., 1999;

Martin Ester, Hans-Peter Kriegel, Jiirg Sander, 1996; Tonini and Abellan, 2014) and HDBSCAN (Carrea et al., 2021; McInnes et al., 2017). None of them managed to provide a significantly better segmentation of the largest landslide and are significantly longer to run than the connected component algorithm we use. We now thoroughly discuss about this in section 5.1.2.

**MC4) Topographic errors propagate through segmentation to introduce a bias towards small landslides.**
**Existing experiments to quantify this bias are insufficient.**

*You say on L559-60 that "our approach has the benefits of more systematically capturing small landslides than traditional approaches". However, this is one of my main problems with the paper given potential propagation of topographic errors through thresholding and segmentation. You show (in SI) that: 1) in the absence of any real topographic change your detection algorithm generates artefact landslides of 1-20 $m_2$ purely due to spatially*
*uncorrelated topographic noise; 2) this noise generates many more small than large landslides; 3) in this experiment artefacts >20 $m_2$ were extremely rare. However, this does not demonstrate that predicted landslide size is insensitive to longer wavelength topographic errors (known to be present in the data); nor even to short wavelength noise in the presence of longer wavelength surface differences (e.g. real landslides).*

*First, even without any real topographic change (i.e. no real landslides), the size distribution of erroneous*
*landslide-like clusters will depend on the spatial correlation length of the difference errors, which in turn depends on the correlation lengths of the errors in the surfaces being differenced. Longer error wavelengths will enable the generation of larger error clusters. Your figures show that topographic errors are clearly not uncorrelated and you recognise this yourselves (L418-22); nor do the errors appear to have a single characteristic wavelength. This is a hard problem but one that you must deal with if you are to convince the reader that the landslide inventory you*
*have generated is not hopelessly biased by these landslide-like artefacts. Second, the problem is not only that clusters of erroneous negative surface difference due to roughness (or other errors) can create artefacts that appear*

*to be landslides, but also that clusters of erroneous positive surface difference are collocated with real topographic change (e.g. due to a landslide); these can interfere negatively with real changes reducing the surface difference below the threshold for detection and breaking a single landslide into multiple patches.*

***Oversampling of small landslides is important because*** *it undermines your most surprising and high impact claim: that rollover reported in previous inventories is due to under detection (L573-4). I am not currently convinced by this claim because you do not exclude the possibility that the lack of a rollover is solely due to detection errors. You need to quantify these detection biases before you can make these claims.*

***Suggested additional analysis:*** *A "more advanced segmentation" (L431-3) may be out of scope for this paper.*
*However, an indication of the impact of the simple segmentation on object based classification skill is an essential requirement of this paper if it is to retain the current approach to identifying discrete landslides. Two possible avenues could be followed to provide such an indication.* ***First,*** *your analysis of topographic changes on the stable surfaces (Fig 3) would allow you to perform the same analysis that you have performed in the SI but using the pre- and post-EQ surfaces for the stable zones identified in Fig 3. This would enable you to identify the size distribution*
*of artefacts that can be generated from topographic errors with a spatial correlation length closer to that for the unstable parts of the study area. This though still does not account for the possibility that the landslide erosion signal itself is altered by the noise (e.g. by disconnecting clusters).* ***Second,*** *you could collect a landslide check dataset using independent observations. This might take the form of an entirely independent inventory but should certainly also involve cross-checking to confirm the existence and characteristics (e.g. area, shape, depth) of your*
*predictions.*

We added further analyses to the method to deal with topographic errors and erroneous landslide. We also added an entirely new 2D landslide inventory as suggested by the reviewer. We are now confident that the actual landslide inventory corresponds to real changes.

Please see our reply to the MC2) and MC8) comments for a detail answer.

**MC5) Findings that differ from previous work**

*There are a number of unusual findings that are worthy of comment because they are some of your most interesting and potentially important findings. It is essential though that each is carefully examined and that the critique that it might have arisen due to methodological errors is dealt with head on.*

***First****, it is not unusual to identify more sources than deposits due to amalgamation of landslide deposits. However, it is unusual for deposit areas to be smaller than source areas (L281-2), and for deposit depths to be thicker than source depths (L265-70 and L460). These result should be compared with results from previous studies.*

*Second, the estimate that Massey et al. (2020) "potentially missed around 169,000" landslides (L491-2), i.e. 92% of the landslides is surprising because of its magnitude. However, to be convincing you need to demonstrate that the landslides that you have detected genuinely are landslides.*

*Third, you identify areas of deposition where there is no upslope erosion (L256 and Fig 4). I don't think these can be real deposition zones but instead must be a consequence of incorrect landslide detection. Their spatial extent and depth distribution would be a useful indication of the precision of the technique.*

*Fourth, it is extremely unusual that locations classed as vegetated in post-event optical imagery but identified as a landslide by another technique are considered by the authors to be genuine landslides (L347-8). Instead, the presence of vegetation at the location strongly suggests a false positive.*

*Fifth, you find no statistically significant difference between total landslide erosion and deposition (L459), does that mean that there was no significant mass loss due to landsliding from the study area? If so, the earthquake's role has been almost exclusively to break up the material and redistribute it within the study area. This seems to differ from the findings of previous studies on the influence of coseismic landslides on earthquake mass balance. It would be very interesting to see the elevation change of the mass (perhaps as elevation pdfs for scars and deposits). This might be a valuable contribution on the instantaneous impact of coseismic landslides on mass balance.*

**First**: contrary to the reviewer experience, this result does not surprise us, in particular when the runout of landslides is not long. The filtered data clearly support this finding.

**Second**: this statement has been removed, and we now use our own 2D inventory to discuss under-detection and the number of missed landslides.

**Third:** We now filter landslides by a signal-to-noise ratio (section 3.3.4) and are confident that the actual landslide inventory corresponds to real changes. Some very small deposit areas may not have upslope erosion as we expect deposit are to be easily detectable by our method than source areas (section 5.1.2).

**Fourth:** we partially disagree with this statement. As now explained in the discussion large landslides that strip out vegetation are obviously mapped in 2D inventory, but small ones that occur on less dense area are extremely difficult to map in 2D imagery as our inventory shows. Moreover, vertical subsidence due to upslope propagation of landslides is entirely missed in forest, while it is detected in our approach. We think the comparison with the new 2D inventory will resolve the reviewer's reserve.

**Fifth:** indeed, we found no statistically significant difference between the total landslide erosion and deposition suggesting that there was no significant mass loss (given the uncertainty of our method). We also observe that most of the landslide material volume remained on hillslope with only 3 large deposit areas located on the river bed. Although we agree that it would be very interesting to look at the mass balance as suggested, we choose to let this question for another study as the MS is already quite long and dense.

**MC6) Areas should be calculated surface parallel**

*I agree that variation in surface normal orientation could introduce bias in the volume estimate (L238-9) and that volume should be calculated normal to the plane on which landslide area is calculated. However, it isn't clear to me that a horizontal plane is most appropriate for area calculation. This will severely underestimate landslide area on the very steep slopes typical of rockfall initiation. Surface parallel area calculations would be more appropriate and orthogonal measurements for volume calculation would still be possible. I don't find the argument based on retaining consistency with other studies (L279-80) convincing. Your inventory appears to contain a significant number of rockfalls and their area and volume are often calculated in a vertical rather than horizontal reference frame. If you do not alter the reference frame you should certainly report the distribution of surface slopes of your landslides and comment on the effect of vertically projected area given this distribution.*

Computing true surface area requires significant new developments of the workflow in relation to segmentation, to better handle complex cases of landslide amalgamation. Surface parallel area calculations is not as trivial as it may seem if it is to be applied automatically over a large number of landslide geometries and configuration. We clearly highlight this issue in the discussion (as we did in the first version of the paper), but have no simple solution to offer (see section 5.1.2). We present the distribution of pre-eq landslide source slopes, and shows that only a very small fraction (1 %) of points haves slopes > 60°. Given that the median of the slope distribution of our landslide source inventory is 34.1, the estimate of the true area gives a total landslide source area of 356,876 m² rather than 286,445 m.

**MC7) Reactivation is actually under-detection on bare or sparsely vegetated slopes**

*I am not convinced that the set of landslides that you are describing are really re-activated landslides (L344-5) but I'm also not convinced that reactivation is really the relevant issue. I think you demonstrate that you are able to detect landslides in vegetation sparse or vegetation free areas that have generally suffered serious underreporting. Some of these areas may be bare due to previous landsliding, others may not but the key point is that inventories have underestimated landslide density due to underdetection in these areas.*

*However, if you do want to focus specifically on reactivation then you need to clearly define reactivated landslides and detail how you classify them as such. For example: Does a landslide need to reoccur within the footprint of an existing landslide to be a reactivation or is retrogression included within reactivation? Within what time window following the first landslide must the second occur for it to be considered reactivation?*

We agree that "reactivation" was not the appropriate definition for what we consider which is more associated with under-detection of landslides. We now describe in details the differences in terms of detection of landslide areas between a 2D manually mapped inventory and the 3D differencing approach in section 4.2.

**MC8) Landslide object detection needs to be tested against an independent dataset**

*The findings in this paper depend critically on the skill with which the proposed method can classify landslide scars*

*and deposits. Thus it is essential that the paper reports testing results that quantify this object based classification skill. At present, "orthophotos were used to visually validate" the classifier (L115-6) but without reporting results of this analysis. I think it is essential that you explicitly explain your sampling and mapping strategy for landslide detection from orthophotos in the methods. You should then include a section in the results where you compare your orthophoto based mapping to the surface differencing approach. However, it is not enough to simply say the*

*orthophoto mapping did not identify landslides that were identified by the surface differencing. You should then go to a carefully chosen (e.g. stratified random) subset of the landslides detected by each method that were not detected by the other (i.e. surface differencing but not ortho-photo mapping and vice versa) to establish as far as possible which of the two methods was in error and why. While finding and mapping thousands of landslides might be time-consuming (L255), confirming their existence and characteristics (e.g. area, shape, depth) would not.*

This lack of comparison between the 3D differencing method and a more classical approach has been addressed by adding a manual mapping of landslides based on 2D images. We added a section that specifically explore the differences between the methods (section 4.2). Moreover, the resulted landslide area distribution mapped manually has been added in the figure in section 4.3 and compared with those obtained with the 3D differencing method.

**Minor Comments**

*L32: You need to define landslides early in the paper, it will influence interpretation of your later findings.*

We now define landslides as "the spatially coherent changes detected by our method on hillslopes that result in at least several decimeter erosion (i.e.,scars or sources) or deposition" (L 95-96).

*L34: "spatial distribution, total volume…"? explaining why these are useful would be helpful because this sets the motivation for everything that follows. Landslide mapping has different requirements depending on its purpose. The purposes you choose here set the metrics against which your own mapping should be evaluated.*

This sentence has been modified by: "Both are important for the understanding of landscape evolution and the management of associated direct and secondary hazards ".

*L35: "associated direct and secondary hazards" What is the connection between mapping, the first three elements of the list and these later two? It would help to make this explicit. As above, this relates to the purpose of the mapping.*

See comment above.

*L43: "process specific": here you recognise the importance of different landslide processes. This is an important point that needs to be reflected in your own analysis (see MC1).*

Potentially has been added to reflect the current knowledge uncertainties on this question. We also discuss what potential processes we detect with our method in section 5.1.2.

*L110&113: It isn't clear what statistic is being referred to here as a metric for vertical accuracy. RMSE? If so it would be better to quote mean error and SDE, giving an indication of the contributions of precision and bias.*

The vertical accuracy of both LiDAR data has been estimated from the difference between the elevation of GPS points and the nearest neighbour LiDAR point elevation. While the mean and the standard deviation of the difference are provided by the 2017 LiDAR dataset report (Aerial survey, 2017), and respectively equal to 0.00 m and 0.04 m, only the standard deviations are mentioned in the 2014 LiDAR dataset report (Dolan, 2014) and range from 0.068 m to 0.165 m.

*L114: "using the classification provided": More detail is needed on the method used to classify ground points.*

The survey report of the LiDAR data only mention that the ground points have been automatically classified using the Terrascan software. The reference to the survey report (Dolan and Rhodes, 2016) have been added to the text.

*L115-6: "orthophotos were used to visually validate": The results of this validation are missing from the paper.*

This part has been replaced by the comparison with a manual mapping based on 2D images. See section 4.2. The text of this section has been modified to introduce this manual mapping.

*L135: "core point": What is the sensitivity of the results to varying core point spacing? The orientation and spacing of the grid of core points should also be described and justified either here or in section 3.2.*

We have added a sentence explaining how to select the core point spacing. However, given that it can only varies between 1 and 2 m, we did not deem important to explore the impact it has on the end result.

*L136: "first dataset": Which is the first dataset? Are your results sensitive to this choice, both in terms of depth area scaling and disagreement in 1) change surfaces and 2) maps of significant positive and negative change?*

The first dataset used in the paper is the pre-event point cloud. To analyse the potential impact of the first dataset to be compared in M3C2, we applied the workflow detailed in the paper with the post-event point cloud as the first dataset and the pre-event as the second one to be compared. We then analysed the results from both versions of the workflow by computing the difference between the 3D-M3C2 distance values obtained on the significant change point clouds. Results show that 98.6% of the significant change point cloud are exactly the same in both cases and that the mean difference between both 3D-M3C2 distance values is 0.001m with a standard deviation of 0.28 m.

As these results show that the choice of the first dataset to be compared do not influence the final results and that the text of the method is already long, we did not include this result in the manuscript.

*L137-8: This is not clear: do you mean that you calculate the centroid of each point cloud in 3D then take the magnitude of the 3D vector that connects these two points; or that (for each point cloud) you take the arithmetic*

*mean of differences from the reference plane (defined by D) in a direction normal to that plane? In either case you are performing a spatial averaging at length scale d/2 assuming a uniform kernel. First, is it problematic to perform averaging over length scales larger than the core point spacing? Second does it make sense to assume equal weight in the average with plane parallel radial distance from the core point, or should some form of inverse distance weighted average be used? I would have assumed a weighted average was more appropriate but it would be useful*

*for you to explain why an unweighted mean is more appropriate.*

To calculate the distance between the two point clouds, the average positions $i_1$ and $i_2$ of the point clouds are first defined and then the distance is computed between the two positions along the normal vector. The average positions are defined as the arithmetic mean of the distance distribution of each point of the subset of points (created by the intersection of the cylinder to the point cloud) to the normal vector (or the cylinder axis; see Lague et al., 2013).

As this is part of the M3C2 algorithm that we did not modify, we don't discuss the choice of a uniform kernel rather than a non-uniform one.

*L139: "if not intercept is found…": This is not clear to me. Do you mean 'if the cylinder does not intersect any points in the second surface? Why would this happen? Does this only occur at the boundaries of the point clouds?*

*How do these intersection failures influence the surface differencing and how do you report them in your later analysis?*

In LiDAR datasets, the density of points is non-uniform over the entire point cloud. Consequently, missing data or very low point density (<5 pt/m²) can occur inside the point cloud due to the absence of laser impact on the ground during the data acquisition. This mostly occur in dense vegetated areas or water surface areas (for topographic lidar). When performing M3C2, it is thus possible that the cylinder cannot intercept points or just a few (< 5). In both cases, the M3C2 distance will not be considered significant. In areas with low point density (<5 pts/m²) a solution is to perform M3C2 with a larger projection scale $d$ to include more points in the distance and statistic calculations.

*L140: "provides uncertainty": what is the basis / justification for the uncertainty estimate taking this particular form? It looks familiar as it has some similarities to a confidence interval but also some differences. This threshold is important to explain and justify in detail because it is used to threshold discrete landslides in the following analysis. Why threshold at 95% confidence? What is the impact on your findings (total volumes and scaling relationships) of thresholding at a difference CI (e.g. 99 or 90% confidence)?*

We refer the reviewer to the original M3C2 paper which has an extensive discussion on the confidence interval, and how to consider surface roughness, point cloud errors, point density and registration in the context of change detection on 3D point clouds. The threshold has been set to 95% to build the segmentation on as many good points as possible. We do not believe that changing this threshold to 90 or 99% significantly change the landslide statistics given the results of the sensitivity analyses of parameters (reg, $D_m$ and SNR threshold) that mainly control the landslide inventory.

*L145: "detrended roughness": I think it would be useful to add that these are standard deviations if this is true.*
We removed this part of the sentence which was unnecessary

*L145: "reg" is quantified using the standard deviation of differences between the surfaces. I think it would be helpful here and elsewhere to use similar notation for the registration error to the other errors being examined here. Why are the local terms converted to standard errors but reg is left as a standard deviation? Finally, the length scale over which reg was calculated would seem to be important here.*
We choose to keep the "reg" notation to specifically refer to the registration error when needed in the manuscript. We now provide an explanation in the text as to why reg is not converted as a standard error (L 166). Reg is not measured over a length scale, it is based on the standard deviation of the 3D-M3C2 distance between 2 clouds. This part is now explained in greated detail in section 3.3.1 where we discuss the notion of intra-survey and inter-survey registration quality.

*L164: "at least 5 data points": Does removing these points from your analysis alter your results on pdfs and scaling?*

The 5 data points considered here is the minimum number of points required to compute the $LoD_{95\%}$ (eq(2)) for each core point at a scale $d$. Even if M3C2 parameters can be changed to evaluate the $LoD_{95\%}$ for fewer points, the test performed by Lague et al. (2013) showed that the statistical model behind the definition of the $LoD_{95\%}$ became incorrect.

*L165: "not deemed interesting": I don't think that this is the right phrase, can you rephrase? What was the impact on your results of applying d=10 m throughout?*

The sentence has been modified. Applying d=10 m increases the landslide source and deposit by 1% and 0.7% respectively. In terms of volume this represents an increase of 2% and 0.89% respectively.

*L173: "may result in…": How do you identify when this has happened? What is the objective that you are optimising?*

This can be identified in the bottom of narrow valleys and top of very steep divides where no evidence of mass movement processing can be identified by visual inspection of orthophotos. These cases are now filtered by using the SNR metric as they have very large $LoD_{95\%}$ (see section 3.3.4).

*L187: "larger than 1 m": perhaps give a range, >1 could be very large but really it is on the order of 1-2 m.*

Updated as suggested (L241).

*L187: "rasterizing the dataset": at what resolution?*

The grid spacing used here have been added in the text.

*L196: "standard deviation…" These stable areas would be an excellent test of your landslide detection method, indicating the scaling relationships, size distributions, and total volumes generated by artefacts alone.*

This analyse has not been performed due to the changes we made on how we manage artefacts with SNR filtering (see section 3.3.4). Applied on stable area, the workflow does not detect any landslide.

*L199-200: "The registration error…": this definition should come earlier. It is important for interpreting eqn 2.*

We disagree with this comment as the registration error "reg" can be defined differently depending on the application of M3C2. In section 3.1, we aim to give a general description of the M3C2 algorithm.

*L200-1: "manually selected": This doesn't seem consistent with L191. Was a threshold of change selected manually then all areas with changes smaller than threshold included? Or, did you take a subset of pixels from the areas with changes smaller than the threshold so that only large patches were considered?*

Actually, after applying a first 3D-M3C2, we selected only areas with a 3D-M3C2 distance less than 1m to help the identification of stable areas. We then manually refined the stable areas to select only areas away from landslides. The fine registration is then applied on these stable areas. The text has been updated.

*L215: 15-20% it might be useful to show the location of these points on one of your maps.*

The percentage of these points on steep hillslope has been revised and actually represents up to 12% of steep slopes.

*L217-9: "all geomorphic processes…": This assumes that all non-fluvial changes are landslides (see MC1).*
The definition of what we consider as landslide has been added in the introduction. Please see the reply to the first minor comment.

*L221-234: "Landslide source and deposit segmentation": This process is much more clearly articulated in the SI. When I finished this paragraph I was totally confused but when I read the SI it became clear. The SI text on this analysis is really important and needs to be included within the main manuscript.*

This section has been clarified and simplified. The segmentation algorithm segments a point cloud into sub-clouds based on the spatial connectivity between each point. A first parameter defines the minimum number of points ($Np$) a sub-cloud must have and a second parameter defines a minimum distance ($D_m$) from which two sub-clouds are considered as two individual landslides. In the new version of the manuscript, we explain how we chose the value of the two parameters. $Np$ has been set to 20 points (or 20 m² given the spatial resolution of core point of 1m) as we average the position of both LiDAR point clouds in a 20 m² window (projection scale d=5m in diameter). As there is no objective way to a priori choose $D_m$, we explored various values and chose $D_m$=2 m as an optimal value between landslide amalgamation and over-segmentation. We also analyse the impact of $D_m$ on the statistical distribution of landslide sources in the discussion (see section 5.2.2, Appendix A and S13, S14 in the supplements).

*L225: "compact": I don't understand what compact means in this context*

This word has been deleted.

*L225: More detail on how this method works is essential here. This step is a key component of your method and it is very important that the algorithm is given the same clear treatment that others have been in previous sections. I guess that Dm sets the distance between points within which two points are considered connected. This is a tuning*

*parameter and it is not clear to me what objective you are seeking to optimise during tuning.*

This question has been addressed in an earlier comment. Please see the answer of the comment for L221-234.

*L226: "amalgamation effect and the over-segmentation": These two effects and your method for quantifying them need to be explained. I don't think this can be included only in SI.*

Indeed, amalgamation errors are important to consider have plagued 2D inventories for quite some time. Yet, these have been published. We now provide a detailed sensitivity analysis, going far beyond we've ever seen, that should help the reviewer evaluate if he has confidence in our results. Also, contrary to a 2D manual analysis, our 3D detection is fully reproducible and data and workflow are provided for the reviewer to try by himself.

*L228: "small artefacts". Do you mean that there are a small number or that they are small in magnitude?*
*We meant that they are small in magnitude.*

We meant they are mall in magnitude. The text has been modified and "small artefacts" has been removed (see section 3.3.4).

*L229: Why would it be the smallest landslides that are particularly affected by "these artefacts"? I had assumed that these were due to things like inclusion of points on either side of ridges.*

The range of artefact area is now given in section 3.3.4.

*L230: "minimum number of points": You haven't explained what this parameter controls. Is it the minimum number*

*of connected components required for a patch to be retained (i.e. minimum landslide size)?*

Np is the minimum number of points required to define a sub-cloud from the original point cloud and thus indeed corresponds to the minimum landslide size. This was previously in SI, and now is in the text

*L231: I must have misunderstood Np, it does not follow that all detected changes will be artefacts if Np simply*

*removes landslides smaller than some minimum size. On reading the SI I now understand, this is because you aren't comparing before and after point clouds. You explain this well in the SI, but need to bring some of that text in here.*

This section has been updated. The choice of the Np value has been addressed in an earlier comment. Please see comments for lines 221-234.

*L231-2: Reference to pre- and post- earthquake data here is confusing unless read in conjunction with the SI.*
This section has been updated.

*L233: "artefacts": What are artefacts? How do you identify them and why do they become smaller/less frequent/less important for larger areas? What does it mean that they are limited (e.g. they make up less than x%*
*of the landslide objects)? The SI text on this analysis is really important point that needs to be included within the main manuscript.*
We define artefacts as either false negative or false positive change detection. We identify them with the SSDS analyses (see section 3.2 and 3.3.4).

*L238-9: I agree that volume should be calculated normal to the plane on which landslide area is calculated. But disagree that a horizontal reference frame is most appropriate. This will underestimate landslide area (severely on the very steep slopes typical of rockfall initiation). Surface parallel calculations seem more appropriate (see MC6).*
Please see reply to the MC6) comment.

*L245-7: "specific to each landslide": This potentially presents a nice opportunity to check your method by comparing source and deposit volumes for individual landslides over your study area.*
We agree that it is something that would be interesting to look at it. However, this require further development to automatically link landslide source with the appropriate deposits which is beyond of the scope of the manuscript.

*L252-3: "Most of the detected changes on hillslopes correspond to…" How do you know this? Is it based on your definition that all non-fluvial change is due to mass movement? Or, are you drawing on additional information to interpret the change map? Ideally you would have some independent check data.*
In this section, we now first describe the pattern of large detected changes for which a source area can be associated to a deposit
area. These large patterns can easily be linked to landslide processes. For smaller patterns, we now compare the landslide inventory obtained by the 3D point cloud differencing method with a landslide inventory manually mapped from visual interpretation of aerial imageries (see section 4.2). We also added in the discussion (section 5.1.2) a paragraph discussing the different processes we inferred from both the 3D-M3C2 distance field and ortho-imageries and the limit of our approach to confidently identify the dominant landslide mechanism for each sources and deposits.

*L254: "previously unstable bare rock": It is useful to know that many are located on bare rock but how do you know that it was previously unstable and what do you mean by previously (this implies a timescale)?*
The text has been modified and 'previously unstable" has been removed.

*L255: "Their large number illustrates how difficult": Why would their large number make them difficult to manually extract? Perhaps time-consuming rather than difficult. An inventory with 1431 sources is a very modest sample size by the standards of modern landslide inventories.*
The text of his section has been subject to significant modifications. This sentence has been removed.

*L256: "deposit areas" Some of these areas do not have any upslope erosion. How do you explain this? See MC5*
Some deposit areas can be detected without an associated upslope erosion. A possible reason is that the surface change associated to these deposit areas is sufficient to be above the $LoD_{95\%}$ but not for the upslope erosion area.

*L259: "stable areas": This argument seems somewhat circular since stable areas were identified from surface*
*differencing rather than independent information.*
We agree but our study deals with data for which such information are not available.

*L259: What do you mean by "artificial changes" here?*
Artificial changes correspond to either false negative and false positive change detection. This definition has been
added to the manuscript.

*L286: It is not clear what "substantial" means in this context.*
This word has been deleted

*L291: A rollover is considered characteristic of landslide, but not of rockfall, distributions I think. As I understand it your inventory includes both. It would be useful to comment on this amalgamation of processes within a single inventory and what it implies for interpretation of the landslide distribution. See MC1.*

Please see replies to the MC1) comment. We also now thoroughly discuss on the lack of a rollover and the effect of the amalgamation of processes on this result in section 5.2.2.

*L292-3: "if we reduce the minimum landslide size to…" I don't think that you can make this statement if you have excluded landslides smaller than 20 m₂ for good reason. Your methods text makes a compelling argument that surface differencing errors will generate landslide-like artefacts that will distort the distribution and thus should be censored.*

We agree that this sentence is confusing. It has been deleted.

*L305-7: This is connected to my earlier concern about planimetric areas and suggests a need either to address the problem as you suggest here or to include a more detailed explanation for when the problem presents itself, what fraction of the landslides in your inventory might suffer from it. See MC6.*

Please see the reply to the MC6) comment.

*L308: Why frame your analysis in terms of volume-area relationships rather than depth-area? Since volume is the product of depth and area the x and y variables are correlated by definition. Others e.g. Larsen have examined depth-area scaling and the translation from one to the other is straightforward. But using depth de-trends the y-*
*axis making it easier to see the scatter in the relationship, which is a major advantage.*
This version of the manuscript now includes both V-A relationship and depth-area relationship (see section 4.3).

*L315: It is not surprising that your depth area scaling relationship is gentler than that of Larsen since you censor core points with difference < 0.33, making it impossible for small landslides to also be shallow.*
Indeed, and we now discuss this. In particular, the relationship is barely different if we consider a SNR=1, meaning that compared to the 2D inventory, we miss 32 shallow landslide sources over an inventory of 1270 sources (SNR >=1). Hence, even if the 3d inventory had captured the very shallow landslides that the 2D mapping did capture, it would hardly change the scaling relationship. One could also say that previous 2D inventories have significantly underevaluated the number of small landslides, which in turns affect the representativity of published V-A
relationships from 2D inventories.

*L325: An analysis of the uncertainty associated with total volume estimates would be very valuable. Your study is unusual in not only being able to resolve scar volumes but also deposit volumes, thus you can calculate a net*

*volume removed from the hillslopes. This value is of considerable interest and comparing it to the estimates that have been made elsewhere using different approaches seems worthwhile. You should however deal with your own uncertainties as well and within those uncertainties should be included those areas of deposition that have no landslide source upslope.*

The volume of landslide sources and deposits of our inventory is not statistically different. However, we believe that the larger volume of deposit is consistent with decompaction and the likelihood for sources to be more statistically filtered out than deposits (see section 5.2.1).

*L344-5: A clear definition of reactivated landslides is needed here. See MC7.*

This section has been removed. We do not address reactivated landslide anymore. This analysis has been replaced by the estimation of under-detected landslides based on the comparison with the manually mapped landslides.

*L347: In many areas "bare rock areas" surfaces do not necessarily indicate recent landsliding (depending on how recent is defined). They may in this landscape though, so it would be useful to say so and evidence the claim.*

This section has been removed. Please see the above comment.

*L349: "following classical approaches": This is not a convincing justification for the method. It argues for the method's validity because others have previously adopted it without any further indication of why it might be valid here.*

This section has been removed. Please see the above comment.

*L384: I don't think you do explain the subsampling exercise in section 3.2 it is currently in the SI as I understand it.*

We added a new section explaining the subsample exercise (see section 3.2), that we call SDDS (Same Data Different Sampling) test.

*L431-3: "more advanced segmentation": an indication of the classification skill of the simple segmentation of object based classification skill is an essential requirement of this paper.*

The classification skill of the segmentation approach can now be assessed with the landslide inventory map show in figure 6b for which each individual landslide has a single color (section 4.2).

*L460: "deposits form more concentrated and thicker patches": This result is surprising and should be compared with expectations from other studies on landslide runout.*

We think it is not surprising given the small runout of the landslides in this area, and is actually backed by the data in terms of mean 3D thickness of the deposits and sources.

*L461-2: I would have expected "very shallow rockfalls" to result in even shallower rockfall deposits because they are initially small in volume and spread out over a wide area during deposition. Reference to rockfalls here increases my concern around use of horizontal (i.e. planform) area calculations.*

Please see response to MC6) comment. Concerning the rockfall deposits, it might be the case if the rockfall is triggered in areas free of previous rockfall/landslides. However, if the rockfall is triggered in an active sliding area thus the likelihood of
the associated deposits to merge with a pre-existing deposit area increase. In such case, the volume of deposit area will appear larger than source area because the deposit zone collects sediments from different sources. Our study are present such cases. To appease the concerns of the reviewer, we also provide the distribution of pre-eq source slope that shows that in the final inventory, slopes larger than 60° are less than 1 %. Rockfall are thus not expected to be a significant contribution in our inventory.

*L465: It is not clear what you mean by "landslide analysis". I suggest cutting this phrase, I don't think it is necessary.*

Updated as suggested.

*L466: Whether the right tail is a "power law" or not is debated. See Medwedeff et al. (2020) among others.*
The reference has been added in the text.

*L473-4: Power law scaling can be sensitive to amalgamation errors such as those in your inventory. As a result I struggle to know how much confidence I can place in these values. See MC3.*
Please see response to the MC3) comment. Obviously amalgamation will affect power-law exponents, and we now provide a systematic analysis of the impact of various parameters of the workflow that affect the final geometry of landslides (reg, and $D_m$). We do not argue that our dataset provides a unique, absolute power-law exponent.

*L488-90: "rollover ... is likely caused by an under detection of small landslides": This is a very important claim*
*with significant implications for our understanding of landslide mechanics. But it depends on both: 1) your findings*

*being robust to error in segmentation; and 2) the inventories that you are discussing having comparable definitions of landslides. Given that the landslide detection method has not been tested and that there are reasons to expect that the method introduces considerable bias in the size distribution I am not convinced that the claim is true. The lack of a rollover in your dataset might instead be due to detection errors.*

We believe we provide now firm evidence that the rollover is related to under-detection, in particular given the strict filters we apply on our inventory. We also demonstrate that under-detection exists, even for relatively large landslide of the inventories for which there is absolutely no doubt that they are landslides (see section 4.2). We now extensively discuss the potential other causes for the occurrence of a rollover and demonstrate through a sensitivity analysis that the segmentation parameter does not change this (note, the segmentation parameter sensitivity analysis was already presented in the first version of the paper; see section 5.2.4).

*L499-500: "most of our inventory is relevant to shallow landsliding": shallow landsliding is usually defined as landslides initiating above the soil-bedrock interface, and thus distinct from rockfall. Can you make this distinction in your inventory?*

We agree that we cannot make this distinction. We now compare, in section 5.2.4, our landslide depth-area and volume-area relationship with previous studies (Larsen et al., 2010; Massey et al., 2020). The similarities with the results obtained for soil-type landslide suggest that our landslide inventory may be consistent with shallow landsliding. We discuss about this suggestion at L-776-781.

*L511: "cannot resolve the amalgamation…": The problem of amalgamation needs a more detailed treatment. Total volume is insensitive to amalgamation but landslide size distributions and scaling relationships are not. See MC3.*

We brought further analysis on how our results are sensitives to amalgamation. See section 5.1.2.

*L535: "a much lower detection level than optical methods": This phrase is unclear, can you rephrase?*

This section has been subject to significant modifications. This sentence has been removed.

*L551: "95% confidence of 0.34 m": This is quoted here as spatially invariant but the method captures spatial variability. The value here doesn't account for the impact of local roughness on the confidence intervals.*

That's right. What we describe here is the significant minimum distance. The text has been updated.

*L558: "Amalgamation in 3D is still a potential issue": Amalgamation needs a more detailed treatment, given its potential impact on your findings it cannot simply be flagged as a potential issue.*

We brought further analysis on how our results are sensitives to amalgamation. See section 5.2.2, appendix A and fig. S13 and S14 in supplements.

*L561: "reactivation" needs to be much more clearly defined. However, your argument doesn't really depend on reactivation but on landslide detection in vegetation sparse or vegetation free areas. I suggest restating in these terms.*

We do not address the landslide "reactivation" issue anymore. This has been replaced by an evaluation of landslide under-detection based on the comparison with the manually mapped landslides.

*L566: "V-A relationship": depth-area should be examined as well as or instead of V-A because it removes spurious correlation with area. All three geometric properties are likely to be very severely affected by detection errors.*

This version of the manuscript now includes both V-A relationship and depth-area relationship (see section 4.3).

*L569-70: "largest and deepest landslides deviate significantly": these slides are also most likely to suffer amalgamation.*

This sentence has been removed.

**Figures**

*Figs: 3,4&6: Though they are very well presented I find the 3D plots (Fig. 3, 4 and 6) very difficult to interpret. I feel strongly that all these Figures should be presented in 2D map view instead of as oblique images.*

We modified these figures with a 3D view to a 2D view. We kept 3D views when it helped in the interpretation (e.g., Fig. 10)

*Fig 4: The colourmap in Fig4a does not match description in the caption or in the text. It appears to show a single colour for erosion and another for deposition. It also appears to apply a filter to areas with non-significant change. Areas of significant change are shown in Fig4b so the mask is not necessary here. By masking non-significant changes you lose the opportunity to compare the magnitude of change before significance tested. The colourbars in both Fig4a and c are a good length but contain little usable information. As I see it there is no change in colour*

*for erosion/deposition >1 m in Fig 4a and 5 m in Fig 4c. In addition, the scaling is asymmetric in 4c making it difficult to compare erosion and deposition.*

The colorbar has been modified to match changes up to 4m and has been saturated for higher values.

*Fig 5: The 3D minimum volume line needs explaining in the caption. I would have expected this line to be oriented parallel to the depth contours since minimum volume for a given area is set by minimum detectable depth. If so the minimum detectable volume might explain the sharp lower boundary to the volume area point cloud.*

Actually, what it is illustrated here is the 3D minimum volume that can be measured here given the minimum area (20 m²) that we consider and the minimum significant depth (~0.4 m).

*Fig 6: The inventory of Massey is useful additional information but scar outlines would be more useful than controids.*

This figure has been removed. Unfortunately we were never able to get the scar outlines of the first or the second inventory.

*Fig 6b: It would be useful to identify the largest four or five landslides in the right hand panel of Fig 6b either by giving them individual outlines or different colours this would give an indication of the degree of amalgamation.*

We now illustrate the amalgamation by showing the polygon of each landslide in figure 7a.

**References**

Aerial Surveys, Aerial photographs derived from two surveys of the study area carried out in 2014 to 2015 and in 2016 to 2017, by Aerial Surveys Ltd, 2017.

Dolan, J.F.: Data collection and processing report LiDAR survey of five fault segments (Estern Clarence, Western Clarence, Central Eastern Awatere, West Wairau and East Hope-Conway) of the Malborough fault system on the Northwestern portion of New Zealand's south island. Ph.D., University of southern California, 11 pp., 2014

Dolan J.F, Rhodes E.J.. Marlborough Fault System, South Island, New Zealand, airborne lidar. National Center for Airborne Laser Mapping (NCALM), distributed by OpenTopography. http://dx.doi.org/10.5069/G9G44N75, 2016.

Ankerst, M., Breunig, M. M., Kriegel, H.-P. and Sander, J.: OPTICS, ACM SIGMOD Rec., 28(2), 49–60, doi:10.1145/304181.304187, 1999.

Benjamin, J., Rosser, N. J. and Brain, M. J.: Emergent characteristics of rockfall inventories captured at a regional scale, Earth Surf. Process. Landforms, 45(12), 2773–2787, doi:10.1002/esp.4929, 2020.

Carrea, D., Abellan, A., Derron, M., Gauvin, N. and Jaboyedoff, M.: MATLAB Virtual Toolbox for Retrospective Rockfall

Source Detection and Volume Estimation Using 3D Point Clouds : A Case Study of a Subalpine Molasse Cliff, 2021.

Lague, D., Brodu, N. and Leroux, J.: Accurate 3D comparison of complex topography with terrestrial laser scanner: Application to the Rangitikei canyon (N-Z), ISPRS J. Photogramm. Remote Sens., 82, 10–26, doi:10.1016/j.isprsjprs.2013.04.009, 2013.

Larsen, I. J., Montgomery, D. R. and Korup, O.: Landslide erosion controlled by hillslope material, Nat. Geosci., 3(4), 247–251, doi:10.1038/ngeo776, 2010.

Martin Ester, Hans-Peter Kriegel, Jiirg Sander, X. X.: A Density-Based Algorithm for Discovering Clusters in Large Spatial Databases with Noise, Compr. Chemom., 96(34), 226–231 [online] Available from: https://www.aaai.org/Papers/KDD/1996/KDD96-037.pdf?source=post_page, 1996.

Massey, C. I., Townsend, D., Jones, K., Lukovic, B., Rhoades, D., Morgenstern, R., Rosser, B., Ries, W., Howarth, J., Hamling, I., Petley, D., Clark, M., Wartman, J., Litchfield, N. and Olsen, M.: Volume characteristics of landslides triggered by the M

W 7.8 2016 Kaikōura Earthquake, New Zealand, derived from digital surface difference modelling , J. Geophys. Res. Earth Surf., 0–3, doi:10.1029/2019jf005163, 2020.

McInnes, L., Healy, J. and Astels, S.: hdbscan: Hierarchical density based clustering, J. Open Source Softw., 2(11), 205, doi:10.21105/joss.00205, 2017.

Tonini, M. and Abellan, A.: Rockfall detection from terrestrial lidar point clouds: A clustering approach using R, J. Spat. Inf.

Sci., 8(1), 95–110, doi:10.5311/JOSIS.2014.8.123, 2014.

**Author's response to referee comments of Bernard et al. 2020 by Alexander Densmore.**

Thomas G. Bernard, Dimitri Lague, Philippe Steer

Univ Rennes, CNRS, Géosciences Rennes - UMR 6118, 35000, Rennes, France

**Summary**

*This is an exciting manuscript that reports on a promising way forward in detection and analysis of landslide inventories. The authors use the M3C2 point cloud analysis approach, previously developed by Lague and colleagues, to detect change in pre- and post-earthquake Lidar point clouds from the Kaikoura earthquake area in New Zealand. This is absolutely the right thing to be doing, and the authors are clear about how their approach gets around some vexing issues with current best practice*

*(which is to map using 2d imagery and to use volume-area scaling to get at landslide volumes). The topic is fully appropriate*

*to the journal and I anticipate that this work will attract a good deal of attention from the journal readership. do have some questions and suggestions for the authors to consider before the manuscript is published. Many of these are fairly minor and are detailed in the attached PDF; I will not repeat those here. But there are a couple of wider issues that are related to the clarity of what the authors have done, and in part to the division of material between the manuscript and the supplemental*

*information. In brief, I don't think it's possible to follow the authors' approach from the manuscript alone, and there are key parts of their analysis that can only be understood by going to the SI. I don't think that's right for a manuscript that purports to document a new methodological approach. These issues come under three headings (but please see the PDF for more detailed comments): First, because the authors are documenting a new approach for mapping landslides, I would expect to see some quantitative comparison of their results with a landslide inventory or inventories prepared in the more traditional*

*way. The authors show some amalgamated results (Fig 5, Table 2), but I think it would be useful to show a more systematic comparison with the Massey et al. results. Given that there are only 27 landslides in the Massey et al. inventory within the study area, this should be pretty straightforward - but I think it is important to demonstrate the extent to which their approach can match (or not) landslides detected by the alternative approach, as well as the additional landslides that they claim to be able to map. The only place I could see the Massey et al. landslides was as barely-visible centroids on a perspective view of*

*the study area in Fig. 6. Second, and somewhat related, the text is very unclear on how individual landslide sources and deposits are segmented and identified. Because of this, all of the resulting statistics of the area and volume distributions are uncertain in the reader's mind. This is more clearly explained in the SI... but again I don't think it's fair to require the reader to go to the SI to understand a methodological advance that is being proposed. I deliberately read the manuscript without going to the SI to see if I could follow it, and there are places (e.g., section 3.3.3) that are very difficult to understand and don't really address what has been done. I'd really recommend that the authors review the balance between the text and SI*

*and try to flesh out the explanations in the main text. The details of the sensitivity analysis (e.g., to the distance threshold D_m) can be left for the SI, for sure. Third, I found the authors' use of the term 'reactivation' potentially confusing, and I'd suggest that they choose a different way to express this. Assessing reactivation (e.g., of coseismic landslides in post-earthquake storms) is definitely a big issue in multitemporal landslide inventory analysis, and a few different approaches have been put forward, none of them very satisfying. Reactivation implies some renewed landslide activity that partly or wholly overlaps with a*

*landslide from a previous epoch - e.g. further erosion within a pre-existing scar, or headward/lateral progression of the scar edges, or erosion within a scar coupled with deposition on a pre-existing deposit... But that's not what the authors are actually talking about here, because they only have two point clouds spanning one epoch, and there's no independent dataset of pre-existing landslides. A better way of framing this part of the analysis might be around the following question: are there*

*landslides that would not have been recognised using the classical approach, either because they occurred on bare bedrock or because the vegetation contrast was too low, but that they can see with their method? That's an important question - but it's not the same as reactivation. Finally, a fairly minor point: I agree that the approach outlined by the authors is the way forward and that, where suitable Lidar data are available, this should be further pursued and developed. But I think it's equally fair to recognise that (1) suitably accurate and high-resolution Lidar data aren't always available, and (2) there are problems*

*and applications for which this approach simply isn't (yet) feasible. For example, for multi-temporal inventory creation over*
*the full landslide affected area, where the goal is understanding patterns of landslide occurrence and hazard but volume*
*estimation is a secondary concern, then a traditional 2d image-based approach might be fine. New Zealand and a few other*
*countries can fly repeat high density Lidar surveys; this capacity doesn't (yet) exist in most landslide-prone regions of the*
*world. I think it would be fair of the authors to acknowledge this - it doesn't detract from their analysis but perhaps places it*
*into a better wider context. To summarise, this is a really exciting piece of work. Once the authors have dealt with these issues,*
*then I look forward to seeing this published.*

**Minor Comments**

*L20: "It's not clear at this point of the ms what this clause means - so you might either expand/explain this point, or leave it*
*out. 90% of what?"*
We brought clarification to this sentence. It is now : "(81% of landslides with area < 300 m²)".

*L58:" You might also add that this effect complicates comparison of inventories that were constructed from different image*
*sources, because of differences in image spatial resolution, spectral resolution, and consequent ability to characterise surface*
*change... In other words, we don't even know if we're comparing like with like"*
The following sentence has been added to the text: "It can be further complicated by the use of different image sources because
of differences in image resolution, spectral resolution, projected shadows and consequent ability to detect surface change".

*L97:" This is an awkward phrase - maybe reword as 'identification of individual landslides'?".*
The text has been modified as suggested (L110).

*L103:" The black box showing the study area is not easy to see at first - perhaps make that white or yellow so it stands out?"*
The box has been colored in yellow as suggested.

L114: *"I'm not sure what this means"*
All LiDAR point cloud data contained information provided by the owner. One of these information is the classification which
attribute for each point of the point cloud, a number depending on the type of surface the point corresponds to (i.e. ethier it is
vegetation, ground, road etc..). Here, we only keep points from the original point cloud that correspond to the class 'ground'.
This sentence was replaced by "For both LiDAR point clouds, only ground points defined by the data provider are selected
(see details of the classification in Dolan, 2014) ".

*L115:" This seems like a really critical part of the analysis, and yet it is given only a single sentence. I'm not sure how a new approach for estimating landslide area and volume can be put forward without some kind of comparison to a more traditionally-created inventory. I agree that those inventories have problems, but otherwise there is no possible way for the reader to evaluate this new approach. So how did the validation work - what did you compare? I think this needs to be more clearly described »*

We added an entire section that compare our landslide inventory by 3D point cloud differencing and a manual mapping based on 2D images. See section 4.2. The text in this section 2. has been modified to introduced this manual mapping.

L121:" From who? »

The contact of the platform hosting the data has been added.

*L132: "I'm not sure what this means".*

It means that as core points represent a regular grid of points and that the results are "stored" on this grid, it can easily be import in any GIS rather than a non-regular LiDAR point cloud. We modified this sentence as follow: "it can be directly reused with 2D GIS as a raster (rather than a non-regular point cloud)" (L152).

*L135: "Just to be clear - are these normals to the pre-event or post-event data? The schematics in Fig 2 suggest post-event - is that the case? Might be worth mentioning that, either way"*

The normals are calculated from the core points of the post-event data. The text has been modified from "first dataset" to "core points". (L155).

*L180:" There is some repeated text here - can you please clarify?"*

The caption of the figure 2 has been modified.

*L182:" The caption and panels (a)-(c) don't seem to match - volume estimation seems to be panel (b), while separation of source and deposit seems to be panel (c). Can this be clarified? »*

The caption has been clarified.

*L186: "Removed? Corrected for? I'm not quite sure what you mean by 'adjusted'"*

The text has been modified as suggested.

*L191-192:" I'm not quite sure how this works - presumably as the threshold is varied the areas that appear 'stable' vary as well, so how can a single threshold value be assigned? Perhaps I'm misunderstanding what you have done - but as identification of 'stable' areas is important for the registration, it would be good to clarify this".*

This section has been re-written and clarified. Actually, after applying a first 3D-M3C2, we selected only areas with a 3D-M3C2 distance less than 1m to help the identification of stable areas. We then manually refined the stable areas to select only areas away from landslides. The fine registration is then applied on these stable areas (see section 3.3.1).

*L193:" is this rotation and translation, or just translation?"*

We let the algorithm free to adjust the data with rotation and translation. However, the resulting transformation matrix of the ICP only show translation. We let "rigid transformation" in the text to indicate that we did not imposed a translation for the co-registration of the two point clouds.

*L200:" This has already been written in the preceding paragraph"*

This section has been re-written and clarified.

*L218-219: "OK - but how is the location of 'the river' determined? How do you know what is fluvial and what is landslide-*
*driven? I am thinking for example of elevation changes at the base of a steep bank or talus that ends in the river channel -*
*these processes could be difficult to separate, no?"*

The river is determined at the bottom of the valley between distinctive banks and where it is free of vegetation. This was done by visual inspection of the orthophotos. Then, fluvial elevation changes were manually separated from 'mass-wasting' elevation changes in the river based on the shape of the patches of negative or positive elevation changes. All elongated patches
in the direction of the flow of elevation changes were considered as fluvial erosion or sedimentation. Deposits from mass-wasting processes was recognized as compact 'round' patches with highest elevation changes. Where fluvial sedimentation meets mass-wasting deposits, the latter was delimited by an abrupt change of the slope. Of course, this procedure comes with uncertainties that are difficult to quantify. The authors do not exclude the possibility that few fluvial changes have been merged with mass-wasting processes but we do not expect a significant impact on the total volume. It does not have any impact of
landslide source statistics.

*L222: "Remaining after what? I'm not sure what this is referring to"*

The term "remaining" was used in reference to the resulting point cloud after the removal of points located in the river. The word has been removed.

*L223:" Why not describe these in the order in which you carried them out? That also would fit with the order in Fig 2. Not clear why these have been switched around in the text"*

The figure 3 (in the current version of the manuscript) showing the workflow have been changed to match the text. However, the vertical-M3C2 needs to be performed before the segmentation of landslide sources and deposits. This aspect is just a matter of workflow efficiency. Doing the vertical-M3C2 after the segmentation would require to perform the calculation on each individual landslides.

*L225: "Is Np the minimum number of points or the number of sub-clouds (which is how it reads now)?"*

Np is the minimum number of points. It has been corrected in the text.

*L226-227:" This is a pretty important part of your analysis, so I think it needs to be better explained in the text. I know what you mean by amalgamation (although I'm not sure how you are assessing it), but I'm not sure what you mean by over-segmentation. I don't think the reader should have to go to the SI to understand this, as it's critical for going from topographic change to landslide process understanding. Can the concept be explained here, even if the details of how it is implemented are*
*left for the SI?"*

This section has been clarified (see section 3.3.3) and all the information is now in the core of the text. We explain how $D_m$ operates. As there is no objective way to a priori choose $D_m$ we explored various values and chose $D_m$=2 m as an optimal value between landslide amalgamation and over-segmentation, that is documented by the number of clusters that are created. We also analyse how the value of this parameter influence the landslide statistics and show that the value of $D_m$ do not significantly
influence the results for $1.5 < D_m < 3$ m (see section 5.2.2, appendix A and S13 and S14 in supplements).

*L228:" I'm not sure what this means - what kind of artefact? Do you mean 'core points that have a very large distance'? How do you know they are artefacts? »*

We use "artefact" to define either negative or positive false detection (L-301). To estimate the proportion of artefacts in the
landslide inventory, we first perform the landslide detection workflow on two subsampled versions of the post-EQ point cloud (see section 3.2). Due to the difference of sampling, our workflow detects artefacts and we define for each of them a mean signal-to-noise (SNR) ratio that is the ratio between the mean 3D-M3C2 distance and the associated uncertainty. We also define a mean signal-to-noise ratio for each landslide in the inventory. We then use a SNR threshold from which the presence of artefact is limited in the landslide inventory (see section 3.3.4).

*L230-233 : « I found this text hard to follow, and it didn't seem to explain to me the focus of the sub-section - which is how sources and deposits were separated. What does the final sentence mean - how was a minimum surface of 20 m2 'imposed'? Imposed on what? What does the final clause of that sentence mean? Up to here the text has been very clear, but I'm really not sure how the source and deposit have been discriminated. Because that's so critical for the volume distribution information*
*and indeed much of the remaining text, I think that needs to be really clear".*

This section has been subject to significant modifications. We clarified how we perform the segmentation of landslide and how the value of the parameters are chosen. By "imposed" we meant that the minimum landslide area is set by the method. We do not use the word anymore (see section 3.3.3).

*L239:" Is this still using D=10 m and d=5 m? More generally, this approach seems to introduce some of the issues created by using a DoD approach, except without the cellwise averaging/interpolation problem. But for steep rock slopes/rockfalls, this seems like it would introduce substantial uncertainties. I haven't thought this through, but I do wonder whether sticking with the 3D normal approach, but introducing a set of rules or hierarchies to deal with shadow zones, would be preferable. At the very least it seems worthwhile to explore - whether in this ms or elsewhere."*

Here, the projection scale is 5 m. We agree that the vertical measurement approach to estimate volume may be incorrect on very steep slopes and that it would be preferable to estimate landslide volume in 3D. This is highlighted in the discussion (L568). However, the median landslide slope measured on the pre-EQ point cloud is 34.6° and only 0.74% of the landslide area have very steep pre-EQ slopes (>60°). In addition, we do not expect that measuring volume in 3D from a constant surface normal direction for each sources and deposits would be better than a vertical measurement when we have very complex amalgamated landslides as we observed in the dataset (Fig. 7c). However, new approaches based on 3D mesh reconstruction seem promising (Benjamin et al., 2020).We discuss about this issue in section 5.1.2.

*L251:" Was there water in the river? If so, are the lidar data measuring the water surface, or is there some penetration, or a mix of both? Do you have any sense of how changes in water level between surveys could contribute to the distances that you*

*estimate?"*

Both LiDAR used to generate the point clouds are topographic LiDAR using NIR wavelength which means that the laser used does not penetrate into the water, and is actually fully absorbed. Hence, where there is water, there is no data. Because the lidar survey were acquired in summer, the water level were extremely low, and do not appear to create significant area without water. We have not added this kind of information in the MS as it is not central to the topic of the paper.

*L252:" I'm not sure about the evidence for this statement - is this based on comparison of the orthophotos? Did you overlay the changes in Fig 4 on the post-event orthophoto to see how they correspond? You mentioned using the orthophotos as a validation tool, but didn't really explain what this meant. This is really critical, because without some independent validation, the reader has no basis for understanding (or accepting) what you consider to be a 'landslide' and what you consider to be an*

*'artefact'."*

This sentence has been changed. We now discuss about large patches for which the association with a process is pretty straithforward. We also added a comparison between the landslide inventory resulting from 3D point cloud differencing and a landslide inventory manually created from interpretation of orthophotos. See section 4.2.

*L254:"'previously' - and what is this statement based on? How do you know that it was unstable?*

The text has been modified and 'previously unstable" has been removed.

L256:" At this point of the manuscript, you haven't explained how you know (or can state with confidence) that areas of positive distance change are landslide deposits. I think it would be better to first document the pattern of change that you
observe, and then demonstrate how you correlate that (or not) with a particular geomorphic process".

In this section, we now first describe the pattern of large detected changes for which a source area can be associated to a deposit area. These large patterns can easily be linked to landslide processes and we do not think there is any ambiguity about that. For smaller patterns of change, the application of the SNR filtering now removes a very large number of positive (and negative) change which could be ambiguous. We also provide details of typical landslide processes that clearly show where deposits
occur (e.g., fig. 10). We have also added in the discussion (section 5.1.2) a paragraph discussing the different processes we inferred from both the 3D-M3C2 distance field and ortho-imageries and the limit of our approach to confidently identify the dominant landslide mechanism for each sources and deposits.

*L259:" What does this mean? How are these areas identified?"*
Artefacts includes both negative and positive false detections (definition L-301). Artefacts are estimated from the Same Surface Different Sampling test (SSDS; section 3.2) and filtered by the signal-to-noise ratio in the landslide inventory.

*L270:" Line 264 gives the same maximum but ascribes it to an erosion area, not a deposit - can you clarify?"*
This mistake has been corrected.

*L271:" Is it intentional that the colorbar is either blue or red? Or is there a gradation at very small +/- values that can't be seen by the reader? If so then I think it would be good to show this, perhaps with a non-linear color bar - because otherwise the pale blue/red colors can't be understood".*
The colorbar has been modified to match changes up to 4m and has been saturated for higher values.

*L275:" This figure looks to show continuous values of erosion and deposition, which is what I'd expect as the output from the vertical M3C2 step. But the caption seems to suggest that sources and deposits have been segmented, which isn't obvious from the figure. Is it necessary to say that? It would be more clear if you showed the unsegmented raw vertical distance change, and THEN the results of the segmentation (which presumably could be shown as grouped polygons?). Related to this, while*
*the perspective views look interesting, they also hide a lot of the features of the dataset. One perspective view early on might be useful, but I'd argue that these results are better shown in map view. Just an opinion, however - other readers might disagree*

"

We agree that the caption can be confusing regarding the map of the vertical-M3C2. The caption has thus been modified by removing the number of landslide sources and deposits. We also added a landslide source map for which each individual landslide is colored by a single color in figure 7 (section 4.2). In addition, 3D view has been changed for map view.

*L279-280:" OK... but surely one of the advantages of this approach is that you are not constrained to only show planimetric areas! For example, it would be great to look at both planimetric and true surface area, to see how wrong we typically are by only using 2d imagery to map from"*

Computing true surface area requires significant new developments of the workflow in relation to segmentation, to better handle complex cases of landslide amalgamation. We have added in the discussion quantitative elements based on the typical slope of landslide sources to discuss the magnitude of the difference between planimetric area and true surface area (see section 5.1.2). Given that the median of the slope distribution of our landslide source inventory is 34.1, the estimate of the true area gives a total landslide source area of 356,876 m² rather than 286,445 m².

*L291:" These are all fairly old refs... I know that Hakan Tanyas has considered this issue recently, so might be worth making reference to that work?".*
The reference to the work of Tanyas et al., 2019 has been added here.

*L302-303:" This text doesn't quite make sense - missing word"*
The text has been clarified: "the minimum volume that we can confidently measure should be 8 m$^3$".

*L310:" I'm not sure that this is worth including, because the goal of any scaling relationship is capturing the ensemble or inventory behaviour rather than the scaling for an individual landslide"*
We suggest to leave the approach here as it is. We illustrate in section 5.1.4 how the method used to fit the data influence the total volume estimation (L655–657).

*L319:" There's not enough information for a reader to understand this sentence - what are cleaned or soil-dominated landslides? Are these landslides mapped by Massey et al. 2020 ? »*
In their paper, Massey et al. (2020) estimated different V-A scalings in function of different landslide type. To simplify things, we removed the V-A scaling relationship estimated for their soil landslide type as it is close to the V-A scaling obtained with their entire landslide inventory.

*L324-325:" This seems like a point that is better made in the discussion"*
This sentence has been transferred to the discussion as suggested (see section 5.2.4).

*L325-328:" Evidence for these statements? Overpredicts or underpredicts relative to what?"*

This sentence has been removed.

*L328-330:" Again - this is better left for the discussion. If you cannot distinguish between 'shallow' and 'deep' landslides, then why bring this up? It strikes me that, with direct measurement, this is no longer a meaningful distinction anyway, as presumably your data are consistent with a distribution of landslide depths..."*

This sentence has been removed. We now show and discuss the landslide depth-area relationship of our landslide inventory (see section 4.3 and 5.2.2).

*L332:" What does this mean?"*

$N_{LT}$ is defined at the beginning of the section 4.3 as the number of landslide sources. However, it is true that the notation was not respected which can be confusing. We corrected this in the text by writing "$N_{LT}$".

*L333:" Are the Massey et al. data limited to landslides within the same study area? What does V2 mean? Given that you are making an explicit comparison between their data and yours, I am surprised that you are only mentioning that study here in the results. It would be good to mention their work in the intro or methods, including enough information about what they did so that the reader can both understand what's plotted and appreciate why their results might be different from yours"*

The data of Massey et al. (2020) are not limited to our study area but include landslides over the Kaikoura region representing 1095 a total area of 6,875 Km². In the paper of Massey et al. (2020), they define the "V2" landslide inventory to refer to the landslide inventory started in 2018 and updated in 2020. To simplify, we removed "V2" in the manuscript. We now refer to their study in the introduction (L.101-102).

*L335:" This should be 'relationships' because more than one is shown. I don't understand the legend or the wording in Table* 1100 *2 - you refer to fits on 'averaged data' and 'raw data', but that's different than the way you've described this in the text (lines 310-315). If that's what you're referring to, can you make these consistent? "*

The text has been corrected as suggested and we modified the legend in Table 3 to be consistent with the text.

*L335:" The green and red colours will be difficult for colour-blind readers to distinguish. Again, without knowing what Massey* 1105 *et al. (2020) did, it's hard to understand the difference between their 'all landslides' and 'soil landslides' scaling relationships - I think this needs to be introduced in the intro sections of the ms"*

The color have been changed.

*L340:" It took me a few reads to understand this - the column headers could be read as a calculation ( log[b-d]/log[alpha] ).*
*I think this could be a little more clear".*

Clarification have been brought. The column header is now: "log b, log d or lod α".

*L346: " Without knowing anything about the study area, this doesn't seem like a convincing definition of 'reactivated landslides'. And I wouldn't necessarily agree that this matches the 'classical approach' - especially because the 'classical approach' of looking for vegetation change, either qualitatively or using NDVI trajectories, is notoriously incapable of spotting reactivation. There is also a question here of what you mean by 'reactivation' - does this include further erosion within a pre-existing scar, or headward/lateral progression of the scar edges, or erosion within a scar coupled with deposition on a pre-existing deposit, or...? I think 'reactivation' is not really what this is about because you don't have an independent dataset of pre-existing landslides, so you can't say what has or has not been reactivated. A better way of framing this part of your analysis might be around the following question: are there landslides that would not have been recognised using the classical approach, either because they occurred on bare bedrock or because the vegetation contrast was too low, but that we can see with our method? That's an important question - but it's not the same as reactivation"*

We agree that "reactivation" is not what we actually looking at. This section has been removed and we added a new section in which we compare our landslide inventory based on 3D point cloud differencing to a manual mapping based on 2D images (See section 4.2). We then compare both approaches in terms of similarities and differences and discuss sources of under-detection by both approaches.

*L356:" This is exactly the kind of comparison that I would expect in order to evaluate both your change detection and your segmentation routine. I think this needs to come earlier in the results. Why are you comparing against Massey et al. (2018) here and in Fig 6, but Massey et al. (2020) in Fig 5? "*

We added a comparison of our result to a manual mapping based on 2D images. See section 4.2. The comparison with the landslide inventory of Massey et al. (2018) here was due to the availability of the data. The landslide inventory of Massey et al. (2020) is not available online and we never received an answer from the authors to our requests to get the extent of their mapped landslide sources and deposits.

*L356:" These are not the same thing - and the issues involved in their detection are not the same either. I don't think you should be grouping these together »*

This section has been replaced by the section 4.2 in the new version of the paper.

*L361: "I'm still not 100% sure how you have separated source and deposit..."*

The segmentation procedure has been clarified, please see section 3.3.3.

*L361: "The centroids are really difficult to see on the perspective view - and will be impossible for colourblind readers to see. Why not show polygon outlines? That's the most direct comparison between your results and theirs, and you could even summarise in terms of an ROC curve or confusion matrix. As before I think the perspective view decreases the utility of this figure rather than increasing it - I'd really suggest showing this in map view as well, or solely in map view perhaps"*

We did not show the landslide inventory of Massey et al. (2018) as polygons because it is not available online and we never received an answer from the authors to our requests to get this information. However, we added a section in which we analyse the results from 3D point cloud differencing compare to a manual mapping based on 2D images. See section 4.2.

*L363:" Are these vertical or perspective views? And they need a separate scale bar".*

This figure has been removed.

*L364: " To help the reader, you could usefully add the dates to these panels - there is plenty of space to do so."*

This figure has been removed.

*L371:" As written above, I'm not sure that you overcome these limitations because your volume numbers come from a vertical application of M3C2!"*

It is true that a vertical distance measurement is not the most appropriate approach to compute volume. However, here we compare the DoD approach to our 3D point cloud differencing approach only in terms of topographic change detection for which with compute distance in 3D.

*L376-380:" This partly restates points that were made in the introduction - I think you can leave this out and just focus on the novelty and importance of your approach (which you cover below)"*

This paragraph has been removed.

*L387:" This doesn't quite make sense because there isn't a single distance measurement here. Do you mean a non-null mean, or median? I think this could be a little clearer"*

The text has been replaced by "a non-null mean distance" as suggested.

*L397:" I think I know what you are saying, but the wording here is a little odd. In the previous sentence you are talking about the benefits of 3D M3C2, but here this sounds like a drawback. Can you clarify? »*

The idea here is that the distance uncertainty compute by M3C2 will be higher in vegetated area due to (1) a lower point density and (2) a higher roughness of the point cloud. Consequently, distance measured by M3C2 in such area needs to be higher than in flat area to be significant to prevent in part for the detection of false positives and false negatives. The text has been clarified.

*L407: "This is a very effective figure. You could get rid of some of the trailing zeros on the y-axis, and also format the numbers in the legend to match the text"*

Updated as suggested.

*L411:" See query above on whether this is translation and rotation, or just translation".*

We let the algorithm free to adjust the data with rotation and translation. However, the resulting transformation matrix of the ICP only show translation. We let "rigid transformation" in the text to indicate that we did not imposed a translation for the co-registration of the two point clouds.

*L438: "What do you mean by this? Spatial density? Or something else?"*

Indeed, we talked about landslide spatial density. However, this paragraph has been modified and integrated to the new section 5.1.2. This sentence has been removed.

*L439-440: "This sentence would make more sense to me if you are talking about landslides for which \*little\* topographic change occurs in the direction of the surface normal. Is that what you mean? The cartoon in Fig 8b shows such a case - there is negative surface elevation change in the source and positive change at the toe, but little/indeterminate change over much of the body of the slide. Can you clarify if this is what you mean? »*

This is exactly what we meant. However, as explained in the previous comment this paragraph has been modified and this sentence does not appear anymore.

*L454 : " repeat the uncertainty given on this number in section 4"*

The uncertainty has been added.

*L460: "I'm not sure what you mean by this - more concentrated in terms of what?"*

We meant that the landslide deposit areas are on average smaller than landslide source area. We have largely rewritten this section to make things clearer (hopefully !)

*L470: "what is N_LT?"*

$N_{LT}$ is the number of landslide source. It is defined in eq (3).

*L475:" You've expressed the exponents as negative numbers so be consistent here as well"*

The text has been modified as suggested.

*L492: "True... but what fraction of the total area?"*

We do not address this question anymore.

*L494: "In what sense? I think this sentence would be more effective if you could be specific. For me, why this potentially matters is that small landslides may be negligible in terms of their contribution to regional-scale erosion or sediment transfer,*

*and yet still collectively have tremendous impacts on the exposed population. Even a very small landslide can cause casualties or infrastructure damage. It's worth reminding the reader of that, perhaps... »*

We now address this question in section 5.2.2 at lines 749-752.

*L501 : « By 'the best estimation' do you mean the estimate that is closest to yours?"*

We removed the V-A scaling relationship of soil landslide type of Massey et al. (2020) and removed this sentence.

*L503: "You could perhaps remind the reader why this is a 'better result'"*

The text has been replaced by: 'gives the closest approximation of the total landslide volume estimation."

*L571: "have observed [and you only cite one study - if there is only one, then why write 'inventories'?]"*

This sentence has been removed.

---

## Referee Report (RR1)

[referee-annotated manuscript omitted]

---

## Referee Report (RR2)

**Second review of Bernard et al. 2020 by David Milledge.**

**This is an excellent paper that makes at least two significant contributions. The first is methodological, detailing a method with which to robustly determine topographic change over large areas that include steep slopes. The second is substantive, demonstrating that the size distributions (volume and area) and geometric scaling relationships for landslides differ from those previously found for landslides. The authors have considerably improved the manuscript since my last review. However, there are a number of major and minor comments from my previous review that have yet to be fully addressed. Below I have retained only these comments and explained (in bold) why I feel that they remain unaddressed. I have also made minor comments on the new draft of the manuscript by commenting on the PDF.**

**The workflow that you have introduced has great potential to improve the quality of landslide inventories. The paper is a significant and rigorous contribution because it: 1) introduces the workflow for a suitable case study, 2) shows that the workflow improves on alternative 3D methods and can detect landslides not detectable in 2D methods; 3) highlights common errors in 2D methods that have been proposed but rarely demonstrated, and 4) demonstrates for the first time that you can calculate a volumetric budget for landslide derived topographic change without the need for volume-area scaling relationships which are known to (and which you show) introduce considerable uncertainty. It also opens up a discussion about what constitutes a landslide and how this differs from the things that we currently map, whether in 2D or 3D.**

**I agree with you when you say in your response that** *"despite its limitations, it currently represents the state of the art in terms of 3D landslide detection and landslide inventory creation"*. **This is an excellent methodological contribution and I have only very minor comments on presentation of the paper in relation to this aspect of the work. As a methodological contribution I agree when you say that:** *"Scientific research is incremental, and we fully expect that our workflow will be improved in coming years by others, as it was the case for 2D landslide inventories.* **In relation to this, it is excellent that you:** *"provide all the elements (code, data) for other researchers to apply the workflow and reproduce our results, or apply the workflow to their data."* **The reason I am so demanding of the checks you apply to the 3D method relative to the 2D method is that the flaws in the 2D method are relatively well rehearsed in the literature but you are presenting the 3D method as a new (and better) technique. To do so you must demonstrate that this is the case.**

**However, I strongly disagree that you** *"clearly demonstrate, at least in our study case, that the rollover in the pdf(A) observed in our 2D inventory but not in the 3D one, is due to a size-dependent under detection in 2D."* **I disagree because I think that non-trivial errors remain in: 1) the detection of small landslides, where some false positives remain due to spatially correlated errors; and 2) defining the boundaries of landslides, where automated segmentation continues to result in amalgamation. I know that I come to this with a bias: I have interpreted my own field observations as indicating a rollover in landslide size and have developed theoretical explanations for that rollover. So I am probably resistant to the idea that landslide size distributions lack a rollover. I'm trying to avoid this bias but may not manage it.**

**You could easily address my outstanding major concerns by softening your claims. For example, you say in your response:** *"its application in comparison with a 2D landslide inventory shows that landslide under-detection in image based inventories is extremely prevalent in our study area"*. **I broadly agree but I would say: 1) that the comparison "suggests" rather than** *"shows"* **because you cannot identify which inventory is in error only argue which is more probably the source of the error; and 2) that the under-detection is "present" rather than** *"extremely prevalent"* **because you can argue that it is very likely that some of the size-dependent bias between inventories is extremely likely to be due to 2D under-detection but some is also extremely likely to be due to 3D over-detection and you cannot currently identify their relative share.**

825 **Two of your key conclusions (stated in the abstract) are that the manually mapped 2D inventory** *"severely underestimates total area and volume"* **[L20] and that there is** *"a systematic size-dependent under-detection in the 2D inventory"* **[L24]. However, both these statements are underpinned by an assumption that the 3D sources are correct (i.e. the ground truth) such that differences between them and the 2D inventory are attributable to error in the 2D inventory. This assertion needs to be justified in the paper but it is not at present. Instead you consistently assert and**
830 **assume that in cases where the two inventories differ it is the 3D inventory that is correct.**

**Another key conclusion of the paper is that the 2D size distribution has a rollover whereas the 3D distribution does not and that this is due to missed landslides in the 2D inventory. However, it is not clear that this is a fair comparison. The rollover is detected in the manual mapping on the basis of a reduced frequency of landslides in the smallest class 13-20**
835 **m$^2$, relative to the class 20-31 m$^2$ (which is the modal class). This smallest class is below the lower limit of detection for the 3D method. If you enforced a single consistent lower size limit for your analysis and censored all landslides smaller than this limit for both datasets then I don't think you would conclude that the manual mapping displayed a rollover. Note, that the x-axis values for the size distributions in Figure 8 are lower bin limits not central values this is potentially confusing and should be adjusted.**
840

**I think you make one further important finding that you could highlight in the abstract: you demonstrate the variety of types of topographic change that occur in response to an earthquake and show that existing 2D landslide mapping captures only a small part of that range. Your results prompt questions about what constitutes a landslide within these landscapes and how we should delimit them. This is particularly important for size distributions because the way that**
845 **you define both your term landslide (to say what is in or out of the class) and the boundaries of your landslide in space on the basis of post failure observations will differ depending on the motivation for examining them. For example, your point on L420 highlights the complexity of mapping post-earthquake topographic change and relating it to processes. Should subsidence / retrogressive slumping upslope of a catastrophic landslide be included within the same source zone? Is this one landslide or two? The processes and perhaps even the timing of movement are quite different. But it**
850 **is a very important point that these movements will not be captured in conventional inventories though there is widespread recognition of the processes you discuss based on field reconnaissance.**

MC1) The paper needs to more clearly define: 1) landslides (i.e. what the inventory includes) and 2) what your inventory can and cannot be used for.

*Early on you introduce the idea that there are different landslide processes (L43, "process specific") but you don't follow this logic through into your results. Instead, your analysis may contain an implicit definition of landslides as all processes responsible for surface change that cannot be attributed to fluvial processes (L217-9). You certainly need to make this definition of landslides explicit in your introduction.*

*The introduction needs a much clearer explanation for what you expect the inventory to be useful for. If it is for understanding landslide mechanics then it is essential that you make an effort to distinguish individual landslides on a mechanistic basis (see MC3 on amalgamation). If that is not an expected application of the inventory you should say so, otherwise there is a real risk it will be misapplied.*

*Appropriate uses of the inventory (e.g. volume estimation or landslide mechanics) depend not only on its purpose but also on entry criteria into, and distinctions within, the inventory. Non-fluvial surface change that might result from earthquake shaking includes: tree-throw, ravel, rockfalls and slides, slow earthflows, rapid soil slides and debris flows. The processes responsible for these surface changes differ from one another to varying degrees. If there is no distinction between them this precludes the inventory's use in analysis of landslide mechanics and therefore prediction. It allows comment on correlation e.g. of volumes and areas of change, but makes it extremely difficult to make any inference about causation. It also opens the work to the criticism that the bulk statistics mask important differences in behaviour between processes. For example: the differing size distributions for rockfalls (where others have reported no detectable rollover) and landslides (where there usually is).*

We added the following definition of what we consider as "landslide" in the introduction of the manuscript: "We use the generic term of "landslide" to define the spatially coherent changes detected by our method on hillslopes that result in at least several decimeter erosion (i.e.,scars or sources) or deposition". The discussion now features an entire paragraph (L596) addressing the various type of landslides that can be detected by our approach. The aim of this paper is not to better understand landslide mechanics as we cannot confidently identify the different landslide processes we detect. We are mainly interested by the estimation of co-seismic volume and to overcome issues such as under-detection and amalgamation on volume estimation. The introduction now clearly integrates these two problematics. We also believe that the new filtering approach that we introduce, which results in 3 time less landslide sources compared to the initial MS, results in a much more robust inventory.

**RE: definitions. Simply defining landslides as decimetre scale change not due to fluvial processes is accurate but will be a very unusual definition to the reader. You can help them to see that your definition of a landslide is consistent with theirs by adding a little more explanation and I think that would be very worthwhile.**

**Crozier suggests that: "*The three most widely used classifications involving landslides (Sharpe, 1938; Varnes, 1958 and 1978; Hutchinson 1988) separate 'mass movements' (Fairbridge, 1968) into two categories: subsidence (which is the vertical sinking of material-see entry on Land Subsidence) and those movements that occur on slopes. These' slope movements' are then usually divided firstly into 'landslides,' as defined above, and secondly into the slower, more widespread and ill-defined movements such as 'creep,'' sagging,' and 'rebound.*" The landslide definition that he refers to is: "*the downward or outward movement of a mass of slope-forming material under the influence of gravity, occurring on discrete boundaries and taking place initially without the aid of water as a transportational agent.*"**

**Crozier M.J. (1999) Landslides. In: Environmental Geology. Encyclopedia of Earth Science. Springer, Dordrecht.**

**I think you can make the case that most of the change that you detect can be classified as landslides following the definitions of Sharpe (1938), Varnes (1958, 1978), Hutchinson (1988), and Crozier (1999). But you need to make that case. If you explain the timescale over which the change occurs and the spatial limits of detection that you will ultimately impose then you can argue that everything that you detect should fall within the class of landslide. It would help your later discussion if you gave a summary of what that might include (e.g. slides where the failed material is entirely removed from the source zone and those where movements that are small relative to the length of their failure surface). It would also help to explain which non-fluvial mass movement processes are not detected, particularly: tree-throw, ravel and other forms of creep (because the movements are either too small, too localised, or too slow).**

900 It would perhaps also be worth saying that this definition differs from those commonly (implicitly) applied in 2D landslide inventories derived from satellite imagery since these rarely (or incompletely) capture slides where material is displaced by only a fraction of the failure surface. These inventories are censored by their ability to detect change from image properties and thus rarely capture rockfall source zones. The same censoring results in under-sampling of small landslides, landslides in bare or sparse vegetation and landslides obscured by forest canopy because these can't be confidently identified.

905

**RE: expected uses.** Clarifying the focus on co-seismic landslide volume estimation is useful as is the section that you have added on the different processes represented in your inventory.

MC2) Elevation errors need to be better quantified and more thoroughly discussed

910 *The manuscript needs a more thorough treatment of errors in the topographic data dealing with both: 1) the properties of the elevation errors that you have identified (e.g. spatial pattern, wavelength, covariation with landscape properties); and 2) the possible sources of error.*

The reviewer suggested many areas to explore that are extremely interesting, but which, for some of them, would constitute an entire paper by themselves, in particular when it comes to the analysis of error properties suggested by the reviewer.

915 **I am not suggesting that all these areas need to be exhaustively explored but that the inferences that you draw from them must be stated with appropriate confidence for the uncertainty in the data that underpins them. Mass balance, which was your primary objective is largely insensitive to the errors that I highlight here. Landslide size distributions and scaling relationships are potentially very sensitive to these errors.**

We also aim at developing a generic workflow applicable to a variety of cases for which users may not necessarily perform
920 extensive error properties analysis.

**Your contribution in developing a workflow is extremely valuable and is not compromised by the continued presence of these errors. However, prospective users will also use this paper as a model for what the workflow can be used to calculate. Your discussion of errors and their implications is therefore important because it will influence not only how people interpret your results but also the capability and limitations of the workflow.**

925 Hence, to improve the paper we have worked on two aspects: (i) improving and better identifying errors in our dataset, (ii) defining a new confidence metrics (the SNR) for each landslide source or deposit to filter out landslide with low confidence.

**Both these aspects have very considerably improved both the workflow and its application in this paper. However, you have not addressed my original concern about spatially correlated error in this comment.**

*RE error properties: The amplitude and correlation length of elevation uncertainty from different sources and how they*
930 *interact to generate a 2D elevation error field with a particular amplitude and wavelength is really important for this particular application, where landslides are identified by thresholding then segmenting differences. The error appears to have a fairly long wavelength in many areas (tens to hundreds of metres). It also appears to have some aspect dependence. The spatial correlation of these errors is important because you assume uniform isotropic 110 registration errors.*

**I do not see where this comment is addressed in the response. As far as I can tell: 1) there are spatially correlated errors**
935 **in your difference surface; 2) these errors will not be captured by the SSDS analysis; 3) correlation of errors implies that if one core point has errors large enough to exceed LoD then there is a non-trivial probability that one of its neighbours will also have errors that exceed LoD; and 4) the distribution of erroneous patches will be strongly right skewed (i.e. smaller patches more probable than larger patches). I would be keen to know whether you agree. On this basis, I think you must quantify the impact of spatially correlated errors on your size distribution if you are to argue**
940 **that your measured size distribution is the 'true' distribution or even that it is more correct than the 2D distribution.**

*RE error sources: It isn't clear to me what you mean by imperfect alignment, nor ICP related errors (L204-5). Identifying errors on Fig 3 and hypothesising the sources from which they derive are useful but need to be discussed in the manuscript as well. You recognise the presence of "internal flight line height mismatch" and indicate that it results in "large scale low amplitude topographic change" (L418-22). They should be introduced earlier in the article with a more complete explanation*
945 *of what they are and how you found them.*

**Errors in our dataset:** entirely revisiting our data, we identified two sources of errors: (1) Remaining LiDAR point cloud misclassification in forest areas, inducing local topographic errors, and (2) imperfect flight line alignments from the pre-EQ data, inducing topographic errors of longer wavelength. To address the first issue, we first removed as much misclassified points as possible by interpolating a surface and remove outlier points (see detail in supplementary material and section 2 in
950 the paper). To address the second, we estimated residual errors of each flight lines due to imperfect flight line alignments composing the pre-EQ point cloud and defined the registration error *reg* based on the maximum residual error of the flight line misalignments (section 3.3.1. and S3 in supplements). Compared to the previous version of the MS, the *reg* is now 3 cm larger (20 cm vs 17 cm). We also show that only 1% of points are detected as significant change in the stable area, validating our choice of $LoD_{95\%}$. While our *reg* is considered uniform over the study area, the $LoD_{95\%}$ (eq.2) also take into account the local

955 point cloud density and roughness which are correlated to the presence of vegetation. The $LoD_{95\%}$ is thus spatially variable. In addition, we are aware that, ideally, a spatially variable model for point cloud error and registration would be preferable for each survey and combined into a more accurate and complete form of *LoD* than what the M3C2 approach currently offers. However, in the absence of the position and attitude information of the sensor (e.g., Smoothed Best Estimate of Trajectory file) and raw LiDAR data - rarely available on LiDAR data repositories -, or of dense ground control which is hard to get in
960 mountainous environment, it is currently impossible. We now discuss this in the discussion (section 5.1.2).
**This is a useful explanation but the decision to assume that registration error is spatially uniform still needs justifying in the text in a way that addresses the concern that long-wavelength errors might combine with short wavelength errors to generate patches of erroneous change in some places and break up patches of true change in others.**

965 *Error consequences: It would be useful to say something about the implications of the topographic errors. I can see two implications: **First**, incorrectly assuming uniform isotropic errors will result in a confidence interval for identifying significant geomorphic change that is too strict on some slopes (e.g. some aspects) and not sufficiently so on others. This in turn will lead to false negative change detection on some slopes and false positives on others.*
*Second, change detection false positives will result in false identification of landslide objects or false representation of their*
970 *geometry. False negatives are equally problematic since they could result in not only changes to landslide geometry but also cluster breakup (biasing the size distribution). These problems are illustrated in Fig 3 where a number of error patches would be identified as landslides by the detection algorithm if these areas were not assumed to be stable. If such false positives exist here it is they likely also exist elsewhere.*
*It is essential that you quantify their impact on your findings.*
975 **Filtering landslides with low confidence with the SNR**: To limit the false detection due to these errors, we also defined a signal-to-noise ratio threshold to efficiently remove suspicious landslides (section 3.3.4 and 4.1). This index is based on the mean ratio between the 3D-M3C2 distance and the $LoD_{95\%}$ for each landslide. We provide a way to evaluate the optimal value of SNR by comparing the number of landslides in the database to a case with no change (i.e., two versions of the same data but with different sampling, a test that we now call Same Data Different Sampling test (section 3.2). This SNR filtering removes
980 a very large number of landslide source and deposits (fig. 5b), and in particular long wavelength low amplitude changes that occurred due flight line misalignment in the pre-EQ data, as well as many small landslides in forested region where point density is very low. We provide a systematic analysis of the impact of SNR (and *reg*) on pdf (A) (fig. 11), pdf (V) (fig. S 13) and V-A relationship (fig. S14).
**This text does not directly address my concern above. SNR filtering is a good addition to the workflow and it does**
985 **appear to successfully remove a very large number of changes that were erroneously identified as landslides before. The only test of whether this filtering is sufficient is the SSDS (Same surface different sampling, SSDS and SDDS used interchangeably in both the paper and the response). If I have understood it correctly, this is a very weak test of the workflow because it examines only the impact of random errors and ignores spatially correlated errors in the surfaces.**

990 *Suggested additional analysis: You could apply your landslide detection method to only the pre-defined 'stable areas' and generate landslide geometries. These geometries and their scaling relationships might indicate the impact of error on your findings (particularly: number and area density, pdfs, and scaling relationships of artefact landslides). If the results are similar to your findings for 'non-stable areas' it would be very difficult to argue that the data support your claims with any certainty (the same results could have been generated purely from topographic errors in the absence of any landslides).*
995 **Did you undertake this analysis? It seems straightforward to do but I don't see any response to this comment in your response. You do not use the mapped distribution as a test for your distribution and I understand why. However, you don't currently offer any independent test of your distribution.**

MC3) You recognise that segmentation results in amalgamation but don't quantify its extent or impact

*Severe amalgamation can result from automated segmentation of a thresholded classifier. In the landslide maps that you show here (e.g. Fig 2), amalgamation appears a severe problem for the largest landslides. The argument that it can be solved by tuning Dm (as you suggest on L426-7) is unconvincing since two separate landslides can be within millimetres of one another but have different failure mechanisms. You later say that you "cannot resolve the amalgamation" (L511) and that it "is still a potential issue" (L558). I would argue that it is not potentially but certainly an issue. Your figures show that amalgamation is present (perhaps pervasive) in your inventory but its extent or impact is not quantified. The total volume is insensitive to amalgamation but your landslide pdfs and scaling relationships are not.*

We agree that the segmentation approach we use certainly do not allow to solve the amalgamation problem, and this is highlighted in many parts of the MS. However, the problem of amalgamation is inherently subjective and plagues all inventories.

**I agree, that segmentation is a subjective problem but you have made it reproducible by removing the subjectivity. The problem is that the best reproducible (i.e. automated) segmentations still perform poorly (with respect to the segmentation that a human mapper would choose).**

Our 3D data reveals a level of complexity, and a density of amalgamated landslides which makes the definition of a single landslide in relation to an ideal rupture mechanism extremely difficult.

**This is a really important point and could be a key contribution of the paper. Your results show that it's complicated. Far more complicated than we capture in conventional 2D inventories.**

Even the segmentation of the 2D inventory proved to be extremely complex and is entirely not reproducible. Hence, we favour a reproducible approach, even if currently limited, that can be applied exhaustively to much larger datasets, than a non-reproducible one (2D manual mapping) that we now demonstrate misses a very large number of landslides and incorrectly map their contour.

**It isn't clear why reproducibility is favoured over skill. If you think that manual segmentation would outperform connected component segmentation it seems strange to continue with automated approach because it is reproducible. You say that 2D manual mapping "misses a very large number of landslides" but this is on the assumption that the 3D inventory is correct.**

In this new version of the paper, the landslide amalgamation can be visualized with a map of the landslide source colored by individual landslide as defined by the method (section 4.2 Fig.7). Moreover, the comparison between both inventories shows that while 171 of 2D-sources are shared with 3D-sources, it represents 144 sources 3D-sources. This highlight that 25 landslide sources are amalgamated in the 3D inventory (L-. As in the previous version of the paper, we perform a sensitivity analysis of the impact of $D_m$ showing that landslide statistics are not severely affected by this parameter for $1.5 < D_m < 3$. We also explored density based spatial clustering algorithm used in 3D rockfall segmentation, derived from DBSCAN (Ankerst et al., 1999; Martin Ester, Hans-Peter Kriegel, Jiirg Sander, 1996; Tonini and Abellan, 2014) and HDBSCAN (Carrea et al., 2021; McInnes et al., 2017). None of them managed to provide a significantly better segmentation of the largest landslide and are significantly longer to run than the connected component algorithm we use. We now thoroughly discuss about this in section 5.1.2.

**The analysis that you have added demonstrates that the problem is not that your particular segmentation approach is worse than the alternatives but that automated segmentation itself is problematic. You make the argument above the subjective segmentation is equally problematic. I think making this point in a more detailed discussion of the problem of segmentation (both automated and manual) would help to address my concern here.**

MC4) Topographic errors propagate through segmentation to introduce a bias towards small landslides. Existing experiments to quantify this bias are insufficient.

*You say on L559-60 that "our approach has the benefits of more systematically capturing small landslides than traditional approaches". However, this is one of my main problems with the paper given potential propagation of topographic errors through thresholding and segmentation.*

*You show (in SI) that: 1) in the absence of any real topographic change your detection algorithm generates artefact landslides of 1-20 $m_2$ purely due to spatially uncorrelated topographic noise; 2) this noise generates many more small than large landslides; 3) in this experiment artefacts >20 $m_2$ were extremely rare. However, this does not demonstrate that predicted landslide size is insensitive to longer wavelength topographic errors (known to be present in the data); nor even to short wavelength noise in the presence of longer wavelength surface differences (e.g. real landslides). First, even without any real topographic change (i.e. no real landslides), the size distribution of erroneous landslide-like clusters will depend on the spatial correlation length of the difference errors, which in turn depends on the correlation lengths of the errors in the surfaces being differenced. Longer error wavelengths will enable the generation of larger error clusters. Your figures show that topographic errors are clearly not uncorrelated and you recognise this yourselves (L418-22); nor do the errors appear to have a single characteristic wavelength. This is a hard problem but one that you must deal with if you are to convince the reader that the landslide inventory you have generated is not hopelessly biased by these landslide-like artefacts. Second, the problem is not only that clusters of erroneous negative surface difference due to roughness (or other errors) can create artefacts that appear to be landslides, but also that clusters of erroneous positive surface difference are collocated with real topographic change (e.g. due to a landslide); these can interfere negatively with real changes reducing the surface difference below the threshold for detection and breaking a single landslide into multiple patches.*

*Oversampling of small landslides is important because it undermines your most surprising and high impact claim: that rollover reported in previous inventories is due to under detection (L573-4). I am not currently convinced by this claim because you do not exclude the possibility that the lack of a rollover is solely due to detection errors. You need to quantify these detection biases before you can make these claims.*

*Suggested additional analysis: A "more advanced segmentation" (L431-3) may be out of scope for this paper. However, an indication of the impact of the simple segmentation on object based classification skill is an essential requirement of this paper if it is to retain the current approach to identifying discrete landslides. Two possible avenues could be followed to provide such an indication. First, your analysis of topographic changes on the stable surfaces (Fig 3) would allow you to perform the same analysis that you have performed in the SI but using the pre- and post-EQ surfaces for the stable zones identified in Fig 3. This would enable you to identify the size distribution of artefacts that can be generated from topographic errors with a spatial correlation length closer to that for the unstable parts of the study area. This though still does not account for the possibility that the landslide erosion signal itself is altered by the noise (e.g. by disconnecting clusters). Second, you could collect a landslide check dataset using independent observations. This might take the form of an entirely independent inventory but should certainly also involve cross-checking to confirm the existence and characteristics (e.g. area, shape, depth) of your predictions.*

We added further analyses to the method to deal with topographic errors and erroneous landslide. We also added an entirely new 2D landslide inventory as suggested by the reviewer. We are now confident that the actual landslide inventory corresponds to real changes. Please see our reply to the MC2) and MC8) comments for a detail answer.

**Most of my concerns in this comment have been left unaddressed. MC2 focusses on topographic errors but your response to MC2 doesn't deal with the problem of spatially correlated errors. Because you are interested in the size of patches generated by thresholding the difference surface it is essential that you examine the spatial structure of the errors. I will deal with each of my comments in turn reflecting on whether they have been addressed in the new manuscript even if they have not directly been addressed by your response to this comment.**

**First, you did not** *"demonstrate that predicted landslide size is insensitive to longer wavelength topographic errors (known to be present in the data); nor even to short wavelength noise in the presence of longer wavelength surface differences (e.g.*

*real landslides)"*. **You remove reference to an SSDS test to set the minimum area but retain that test to optimise the** **SNR. However, when you introduce it you do not recognise that it synthesises uncorrelated noise while the two surfaces that you are differencing both include spatially correlated elevation errors. You do not include any description or explanation in the text for the "stable areas error" shown in Figure 4, these errors appear to be spatially structured on multiple length scales from tens to hundreds of metres.**

**Second, I argued that** *"Oversampling of small landslides is important because* it undermines your most surprising and high impact claim: that rollover reported in previous inventories is due to under detection"* **and that** *"You need to quantify these detection biases before you can make these claims."* **I'm still not convinced by this claim because I still don't think you have excluded the possibility that the lack of rollover is solely due to detection errors. You need to quantify the size dependent detection bias in the 3D inventory and/or to considerably tone down your claims about rollover in this and other 2D inventories being due to under-detection.**

**Third, I suggested that you:** *"perform the same analysis that you have performed in the SI but using the pre- and post-EQ surfaces for the stable zones identified in Fig 3. This would enable you to identify the size distribution of artefacts that can be generated from topographic errors with a spatial correlation length closer to that for the unstable parts of the study area.* **I don't see a response to this suggestion here but when I raised the same point in a minor comment (related to L196). you responded that** *"Applied on stable area, the workflow does not detect any landslide."* **This result would definitely be worth reporting! However, I think we must have misunderstood one another, I can see many patches of significant change (>150) within the stable areas, most of these patches of significant change are removed in Figure 5c. Is that because they are smaller than 20 m$^2$? However, even after this filtering I can still see several landslides within the stable zones in Figure 5c. I phrased this as a suggestion in my previous review but I really think this is one of the few opportunities that you have to build confidence in your method. It remains a weak test because you chose the stable areas based on areas of limited change in the difference maps and because they oversample non-forest vegetation but in the absence of field checks to the inventory this remains one of the best tests I can come up with. My second suggestion to** *"collect a landslide check dataset using independent observations"* **is dealt with in a separate major comment MC8 and doesn't need further discussion here.**

1110     MC5) Findings that differ from previous work

*There are a number of unusual findings that are worthy of comment because they are some of your most interesting and potentially important findings. It is essential though that each is carefully examined and that the critique that it might have arisen due to methodological errors is dealt with head on.*

*First, it is not unusual to identify more sources than deposits due to amalgamation of landslide deposits. However, it is unusual*
1115     *for deposit areas to be smaller than source areas (L281-2), and for deposit depths to be thicker than source depths (L265-70 and L460). These result should be compared with results from previous studies.*

**First**: contrary to the reviewer experience, this result does not surprise us, in particular when the runout of landslides is not long. The filtered data clearly support this finding.

**This has not been addressed. I commented that you needed to compare your results with those from previous studies**
1120     **with respect to scar and deposit depths and scar and deposit areas. I didn't find this new discussion nor a response to explain why it was not necessary.**

*Third, you identify areas of deposition where there is no upslope erosion (L256 and Fig 4). I don't think these can be real deposition zones but instead must be a consequence of incorrect landslide detection. Their spatial extent and depth distribution would be a useful indication of the precision of the technique.*
1125     **Third:** We now filter landslides by a signal-to-noise ratio (section 3.3.4) and are confident that the actual landslide inventory corresponds to real changes. Some very small deposit areas may not have upslope erosion as we expect deposit are to be easily detectable by our method than source areas (section 5.1.2).

**Why do you expect that deposits are more easily detectable than source zones? I would have expected the opposite. In my experience deposits can be very thin, (<50 mm) patchy and extensive whereas source zones are far more coherent.**
1130     *Fourth, it is extremely unusual that locations classed as vegetated in post-event optical imagery but identified as a landslide by another technique are considered by the authors to be genuine landslides (L347-8). Instead, the presence of vegetation at the location strongly suggests a false positive.*

**Fourth:** we partially disagree with this statement. As now explained in the discussion large landslides that strip out vegetation are obviously mapped in 2D inventory, but small ones that occur on less dense area are extremely difficult to map in 2D
1135     imagery as our inventory shows. Moreover, vertical subsidence due to upslope propagation of landslides is entirely missed in forest, while it is detected in our approach. We think the comparison with the new 2D inventory will resolve the reviewer's reserve.

**How do you know that small ones that occur on less dense area are difficult to map? Is this on the assumption that your LiDAR inventory is correct? If you go to the landslide locations predicted by the 3D method do you find evidence in**
1140     **the orthophotos that there is indeed a landslide at that location (even if it wasn't independently mappable)? This would help to build confidence in your method. The point about vertical subsidence is important and you do a nice job of demonstrating the plausibility of the claim that this is real change. It prompts a series of questions about representation of these landslides within an inventory derived from surface change but you deal with this nicely in the discussion. My only suggestion is that you prepare readers for this finding in the introduction by adding a more complete explanation**
1145     **of the types of landslides that 2D and 3D inventories might include (see MC1).**

MC8) Landslide object detection needs to be tested against an independent dataset

*The findings in this paper depend critically on the skill with which the proposed method can classify landslide scars and deposits. Thus it is essential that the paper reports testing results that quantify this object based classification skill. At present, "orthophotos were used to visually validate" the classifier (L115-6) but without reporting results of this analysis. I think it is essential that you explicitly explain your sampling and mapping strategy for landslide detection from orthophotos in the methods. You should then include a section in the results where you compare your orthophoto based mapping to the surface differencing approach.*

*However, it is not enough to simply say the orthophoto mapping did not identify landslides that were identified by the surface differencing. You should then go to a carefully chosen (e.g. stratified random) subset of the landslides detected by each method that were not detected by the other (i.e. surface differencing but not ortho-photo mapping and vice versa) to establish as far as possible which of the two methods was in error and why. While finding and mapping thousands of landslides might be timeconsuming (L255), confirming their existence and characteristics (e.g. area, shape, depth) would not.*

This lack of comparison between the 3D differencing method and a more classical approach has been addressed by adding a manual mapping of landslides based on 2D images. We added a section that specifically explore the differences between the methods (section 4.2). Moreover, the resulted landslide area distribution mapped manually has been added in the figure in section 4.3 and compared with those obtained with the 3D differencing method.

**The 2D inventory considerably strengthens the paper and addresses the comments in the first paragraph above. However, much of the second paragraph remains unaddressed. You have added analysis of the two inventories and discuss false positives for the 2D inventory where deposit is incorrectly mapped as source zone, this is a secure result and is exactly the type of analysis I was looking for. You use your observations and theory/logic to argue that one method is correct and the other is in error. The remaining areas of disagreement you assign as false negatives for the method that has not identified a landslide at that location. This is not a secure result. You have no objective way of establishing which method is in error (i.e. whether this is a false positive for one or a false negative for the other) and you don't provide any justification for why disagreements should always be treated as false negatives. In fact there is good evidence to suggest that the 3D inventory should contain false positives and that these false positives likely have a strong size bias. First, the SSDS test (which itself is a very weak test because it assumes that errors are uncorrelated in space) results in artefact landslides. Second, the inventory contains landslides in the 'stable' areas of the study area. This problem propagates into the discussion where you describe disagreement between inventories as error in the 2D inventory under the implicit (but untested) assumption that the 3D inventory is correct. I don't think you could use this language in the paper even if the tests above generate only small numbers of false positives from the 3D inventory.**

*L114: "using the classification provided": More detail is needed on the method used to classify ground points.*

1180 The survey report of the LiDAR data only mention that the ground points have been automatically classified using the Terrascan software. The reference to the survey report (Dolan and Rhodes, 2016) have been added to the text.

**I don't see Dolan and Rhodes (2016) cited in the text. The sentence above should be added to the text. You have added a manual quality check and reprocessed the data to remove non-ground-points as a result. That is a good additional step that you have introduced since the last version. However, I don't think you can simply point to the SI for the details**

1185 **of this analysis. This is a key step in your method and should be included in the article itself.**

*L137-8: This is not clear: do you mean that you calculate the centroid of each point cloud in 3D then take the magnitude of the 3D vector that connects these two points; or that (for each point cloud) you take the arithmetic mean of differences from the reference plane (defined by D) in a direction normal to that plane? In either case you are performing a spatial averaging*

1190 *at length scale d/2 assuming a uniform kernel. First, is it problematic to perform averaging over length scales larger than the core point spacing? Second does it make sense to assume equal weight in the average with plane parallel radial distance from the core point, or should some form of inverse distance weighted average be used? I would have assumed a weighted average was more appropriate but it would be useful for you to explain why an unweighted mean is more appropriate.*

To calculate the distance between the two point clouds, the average positions $i_1$ and $i_2$ of the point clouds are first defined and

1195 then the distance is computed between the two positions along the normal vector. The average positions are defined as the arithmetic mean of the distance distribution of each point of the subset of points (created by the intersection of the cylinder to the point cloud) to the normal vector (or the cylinder axis; see Lague et al., 2013). As this is part of the M3C2 algorithm that we did not modify, we don't discuss the choice of a uniform kernel rather than a non-uniform one.

**You have not changed the manuscript in response to my comment. Your response is useful, particularly: *"the distance***

1200 ***is computed between the two positions along the normal vector"* and *"average positions are defined as the arithmetic mean"*. You could easily amend the sentence to clarify this: *"…as the distance of the arithmetic mean positions of the two point clouds along the normal vector".***

*L139: "if not intercept is found…": This is not clear to me. Do you mean 'if the cylinder does not intersect any points in the*

1205 *second surface? Why would this happen? Does this only occur at the boundaries of the point clouds? How do these intersection failures influence the surface differencing and how do you report them in your later analysis?*

In LiDAR datasets, the density of points is non-uniform over the entire point cloud. Consequently, missing data or very low point density (<5 pt/m²) can occur inside the point cloud due to the absence of laser impact on the ground during the data acquisition. This mostly occur in dense vegetated areas or water surface areas (for topographic lidar). When performing M3C2,

1210 it is thus possible that the cylinder cannot intercept points or just a few (< 5). In both cases, the M3C2 distance will not be considered significant. In areas with low point density (<5 pts/m²) a solution is to perform M3C2 with a larger projection scale *d* to include more points in the distance and statistic calculations.

**You have not updated the text to reflect this discussion. You do mean: 'if the cylinder intersects <5 points in the second point cloud'. If so, this should be added to the paper. Your explanation above is useful but I understand it to mean**

1215 **something different to what you say in the paper.**

*L140: "provides uncertainty": what is the basis / justification for the uncertainty estimate taking this particular form? It looks familiar as it has some similarities to a confidence interval but also some differences. This threshold is important to explain and justify in detail because it is used to threshold discrete landslides in the following analysis. Why threshold at 95%*

1220 *confidence? What is the impact on your findings (total volumes and scaling relationships) of thresholding at a difference CI (e.g. 99 or 90% confidence)?*

We refer the reviewer to the original M3C2 paper which has an extensive discussion on the confidence interval, and how to consider surface roughness, point cloud errors, point density and registration in the context of change detection on 3D point clouds. The threshold has been set to 95% to build the segmentation on as many good points as possible. We do not believe that changing this threshold to 90 or 99% significantly change the landslide statistics given the results of the sensitivity analyses of parameters (reg, $D_m$ and SNR threshold) that mainly control the landslide inventory.

**First, I think you can evidence your statement above from your sensitivity analysis and doing so will strengthen the paper. You demonstrate that changes to the LoD do not alter your main findings and the changes you explore cover the range that you would expect from changing the threshold CI from 90-99% (i.e. 0.39-0.52 m). This strengthens your argument that the statistics are robust to model choices.**

**Second, having read the original M3C2 paper (Lague et al., 2013), it discusses the confidence interval and registration error but took a different approach to estimating registration error so it is difficult to translate directly between the two papers. They provide only a brief description of the theoretical basis for the LoD equation citing a statistics textbook.**

**I found the description of James et al (2017) who you cite and who cite Lague et al. (2013) very useful: *"where reg is the relative overall registration error between the surveys, assumed isotropic and spatially uniform (Lague et al., 2013). Note that Lague et al. (2013) took a conservative approach by adding reg directly (as a potential systematic bias)"*. A similar statement would be useful in your paper to explain that you estimate local uncorrelated random errors using the first two terms and systematic errors under the assumption that they introduce a spatially uniform bias with the final term.**

**Anderson (2019) describes an approach similar to yours, but highlights spatially correlated random errors as a key component within error analysis for surface differencing and includes this as a term in his total error calculations (eqn 21). This term is missing from your error propagation but seems likely to be very important, particularly because you are interested in the size of thresholded difference patches. Anderson (2019) also argues for direct characterisation of errors within 'stable areas' similar to my suggestion in MC4.**

**Anderson, S.W., 2019. Uncertainty in quantitative analyses of topographic change: error propagation and the role of thresholding. *Earth Surface Processes and Landforms*, *44*(5), pp.1015-1033.**

*L145: "reg" is quantified using the standard deviation of differences between the surfaces. I think it would be helpful here and elsewhere to use similar notation for the registration error to the other errors being examined here. Why are the local terms converted to standard errors but reg is left as a standard deviation? Finally, the length scale over which reg was calculated would seem to be important here.*

We choose to keep the "reg" notation to specifically refer to the registration error when needed in the manuscript. We now provide an explanation in the text as to why reg is not converted as a standard error (L 166). Reg is not measured over a length scale, it is based on the standard deviation of the 3D-M3C2 distance between 2 clouds. This part is now explained in greated detail in section 3.3.1 where we discuss the notion of intra-survey and inter-survey registration quality.

**You now say: *"The M3C2 definition of the LoD95% makes the conservative choice of adding the registration error to the combined standard error related to point cloud roughness, rather than taking the square root of the sum of squared standard error and squared registration."***

**First, I agree that your equation is conservative in the sense that it results in a larger LoD but I don't see a justification for this functional form either here or in Lague et al. (2013). The best description of a framework for propagation of both random and systematic errors that I can find is in Anderson (2018). It is similar to yours in its approximation of random and systematic errors (see Anderson's eqns 12 and 20) but differs from yours in how these are combined (see eqns 21-22). Can you explain the difference?**

**Second, you have not explained why standard deviation rather than standard error is used for the systematic error (reg). I think this is because standard errors are used to approximate random errors (under the assumption that they are uncorrelated) but standard deviations are used for systematic errors under the assumption that these are perfectly correlated (see Anderson's eqns 1-4, 12 and 20). If that is the case it would be helpful to explain it in the text.**

**Third, I'm happy for you to retain a notation that is specific to registration error but suggest sigma with reg as subscript would make it clearer that this is a standard deviation.**

*L165: "not deemed interesting": I don't think that this is the right phrase, can you rephrase? What was the impact on your results of applying d=10 m throughout?*
The sentence has been modified. Applying d=10 m increases the landslide source and deposit by 1% and 0.7% respectively. In terms of volume this represents an increase of 2% and 0.89% respectively.
**I do not see where you have modified the text to reflect this response.**

*L173: "may result in…": How do you identify when this has happened? What is the objective that you are optimising?*
This can be identified in the bottom of narrow valleys and top of very steep divides where no evidence of mass movement processing can be identified by visual inspection of orthophotos. These cases are now filtered by using the SNR metric as they have very large $LoD_{95\%}$ (see section 3.3.4).
**This is still not clear both in the response and in the text (which has changed very little). L211 you say *"This is generally obtained by trial and error"*. This was what I was referring to when I asked *"what is the objective was that you are optimising"* by trial and error. Re-reading the text I am not sure whether it was obtained by trial and error in this paper. If not then how did you find out that the maximum observed change in the study area was 30 m? If $p_{max}$ is designed to prevent anomalously large changes how do you verify that a change is an anomaly? Something seems circular in the argument as it is currently presented. I don't understand the connection to SNR filtering and that doesn't currently feature in the text.**

*L196: "standard deviation…" These stable areas would be an excellent test of your landslide detection method, indicating the scaling relationships, size distributions, and total volumes generated by artefacts alone.*
This analyse has not been performed due to the changes we made on how we manage artefacts with SNR filtering (see section 3.3.4). Applied on stable area, the workflow does not detect any landslide.
**This result is definitely worth reporting. If there are any remaining patches then you should report the size distribution of these patches as they give an indication of the expected bias that elevation errors will introduce into your size distribution.**

*L199-200: "The registration error…": this definition should come earlier. It is important for interpreting eqn 2.*
We disagree with this comment as the registration error "reg" can be defined differently depending on the application of M3C2. In section 3.1, we aim to give a general description of the M3C2 algorithm.
**I agree that it can be defined differently but you are reporting your method rather than the method in general so you should define it as you have used it.**

*L215: 15-20% it might be useful to show the location of these points on one of your maps.*

The percentage of these points on steep hillslope has been revised and actually represents up to 12% of steep slopes.

**It would still be useful to show the location of the points on one of your maps.**

*L256: "deposit areas" Some of these areas do not have any upslope erosion. How do you explain this? See MC5*

Some deposit areas can be detected without an associated upslope erosion. A possible reason is that the surface change associated to these deposit areas is sufficient to be above the $LoD_{95\%}$ but not for the upslope erosion area.

**This is useful discussion, I couldn't see it in the revised manuscript. It would be useful in your description of Figure 5.**

*L315: It is not surprising that your depth area scaling relationship is gentler than that of Larsen since you censor core points with difference < 0.33, making it impossible for small landslides to also be shallow.*

Indeed, and we now discuss this. In particular, the relationship is barely different if we consider a SNR=1, meaning that compared to the 2D inventory, we miss 32 shallow landslide sources over an inventory of 1270 sources (SNR >=1). Hence, even if the 3d inventory had captured the very shallow landslides that the 2D mapping did capture, it would hardly change the scaling relationship. One could also say that previous 2D inventories have significantly underevaluated the number of small landslides, which in turns affect the representativity of published V-A relationships from 2D inventories.

**This is useful discussion, I couldn't see it in the revised manuscript but I wasn't sure whether I was looking for parts of the text in your response above or something different.**

*L460: "deposits form more concentrated and thicker patches": This result is surprising and should be compared with expectations from other studies on landslide runout.*

We think it is not surprising given the small runout of the landslides in this area, and is actually backed by the data in terms of mean 3D thickness of the deposits and sources.

**This comment has not been addressed. I cannot find new text comparing your results with expectations from other studies.**

*L466: Whether the right tail is a "power law" or not is debated. See Medwedeff et al. (2020) among others.*

The reference has been added in the text.

**This comment has not been addressed. You have not altered the text to recognise the debate around the form of the right tail. I don't think you can simply add Medwedeff to the current citation. It reads as though Medwedeff et al. are among those arguing for power law scaling when I understand their paper to argue the opposite.**

*Figures*

*Fig 5: The 3D minimum volume line needs explaining in the caption. I would have expected this line to be oriented parallel to the depth contours since minimum volume for a given area is set by minimum detectable depth. If so the minimum detectable volume might explain the sharp lower boundary to the volume area point cloud.* Actually, what it is illustrated here is the 3D minimum volume that can be measured here given the minimum area (20 m²) that we consider and the minimum significant depth (~0.4 m).

**The minimum area and depth act together to set an absolute minimum detectable volume (when both area and depth are minimised). However, there is also a depth dependent minimum detectable volume that is set by the depth constraint alone. At present the horizontal line that you use to highlight the absolute minimum volume might be misinterpreted by some to be the area dependent minimum volume. You can easily fix this by adding the area dependent minimum volume. It will be a straight line in log-log space and will pass through the points (20,8 and 20000,8000). It would be useful to include this on the Figure and would also bring make the interpretation of the dashed lines in the figure and inset internally consistent.**

**Supplement**

**S1:** this text should definitely be included within the paper itself this is a key part of your method. You should also report the parameters that you used for this analysis and the parameter values that you chose, preferably with a justification.

**L11:** "4 standard deviations" I have three questions here: 1) what sample is the standard deviation being calculated from? 2) Do you remove points that differ by >4sigma in positive and negative? I'd expect vegetation to result in only positive residuals. 3) How did you choose the 4 standard deviations threshold?

**L12:** did you repeat three times because there were no outliers after that? If so you should report this, if not you should explain why you chose to stop after three iterations.

**Figure S2:** This is a useful Figure but I can't distinguish the flight lines based on the legend information, the line styles are not sufficiently distinct. Dashed lines are clearly visible and can be distinguished from the solid line. I think you will need to use different line styles to enable the reader to distinguish this many lines.

I don't understand how you can have only one reference line in Figure S2 and Table S3. Is this a flight line from the post-EQ set? Or is this some combination of points from multiple lines? Either way I think you need to explain this, it will affect how the reader interprets Table S3.

**S9:** "Determination of forested area" Is this a standard technique? It would help if you could give a reference for the technique. I would like to know how returns are classified (i.e. how different do returns need to be to be classified as two distinct returns). However, if this is a very standard exercise it is fine for you simply to point to a reference.

**Figure S10:** It would be useful to give more detail in the caption. Something like "corresponding to the number of targets a laser pulse has intercepted" would probably be sufficient.

**Figure S12:** This is an interesting plot. How do you calculate slope here? Is this based on the gradient of the core points? How do you explain the large number of sources with very low slope? Are these all associated with deep seated failures? This slope data provides another really useful way to check your dataset and therefore to build confidence in your results. You should plot slope for each core point against the size of the patch to which each core point belongs. Core points with low slopes associated with large patches may indicate deep seated landslides; those associated with small patches are likely to be errors because landslides require a steeply dipping failure surface (>20 degrees?) to move. Another useful approach would be to examine average slope for each patch plotted against patch size. As above, small patches with gentle gradients are likely to be errors.

[revised manuscript text omitted]

---

## Author Response (AR2)

**1** Global response highlighting the main modifications to the paper

Note: in the following : black = referee's comment, red : quote by referee 2 from our previous answer to
 referees, BLUE= our new answer. We refer to lines and sections of the revised MS with track changes.

5 Dear Editors and referees,

2

6

23

28

7 We thank both referees for the time and attention spent in evaluating again our work, and the editors for
8 handling and MS. Here is a synthesis of our actions following the previous round of reviews:

- All comments by referee 1 has been addressed, with answers to these along each of referee 1's comments in the pdf he provided.
- 11 In response to referee 2's comment we have significantly improved our workflow which resulted in a 12 new element in the MS dealing, mainly, with the treatment of false detections. We believe these 13 additions, which do not alter our main conclusion, but rather strengthen them will answer the main 14 issues that referee 2 raised. We also addressed minor comments (old ones and new ones) that we 15 thought were fully relevant to the MS, and were helping its readability. Old ones are answered in this 16 document; new ones are answered directly on the pdf file the referee provided. We are extremely 17 thankful to referee 2 for his detailed and extensive review. His persistence has challenged us, but we 18 think that the level of detail or extra work that he requests is sometime unnecessary, and would 19 hamper the readability of the MS. So not all his comments have resulted in change to the text, and we 20 have tried to explain why. The length of his reply has been also a challenge to account for, and actually 21 writing the response to reviewer has been indeed, more complex, than revising the MS itself, while it 22 should, generally be the opposite!

We synthesize below the major modifications made to the MS. They are presented in greater detail attached to
the comment of referee 2 with line number referring to the MS with track changes.

- 27 We have made 4 significant improvements to our workflow and demonstration:
  - 1. The M3C2 algorithm has been improved on two aspects:
- 29 a. the uncertainty model accounting for point cloud roughness, point density and registration 30 error (LoD95%) adheres more strictly to the two-tailed t-statistics when the number of points 31 intercepted by the projection cylinder is smaller than 20, resulting in the LoD95% being up to 32 46 % larger in the lowest point density area (see L259). This model was originally presented in 33 Lague et al. (2013) but simplified by using a constant value of 1.96 for the t-statistics. This 34 slightly reduces the fraction of surface detected as significant change before segmentation. 35 This has a marginal effect on the end result, but is more statistically robust when working 36 with low point density data such as the pre-EQ lidar survey.
- b. a new iterative procedure detects the rare cases for which the projection cylinder could
  intercept twice the same point cloud, for instance in very narrow valleys or near ridges. If not
  corrected, the M3C2 distance was incorrect and would generally indicate a significant change
  when there was not. The iterative procedure checks for the consistency of the M3C2 distance
  for various cylinder depth (see L238).
- The combination of these 2 changes results in 1118 potential landslide sources after segmentation
  compared to 1270 in the previous version of the MS. These changes to the algorithm will be
  implemented in a new version of the M3C2 plugin in Cloudcompare that will be released when the
  paper will be published.
- 47 2. A manually labelled 3D source inventory corresponding to 66 % of all potential landslide sources has 48 been created with 2 classes: true landslide sources (384) and false detections (355) (fig 6c) (see new 49 method section 3.5, L475). The labelling and interpretation was done using orthoimagery, the 3D 50 M3C2 field (to identify nearby deposits) and detailed inspection of the point cloud. The sampling 51 strategy was as follow: all clusters larger than 200 m2 were labelled (as they are the most important in 52 terms of volume). The remaining smaller clusters were labelled through stratified sampling ensuring a 53 uniform spatial distribution and no area bias. All area intervals below 200 m2 have 60 % of the clusters 54 labelled to avoid impacting the scaling behavior of pdf(A) and the occurrence of a rollover. This 55 labelled source inventory is used as follow:

| 56                   |        | a. to quantify the prevalence of false detections in the inventory immediately after                         |
|----------------------|--------|--------------------------------------------------------------------------------------------------------------|
| 57                   |        | segmentation (section 4.2.1, Fig. 6.7) and evaluate their potential impact on scaling                        |
| 58                   |        | properties if they are not properly removed (section 5.3.2 of the discussion, fig. 14a). We                  |
| 59                   |        | document as expected by referee 2 a prevalence of false detections decreasing with size. In                  |
| 60                   |        | forest free areas, the prevalence of false detections is relatively low (24.4.%) and corresponds             |
| 61                   |        | to correlated elevation errors of the LiDAP data due to intra survey lider errors and inter                  |
| 61                   |        | to correlated elevation errors of the LiDAR data due to intra-survey lidar errors and inter-                 |
| 62                   |        | survey registration error. In forested areas, faise detections dominates the inventory (80 %)                |
| 63                   |        | due to ground classification errors in the pre-EQ data. As explained in the method section 2                 |
| 64                   |        | (L171) we did not try to further improve the ground classification of the pre-EQ data because                |
| 65                   |        | (i) we think it is important to clearly highlight the issue which is expected to be frequent in              |
| 66                   |        | legacy LiDAR data of steep hillslopes with dense evergreen forest and low shot density, and                  |
| 67                   |        | (ii) we do not think the classification can be further improved because laser penetration in                 |
| 68                   |        | the forest was too poor to correctly identify the ground                                                     |
| 69                   |        | b The labelled source inventory is used to evaluate the performance of filtering metrics (section            |
| 70                   |        | 12.28(12.3) we apply to create the final inventory on which we compute total volume and                      |
| 70                   |        | 4.2.2 & 4.2.5) we apply to create the initial inventory on which we compute total volume and                 |
| 71                   |        | geometric characteristics of failustice sources (see point 5 below)                                          |
| 12                   |        | c. The tabelled source inventory is used as a reference case for statistics on fandslide sources             |
| /3                   |        | (pdf(A), pdf(V)) which are unambiguously true labelled landslides, as opposed to predicted                   |
| 74                   |        | one (in which a small fraction of false detection may remain). It is thus used in 3 figures (11a,            |
| 75                   |        | 11b, 14a)                                                                                                    |
| 76                   | 3.     | A new filtering metric is introduced: the closest deposit distance (CDD), calculated along the               |
| 77                   |        | downslope distance of any potential landslide. Along with other metrics derived from M3C2 and                |
| 78                   |        | computed for each landslides (fig. 8), we use the labelled 3D source inventory to show how to obtain         |
| 79                   |        | an optimal balance between preservation of true landslide sources, and removal of false detection.           |
| 80                   |        | We show that the CDD in combination of the SNR is a very effective way of filtering out false                |
| 81                   |        | detections while retaining a very large amount of true landslide sources in forest free areas (table 2.      |
| 82                   |        | balanced accuracy = $0.93$ ). In forested area, the results are slightly worse (balanced accuracy = $0.8.50$ |
| 83                   |        | % of true landslide are preserved. 94 % of false detections are removed) owing to the poor quality of        |
| 0.0                  |        | the pro EQ data classification. We also demonstrate that after filtering, false positives do not exhibit a   |
| 0 <del>4</del>
0F |        | the pre-LQ data classification. We also demonstrate that after intering, faise positives do not exhibit a    |
| 00                   |        | size dependency in forest-free areas and only a weak size-dependent effect in forested area, so that if      |
| 80                   |        | a rollover where to exist at those areas in the real landslide data, it would be largely preserved and hot   |
| 8/                   |        | obscured by the increased occurrence of false positives (fig. 7, section 4.2.1). Finally, deposits, which    |
| 88                   |        | we recall are not the core of our study, and have thus not be manually labelled, are filtered after the      |
| 89                   |        | sources, using as condition that a true deposit must be connected to an upstream valid source (L538).        |
| 90                   | 4.     | A new predicted inventory has been calculated based on the optimal filtering metrics, and                    |
| 91                   |        | represents the reference landslide source inventory that we use for total volume calculation (source         |
| 92                   |        | and deposits) and source statistics.                                                                         |
| 93                   |        |                                                                                                              |
| 94                   | Conseq | uently the main changes in the organization of the MS are:                                                   |
| 95                   | 1.     | The issue of false detection is presented in the abstract (L17-21) and in the introduction to emphasize      |
| 96                   |        | the critical importance of accounting for these issues (L140-145)                                            |
| 97                   | 2      | Method section:                                                                                              |
| 98                   |        | a A more extensive presentation of error models in relation to comment MC1 of referee 2 and                  |
| 00                   |        | the new elements of the M2C2 elevithm has been added in the method section (section 2.1                      |
| 100                  |        |                                                                                                              |
| 100                  |        | $LZ43^{-}Z3Z$                                                                                                |
| 101                  |        | b. a new section presenting the treatment of faise detection has been added (section 3.5, L 475)             |
| 102                  |        | with the following sub-sections: section 3.5.1 : Construction of a labelled inventory, 3.5.2                 |
| T03                  |        | Definition of filtering metrics, 3.5.3 Definition of a classification performance index. The                 |
| 104                  |        | nature and type of false detection that can occur with topographic change detection applied                  |
| 105                  |        | to landslide detection is described (section 3.5 fig 5). Figure 3 presenting the workflow has                |
| 106                  |        | been revised to further highlights the treatment of false detection AFTER the segmentation.                  |
| 107                  | 3.     | Result section:                                                                                              |
| 108                  |        | a. section 4.1 is now restricted to the presentation of the result of the segmentation before the            |
| 109                  |        | application of filtering metrics. Figure 6 presenting the map views of 3D distances and                      |
| 110                  |        | segmented data has been recalculated to reflect the results of the new M3C2, the significant                 |
| 111                  |        | change map with the map of forested area (to answer one of the request of reviewer 2) as                     |
| 112                  |        | well as the significant change corresponding to the river bed. Fig. 6c is actually now it                    |
|                      |        | wen as the significant change corresponding to the river bed. Fig. of is actually new. It                    |

| 113
114
115
116
117        |                                |                                                                         | presents the labelled dataset, the unlabeled data and the deposits. It allows the reader to see
the location of labelled false detections. We have added a detailed view at each of these
steps for the reader to better understand the segmentation and filtering of small landslides in
forested area. The final inventory on which we perform statistical analysis, is now in a
dedicated figure 9.                                                                                                                                                                |
|----------------------------------------|--------------------------------|-------------------------------------------------------------------------|-----------------------------------------------------------------------------------------------------------------------------------------------------------------------------------------------------------------------------------------------------------------------------------------------------------------------------------------------------------------------------------------------------------------------------------------------------------------------------------------------------------------------------------------------------------------------------------|
| 118
119
120
121
122
123 |                                | b.                                                                      | An entire new section 4.2 Removal of false detections and the 3D predicted inventory (L631) has been added with 2 new figures: fig 7 presents the proportion of false detections before and after filtering as a function of landslide area. Fig. 8 shows the cumulative distribution functions of filtering metrics for labelled landslide sources and false detections. Section 4.2.2 Optimal filtering metrics explores the best combination of filtering metrics. A new table 2, synthesizes the results. Figure 6 of the previous MS (selection of the optimal SNR) has been |
| 124
125
126
127
128        |                                |                                                                         | removed. Section 4.2.3 The 3D predicted inventory summarizes the characteristics of the predicted inventory we obtain after application of the filtering metrics, with a dedicated new figure 9 showing the map of predicted true landslide sources and deposits and the predicted false detection and a new table 3 summarizing the number, area, and volume of predicted sources and deposits, and predicted false detection (sources and deposits).                                                                                                                            |
| 129
130
131
132               |                                | c.                                                                      | The subsequent part of the results (4.3 comparison with the 2D inventory, 4.4 Landslide sources area, depth and volume analysis) use the predicted landslide source inventory containing 433 sources as opposed to 524 sources in the previous version of the MS. A billshade has been added to figure 10c to better highlight the presence of retrogressive                                                                                                                                                                                                                      |
| 133
134
135                      |                                | e.                                                                      | scars in the forested area coherent with the pattern of deformation we detect (to reflect a comment by referee #1).
Figure 11 has been updated with the new inventory and comments by referee 2 regarding                                                                                                                                                                                                                                                                                                                                                                      |
| 136
137
138
139
140        |                                |                                                                         | centered bin in X axis (as in all figures now), comparison of depth/area with the relationships predicted by Massey et al. and Larsen. The pdf(A) of the new predicted inventory is extremely similar to our previous MS version. We have added the labelled sources data to the pdf(A) and pdf(V) which follow almost exactly the same trends than the predicted inventory sources. With this new manually labelled data, the lack of rollover above 20 m 2 in the 3D inventory is                                                                                    |
| 141
142
143
144               | 1                              | The disc                                                                | confirmed. The new inventory has only 4 points below the 3D minimum volume, and the mean-depth/area data no more hints at the existence of two trends in the D-A relationship. We have thus remove these elements from the MS.                                                                                                                                                                                                                                                                                                                                             |
| 145
146
147
148               |                                | to create
compari
the false                                       | and slide inventories" where we have centralized most of the aspect pertaining to the
son with 2D inventories. A new section has been added dedicated to discussing the results of
detections (section 5.1.3 False detections and filtering metrics)                                                                                                                                                                                                                                                                                                                        |
| 149                                    | Other n                        | ninor char                                                              | Iges:                                                                                                                                                                                                                                                                                                                                                                                                                                                                                                                                                                             |
| 150
151
152
152               | •                              | 2D-source
core poi
areas (fr                                      | ces areas are calculated and are now estimated in the same way than the 3D method (sum of nts) so that both source areas can be correctly compared. This slightly reduces the 2D-source om 149,039 m 2 to 146,641 m 2 ). Moreover, the maximum 2D source area of 40,679 m 2 was a and has been corrected in the new version of the MS."                                                                                                                                                                                                          |
| 154
155
156                      | ٠                              | The new
updated                                                      | p(V) shows hint of a rollover above the theoretical lower limit of volume detection. We have this part of the discussion (section 5.3.3)                                                                                                                                                                                                                                                                                                                                                                                                                                          |
| 157
158
159
160               | To facil
main co
provide | itate the r
omments o
ed in the p                                 | eview process, we synthesize here our answers/modification in response to the outstanding
of referee 2 that he considered were not properly answered. More detailed answers are
oint by point response to reviewer 2.                                                                                                                                                                                                                                                                                                                                                       |
| 161
162
163                      | 1.                             | can and
some of
MC2: Ele                                          | cannot be used for. We have changed the text in the introduction and discussion following
the referee recommendations.
evation errors need to be better quantified and more thoroughly discussed                                                                                                                                                                                                                                                                                                                                                                            |
| 164
165
166
167               |                                |  <li>New generation</li> <li>caning</li> <li>choi</li>  | v elements in the MS on elevation errors (including classification errors) and how they may erate false detections due to spatially correlated errors. We also explain why elevation errors not be precisely quantified with the limited information we have. We also better explain our ice of a uniform registration error and the actual formulation of the LoD95%                                                                                                                                                                                                      |
| 168
169                             |                                | • We sign                                                               | use our new labelled 3D inventory to demonstrate that the base inventory contains a ificant fraction of false detections due to the above correlated errors with a larger prevalence                                                                                                                                                                                                                                                                                                                                                                                              |

| 170 |    | in forested due to classification errors. This highlights the importance of developing filtering           |
|-----|----|------------------------------------------------------------------------------------------------------------|
| 171 |    | procedures after the segmentation to remove as much of these false detections as possible.                 |
| 172 |    | • The discussion has been expanded to emphasize the importance of developing better elevation              |
| 173 |    | error models, the importance of point density and laser penetration for ground detection in forest         |
| 174 |    | area and the importance of post-segmentation filtering to remove false detections.                         |
| 175 | 3. | MC3 : Segmentation results in amalgamation but don't quantify its extent or impact : we have slightly      |
| 176 |    | changed the MS in the discussion to further highlight the complexity of amalgamated landslides that        |
| 177 |    | the 3D inventory exhibit. However, we consider that we have adequately answered this point in the          |
| 178 |    | previous iteration, even if it is not to the satisfaction of the referee, by showing that no current state |
| 179 |    | of the art 3D segmentation approach manages to correctly segment complex amalgamated landslides,           |
| 180 |    | nor does the 2D inventory. As explained in our previous answer, even if we were to provide a manual        |
| 181 |    | segmentation, it would be highly subjective such that its relevance would be equally as questionable       |
| 182 |    | as the automated segmentation. Our MS was already emphasizing strongly the need to develop better          |
| 183 |    | 3D segmentation approaches, but also demonstrating that the current limitations are not impacting          |
| 184 |    | the total volume, nor the lack of rollover in the 3D data.                                                 |
| 185 | 4. | MC4 : Topographic errors propagate through segmentation to introduce a bias towards small                  |
| 186 |    | landslides. Existing experiments to quantity this bias are insufficient. The reviewer was correct: the     |
| 187 |    | new addition of a manually labelled 3D inventory covering 66 % of the raw source inventory show that       |
| 188 |    | correlated elevation errors do produce a significant amount of false detection with a bias towards         |
| 189 |    | small landslides. This is something we were not able to properly quantify in the previous version of the   |
| 190 |    | MS, even though the SNR was already a good way to get rid of false detections in forest-free areas.        |
| 191 |    | The addition of the labelled dataset allows us to properly demonstrate that our filtering approach         |
| 192 |    | removes a large fraction of the false detections and that there is no bias left towards small landslides.  |
| 193 | 5. | MC5: Findings that differ from previous work. We have addressed all outstanding comments raised by         |
| 194 |    | the referee when we deemed they were relevant.                                                             |
| 195 | 6. | MC8: Landslide object detection needs to be tested against an independent dataset. The new                 |
| 196 |    | manually labelled 3D inventory and its exploitation answers the referee comment                            |
| 197 |    |                                                                                                            |

198

**199 Response to Referee 2 – David Milledge**

200 All the text from Referee 2 is included below. Because the new very detailed review of referee 2, adds to 201 another detailed review, I (D. Lague) must admit that never in my life as a scientist or associate editor did I 202 have to write or supervise a response to reviewer that is so long and complex. The way referee 2 responds by 203 quoting us then adding new elements makes it extremely complex to answer concisely, and not turn our 204 response into a discussion forum which is not the point of a review. Also, we feel that referee 2 has difficulties 205 accepting that our work has limitations. We consider that there are limits to the details and additional work a 206 referee should request for a paper that he otherwise qualifies as excellent. We have stated where we 207 considered that this threshold was crossed

208

209 This is an excellent paper that makes at least two significant contributions. The first is methodological, 210 detailing a method with which to robustly determine topographic change over large areas that include steep 211 slopes. The second is substantive, demonstrating that the size distributions (volume and area) and geometric 212 scaling relationships for landslides differ from those previously found for landslides. The authors have 213 considerably improved the manuscript since my last review. However, there are a number of major and 214 minor comments from my previous review that have yet to be fully addressed. Below I have retained only 215 these comments and explained (in bold) why I feel that they remain unaddressed. I have also made minor 216 comments on the new draft of the manuscript by commenting on the PDF.

217 The workflow that you have introduced has great potential to improve the quality of landslide inventories.
218 The paper is a significant and rigorous contribution because it: 1) introduces the workflow for a suitable
219 case study, 2) shows that the workflow improves on alternative 3D methods and can detect landslides not
220 detectable in 2D methods; 3) highlights common errors in 2D methods that have been proposed but rarely

- demonstrated, and 4) demonstrates for the first time that you can calculate a volumetric budget for
- 222 landslide derived topographic change without the need for volume-area scaling relationships which are

- 223 known to (and which you show) introduce considerable uncertainty. It also opens up a discussion about
- what constitutes a landslide and how this differs from the things that we currently map, whether in 2D or
   3D.
- 226 I agree with you when you say in your response that "despite its limitations, it currently represents the state of
- 227 the art in terms of 3D landslide detection and landslide inventory creation". This is an excellent methodological
- 228 contribution and I have only very minor comments on presentation of the paper in relation to this aspect of
- the work. As a methodological contribution I agree when you say that: "Scientific research is incremental, and
- 230 we fully expect that our workflow will be improved in coming years by others, as it was the case for 2D landslide
- 231 *inventories.* In relation to this, it is excellent that you: "provide all the elements (code, data) for other
- researchers to apply the workflow and reproduce our results, or apply the workflow to their data." The reason I
- am so demanding of the checks you apply to the 3D method relative to the 2D method is that the flaws in
- the 2D method are relatively well rehearsed in the literature but you are presenting the 3D method as a new
- 235 (and better) technique. To do so you must demonstrate that this is the case.
- 236 We think the new additions demonstrate that this is the case.
- However, I strongly disagree that you "clearly demonstrate, at least in our study case, that the rollover in the
- pdf(A) observed in our 2D inventory but not in the 3D one, is due to a size-dependent under detection in 2D."
- disagree because I think that non-trivial errors remain in: 1) the detection of small landslides, where some
- false positives remain due to spatially correlated errors; and 2) defining the boundaries of landslides, where
- automated segmentation continues to result in amalgamation. I know that I come to this with a bias: I have
- 242 interpreted my own field observations as indicating a rollover in landslide size and have developed
- theoretical explanations for that rollover. So I am probably resistant to the idea that landslide size
- distributions lack a rollover. I'm trying to avoid this bias but may not manage it.
- 245 We too have come up with a theoretical model for the origin of the rollover (Jeandet et al., 2018) and the
- results we show in this paper largely contradict these. We do not look specifically for controversy: we try to be
- as rigorous as possible given the data we have and processing tools we have developed. The referee himself
- 248 acknowledges that our approach is rigorous. It is even more rigorous now, but the results we obtain are
- essentially the same. We are however extremely worried that the referee acknowledge himself he has a
- 250 potential bias in assessing our work. As for the 2 issues raised by the referee:
- 1: we answer to this point in details in section MC2/MC4 of the present response: our new additions shouldclear things up for the referee.
- 253 2: this point was highlighted in the new discussion of the previous version of the paper and the answer to the 254 reviewer (we again answer to this in detail below). In essence, we fundamentally disagree with the referee, and 255 as explained before, the lack of a rollover is not due to the segmentation approach as amalgamation tend to 256 limit the number of small landslides and favor the occurrence of a rollover (something that can be hinted at in 257 figure 14c as Dm is increased). It obviously impacts the scaling exponent for large landslides, a fact that we 258 illustrated in the MS, and still illustrate in fig. 14c. We made very clear in the previous version of the MS that
- 259 our segmentation approach was far from perfect, and we are certainly not trying to hide this fact under the
- 260 carpet !
- 261
- You could easily address my outstanding major concerns by softening your claims. For example, you say in
   your response: *"its application in comparison with a 2D landslide inventory shows that landslide under-*
- 264 *detection in image based inventories is extremely prevalent in our study area"*. I broadly agree but I would say:
- 265 1) that the comparison "suggests" rather than "shows" because you cannot identify which inventory is in
- error only argue which is more probably the source of the error; and 2) that the under-detection is "present"
- rather than "extremely prevalent" because you can argue that it is very likely that some of the size-
- dependent bias between inventories is extremely likely to be due to 2D under-detection but some is also
- 269 extremely likely to be due to 3D over-detection and you cannot currently identify their relative share.
- Two of your key conclusions (stated in the abstract) are that the manually mapped 2D inventory *"severely underestimates total area and volume"* [L20] and that there is *"a systematic size-dependent under-detection*

- 272 in the 2D inventory" [L24]. However, both these statements are underpinned by an assumption that the 3D
- sources are correct (i.e. the ground truth) such that differences between them and the 2D inventory are
- attributable to error in the 2D inventory. This assertion needs to be justified in the paper but it is not at
- present. Instead you consistently assert and assume that in cases where the two inventories differ it is the
- **3D inventory that is correct.**
- 277 Following the reviewer requests we have soften some of our claims (see detailed comments). However the
- addition of a manual labelling of a large fraction of the 3D dataset allow us to better support our claims when it
- comes to the comparison between the 2D and 3D inventory. In particular, we can now better demonstrate the
- 280 systematic size dependent under-detection because we obtain a similar tendency (fig. 15) if we use the labelled
- 3D sources or the predicted 3D inventory. We do not (and did not) claim that the 3D dataset was completely
- exhaustive, as we showed that it misses some of the landslides detected in 2D, and it cannot detect landslide
- smaller than 20m2 while the 2D inventory can.
- 284 Another key conclusion of the paper is that the 2D size distribution has a rollover whereas the 3D
- distribution does not and that this is due to missed landslides in the 2D inventory. However, it is not clear
- that this is a fair comparison. The rollover is detected in the manual mapping on the basis of a reduced
- frequency of landslides in the smallest class 13-20 m2, relative to the class 20-31 m2 (which is the modal
- class). This smallest class is below the lower limit of detection for the 3D method. If you enforced a single
- 289 consistent lower size limit for your analysis and censored all landslides smaller than this limit for both
- 290 datasets then I don't think you would conclude that the manual mapping displayed a rollover. Note, that the
- 291 x-axis values for the size distributions in Figure 8 are lower bin limits not central values this is potentially
- 292 confusing and should be adjusted.
- **293** First, we now use central bin values as suggested by the reviewer, as opposed to the lower bin limit for all
- 294 graphs. For the pdf(A) (fig.11a), it results in an even clearer rollover of the 2D labelled inventory as the first two 295 points shows an increase in frequency with size. Second, it seems important to show that the 2D approach has
- a higher resolution than the 3D, and thus we do not see why we should censor this graph to the same
- 297 minimum value than the 3D data. However, it is fair to say that we cannot state that the 3D data lacks the
- 298 rollover that the 2D data exhibit. The only rigorous and strict conclusion is: if the 3D data were to exhibit a
- rollover if would be for sizes smaller than 20 m2, a value much lower than any of the previously published
- 300 values (see review by Tanyas et al, 2019). We have modified the text (in places highlighted by the detailed
- 301 comments of the reviewer) to make things clearer in particular in the discussion section 5.3.2.
- 302 I think you make one further important finding that you could highlight in the abstract: you demonstrate the
- 303 variety of types of topographic change that occur in response to an earthquake and show that existing 2D
- 304 landslide mapping captures only a small part of that range.
- 305 Indeed, and we have added a sentence relating to that in the abstract (L38)
- 306 Your results prompt questions about what constitutes a landslide within these landscapes and how we 307 should delimit them. This is particularly important for size distributions because the way that you define 308 both your term landslide (to say what is in or out of the class) and the boundaries of your landslide in space 309 on the basis of post failure observations will differ depending on the motivation for examining them. For 310 example, your point on L420 highlights the complexity of mapping post-earthquake topographic change and 311 relating it to processes. Should subsidence / retrogressive slumping upslope of a catastrophic landslide be 312 included within the same source zone? Is this one landslide or two? The processes and perhaps even the 313 timing of movement are quite different. But it is a very important point that these movements will not be 314 captured in conventional inventories though there is widespread recognition of the processes you discuss 315 based on field reconnaissance.
- This is a very interesting comment. It addresses the question of the subjectivity of landslide delineation thatmakes automated landslide segmentation even more complex. We do not know if this comment was a request
- 318 for further modification of the MS, or just a general comment.

- 319 In the following, to avoid adding further comments to our previous comments lacking line numbers, we have
- 320 added in blue and bold line number that were missing. These line number refer to the new version of the
- 321 manuscript with tracked change.

322

**MC1) The paper needs to more clearly define: 1) landslides (i.e. what the inventory includes) and 2) what 323 your inventory can and cannot be used for. 324**

- Early on you introduce the idea that there are different landslide processes (L43, "process specific") but you 325
- 326 don't follow this logic through into your results. Instead, your analysis may contain an implicit definition of
- 327 landslides as all processes responsible for surface change that cannot be attributed to fluvial processes (L217-
- 328 9). You certainly need to make this definition of landslides explicit in your introduction.
- 329 The introduction needs a much clearer explanation for what you expect the inventory to be useful for. If it is for
- 330 understanding landslide mechanics then it is essential that you make an effort to distinguish individual
- 331 landslides on a mechanistic basis (see MC3 on amalgamation). If that is not an expected application of the 332 inventory you should say so, otherwise there is a real risk it will be misapplied.
- 333 Appropriate uses of the inventory (e.g. volume estimation or landslide mechanics) depend not only on its
- 334 purpose but also on entry criteria into, and distinctions within, the inventory. Non-fluvial surface change that
- 335 might result from earthquake shaking includes: tree-throw, ravel, rockfalls and slides, slow earthflows, rapid
- 336 soil slides and debris flows. The processes responsible for these surface changes differ from one another to
- 337 varving degrees. If there is no distinction between them this precludes the inventory's use in analysis of landslide
- 338 mechanics and therefore prediction. It allows comment on correlation e.g. of volumes and areas of change, but 339
- makes it extremely difficult to make any inference about causation. It also opens the work to the criticism that 340 the bulk statistics mask important differences in behaviour between processes. For example: the differing size
- 341 distributions for rockfalls (where others have reported no detectable rollover) and landslides (where there
- 342 usually is).
- We added the following definition of what we consider as "landslide" in the introduction of the manuscript 343
- 344 (L.127): "We use the generic term of "landslide" to define the spatially coherent changes detected by our method
- 345 on hillslopes that result in at least several decimeter erosion (i.e., scars or sources) or deposition". The discussion 346 now features an entire paragraph ( $\frac{1.596}{1.596}$ ) addressing the various type of landslides that can be detected by our
- 347 approach. The aim of this paper is not to better understand landslide mechanics as we cannot confidently identify
- 348 the different landslide processes we detect. We are mainly interested by the estimation of co-seismic volume and
- 349 to overcome issues such as under-detection and amalgamation on volume estimation. The introduction now
- 350 clearly integrates these two problematics. We also believe that the new filtering approach that we introduce,
- 351 which results in 3 time less landslide sources compared to the initial MS, results in a much more robust
- 352 inventory.
- 353 RE: definitions. Simply defining landslides as decimetre scale change not due to fluvial processes is
- 354 accurate but will be a very unusual definition to the reader. You can help them to see that your definition of a landslide is consistent with theirs by adding a little more explanation and I think that would be very
- 355 356 worthwhile.
- 357 Crozier suggests that: "The three most widely used classifications involving landslides (Sharpe, 1938;
- 358 Varnes, 1958 and 1978; Hutchinson 1988) separate 'mass movements' (Fairbridge, 1968) into two categories:
- 359 subsidence (which is the vertical sinking of material-see entry on Land Subsidence) and those movements that
- 360 occur on slopes. These' slope movements' are then usually divided firstly into 'landslides,' as defined above,
- 361 and secondly into the slower, more widespread and ill-defined movements such as 'creep,' sagging,' and
- 362 'rebound." The landslide definition that he refers to is: "the downward or outward movement of a mass of
- 363 slope-forming material under the influence of gravity, occurring on discrete boundaries and taking place
- 364 initially without the aid of water as a transportational agent."
- 365 Crozier M.J. (1999) Landslides. In: Environmental Geology. Encyclopedia of Earth Science. Springer, Dordrecht.
- 366
- 367 I think you can make the case that most of the change that you detect can be classified as landslides
- 368 following the definitions of Sharpe (1938), Varnes (1958, 1978), Hutchinson (1988), and Crozier (1999). But
- 369 you need to make that case. If you explain the timescale over which the change occurs and the spatial limits
- 370 of detection that you will ultimately impose then you can argue that everything that you detect should fall
- 371 within the class of landslide. It would help your later discussion if you gave a summary of what that might
- 372 include (e.g. slides where the failed material is entirely removed from the source zone and those where
- 373 movements that are small relative to the length of their failure surface). It would also help to explain which
- 374 non-fluvial mass movement processes are not detected, particularly: tree-throw, ravel and other forms of
- 375 creep (because the movements are either too small, too localised, or too slow).

- 376 First we note that the definition of what constitute a landslide is highly variable (e.g., Tanyas et al., 2019), and
- that referee 1 did not have any trouble with the definition we propose. Authors publishing 2D inventories of
- 378 landslides generally use a very loose description of "what's measurable" in terms of optical difference on
- hillslopes. While we can understand the need for the referee and some readers to place our results in the
- 380 framework of existing classifications, we think that some of these classifications were based on a descriptive
- approach which favored highlighting the heterogeneity and peculiarity of sites, while in the end a landslide can
- be described simply as a large amount of earth and rocks falling down a cliff or the side of a mountain (Collins
   dictionary). We do not want to err too much on the descriptive side of things in the introduction and prefer to
- 384 stick with a simple definition on which we elaborate during the discussion as a function of our observations.
- 385 Because we now use the proximity of deposits in the manual labelling and through the CDD, and because the
- 386 signal we measure is topographic change which can corresponds to both erosion and subsidence, we have
- 387 updated this definition in the introduction as follow (L127 to L138):
- 388 *"We use the generic term of "landslide" to define the spatially coherent changes detected by our*
- 389 *method on hillslopes that result in at least several decimeters of negative topographic change*
- *associated with a downstream positive topographic change. Patches of negative (resp. positive) associated with a downstream positive topographic change. Patches of negative (resp. positive)*
- **391** *topographic change are called sources (resp. deposits) and correspond to erosion (resp.*
- *sedimentation) or subsidence (resp. accumulation). This definition therefore includes all the types of*
- 393 mass wasting processes involving the downward or outward movement of soil, rocks and debris under
- the influence of gravity, occurring on discrete boundaries and taking place initially without the aid of
- **395** water as a transportational agent (Crozier, 1999)"
- **396** We have embedded the definition of Crozier and reference to Crozier as suggested by the reviewer.
- 397 The variety of landslide processes that we capture is discussed in section 5.2.1 and underlined in the abstract398 (L30) and conclusion (L1297).
- 399 It would perhaps also be worth saying that this definition differs from those commonly (implicitly) applied
- in 2D landslide inventories derived from satellite imagery since these rarely (or incompletely) capture
   slides where material is displaced by only a fraction of the failure surface. These inventories are censored
- 402 by their ability to detect change from image properties and thus rarely capture rockfall source zones. The 403 same censoring results in under-sampling of small landslides, landslides in bare or sparse vegetation and
- 404 landslides obscured by forest canopy because these can't be confidently identified.
- 405
- 406 This is a very good point, but we think that stating this in the introduction would probably be confusing, because
  407 most readers will have no idea of what the 3D topographic differencing actually generate. We have added the
  408 following sentence in the discussion section 5.1.3:
- 409 "These inventories are also censored in the variety of landslide processes they can capture as they
  410 rarely capture rockfall source zones on very steep hillslopes and rotational/translational landslides
  411 where material is displaced by only a fraction of the failure surface."
- 411 where material is alsplaced by 412
- 413 RE: expected uses. Clarifying the focus on co-seismic landslide volume estimation is useful as is the section
   414 that you have added on the different processes represented in your inventory.
- 415
  416 We did not understand to which part of the manuscript it referred or if it was a comment or a request for change.
  417 We did answer to one comment of the referee in the pdf version (L50) for the introduction which seems to relate
- 417 We did 418 to this.
- 419

**420 MC2) Elevation errors need to be better quantified and more thoroughly discussed**

- 421 The manuscript needs a more thorough treatment of errors in the topographic data dealing with both: 1) the
- 422 properties of the elevation errors that you have identified (e.g. spatial pattern, wavelength, covariation with
  423 landscape properties); and 2) the possible sources of error.
- 424 The reviewer suggested many areas to explore that are extremely interesting, but which, for some of them, would
- 425 constitute an entire paper by themselves, in particular when it comes to the analysis of error properties suggested426 by the reviewer.
- 427 I am not suggesting that all these areas need to be exhaustively explored but that the inferences that you
- 428 draw from them must be stated with appropriate confidence for the uncertainty in the data that
- 429 underpins them. Mass balance, which was your primary objective is largely insensitive to the errors that I

- highlight here. Landslide size distributions and scaling relationships are potentially very sensitive to these
   errors.
- 432 We also aim at developing a generic workflow applicable to a variety of cases for which users may not
- 433 necessarily perform extensive error properties analysis.
- 434 Your contribution in developing a workflow is extremely valuable and is not compromised by the
- 435 continued presence of these errors. However, prospective users will also use this paper as a model for what
- 436 the workflow can be used to calculate. Your discussion of errors and their implications is therefore
- important because it will influence not only how people interpret your results but also the capability and
  limitations of the workflow.
- 439 Hence, to improve the paper we have worked on two aspects: (i) improving and better identifying errors in our
- dataset, (ii) defining a new confidence metrics (the SNR) for each landslide source or deposit to filter out landslide with low confidence.
- 442 Both these aspects have very considerably improved both the workflow and its application in this paper.
- 443 However, you have not addressed my original concern about spatially correlated error in this comment.
- 444 *RE error properties: The amplitude and correlation length of elevation uncertainty from different sources and* 445 *how they interact to generate a 2D elevation error field with a particular amplitude and wavelength is really*
- how they interact to generate a 2D elevation error field with a particular amplitude and wavelength is really
  important for this particular application, where landslides are identified by thresholding then segmenting
- 446 *important for this particular application, where tanasities are taentified by inresholding then segmenting* 447 *differences. The error appears to have a fairly long wavelength in many areas (tens to hundreds of metres). It*
- 447 alignmences. The error appears to have a fairly long wavelength in many areas (tens to hundreds of metres). It 448 also appears to have some aspect dependence. The spatial correlation of these errors is important because you
- also appears to have some aspect dependence. The spatial correlation of these errors is important because ye
- 449 assume uniform isotropic registration errors.
- 450 I do not see where this comment is addressed in the response.
- 451 This comment was initially addressed in the general answer to the referee comment L 955 to L 960 and in
- 452 section 5.1.2 of the discussion, specifically in L718 to L724 of the previous MS with tracked change:453
- 454 Now, we have added new elements in our MS regarding elevation errors:
- 455 1. A more extended presentation of the potential source of elevation errors in airborne lidar data in section 456 3.1 separating uncorrelated errors (survey noise, surface roughness) and correlated errors (time 457 dependent attitude and position error; intra-survey registration error of flight lines; inter-survey rigid 458 registration) (L276-292). These are introduced in the context of the distance uncertainty model we use 459 (eq. (2) and how they have been evaluated in previous work by (i) a spatially explicit direct error model 460 propagation, or (ii) via an empirical analysis on areas that are perfectly stable, horizontal and smooth 461 (e.g., with a classical variogram approach as in Anderson, 2019 that the referee suggested). We explain 462 why neither approach is feasible in our study site given the lack of (i) trajectory data (for a direct 463 model), and (ii) of large enough stable and flat surface (for an empirical approach). Hence, and let me 464 stress that because it seems that the referee think we are lazy or uniformed when it comes to LiDAR 465 data generation and analysis: it is not that we do not want to quantitatively study the properties of 466 elevation errors or generate a non-uniform reg model. It cannot simply be done at present, and we 467 expect that it will always be the case when future researchers will apply our workflow for landslide 468 detection.
- 469
  470
  470
  470
  471
  471
  471
  471
  472
  472
  473
  474
  474
  474
  475
  476
  476
  477
  477
  478
  479
  479
  470
  470
  470
  470
  470
  471
  471
  471
  472
  473
  474
  474
  474
  474
  474
  474
  474
  474
  474
  474
  474
  474
  474
  474
  474
  474
  474
  474
  474
  474
  474
  474
  474
  474
  474
  474
  474
  474
  474
  474
  474
  474
  474
  474
  474
  474
  474
  474
  474
  474
  474
  474
  474
  474
  474
  474
  474
  474
  474
  474
  474
  474
  474
  474
  474
  474
  474
  474
  474
  474
  474
  474
  474
  474
  474
  474
  474
  474
  474
  474
  474
  474
  474
  474
  475
  474
  474
  474
  474
  474
  474
  474
  474
  474
  474
  474
  474
  474
  474
  474
  474
  474
  474
  475
  474
  476
  476
  477
  476
  476
  476
  477
  476
  476
  476
  477
  476
  476
  476
  476
  477
  476
  476
  476
  477
  476
  476
  476
  476
  476
  476
  476
  476
  476
  476
- 475
  476
  476
  476
  477
  3. The addition of the manual labelling of true landslide sources and false detections allow us to highlight the occurrence of these errors and study their prevalence, their size distribution, and highlighting the critical role of ground classification errors in forested area (a new result).
- 478 As far as I can tell: 1) there are spatially correlated errors in your difference surface; 2) these errors will 479 not be captured by the SSDS analysis; 3) correlation of errors implies that if one core point has errors 480 large enough to exceed LoD then there is a non-trivial probability that one of its neighbours will also have 481 errors that exceed LoD; and 4) the distribution of erroneous patches will be strongly right skewed (i.e. 482 smaller patches more probable than larger patches). I would be keen to know whether you agree. On this 483 basis, I think you must quantify the impact of spatially correlated errors on your size distribution if you 484 are to argue that your measured size distribution is the 'true' distribution or even that it is more correct 485 than the 2D distribution.
- 486
- 487 Answers to the above comments:

- 488 1) we agree there are spatially correlated topographic change associated to elevation errors and this was stated in 489 all versions of the MS, in particular when it comes to flight line misalignments and to ground classification 490 errors of the pre-EQ data. In the previous MS with tracked changes this was indicated: 491 in section 3.3.1 (registration error estimate) in L291 to L295 given that bias and error between 492 flight lines for each survey are spatially correlated errors. 493 In the result section 4.1 : L461-464 (effect of SNR filtering in removing low amplitude 494 topographic change related to flight line misalignment) 495 In the discussion : L709-710 496 But what matters is the new version in which the above parts have been largely rewritten and extended: 497 we have added a detailed description of some typical false detection (new fig. 5) due to spatially 498 correlated errors originating from incorrect ground classification data and flight line misalignment in 499 the pre-EQ. Fig. 6c now shows the labelled false source detections. 500 Section 4.2.1 presents the prevalence of these false detections and their size characteristics, and other M3C2 derived metrics. The prevalence of false detections in forest-free area is relatively low (24.4 %). 501 502 It is however difficult to separate the influence of intra-line, intra-survey and inter-survey errors on 503 these and we did not try to separate these, and we do not see how it would be easily feasible. In forested 504 areas, false detections dominates the inventory (80%) and are mostly due to ground classification errors 505 in the pre-EQ data, an important result of the new MS highlighting the importance of high quality 506 LiDAR when working on evergreen forests. 507 A new discussion section 5.1.3., further address the occurrence of these errors and the importance of • 508 filtering. 509 2) yes, the SDDS is not designed to evaluate these effects. It is now only used to evaluate the optimal parameters 510 of normal scale and projection scale for M3C2. 511 3) yes, but these errors are largely filtered out. 4) indeed, and using the labelled data this now shown in the new figure 7 highlighting the prevalence of false 512 513 detections as function of patch size, as well as in fig. 14a of the pdf(A) of false detection compared to true 514 landslides 515 516 RE error sources: It isn't clear to me what you mean by imperfect alignment, nor ICP related errors (L204-5). 517 Identifying errors on Fig 3 and hypothesising the sources from which they derive are useful but need to be 518 discussed in the manuscript as well. You recognise the presence of "internal flight line height mismatch" and 519 indicate that it results in "large scale low amplitude topographic change" (L418-22). They should be introduced 520 earlier in the article with a more complete explanation of what they are and how you found them. 521 Errors in our dataset: entirely revisiting our data, we identified two sources of errors: (1) Remaining LiDAR 522 point cloud misclassification in forest areas, inducing local topographic errors, and (2) imperfect flight line 523 alignments from the pre-EQ data, inducing topographic errors of longer wavelength. To address the first issue, 524 we first removed as much misclassified points as possible by interpolating a surface and remove outlier points 525 (see detail in supplementary material and section 2 in the paper). To address the second, we estimated residual 526 errors of each flight lines due to imperfect flight line alignments composing the pre-EQ point cloud and defined 527 the registration error reg based on the maximum residual error of the flight line misalignments (section 3.3.1. 528 and S3 in supplements). Compared to the previous version of the MS, the reg is now 3 cm larger (20 cm vs 17 529 cm). We also show that only 1% of points are detected as significant change in the stable area, validating our 530 choice of LoD95%,. While our reg is considered uniform over the study area, the LoD95% (eq.2) also take into 531 account the local point cloud density and roughness which are correlated to the presence of vegetation. The 532 LoD95% is thus spatially variable. In addition, we are aware that, ideally, a spatially variable model for point 533 cloud error and registration would be preferable for each survey and combined into a more accurate and 534 complete form of LoD than what the M3C2 approach currently offers. However, in the absence of the position 535 and attitude information of the sensor (e.g., Smoothed Best Estimate of Trajectory file) and raw LiDAR data -536 rarely available on LiDAR data repositories -, or of dense ground control which is hard to get in mountainous 537 environment, it is currently impossible. We now discuss this in the discussion (section 5.1.2).
- 538 This is a useful explanation but the decision to assume that registration error is spatially uniform still needs
- justifying in the text in a way that addresses the concern that long-wavelength errors might combine with
- short wavelength errors to generate patches of erroneous change in some places and break up patches of
- 541 true change in others.

- 542 This is a case where we consider that the referee do not know where to stop in his requests. We will not add
- another page of MS to address this. We already discussed at length in the previous version (see response
- above) why we cannot generate a non-uniform registration error, and highlighted in the discussion that it
- would be an important development to have one. We show that our main results are not altered by increasing
- reg (fig. 14b). And most importantly we now show that our filtering approach get rid of most of the false
- 547 detections. That being said the new version of the method section in which we present the LoD contains a
- 548 justification of our choice of reg that should answer the reviewer requests.
- 549 MC3) You recognise that segmentation results in amalgamation but don't quantify its extent or impact
- 550 Severe amalgamation can result from automated segmentation of a thresholded classifier. In the landslide maps
- that you show here (e.g. Fig 2), amalgamation appears a severe problem for the largest landslides. The
- argument that it can be solved by 1000 tuning Dm (as you suggest on L426-7) is unconvincing since two
- separate landslides can be within millimetres of one another but have different failure mechanisms. You later say
- that you "cannot resolve the amalgamation" (L511) and that it "is still a potential issue" (L558). I would argue
- that it is not potentially but certainly an issue. Your figures show that amalgamation is present (perhaps
- pervasive) in your inventory but its extent or impact is not quantified. The total volume is insensitive to
   amalgamation but your landslide pdfs and scaling relationships are not. 1005
- 558 We agree that the segmentation approach we use certainly do not allow to solve the amalgamation problem, and
- the agree that the segmentation approach we use certainly do not allow to solve the analgamation protein, and
   this is highlighted in many parts of the MS. However, the problem of amalgamation is inherently subjective and
   plagues all inventories.
- 561 I agree, that segmentation is a subjective problem but you have made it reproducible by removing the
- subjectivity. The problem is that the best reproducible (i.e. automated) segmentations still perform poorly
   (with respect to the segmentation that a human mapper would choose).
- 564 Our 3D data reveals a level of complexity, and a density of amalgamated landslides which makes the definition 565 of a single landslide in relation to an ideal rupture mechanism extremely difficult.
- 566 This is a really important point and could be a key contribution of the paper. Your results show that it's 567 complicated. Far more complicated than we capture in conventional 2D inventories.
- 568 Even the segmentation of the 2D inventory proved to be extremely complex and is entirely not reproducible.
- Hence, we favour a reproducible approach, even if currently limited, that can be applied exhaustively to much
- 570 larger datasets, than a non-reproducible one (2D manual mapping) that we now demonstrate misses a very large571 number of landslides and incorrectly map their contour.
- 572 It isn't clear why reproducibility is favoured over skill. If you think that manual segmentation would
- 573 outperform connected component segmentation it seems strange to continue with automated approach
- because it is reproducible. You say that 2D manual mapping "misses a very large number of landslides"
  but this is on the assumption that the 3D inventory is correct.
- 576 In this new version of the paper, the landslide amalgamation can be visualized with a map of the landslide source 577 colored by individual landslide as defined by the method (section 4.2 Fig.7). Moreover, the comparison between
- 578 both inventories shows 1025 that while 171 of 2D-sources are shared with 3D-sources, it represents 144 sources
- 579 3D-sources. This highlight that 25 landslide sources are amalgamated in the 3D inventory (L-. As in the previous
- version of the paper, we perform a sensitivity analysis of the impact of  $D_m$  showing that landslide statistics are not severely affected by this parameter for  $1.5 < D_m < 3$ . We also explored density based spatial clustering
- solution severely affected by this parameter for  $1.3 < D_m < 3$ . We also explored density based spatial clustering algorithm used in 3D rockfall segmentation, derived from DBSCAN (Ankerst et al., 1999; Martin Ester, Hans-
- Peter Kriegel, Jiirg Sander, 1996; Tonini and Abellan, 2014) and HDBSCAN (Carrea et al., 2021; McInnes 1030)
- et al., 2017). None of them managed to provide a significantly better segmentation of the largest landslide and
   are significantly longer to run than the connected component algorithm we use. We now thoroughly discuss
- 586 about this in section 5.1.2.
- 587 The analysis that you have added demonstrates that the problem is not that your particular segmentation
- 588 approach is worse than the alternatives but that automated segmentation itself is problematic. You make
- 589 the argument above the subjective segmentation is equally problematic. I think making this point in a more
- 590 detailed discussion of the problem of segmentation (both automated and manual) would help to address my
- 591 concern here.
- 592 We have a fundamental disagreement with the referee on the value of adding a manual segmentation to our
- approach and we consider that the requests for further discussions are unnecessary diversions from the core of
- the paper and would impact the readability of our work. We already have put a strong emphasis in saying that
- 595 our segmentation approach is far from perfect and needs to be improved in many places of the MS: abstract
- 596 (L35), discussion (L916-942), conclusion (L1307). Referee 1 has strictly no problem with the limitations of our
- approach. We also explain(ed) (L1169-1171) that the lack of rollover cannot be related to the current tendency

- 598 of the segmentation to generate amalgamation as this tend to reduce the number of small landslides and
- 599 would indeed favor the occurrence of a rollover. And we show (fig. 14c, fig S12 and S13) that, obviously, the
- 600 parameter *Dm* of the segmentation changes the values of scaling exponents.
- 601 Now, with the new elements on the prevalence of false detection and the importance of filtering, we have
- added new elements in the discussion highlighting that automatic segmentation is a necessary step to generate
- a preliminary set of potential sources and deposits on which filtering is applied (L925) : "As we aim to apply our
- 604 workflow to very large dataset with potentially several tens of thousands of landslides, and that false detections
- 605 filtering need to operate on segmented patches, automatic segmentation is a mandatory step.". We have also
- added (L938) : "Manual segmentation can also be envisioned as a refinement after the predicted true landslide
- 607 sources and deposits have been produced, but is non reproducible and time consuming when applied to very
- 608 large datasets."
- 609 MC4) Topographic errors propagate through segmentation to introduce a bias towards small landslides. Existing
   610 experiments to quantify this bias are insufficient.
- 611 You say on L559-60 that "our approach has the benefits of more systematically capturing small landslides than 612 traditional approaches". However, this is one of my main problems with the paper given potential propagation
- 613 of topographic errors through thresholding and segmentation.
- 614 *You show (in SI) that: 1) in the absence of any real topographic change your detection algorithm generates*
- artefact landslides of 1-20 m2 purely due to spatially uncorrelated topographic noise; 2) this noise generates
- 616 *many more small than large* 1045 *landslides; 3) in this experiment artefacts* >20 *m*2 *were extremely rare.*
- 617 However, this does not demonstrate that predicted landslide size is insensitive to longer wavelength topographic
- 618 errors (known to be present in the data); nor even to short wavelength noise in the presence of longer
- 619 wavelength surface differences (e.g. real landslides). First, even without any real topographic change (i.e. no
- 620 real landslides), the size distribution of erroneous landslide-like clusters will depend on the spatial correlation
- 621 length of the difference errors, which in turn depends on the correlation lengths of the errors in the surfaces
  622 being differenced. Longer error wavelengths will enable the generation of larger error clusters. Your figures
- 623 show that topographic errors are clearly not uncorrelated and you recognise this yourselves (L418-22); nor do
- 624 the errors appear to have a single characteristic wavelength. This is a hard problem but one that you must deal
- 625 with if you are to convince the reader that the landslide inventory you have generated is not hopelessly biased by
- 626 these landslide-like artefacts. Second, the problem is not only that clusters of erroneous negative surface
- 627 difference due to roughness (or other errors) can create artefacts that appear to be landslides, but also that
- 628 clusters of erroneous positive surface difference are collocated with real topographic change (e.g. due to a
- 629 *landslide*); these can interfere negatively with real changes reducing the surface difference below the threshold630 for detection and breaking a single landslide into multiple patches.
- 631 *Oversampling of small landslides is important because it undermines your most surprising and high impact*
- 632 claim: that rollover reported in previous inventories is due to under detection (L573-4). I am not currently
- 633 convinced by this claim because you do not exclude the possibility that the lack of a rollover is solely due to
  634 detection errors. You need to quantify these detection biases before you can make these claims.
- 635 Suggested additional analysis: A "more advanced segmentation" (L431-3) may be out of scope for this paper.
- 636 *However, an indication of the impact of the simple segmentation on object based classification skill is an*
- 637 essential requirement of this paper if it is to retain the current approach to identifying discrete landslides. Two
- 638 possible avenues could be followed to provide such an indication. **First**, your analysis of topographic changes
- 639 on the stable surfaces (Fig 3) would allow you to perform the same analysis that you have performed in the SI
- 640 but using the pre- and post-EQ surfaces for the stable zones identified in Fig 3. This would enable you to identify
- 641 the size distribution of artefacts that can be generated from topographic errors with a spatial correlation length
- 642 *closer to that for the unstable parts of the study area. This though still does not account for the possibility that*
- the landslide erosion signal itself is altered by the noise (e.g. by disconnecting clusters). Second, you could
   collect a landslide check dataset using independent observations. This might take the form of an entirely
- 645 independent inventory but should certainly also involve cross-checking to confirm the existence and
- 646 *characteristics (e.g. area, shape, depth) of your predictions.*
- 647 We added further analyses to the method to deal with topographic errors and erroneous landslide. We also added
- an entirely new 2D landslide inventory as suggested by the reviewer. We are now confident that the actual
- landslide inventory corresponds to real changes. Please see our reply to the MC2) and MC8) comments for adetail answer.
- 651 Most of my concerns in this comment have been left unaddressed. MC2 focusses on topographic errors but
- 652 your response to MC2 doesn't deal with the problem of spatially correlated errors. Because you are
- 653 interested in the size of patches generated by thresholding the difference surface it is essential that you
- 654 examine the spatial structure of the errors. I will deal with each of my comments in turn reflecting on

655 whether they have been addressed in the new manuscript even if they have not directly been addressed by

656 your response to this comment.

657 First, you did not "demonstrate that predicted landslide size is insensitive to longer wavelength topographic 658 errors (known to be present in the data); nor even to short wavelength noise in the presence of longer wavelength surface differences (e.g. 9 real landslides)". You remove reference to an SSDS test to set the 659 660 minimum area but retain that test to optimise the SNR. However, when you introduce it you do not 661 recognise that it synthesises uncorrelated noise while the two surfaces that you are differencing both 662 include spatially correlated elevation errors. You do not include any description or explanation in the text 663 for the "stable areas error" shown in Figure 4, these errors appear to be spatially structured on multiple 664 length scales from tens to hundreds of metres.

665

In essence, there is no way for us to evaluate how the various elevation errors impact landslide detection,
because we cannot evaluate properly the spatial structure of these elevation errors (see response to MC2). We
could play with synthetic data, adding another 3 pages to the MS showing that if we add a 2D sinusoidal error

to the two dataset with a specific wavelength we would certainly detect significant change if we do not

670 properly set the registration error, and if not filtered out it would generate a landslide size with a certain

distribution. However, we will not do it because: (i) we have no more time to do it in the time frame of the PhD

672 project of Thomas Bernard which finishes soon; (ii) it is not necessary to support our conclusions; (iii) it would

not help in getting better inventories, because in the end the critical step is the identification of true landslides

and false detections which requires well-chosen filtering metrics.

675 We now show examples of false detection occurring on spatially stable areas (fig 5b, fig. 6c, section 3.5.1,

section 4.2), but also the much more predominant source of error related to ground classification errors in the

677 pre-eq forested area (fig. 5a and 6c). What matters most, is that our new filtering metrics (CDD+SNR) in non-

678 forested areas results in excellent removal of false detections (97% precision) while preserving a large part of

- true landslides (79%) and all true landslides above 40 m2. The new final predicted inventory does not contain
- 680 any source or deposit in the stable areas (L709).
- 681

682 Second, I argued that "Oversampling of small landslides is important because it undermines your most 683 surprising and high impact claim: that rollover reported in previous inventories is due to under detection" and 684 that "You need to quantify these detection biases before you can make these claims." I'm still not convinced by 685 this claim because I still don't think you have excluded the possibility that the lack of rollover is solely due 686 to detection errors. You need to quantify the size dependent detection bias in the 3D inventory and/or to 687 considerably tone down your claims about rollover in this and other 2D inventories being due to under-688 detection.

689
690 Using the labelled 3D sources and false detections, we now demonstrate that when considering only manually
691 validated landslide source, the rollover is still lacking (fig. 11a). We also show that if no filtering were applied,
692 then the statistics of small landslides would be indeed significantly biased towards small landslides as expected
693 by the referee (fig. 14a), in particular in forest area given the large prevalence of ground classification errors.

694 We hope the addition of the labelled data will convince the referee. However, we will not tone down our claim

695 because we are even more confident that the lack of rollover above 20 m2 is a real feature of our dataset.

696

697 Third, I suggested that you: "perform the same analysis that you have performed in the SI but using the pre-698 and post-EQ surfaces for the stable zones identified in Fig 3. This would enable you to identify the size 699 distribution of artefacts that can be generated from topographic errors with a spatial correlation length closer to 700 that for the unstable parts of the study area". I don't see a response to this suggestion here but when I raised 701 the same point in a minor comment (related to L196). you responded that "Applied on stable area, the 702 workflow does not detect any landslide." This result would definitely be worth reporting! However, I think 703 we must have misunderstood one another, I can see many patches of significant change (>150) within the 704 stable areas, most of these patches of significant change are removed in Figure 5c. Is that because they are 705 smaller than 20 m2? However, even after this filtering I can still see several landslides within the stable 706 zones in Figure 5c. I phrased this as a suggestion in my previous review but I really think this is one of the 707 few opportunities that you have to build confidence in your method. It remains a weak test because you 708 chose the stable areas based on areas of limited change in the difference maps and because they

oversample non-forest vegetation but in the absence of field checks to the inventory this remains one of the
 best tests I can come up with. My second suggestion to *"collect a landslide check dataset using independent observations"* is dealt with in a separate major comment MC8 and doesn't need further discussion here.

712

713 We believe our manual labelling of true landslide and false detection offer a richer and more complete approach
714 to the issue of false detections due to elevation errors than just looking at the statistics of false detections only on
715 stable areas. As stated above there are no source on stable area in the final predicted inventory (L680)

- 716 717
- 718 MC5) Findings that differ from previous work
- 719 There are a number of unusual findings that are worthy of comment because they are some of your most
   720 interesting and potentially important findings. It is essential though that each is carefully examined and that the
- 720 interesting and potentially important findings. It is essential mough that each is carefull. 721 critique that it might have arisen due to methodological errors is dealt with head on.
- 722
- 723 *First, it is not unusual to identify more sources than deposits due to amalgamation of landslide deposits.*
- However, it is unusual for deposit areas to be smaller than source areas (L281-2), and for deposit depths to be
   thicker than source depths (L265-70 1115 and L460). These result should be compared with results from
- 726 previous studies.
- First: contrary to the reviewer experience, this result does not surprise us, in particular when the runout oflandslides is not long. The filtered data clearly support this finding.
- 729 This has not been addressed. I commented that you needed to compare your results with those from 730 previous studies with respect to scar and deposit depths and scar and deposit areas. I didn't find this new
- 731 discussion nor a response to explain why it was not necessary.
- 732 Because our study focuses on landslide sources, and our analysis of deposits has not been as detailed as the
- sources, and the discussion is already quite long, we choose to remove this part.
- 734 Third, you identify areas of deposition where there is no upslope erosion (L256 and Fig 4). I don't think these
- can be real deposition zones but instead must be a consequence of incorrect landslide detection. Their spatial
- extent and depth distribution would be a useful indication of the precision of the technique.
- **737** Third: We now filter landslides by a signal-to-noise ratio (section 3.3.4) and are confident that the actual
- 1 landslide inventory corresponds to real changes. Some very small deposit areas may not have upslope erosion aswe expect deposit are to be easily detectable by our method than source areas (section 5.1.2).
- 740 Why do you expect that deposits are more easily detectable than source zones? I would have expected the
- opposite. In my experience deposits can be very thin, (<50 mm) patchy and extensive whereas source zones</li>
   are far more coherent.
- 743 Deposits now makes it to the final inventory only if they are connected to an upstream source (L539). This
- resolve the initial issue that the referee had.
- 745 Fourth, it is extremely unusual that locations classed as vegetated in post-event optical imagery but identified as
- a landslide by another technique are considered by the authors to be genuine landslides (L347-8). Instead, the
- 747 presence of vegetation at the location strongly suggests a false positive.
- **Fourth:** we partially disagree with this statement. As now explained in the discussion large landslides that strip
- out vegetation are obviously mapped in 2D inventory, but small ones that occur on less dense area are extremely
- 750 difficult to map in 2D imagery as our inventory shows. Moreover, vertical subsidence due to upslope
- **751** propagation of landslides is entirely missed in 1135 forest, while it is detected in our approach. We think the comparison with the new 2D inventory will resolve the reviewer's recerve
- 752 comparison with the new 2D inventory will resolve the reviewer's reserve.
- 753 How do you know that small ones that occur on less dense area are difficult to map? Is this on the
- assumption that your LiDAR inventory is correct? If you go to the landslide locations predicted by the 3D
- method do you find evidence in the orthophotos that there is indeed a landslide at that location (even if it
- 756 wasn't independently mappable)? This would help to build confidence in your method. The point about
- vertical subsidence is important and you do a nice job of demonstrating the plausibility of the claim that
- this is real change. It prompts a series of questions about representation of these landslides within an
- inventory derived from surface change but you deal with this nicely in the discussion. My only suggestion is that you prepare readers for this finding in the introduction by adding a more complete explanation of
- 761 the types of landslides that 2D and 3D inventories might include (see MC1).
- 762 The referee was correct in doubting the reality of the landslides in forested area! Manual labelling showed a high
- 763 prevalence of false detection (80%) due to ground classification errors in the pre-eq lidar. We now describe in
- 764 greater detail these errors (fig 5a, 6c), and emphasize in the results section 4.2.1 and the discussion section 5.1.3
- the importance of high point density LiDAR, conducive of less ground classification errors, for change detection
- in evergreen forested area. We however emphasize, and now better describe with the addition of fig. 10 c
- 767 (hillshade view) that large subsidence patterns consistent with the occurrence of scars visible in the post-eq
- 768 hillshade DEM can be detected with our approach

- 769
- 770 MC8) Landslide object detection needs to be tested against an independent dataset

771 The findings in this paper depend critically on the skill with which the proposed method can classify landslide

772 scars and deposits. Thus it is essential that the paper reports testing results that quantify this object based

classification skill. At present, "orthophotos were used to visually validate" the classifier (L115-6) but without 773

774 reporting results of this analysis. I think it is essential that you explicitly explain your sampling and mapping

- 775 strategy for landslide detection from orthophotos in the methods. You should then include a section in the results 776 where you compare your orthophoto based mapping to the surface differencing approach.
- 777 However, it is not enough to simply say the orthophoto mapping did not identify landslides that were identified
- 778 by the surface differencing. You should then go to a carefully chosen (e.g. stratified random) subset of the
- 779 landslides detected by each method that were not detected by the other (i.e. surface differencing but not ortho-
- 780 photo mapping and vice versa) to establish as far as possible which of the two methods was in error and why.
- 781 While finding and mapping thousands of landslides might be timeconsuming (L255), confirming their existence
- 782 and characteristics (e.g. area, shape, depth) would not.
- 783 This lack of comparison between the 3D differencing method and a more classical approach has been addressed
- 784 by adding a manual mapping of landslides based on 2D images. We added a section that specifically explore the 785 differences between the 1160 methods (section 4.2). Moreover, the resulted landslide area distribution mapped
- 786 manually has been added in the figure in section 4.3 and compared with those obtained with the 3D differencing 787 method.
- 788 The 2D inventory considerably strengthens the paper and addresses the comments in the first paragraph
- 789 above. However, much of the second paragraph remains unaddressed. You have added analysis of the two
- 790 inventories and discuss false positives for the 2D inventory where deposit is incorrectly mapped as source
- 791 zone, this is a secure result and is exactly the type of analysis I was looking for. You use your observations
- 792 and theory/logic to argue that one method is correct and the other is in error. The remaining areas of
- 793 disagreement you assign as false negatives for the method that has not identified a landslide at that
- 794 location. This is not a secure result. You have no objective way of establishing which method is in error (i.e.
- 795 whether this is a false positive for one or a false negative for the other) and you don't provide any
- 796 justification for why disagreements should always be treated as false negatives. In fact there is good
- 797 evidence to suggest that the 3D inventory should contain false positives and that these false positives likely
- 798 have a strong size bias. First, the SSDS test (which itself is a very weak test because it assumes that errors are
- 799 uncorrelated in space) results in artefact landslides. Second, the inventory contains landslides in the 'stable'
- 800 areas of the study area. This problem propagates into the discussion where you describe disagreement
- 801 between inventories as error in the 2D inventory under the implicit (but untested) assumption that the 3D 802
- inventory is correct. I don't think you could use this language in the paper even if the tests above generate
- 803 only small numbers of false positives from the 3D inventory.
- 804 The addition of the manually labelled dataset (which contrary to what the referee states, was extremely time
- 805 consuming) answers the problems raised by the referee as we have now a very high degree of confidence in
- 806 the 3D source inventory that we use to compare with the 2D inventory. However, the text comparing the 2D
- 807 and 3D inventory has not greatly changed because the 3D inventory we used in the previous iteration of the 808
- MS was very close to the present one. We have mostly only updated numbers. We consider that the
- 809 comparison between the 3D and 2D inventory (result section 4.3 and discussion section) are covering all the 810 topics above and clearly highlighting the limits of the two types inventories.
- 811 As for tests, the SDDS is no more used to validate the approach (only for parameter estimates for M3C2), and
- 812 we clearly state in the text that the final inventory of sources does not contain sources in the stable area (L709)
- 813

**Outstanding Minor Comments from previous review**

- 814 L114: "using the classification provided": More detail is needed on the method used to classify ground points.
- 815 The survey report of the LiDAR data only mention that the ground points have been automatically classified 816 using the Terrascan software. The reference to the survey report (Dolan and Rhodes, 2016) have been added to
- 817 the text.
- 818 I don't see Dolan and Rhodes (2016) cited in the text. The sentence above should be added to the text. You
- 819 have added a manual guality check and reprocessed the data to remove non-ground-points as a result. That
- 820 is a good additional step that you have introduced since the last version. However, I don't think you can

**simply point to the SI for the details of this analysis. This is a key step in your method and should be included in the article itself.**

823 Dolan and Rohdes (2016) was an incorrect reference. The report is Dolan (2014) and is now in the bibliography.

824 It turns out that our attempt at improving the classification was not successful as a large number of ground

825 classification errors remain (e;g., fig 5a and section 4.2.1). The classification improvement is not a key step of

826 our workflow because the user faces two configurations: (i) either the two datasets are high quality lidar survey

827 (as the post-eq data) in which case the classification provided does not need to be improved (ii) either it has

828 poor legacy data as the pre-eq data, where there's basically not much to do. We have added more details in

- 829 section 2 on the classification (L204-208), and added the following text:
- 830 "We did not attempt to further improve the classification as these errors are expected to occur in low point-
- 831 *density LiDAR survey of evergreen forested areas and will generate false landslide sources that our workflow*
- 832 should detect and filter out We note that the classification refinement is not a critical component of our
- 833 workflow and that other classifications algorithms (e.g., Sithole and Vosselman, 2004) could be used to improve
- 834 or check the quality of the LiDAR ground points before the application of the workflow."
- 835 The discussion also contains reference to the issue of ground classification in relation to the quality of LiDAR836 datasets (section 5.1.3).
- 837 *L137-8: This is not clear: do you mean that you calculate the centroid of each point cloud in 3D then take the*
- 838 magnitude of the 3D vector that connects these two points; or that (for each point cloud) you take the arithmetic

839 *mean of differences from the reference plane (defined by D) in a direction normal to that plane? In either case*

840 you are performing a spatial averaging at length scale d/2 assuming a uniform kernel. First, is it problematic to

- perform averaging over length scales larger than the 1190 core point spacing? Second does it make sense to
- assume equal weight in the average with plane parallel radial distance from the core point, or should some form
  of inverse distance weighted average be used? I would have assumed a weighted average was more appropriate
- 844 but it would be useful for you to explain why an unweighted mean is more appropriate.

845 To calculate the distance between the two point clouds, the average positions i1 and i2 of the point clouds are first

defined and then the distance is computed between the two positions along the normal vector. The average

**847** positions are defined as the arithmetic mean of the distance distribution of each point of the subset of points

(created by the intersection of the cylinder to the point cloud) to the normal vector (or the cylinder axis; see
Lague et al., 2013). As this is part of the M3C2 algorithm that we did not modify, we don't discuss the choice of

850 a uniform kernel rather than a non-uniform one.

851 You have not changed the manuscript in response to my comment. Your response is useful, particularly: *"the*

- distance is computed between the two positions along the normal vector" and "average positions are
- 853 *defined as the arithmetic mean*". You could easily amend the sentence to clarify this: "...as the distance of
- 854 the arithmetic mean positions of the two point clouds along the normal vector".
- 855 Indeed, as we explained in our answer, we did not change the text as this is presented in the original M3C2

paper which we consider is a classical paper (> 700 citations) that should be read by anyone wanting to use our

857 workflow. So this should not have been a surprise for the referee that we did not change the text. That being

858 said, we have modified the sentence presenting this aspect of M3C2 following the suggestion of the referee

- 859 (L230)
- 860 *L139: "if not intercept is found...": This is not clear to me. Do you mean 'if the cylinder does not intersect any*

861 points in the second surface? Why would this happen? Does this only occur at the boundaries of the point

862 clouds? How do these intersection failures influence the surface differencing and how do you report them in863 your later analysis?

- 864 In LiDAR datasets, the density of points is non-uniform over the entire point cloud. Consequently, missing data
- $\frac{865}{1000}$  or very low point density (<5 pt/m2) can occur inside the point cloud due to the absence of laser impact on the

ground during the data acquisition. This mostly occur in dense vegetated areas or water surface areas (fortopographic lidar). When performing M3C2, it is thus possible that the cylinder cannot intercept points or just a

- few (< 5). In both cases, the M3C2 distance will not be considered significant. In areas with low point density
- $(<5 \text{ pts/m}^2)$  a solution is to perform M3C2 with a larger projection scale *d* to include more points in the distance
- and statistic calculations.

- 871 You have not updated the text to reflect this discussion. You do mean: 'if the cylinder intersects <5 points in
- the second point cloud'. If so, this should be added to the paper. Your explanation above is useful but I
- 873 understand it to mean something different to what you say in the paper.
- 874 We have clarified this classifiable feature of the M3C2 algorithm by specifying that no distance is computed in
- 875 the two point clouds do not overlap or if one of the point cloud has missing data (L233). A distance is always
- 876 computed if there is at least one point in each cloud, however a distance uncertainty is calculated only if there
- 877 are more than 5 points in each cloud. This point is mentioned later in the paragraph (L268)
- 878 L140: "provides uncertainty": what is the basis / justification for the uncertainty estimate taking this particular
- 879 form? It looks familiar as it has some similarities to a confidence interval but also some differences. This
- threshold is important to explain and justify in detail because it is used to threshold discrete landslides in the
- 881 following analysis. Why threshold at 95% confidence? What is the impact on your findings (total volumes and
- scaling relationships) of thresholding at a difference CI (e.g. 99 or 90% confidence)?
- 883 We refer the reviewer to the original M3C2 paper which has an extensive discussion on the confidence interval,
- and how to consider surface roughness, point cloud errors, point density and registration in the context of change
- detection on 3D point clouds. The threshold has been set to 95% to build the segmentation on as many good
  points as possible. We do not believe that changing this threshold to 90 or 99% significantly change the landslide
- statistics given the results of the sensitivity analyses of parameters (reg,  $D_m$  and SNR threshold) that mainly
- 888 control the landslide inventory.
- First, I think you can evidence your statement above from your sensitivity analysis and doing so will
- 890 strengthen the paper. You demonstrate that changes to the LoD do not alter your main findings and the
- changes you explore cover the range that you would expect from changing the threshold CI from 90-99%
- 892 (i.e. 0.39-0.52 m). This strengthens your argument that the statistics are robust to model choices.
- 893 We do not think it is relevant or necessary to discuss why choosing 95% is better than another confidence level
- as this is in the end an arbitrary choice. As the referee emphasizes himself, we show in the discussion that ourresults are robust to the choice of reg (which translates into much larger LoD). The MS is already quite long, so
- we have not modified the text.
- 897 Second, having read the original M3C2 paper (Lague et al., 2013), it discusses the confidence interval and
- registration error but took a different approach to estimating registration error so it is difficult to
- translate directly between the two papers. They provide only a brief description of the theoretical basis for the LoD equation citing a statistics textbook.
- 901 I found the description of James et al (2017) who you cite and who cite Lague et al. (2013) very useful:
- 902 *"where reg is the relative overall registration error between the surveys, assumed isotropic and spatially*
- 903 uniform (Lague et al., 2013). Note that Lague et al. (2013) took a conservative approach by adding reg
- 904 directly (as a potential systematic bias)". A similar statement would be useful in your paper to explain that 905 you estimate local uncorrelated random errors using the first two terms and systematic errors under the 906 assumption that they introduce a spatially uniform bias with the final term.
- 907 Anderson (2019) describes an approach similar to yours, but highlights spatially correlated random errors
- 908 as a key component within error analysis for surface differencing and includes this as a term in his total
- 909 error calculations (eqn 21). This term is missing from your error propagation but seems likely to be very
- 910 important, particularly because you are interested in the size of thresholded difference patches. Anderson
- 911 (2019) also argues for direct characterisation of errors within 'stable areas' similar to my suggestion in
   912 MC4.
- 913 Anderson, S.W., 2019. Uncertainty in quantitative analyses of topographic change: error propagation and the 914 role of 1245 thresholding. *Earth Surface Processes and Landforms*, 44(5), pp.1015-1033.
- 915 Here we have added much more justification and explanations on our choice of *reg*, as well as the form of the
- 916 LoD95% in section 3.1. This changes are also tightly linked to our modifications in relation to the evaluation of
- 917 elevation errors. We now cite Anderson 2019. His approach is interesting, but also amount at a spatially
- 918 uniform *reg* as we explain in the text. The best way to improve the level of detection would be, as already
- 919 stated in the discussion (section 5.1.2 registration error), to build a spatially explicit direct elevation error
- 920 models for each survey.
- 921 *L145: "reg" is quantified using the standard deviation of differences between the surfaces. I think it would be*
- 922 helpful here and elsewhere to use similar notation for the registration error to the other errors being examined
- 923 here. Why are the local terms converted to standard errors but reg is left as a standard deviation? Finally, the
- 924 *length scale over which reg was calculated would seem to be important here.*

- 925 We choose to keep the "reg" notation to specifically refer to the registration error when needed in the
- manuscript. We now provide an explanation in the text as to why reg is not converted as a standard error (L 166).
- 927 Reg is not measured over a length scale, it is based on the standard deviation of the 3D-M3C2 distance between
- 2 clouds. This part is now explained in greated detail in section 3.3.1 where we discuss the notion of intra-survey and inter-survey registration quality.
- 930 You now say: "The M3C2 definition of the LoD95% makes the conservative choice of adding the registration 931 error to the combined standard error related to point cloud roughness, rather than taking the square root of
- 932 the sum of squared standard error and squared registration."
- 933 First, I agree that your equation is conservative in the sense that it results in a larger LoD but I don't see a
- 934 justification for this functional form either here or in Lague et al. (2013).
- 935 This is an arbitrary choice to make the uncertainty model more conservative. This is now clearly stated L307:
- 936 This arbitrary choice similar to Lague et al. (2013) ensures that the frequency of false detection of statistically
- 937 significant change is below 5%, at the expense of a reduced capacity to detect real small topographic changes
  938 close to the LoD95%.
- 939 The best description of a framework for propagation of both random and systematic errors that I can find is
- 940 in Anderson (2018). It is similar to yours in its approximation of random and systematic errors (see
- Anderson's eqns 12 and 20) but differs from yours in how these are combined (see eqns 21-22). Can you
- 942 explain the difference?
- 943 As above this is an arbitrary choice and we were explaining it in the previous version of the MS (L304): *the*
- 944 *M3C2* definition of the LoD95% makes the conservative choice of adding reg to the combined standard
- 945 *error related to point cloud roughness, rather than taking the square root of the sum of squared*
- **946** *standard error and squared registration error (e.g., Anderson, 2019; Joerg et al., 2012)*
- 947 Second, you have not explained why standard deviation rather than standard error is used for the
  948 systematic error (reg). I think this is because standard errors are used to approximate random errors
  949 (under the assumption that they are uncorrelated) but standard deviations are used for systematic errors
  950 under the assumption that these are perfectly correlated (see Anderson's eqns 1-4, 12 and 20). If that is
  951 the case it would be helpful to explain it in the text.
- Indeed, this was implicit, because taking the standard error with > 1000 core points over which we calculate
   M3C2 would result in reg ~0. We now state explicitly that we treat reg as a systematic error (L276) and that it is
   the standard deviation of M3C2 distances (L301).
- 956
- Third, I'm happy for you to retain a notation that is specific to registration error but suggest sigma with reg
   as subscript would make it clearer that this is a standard deviation.
- 959 Thanks for the suggestion, but we did not follow it up, because adding a sigma to reg could be confused as the 960 standard deviation of *reg*.
- 961 *L165: "not deemed interesting": I don't think that this is the right phrase, can you rephrase? What was the*
- 962 *impact on your results of applying* d=10 *m throughout?*
- 963 The sentence has been modified. Applying d=10 m increases the landslide source and deposit by 1% and 0.7%
- 964 respectively. In terms of volume this represents an increase of 2% and 0.89% respectively.
- 965 I do not see where you have modified the text to reflect this response.
- 966 We did not change the text, and we still haven't because we think these are unnecessary details.
- 967 *L173: "may result in ...": How do you identify when this has happened? What is the objective that you are optimising?*
- 969 This can be identified in the bottom of narrow valleys and top of very steep divides where no evidence of mass
- 970 movement processing can be identified by visual inspection of orthophotos. These cases are now filtered by971 using the SNR metric as they have very large LoD95% (see section 3.3.4).
- 972 This is still not clear both in the response and in the text (which has changed very little). L211 you say "This is
- 973 generally obtained by trial and error". This was what I was referring to when I asked "what is the objective
- 974 *was that you are optimising"* by trial and error. Re-reading the text I am not sure whether it was obtained by
- 975 trial and error in this paper. If not then how did you find out that the maximum observed change in the study

- area was 30 m? If pmax is designed to prevent anomalously large changes how do you verify that a change is
- 977 an anomaly? Something seems ircular in the argument as it is currently presented. I don't understand the
- 978 connection to SNR filtering and that doesn't currently feature in the text.

979 The rare issues we had with the projection cylinder intercepting twice the same cloud has been sorted out with
980 a modification of the M3C2 algorithm. We have removed the line referring to this issue, and added information
981 on the algorithm at lines L237-241.

L196: "standard deviation..." These stable areas would be an excellent test of your landslide detection method,
indicating the scaling relationships, size distributions, and total volumes generated by artefacts alone.
This analyse has not been performed due to the changes we made on how we manage artefacts with SNR

filtering (see section 3.3.4). Applied on stable area, the workflow does not detect any landslide.

- 986 This result is definitely worth reporting. If there are any remaining patches then you should report the size 987 distribution of these patches as they give an indication of the expected bias that elevation errors will
- 988 introduce into your size distribution.
- 989 There are no landslide source remaining on the stable area in the final inventory. False detections on stable

areas are studied through the metrics we analyze in comparison to true landslides. We also now report the

- pdf(A) of false detections (fig. 14a) for all areas (there are not enough on stable areas to construct a robustpdf(A)
- L199-200: "The registration error...": this definition should come earlier. It is important for interpreting eqn 2.
  We disagree with this comment as the registration error "reg" can be defined differently depending on the
- application of M3C2. In section 3.1, we aim to give a general description of the M3C2 algorithm.
- 996 I agree that it can be defined differently but you are reporting your method rather than the method in 997 general so you should define it as you have used it.
- reg is now define much more extensively and in the first section of the workflow (section 3.1, L244-276)
- 999 *L215: 15-20% it might be useful to show the location of these points on one of your maps.*

1000 The percentage of these points on steep hillslope has been revised and actually represents up to 12% of steep

- 1001 slopes.
- 1002 It would still be useful to show the location of the points on one of your maps.
- 1003 Useful maybe, but we don't think it is necessary nor actually feasible to show them clearly on a map.

L256: "deposit areas" Some of these areas do not have any upslope erosion. How do you explain this? See MC5
 Some deposit areas can be detected without an associated upslope erosion. A possible reason is that the surface
 change associated to these deposit areas is sufficient to be above the LoD95% but not for the upslope erosion area.
 This is useful discussion, I couldn't see it in the revised manuscript. It would be useful in your description of
 Figure 5.

- 1009 With our new filtering metrics, a deposit of the final inventory cannot exist if it does not have a upstream valid1010 source
- 1011 *L315: It is not surprising that your depth area scaling relationship is gentler than that of Larsen since you*
- 1012 censor core points with difference < 0.33, making it impossible for small landslides to also be shallow.
- 1013 Indeed, and we now discuss this. In particular, the relationship is barely different if we consider a SNR=1,
- 1014 meaning that compared to the 2D inventory, we miss 32 shallow landslide sources over an inventory of 1270
- sources (SNR >=1). Hence, even if the 3d inventory had captured the very shallow landslides that the 2D
- 1016 mapping did capture, it would hardly change the scaling relationship. One could also say that previous 2D
- 1017 inventories have significantly underevaluated the number of small landslides, which in turns affect the
- 1018 representativity of published V-A relationships from 2D inventories.
- 1019 This is useful discussion, I couldn't see it in the revised manuscript but I wasn't sure whether I was looking
- 1020 for parts of the text in your response above or something different.
- 1021 With our new cleaned inventory, our D-A scaling is very close to the Larsen et al. (2020) relationship for soils
- 1022 (fig 11c) The original comment is irrelevant. We do not want to discuss potential causes for differences, when
- 1023 we do not observe one.

| 1025                                                                                                                                                 | L460: "deposits form more concentrated and thicker patches": This result is surprising and should be compared
with expectations from other studies on landslide runout.                                                                                                                                                                                                                                                                                                                                                                                                                                                                                                                                                                                                                                                                                                                                                                                                                                                                                                                                                                                                                                                                                                                                                                                                                                                                                                                                                                                                                                                                                                                                                                                                                                                                                                                                                                                                                                                                                                                                           |
|------------------------------------------------------------------------------------------------------------------------------------------------------|----------------------------------------------------------------------------------------------------------------------------------------------------------------------------------------------------------------------------------------------------------------------------------------------------------------------------------------------------------------------------------------------------------------------------------------------------------------------------------------------------------------------------------------------------------------------------------------------------------------------------------------------------------------------------------------------------------------------------------------------------------------------------------------------------------------------------------------------------------------------------------------------------------------------------------------------------------------------------------------------------------------------------------------------------------------------------------------------------------------------------------------------------------------------------------------------------------------------------------------------------------------------------------------------------------------------------------------------------------------------------------------------------------------------------------------------------------------------------------------------------------------------------------------------------------------------------------------------------------------------------------------------------------------------------------------------------------------------------------------------------------------------------------------------------------------------------------------------------------------------------------------------------------------------------------------------------------------------------------------------------------------------------------------------------------------------------------------------------------------------|
| 1026                                                                                                                                                 | data in terms of mean 3D thickness of the denosits and sources                                                                                                                                                                                                                                                                                                                                                                                                                                                                                                                                                                                                                                                                                                                                                                                                                                                                                                                                                                                                                                                                                                                                                                                                                                                                                                                                                                                                                                                                                                                                                                                                                                                                                                                                                                                                                                                                                                                                                                                                                                                       |
| 1027                                                                                                                                                 | This comment has not been addressed. I cannot find new text comparing your results with expectations from                                                                                                                                                                                                                                                                                                                                                                                                                                                                                                                                                                                                                                                                                                                                                                                                                                                                                                                                                                                                                                                                                                                                                                                                                                                                                                                                                                                                                                                                                                                                                                                                                                                                                                                                                                                                                                                                                                                                                                                                            |
| 1029                                                                                                                                                 | other studies.                                                                                                                                                                                                                                                                                                                                                                                                                                                                                                                                                                                                                                                                                                                                                                                                                                                                                                                                                                                                                                                                                                                                                                                                                                                                                                                                                                                                                                                                                                                                                                                                                                                                                                                                                                                                                                                                                                                                                                                                                                                                                                       |
| 1030                                                                                                                                                 | We have removed this sentence because our analysis and treatment of deposits is not as detailed as sources.                                                                                                                                                                                                                                                                                                                                                                                                                                                                                                                                                                                                                                                                                                                                                                                                                                                                                                                                                                                                                                                                                                                                                                                                                                                                                                                                                                                                                                                                                                                                                                                                                                                                                                                                                                                                                                                                                                                                                                                                          |
| 1031
1032                                                                                                                                         | L466: Whether the right tail is a "power law" or not is debated. See Medwedeff et al. (2020) among others. The reference has been added in the text.                                                                                                                                                                                                                                                                                                                                                                                                                                                                                                                                                                                                                                                                                                                                                                                                                                                                                                                                                                                                                                                                                                                                                                                                                                                                                                                                                                                                                                                                                                                                                                                                                                                                                                                                                                                                                                                                                                                                                                 |
| 1033                                                                                                                                                 | This comment has not been addressed. You have not altered the text to recognise the debate around the                                                                                                                                                                                                                                                                                                                                                                                                                                                                                                                                                                                                                                                                                                                                                                                                                                                                                                                                                                                                                                                                                                                                                                                                                                                                                                                                                                                                                                                                                                                                                                                                                                                                                                                                                                                                                                                                                                                                                                                                                |
| 1034                                                                                                                                                 | form of the right tail. I don't think you can simply add Medwedeff to the current citation. It reads as though                                                                                                                                                                                                                                                                                                                                                                                                                                                                                                                                                                                                                                                                                                                                                                                                                                                                                                                                                                                                                                                                                                                                                                                                                                                                                                                                                                                                                                                                                                                                                                                                                                                                                                                                                                                                                                                                                                                                                                                                       |
| 1035                                                                                                                                                 | Medwedeff et al. are among those arguing for power law scaling when I understand their paper to argue the                                                                                                                                                                                                                                                                                                                                                                                                                                                                                                                                                                                                                                                                                                                                                                                                                                                                                                                                                                                                                                                                                                                                                                                                                                                                                                                                                                                                                                                                                                                                                                                                                                                                                                                                                                                                                                                                                                                                                                                                            |
| 1036                                                                                                                                                 | opposite.                                                                                                                                                                                                                                                                                                                                                                                                                                                                                                                                                                                                                                                                                                                                                                                                                                                                                                                                                                                                                                                                                                                                                                                                                                                                                                                                                                                                                                                                                                                                                                                                                                                                                                                                                                                                                                                                                                                                                                                                                                                                                                            |
| 1037                                                                                                                                                 | Indeed, this was ambiguous. We have added the following text: "and although the power-law nature of the                                                                                                                                                                                                                                                                                                                                                                                                                                                                                                                                                                                                                                                                                                                                                                                                                                                                                                                                                                                                                                                                                                                                                                                                                                                                                                                                                                                                                                                                                                                                                                                                                                                                                                                                                                                                                                                                                                                                                                                                              |
| 1038                                                                                                                                                 | tail is debated (Jeandet et al., 2019; Medwedeff et al., 2020)"                                                                                                                                                                                                                                                                                                                                                                                                                                                                                                                                                                                                                                                                                                                                                                                                                                                                                                                                                                                                                                                                                                                                                                                                                                                                                                                                                                                                                                                                                                                                                                                                                                                                                                                                                                                                                                                                                                                                                                                                                                                      |
| 1039                                                                                                                                                 |                                                                                                                                                                                                                                                                                                                                                                                                                                                                                                                                                                                                                                                                                                                                                                                                                                                                                                                                                                                                                                                                                                                                                                                                                                                                                                                                                                                                                                                                                                                                                                                                                                                                                                                                                                                                                                                                                                                                                                                                                                                                                                                      |
| 1040                                                                                                                                                 | Figures                                                                                                                                                                                                                                                                                                                                                                                                                                                                                                                                                                                                                                                                                                                                                                                                                                                                                                                                                                                                                                                                                                                                                                                                                                                                                                                                                                                                                                                                                                                                                                                                                                                                                                                                                                                                                                                                                                                                                                                                                                                                                                              |
| 1041                                                                                                                                                 | Fig 5: The 3D minimum volume line needs explaining in the caption. I would have expected this line to be                                                                                                                                                                                                                                                                                                                                                                                                                                                                                                                                                                                                                                                                                                                                                                                                                                                                                                                                                                                                                                                                                                                                                                                                                                                                                                                                                                                                                                                                                                                                                                                                                                                                                                                                                                                                                                                                                                                                                                                                             |
| 1042                                                                                                                                                 | oriented parallel to the depth contours since minimum volume for a given area is set by minimum detectable                                                                                                                                                                                                                                                                                                                                                                                                                                                                                                                                                                                                                                                                                                                                                                                                                                                                                                                                                                                                                                                                                                                                                                                                                                                                                                                                                                                                                                                                                                                                                                                                                                                                                                                                                                                                                                                                                                                                                                                                           |
| 1043                                                                                                                                                 | depth. If so the minimum detectable volume might explain the sharp lower boundary to the volume area point                                                                                                                                                                                                                                                                                                                                                                                                                                                                                                                                                                                                                                                                                                                                                                                                                                                                                                                                                                                                                                                                                                                                                                                                                                                                                                                                                                                                                                                                                                                                                                                                                                                                                                                                                                                                                                                                                                                                                                                                           |
| 1044                                                                                                                                                 | cloud. Actually, what it is illustrated here is the 3D minimum volume that can be measured here given the minimum area $(20 \text{ m}^2)$ that we consider and the minimum configuration $1240 \text{ denth}$ ( $0.4 \text{ m}$ ).                                                                                                                                                                                                                                                                                                                                                                                                                                                                                                                                                                                                                                                                                                                                                                                                                                                                                                                                                                                                                                                                                                                                                                                                                                                                                                                                                                                                                                                                                                                                                                                                                                                                                                                                                                                                                                                                            |
| 1045                                                                                                                                                 | The minimum area and denth act together to set an absolute minimum detectable volume (when both area                                                                                                                                                                                                                                                                                                                                                                                                                                                                                                                                                                                                                                                                                                                                                                                                                                                                                                                                                                                                                                                                                                                                                                                                                                                                                                                                                                                                                                                                                                                                                                                                                                                                                                                                                                                                                                                                                                                                                                                                                 |
| 1040                                                                                                                                                 | and denth are minimised). However, there is also a denth dependent minimum detectable volume that is set                                                                                                                                                                                                                                                                                                                                                                                                                                                                                                                                                                                                                                                                                                                                                                                                                                                                                                                                                                                                                                                                                                                                                                                                                                                                                                                                                                                                                                                                                                                                                                                                                                                                                                                                                                                                                                                                                                                                                                                                             |
| 1047                                                                                                                                                 | by the denth constraint alone. At present the horizontal line that you use to highlight the absolute minimum                                                                                                                                                                                                                                                                                                                                                                                                                                                                                                                                                                                                                                                                                                                                                                                                                                                                                                                                                                                                                                                                                                                                                                                                                                                                                                                                                                                                                                                                                                                                                                                                                                                                                                                                                                                                                                                                                                                                                                                                         |
| 1040                                                                                                                                                 | volume might be misinterpreted by some to be the area dependent minimum volume. You can easily fix this                                                                                                                                                                                                                                                                                                                                                                                                                                                                                                                                                                                                                                                                                                                                                                                                                                                                                                                                                                                                                                                                                                                                                                                                                                                                                                                                                                                                                                                                                                                                                                                                                                                                                                                                                                                                                                                                                                                                                                                                              |
| 1050                                                                                                                                                 | by adding the area dependent minimum volume. It will be a straight line in log-log snace and will nass                                                                                                                                                                                                                                                                                                                                                                                                                                                                                                                                                                                                                                                                                                                                                                                                                                                                                                                                                                                                                                                                                                                                                                                                                                                                                                                                                                                                                                                                                                                                                                                                                                                                                                                                                                                                                                                                                                                                                                                                               |
| 1051                                                                                                                                                 | through the points (20.8 and 20000.8000). It would be useful to include this on the Figure and would also                                                                                                                                                                                                                                                                                                                                                                                                                                                                                                                                                                                                                                                                                                                                                                                                                                                                                                                                                                                                                                                                                                                                                                                                                                                                                                                                                                                                                                                                                                                                                                                                                                                                                                                                                                                                                                                                                                                                                                                                            |
| 1052                                                                                                                                                 | bring make the interpretation of the dashed lines in the figure and inset internally consistent.                                                                                                                                                                                                                                                                                                                                                                                                                                                                                                                                                                                                                                                                                                                                                                                                                                                                                                                                                                                                                                                                                                                                                                                                                                                                                                                                                                                                                                                                                                                                                                                                                                                                                                                                                                                                                                                                                                                                                                                                                     |
| 1053                                                                                                                                                 | This has been added to figure 11C (initially figure 5 in the first version of the MS)                                                                                                                                                                                                                                                                                                                                                                                                                                                                                                                                                                                                                                                                                                                                                                                                                                                                                                                                                                                                                                                                                                                                                                                                                                                                                                                                                                                                                                                                                                                                                                                                                                                                                                                                                                                                                                                                                                                                                                                                                                |
| 1054                                                                                                                                                 |                                                                                                                                                                                                                                                                                                                                                                                                                                                                                                                                                                                                                                                                                                                                                                                                                                                                                                                                                                                                                                                                                                                                                                                                                                                                                                                                                                                                                                                                                                                                                                                                                                                                                                                                                                                                                                                                                                                                                                                                                                                                                                                      |
| 1055                                                                                                                                                 | Supplement                                                                                                                                                                                                                                                                                                                                                                                                                                                                                                                                                                                                                                                                                                                                                                                                                                                                                                                                                                                                                                                                                                                                                                                                                                                                                                                                                                                                                                                                                                                                                                                                                                                                                                                                                                                                                                                                                                                                                                                                                                                                                                           |
| 1056                                                                                                                                                 |                                                                                                                                                                                                                                                                                                                                                                                                                                                                                                                                                                                                                                                                                                                                                                                                                                                                                                                                                                                                                                                                                                                                                                                                                                                                                                                                                                                                                                                                                                                                                                                                                                                                                                                                                                                                                                                                                                                                                                                                                                                                                                                      |
|                                                                                                                                                      | C1. this tast should definitely be included within the neuron itself this is a last neut of your method. You                                                                                                                                                                                                                                                                                                                                                                                                                                                                                                                                                                                                                                                                                                                                                                                                                                                                                                                                                                                                                                                                                                                                                                                                                                                                                                                                                                                                                                                                                                                                                                                                                                                                                                                                                                                                                                                                                                                                                                                                         |
| 1057                                                                                                                                                 | S1: this text should definitely be included within the paper itself this is a key part of your method. You should also report the parameters that you used for this analysis and the parameter values that you chose                                                                                                                                                                                                                                                                                                                                                                                                                                                                                                                                                                                                                                                                                                                                                                                                                                                                                                                                                                                                                                                                                                                                                                                                                                                                                                                                                                                                                                                                                                                                                                                                                                                                                                                                                                                                                                                                                                 |
| 1057
1058
1059                                                                                                                                 | S1: this text should definitely be included within the paper itself this is a key part of your method. You should also report the parameters that you used for this analysis and the parameter values that you chose, preferably with a justification.                                                                                                                                                                                                                                                                                                                                                                                                                                                                                                                                                                                                                                                                                                                                                                                                                                                                                                                                                                                                                                                                                                                                                                                                                                                                                                                                                                                                                                                                                                                                                                                                                                                                                                                                                                                                                                                               |
| 1057
1058
1059
1060                                                                                                                         | S1: this text should definitely be included within the paper itself this is a key part of your method. You should also report the parameters that you used for this analysis and the parameter values that you chose, preferably with a justification.                                                                                                                                                                                                                                                                                                                                                                                                                                                                                                                                                                                                                                                                                                                                                                                                                                                                                                                                                                                                                                                                                                                                                                                                                                                                                                                                                                                                                                                                                                                                                                                                                                                                                                                                                                                                                                                               |
| 1057
1058
1059
1060
1061                                                                                                                 | S1: this text should definitely be included within the paper itself this is a key part of your method. You should also report the parameters that you used for this analysis and the parameter values that you chose, preferably with a justification.
As explained above (response to minor comment L63), the improvement of the classification is not a key part of                                                                                                                                                                                                                                                                                                                                                                                                                                                                                                                                                                                                                                                                                                                                                                                                                                                                                                                                                                                                                                                                                                                                                                                                                                                                                                                                                                                                                                                                                                                                                                                                                                                                                                                                             |
| 1057
1058
1059
1060
1061
1062                                                                                                         | <li>S1: this text should definitely be included within the paper itself this is a key part of your method. You should also report the parameters that you used for this analysis and the parameter values that you chose, preferably with a justification.</li><li>As explained above (response to minor comment L63), the improvement of the classification is not a key part of our method. To keep the method part of the article as compact as possible, and given that we have significantly</li>                                                                                                                                                                                                                                                                                                                                                                                                                                                                                                                                                                                                                                                                                                                                                                                                                                                                                                                                                                                                                                                                                                                                                                                                                                                                                                                                                                                                                                                                                                                                                                                                      |
| 1057
1058
1059
1060
1061
1062
1063                                                                                                 | S1: this text should definitely be included within the paper itself this is a key part of your method. You should also report the parameters that you used for this analysis and the parameter values that you chose, preferably with a justification. As explained above (response to minor comment L63), the improvement of the classification is not a key part of our method. To keep the method part of the article as compact as possible, and given that we have significantly lengthen the MS with new additions, we have kept this part in the supplement. Our description of the method and have a supplement be available for even the supplement.                                                                                                                                                                                                                                                                                                                                                                                                                                                                                                                                                                                                                                                                                                                                                                                                                                                                                                                                                                                                                                                                                                                                                                                                                                                                                                                                                                                                                                                        |
| 1057
1058
1059
1060
1061
1062
1063
1064                                                                                         | S1: this text should definitely be included within the paper itself this is a key part of your method. You should also report the parameters that you used for this analysis and the parameter values that you chose, preferably with a justification. As explained above (response to minor comment L63), the improvement of the classification is not a key part of our method. To keep the method part of the article as compact as possible, and given that we have significantly lengthen the MS with new additions, we have kept this part in the supplement. Our description of the method and how we choose parameters should be enough for anyone to reproduce. We also now provide a classical reference to grow the interacted reacter can be prevented by a supervised of the prevented reacter can be prevented on the prevented of the prevented reacter can be prevented on the prevented of the prevented reacter can be prevented on the prevented of the prevented reacter can be prevented on the prevented of the prevented reacter can be prevented on the prevented of the prevented reacter can be prevented on the prevented of the prevented reacter can be prevented on the prevented of the prevented reacter can be prevented on the pre |
| 1057
1058
1059
1060
1061
1062
1063
1064
1065
1066                                                                         | S1: this text should definitely be included within the paper itself this is a key part of your method. You should also report the parameters that you used for this analysis and the parameter values that you chose, preferably with a justification. As explained above (response to minor comment L63), the improvement of the classification is not a key part of our method. To keep the method part of the article as compact as possible, and given that we have significantly lengthen the MS with new additions, we have kept this part in the supplement. Our description of the method and how we choose parameters should be enough for anyone to reproduce. We also now provide a classical reference to ground classification in the main text that the interested reader can follow.                                                                                                                                                                                                                                                                                                                                                                                                                                                                                                                                                                                                                                                                                                                                                                                                                                                                                                                                                                                                                                                                                                                                                                                                                                                                                                                  |
| 1057
1058
1059
1060
1061
1062
1063
1064
1065
1066
1067                                                                 |  <li>S1: this text should definitely be included within the paper itself this is a key part of your method. You should also report the parameters that you used for this analysis and the parameter values that you chose, preferably with a justification.</li> <li>As explained above (response to minor comment L63), the improvement of the classification is not a key part of our method. To keep the method part of the article as compact as possible, and given that we have significantly lengthen the MS with new additions, we have kept this part in the supplement. Our description of the method and how we choose parameters should be enough for anyone to reproduce. We also now provide a classical reference to ground classification in the main text that the interested reader can follow.</li> <li>L11: "4 standard deviations" I have three questions here: 1) what sample is the standard deviation being</li>                                                                                                                                                                                                                                                                                                                                                                                                                                                                                                                                                                                                                                                                                                                                                                                                                                                                                                                                                                                                                                                                                                                                                                    |
| 1057
1058
1059
1060
1061
1062
1063
1064
1065
1066
1067
1068                                                         |  <li>S1: this text should definitely be included within the paper itself this is a key part of your method. You should also report the parameters that you used for this analysis and the parameter values that you chose, preferably with a justification.</li> <li>As explained above (response to minor comment L63), the improvement of the classification is not a key part of our method. To keep the method part of the article as compact as possible, and given that we have significantly lengthen the MS with new additions, we have kept this part in the supplement. Our description of the method and how we choose parameters should be enough for anyone to reproduce. We also now provide a classical reference to ground classification in the main text that the interested reader can follow.</li> <li>L11: "4 standard deviations" I h